# Implicit Bias of Gradient Descent for Non-Homogeneous Deep Networks

**Yuhang Cai** [* 1]  **Kangjie Zhou** [* 2]  **Jingfeng Wu** [1]  **Song Mei** [1]  **Michael Lindsey** [1 3]  **Peter L. Bartlett** [1 4]

## Abstract

We establish the asymptotic implicit bias of gradient descent (GD) for generic non-homogeneous deep networks under exponential loss. Specifically, we characterize three key properties of GD iterates starting from a sufficiently small empirical risk, where the threshold is determined by a measure of the network's non-homogeneity. First, we show that a normalized margin induced by the GD iterates increases nearly monotonically. Second, we prove that while the norm of the GD iterates diverges to infinity, the iterates themselves converge in direction. Finally, we establish that this directional limit satisfies the Karush–Kuhn–Tucker (KKT) conditions of a margin maximization problem. Prior works on implicit bias have focused exclusively on homogeneous networks; in contrast, our results apply to a broad class of non-homogeneous networks satisfying a mild near-homogeneity condition. In particular, our results apply to networks with residual connections and non-homogeneous activation functions, thereby resolving an open problem posed by Ji & Telgarsky (2020).

## 1. Introduction

Deep networks often have an enormous amount of parameters and are theoretically capable of *overfitting* the training data. However, in practice, deep networks trained via *gradient descent* (GD) or its variants often generalize well. This is commonly attributed to the *implicit bias* of GD, in which GD finds a certain solution that prevents overfitting (Zhang et al., 2021; Neyshabur et al., 2017; Bartlett et al., 2021). Understanding the implicit bias of GD is one of the central topics in deep learning theory.

The implicit bias of GD is relatively well-understood when the network is *homogeneous* (see Soudry et al., 2018; Ji & Telgarsky, 2018; Lyu & Li, 2020; Ji & Telgarsky, 2020; Wu et al., 2023, and references therein). For linear networks trained on linearly separable data, GD diverges in norm while converging in direction to the maximum margin solution (Soudry et al., 2018; Ji & Telgarsky, 2018; Wu et al., 2023). Similar results have been established for generic homogeneous networks that include a class of deep networks, assuming that the network at initialization can separate the training data. Specifically, Lyu & Li (2020) showed that the normalized margin induced by GD increases nearly monotonically, and that the limiting direction of a subsequence of the GD iterates satisfies the Karush-Kuhn-Tucker (KKT) condition of a margin maximization problem. Moreover, if the network is definable in an o-minimal structure (see Section 2, which is satisfied for most networks), Ji & Telgarsky (2020) showed that *gradient flow* (GF, that is, GD with infinitesimal stepsizes) converges in direction and that the limiting direction aligns with the direction of the gradient.

However, the implicit bias of GD remains largely unknown when the network is *non-homogeneous*, an arguably more common case in deep learning (there are a few exceptions, which will be discussed later in Section 1.1). For instance, networks with residual connections or non-homogeneous activation functions are inherently non-homogeneous. As posted as an open problem by Ji & Telgarsky (2020), it is unclear whether the implicit bias results developed for homogeneous networks can be extended to non-homogeneous networks.

**Contributions.**   In this work, we establish the implicit bias of GD for a broad set of non-homogeneous but definable networks under exponential loss. Starting from the simpler case of *gradient flow* (GF), we identify two natural conditions of *near-homogeneity* and *strong separability*, respectively. The former condition requires the homogeneous error to grow slower than the output of the network, and the latter condition requires GF to attain a sufficiently small empirical risk depending on the homogeneous error of the network. Under these two conditions, we prove the following implicit bias results of GF for non-homogeneous networks:

1. GF induces a normalized margin that increases nearly

---

*Equal contribution   [1]University of California, Berkeley  [2]Columbia University  [3]Lawrence Berkeley National Laboratory  [4]Google DeepMind. Correspondence to: Yuhang Cai <willcai@berkeley.edu>, Kangjie Zhou <kz2326@columbia.edu>, Peter L. Bartlett <peter@berkeley.edu>.

*Proceedings of the $42^{nd}$ International Conference on Machine Learning*, Vancouver, Canada. PMLR 267, 2025. Copyright 2025 by the author(s).

monotonically.

2. GF converges in its direction while diverging in its norm.

3. The limiting direction of GF satisfies the KKT conditions of a margin maximization problem.

Our near-homogeneity condition covers many commonly used deep networks. In particular, it applies to networks with residual connections and nearly homogeneous activation functions. In addition, we provide structural rules for computing the near-homogeneity order of a network based on that of each layer in the network.

Our strong separability condition is a generalization of the separability condition used in the prior homogeneous analysis (Lyu & Li, 2020; Ji & Telgarsky, 2020). In particular, it reduces to the separability condition when the network is homogeneous. Later in Section 5, we demonstrate that this condition is satisfiable by any non-degenerate near-homogeneous network. Moreover, we show that GF reaches this strong separability condition from zero initialization for training a two-layer network with residual connections.

Finally, we extend the above results from GF to GD with an arbitrarily large stepsize under additional technical conditions. Altogether, we extend the implicit bias of GD from homogeneous cases (Lyu & Li, 2020; Ji & Telgarsky, 2020) to non-homogeneous cases, thereby addressing the open problem posted by Ji & Telgarsky (2020).

**Notation.** For two positive-valued functions $f(x)$ and $g(x)$, we write $f(x) \lesssim g(x)$ (or $f(x) \gtrsim g(x)$) if $f(x) \leq cg(x)$ (or $f(x) \geq cg(x)$) for some constant $c \in (0, +\infty)$. We write $f(x) \asymp g(x)$ if $f(x) \lesssim g(x) \lesssim f(x)$. We use $\mathbb{R}_{\geq 0}[x]$ to denote the set of all univariate polynomials with non-negative real coefficients. For a polynomial $\mathsf{p}$, we use $\deg \mathsf{p}$ to denote its degree. We use $\nabla$ to denote the gradient of a scalar function or the Jacobian of a vector-valued function. We use $\| \cdot \|$ to denote the $\ell^2$-norm of a vector or the operator norm of a matrix. We use $[n]$ to denote the set $\{1, 2, \ldots, n\}$. We use $\mathrm{conv}\mathcal{A}$ to denote the convex hull of a set $\mathcal{A}$ in Euclidean space. Throughout the paper, we define $\phi(x) := \log(1/(nx))$ and $\Phi(x) := \log \phi(x) - 2/\phi(x)$, where $n$ is the number of samples.

### 1.1. Related Works

We discuss related papers in the remainder of this section.

**Homogeneous Networks.** We first review prior implicit bias results for deep homogeneous networks. In this case, Lyu & Li (2020) showed that GD induces a nearly increasing normalized margin, and the direction of a subsequence of GD iterates converges, the limit of which can be characterized by the KKT conditions of a margin maximization

problem. Part of these results are generalized to *steepest descent* by Tsilivis et al. (2024). Using the notion of definability, Ji & Telgarsky (2020) further showed the directional convergence and alignment for gradient flow. In the special case of two-layer homogeneous networks, Chizat & Bach (2020) characterized the limiting direction of (Wasserstein) gradient flow in the large-width limit. For a two-layer Leaky ReLU network with symmetric data, Lyu et al. (2021) showed that gradient flow eventually converges to a linear classifier. Different from these works, we aim to establish implicit bias of GD for non-homogeneous deep networks.

**Non-homogeneous Networks.** Before our work, there are a few papers that extend the implicit bias results from homogeneous cases to certain special non-homogeneous cases (Nacson et al., 2019; Chatterji et al., 2021; Kunin et al., 2023; Cai et al., 2025). The works by Nacson et al. (2019); Kunin et al. (2023) considered a special non-homogeneous network, which is homogeneous when viewed as a function of a subset of the trainable parameters while other parameters are fixed. The homogeneity orders for different subsets of parameters might be different. However, their results cannot cover networks with many commonly used non-homogeneous activation functions. The work by Chatterji et al. (2021) showed the margin improvement for GD with small stepsizes for MLPs with a special type of near-homogeneous activation functions. As a consequence, their results do not allow networks that use general non-homogeneous activation functions or residual connections. In comparison, we handle a large class of non-homogeneous networks satisfying a natural definition of near-homogeneity, covering far more commonly used deep networks.

The work by Cai et al. (2025) is most relevant to us, in which they proved the margin improvement of GD for near-1-homogeneous networks (see their Assumption 1 and our Definition 1 in Section 2). Our work can be viewed as an extension of theirs by handling near-$M$-homogeneous networks for general $M \geq 1$, as well as proving that GF and large-stepsize GD converge in direction to the KKT point of a margin maximization problem.

## 2. Preliminaries

In this section, we set up the problem and introduce basic mathematical tools used in our analysis.

**Locally Lipschitz Functions and Clarke Subdifferential.** For a function $f : D \to \mathbb{R}$ defined on an open set $D$, we say $f$ is *locally Lipschitz* if, for every $x \in D$, there exists a neighborhood $U$ of $x$ such that $f|_U$ is Lipschitz continuous. By Rademacher's theorem, a locally Lipschitz function is differentiable almost everywhere (Borwein et al., 2000). The *Clarke subdifferential* of a locally Lipschitz function $f$

at $x \in D$ is defined as

$$\partial f(x) := \mathrm{conv}\bigg\{ \lim_{i \to \infty} \nabla f(x_i) : x = \lim_{i \to \infty} x_i,$$

$$\text{where } x_i \in D \text{ and } \nabla f(x_i) \text{ exists}\bigg\},$$

which is nonempty, convex, and compact (Clarke, 1975). In particular, if $f$ is continuously differentiable at $x$, then $\partial f(x) = \{\nabla f(x)\}$. Elements of $\partial f(x)$ are called *subgradients*.

**Gradient Flow.** Let $(\mathbf{x}_i, y_i)_{i=1}^n$ be a binary classification dataset, where $\mathbf{x}_i \in \mathbb{R}^d$ and $y_i \in \{\pm 1\}$ for all $i \in [n]$. We denote a network by $f(\boldsymbol{\theta}; \cdot) : \mathbb{R}^d \to \mathbb{R}$, where $\boldsymbol{\theta} \in \mathbb{R}^D$ are the trainable parameters. Thoughout the paper, we assume $f(\boldsymbol{\theta}; \mathbf{x}_i)$ is locally Lipschitz with respect to $\boldsymbol{\theta}$ for every $i \in [n]$. We focus on the empirical risk under the exponential loss defined as

$$\mathcal{L}(\boldsymbol{\theta}) := \frac{1}{n} \sum_{i=1}^n \ell\big(y_i f(\boldsymbol{\theta}; \mathbf{x}_i)\big), \quad \ell(x) := e^{-x}. \quad (1)$$

A curve $z$ from an interval $I$ to some Euclidean space $\mathbb{R}^m$ is called an *arc* if it is absolutely continuous on any compact subinterval of $I$. Clearly, the composition of an arc and a locally Lipschitz function is still an arc. Following Lyu & Li (2020); Ji & Telgarsky (2020), we define *gradient flow* as an arc from $[0, +\infty)$ to $\mathbb{R}^D$ that satisfies

$$\frac{\mathrm{d}\boldsymbol{\theta}_t}{\mathrm{d}t} \in -\partial\mathcal{L}(\boldsymbol{\theta}_t), \quad \text{for almost every } t \geq 0. \quad (\text{GF})$$

**Homogeneity and Near-Homogeneity.** Let $M \geq 1$ be an integer. Recall that a locally Lipschitz network $f(\boldsymbol{\theta}; \mathbf{x})$ is *M-homogeneous* (Lyu & Li, 2020; Ji & Telgarsky, 2020) if for every $\mathbf{x} \in (\mathbf{x}_i)_{i=1}^n$,

$$\text{for all } a > 0 \text{ and } \boldsymbol{\theta} \in \mathbb{R}^D, \; f(a\boldsymbol{\theta}; \mathbf{x}) = a^M f(\boldsymbol{\theta}; \mathbf{x}). \quad (2)$$

One can verify that the above is equivalent to: for every $\mathbf{x} \in (\mathbf{x}_i)_{i=1}^n$, $\boldsymbol{\theta} \in \mathbb{R}^D$, and $\boldsymbol{h} \in \partial_{\boldsymbol{\theta}} f(\boldsymbol{\theta}; \mathbf{x})$,

$$\langle \boldsymbol{h}, \boldsymbol{\theta} \rangle - M f(\boldsymbol{\theta}; \mathbf{x}) = 0. \quad (3)$$

We generalize this definition by introducing the following near-homogeneous condition, which provides a natural quantification of the homogeneity error of the network $f$:

**Definition 1** (Near-$M$-homogeneity). Let $M \geq 1$ be an integer. A network $f(\boldsymbol{\theta}; \mathbf{x})$ is called *near-M-homogeneous*, if there exist $\mathsf{p}, \mathsf{q} \in \mathbb{R}_{\geq 0}[x]$ with $\deg \mathsf{p}, \deg \mathsf{q} \leq M$ such that the following holds for every $\mathbf{x} \in (\mathbf{x}_i)_{i=1}^n$, $\boldsymbol{\theta} \in \mathbb{R}^D$, and $\boldsymbol{h} \in \partial_{\boldsymbol{\theta}} f(\boldsymbol{\theta}; \mathbf{x})$:

(A1). $|\langle \boldsymbol{h}, \boldsymbol{\theta} \rangle - M f(\boldsymbol{\theta}; \mathbf{x})| \leq \mathsf{p}'(\|\boldsymbol{\theta}\|)$;

(A2). $\|\boldsymbol{h}\| \leq \mathsf{q}'(\|\boldsymbol{\theta}\|)$;

(A3). $|f(\boldsymbol{\theta}; \mathbf{x})| \leq \mathsf{q}(\|\boldsymbol{\theta}\|)$.

We make a few remarks on Definition 1. First, our near-homogeneity condition is modified from (3) instead of (2). In this way, our near-homogeneity condition implicitly puts regularity conditions on the subgradient of the network, which will be useful in our analysis.

Second, every $M$-homogeneous (see (2)) network for $M \geq 1$ is also near-$M$-homogeneous. We see this by setting $\mathsf{p}(x) = 0$ and $\mathsf{q}(x) = C(x^M + 1)$ for a sufficiently large constant $C > 1$ in Definition 1.

Third, a near-$M$-homogeneous network is also near-$(M + 1)$-homogeneous according to Definition 1. Thus when we say a network is near-$M$-homogeneous, the degree $M$ should be interpreted as the minimum degree $M$ such that Definition 1 is satisfied. We will see that this interpretation is necessary when we introduce the strong separability condition in Section 3.

Finally, we point out that many commonly used deep networks are near-homogeneous but not homogeneous. Examples include networks using residual connections or non-homogeneous activation functions. This will be further elaborated in Section 4.

**O-minimal Structure and Definable Functions.** The o-minimal structure and definable functions were introduced to deep learning theory by Davis et al. (2020); Ji & Telgarsky (2020) for studying the convergence and implicit bias of subgradient methods. We briefly review these notions below.

An o-minimal structure is a collection $\mathcal{S} = (\mathcal{S}_n)_{n=1}^\infty$, where each $\mathcal{S}_n$ is a collection of subsets of $\mathbb{R}^n$ that satisfies the following properties. First, $\mathcal{S}_n$ contains all algebraic sets in $\mathbb{R}^n$, i.e., zero sets of real-coefficient polynomials on $\mathbb{R}^n$. Second, $\mathcal{S}_n$ is closed under finite union, finite intersection, complement, Cartesian product, and projection. Third, $\mathcal{S}_1$ consists of finite unions of open intervals and points in $\mathbb{R}^1$. A set $A \subset \mathbb{R}^n$ is *definable* if $A \in \mathcal{S}_n$. A function $f : D \to \mathbb{R}^m$ with $D \subset \mathbb{R}^n$ is *definable* if its graph is in $\mathcal{S}_{n+m}$. See Appendix A for a more detailed introduction.

By the definition of o-minimal structure, the set of definable functions is closed under algebraic operations, composition, inversion, and taking maxima or minima. The work by Wilkie (1996) established the existence of an o-minimal structure in which polynomials and the exponential function are definable. By the closure property, commonly used mappings in deep learning are all definable with respect to this o-minimal structure, including fully connected layers, convolutional layers, ReLU activation, max-pooling layers, residual connections, and cross-entropy loss. We refer the readers to Ji & Telgarsky (2020) for more discussion.

Throughout this paper, we assume that there exists an o-minimal structure such that $t \mapsto \exp(t)$ and the network $\boldsymbol{\theta} \mapsto f(\boldsymbol{\theta}; \mathbf{x}_i)$, $i = 1, \ldots, n$, are all definable. This assumption, together with the local Lipschitzness, allows us to apply the chain rule (see Lemma A.6 in Appendix A) to GF defined by subgradients. Moreover, this assumption allows us to leverage the desingularizing function (see Definition 6 in Appendix C.6, and Ji & Telgarsky (2020)) to show the directional convergence. Specifically, we apply two Kurdyka–Łojasiewicz inequalities (See Lemmas C.19 and C.20) under the o-minimal structure to establish the existence of the desingularizing function.

## 3. Implicit Bias of Gradient Flow

In this section, we establish the implicit bias of gradient flow for near-homogeneous networks. Our first assumption is that the network is near-$M$-homogeneous (see Definition 1).

**Assumption 1** (Near-homogeneous network). *Let $M \geq 1$. We assume that the network $f$ is near-$M$-homogeneous with polynomials $\mathsf{p}(\cdot)$ and $\mathsf{q}(\cdot)$, as described in Definition 1.*

Under Assumption 1, let $\mathsf{p}(x) := \sum_{i=0}^{M} a_i x^i$. The following function is handy for our presentation:

$$\mathsf{p}_a(x) := \sum_{i=1}^{M-1} \frac{(i+1)a_{i+1}}{M-i} x^i + \frac{a_1}{M-1/2}. \quad (4)$$

One intuition behind the choice of $\mathsf{p}_a$ is that, for any $f$ satisfying Assumption 1, $g := f - p_a$ satisfies a one-sided inequality of the homogeneous condition (3). This is demonstrated in Appendix C.2.

**Normalized and Modified Margins.** Under Assumption 1, we define the *normalized margin* (Lyu & Li, 2020) as

$$\gamma(\boldsymbol{\theta}) := \min_{i \in [n]} \frac{y_i f(\boldsymbol{\theta}; \mathbf{x}_i)}{\|\boldsymbol{\theta}\|^M}. \quad (5)$$

The normalized margin is hard to analyze directly, due to the hard minimum in its definition. Instead, we analyze a *modified margin*, which increases monotonically and approximates the normalized margin well. Specifically, the *modified margin* for gradient flow is defined as

$$\gamma^{\mathsf{GF}}(\boldsymbol{\theta}) := \frac{\phi(\mathcal{L}(\boldsymbol{\theta})) - \mathsf{p}_a(\|\boldsymbol{\theta}\|)}{\|\boldsymbol{\theta}\|^M}, \quad (6)$$

where $\phi(x) := \log\left(1/(nx)\right)$ and $\mathsf{p}_a$ is given by (4).

In the definition of the modified margin, the $\phi(\mathcal{L}(\boldsymbol{\theta}))$ term produces a soft minimum of $(y_i f(\boldsymbol{\theta}; \mathbf{x}_i))_{i=1}^n$, which approximates the hard margin $\min_{i \in [n]} y_i f(\boldsymbol{\theta}; \mathbf{x}_i)$. This idea appears in prior analysis for homogeneous networks (Lyu & Li, 2020; Ji & Telgarsky, 2020). The offset term $\mathsf{p}_a(\|\boldsymbol{\theta}\|)$ is

our innovation, which controls the homogeneous error when the network is non-homogeneous. Note that as $\|\boldsymbol{\theta}\|$ grows, the offset term $\mathsf{p}_a(\|\boldsymbol{\theta}\|)$ grows slower than the main term $\phi(\mathcal{L}(\boldsymbol{\theta}))$ according to Definition 1. Therefore our modified margin is a good approximation of the normalized margin.

Our second assumption ensures GF can reach a state in which the network strongly separates the data.

**Assumption 2** (Strong separability condition). *Let $f(\boldsymbol{\theta}; \mathbf{x})$ be a network satisfying Assumption 1. Assume that there exists a time $s > 0$ such that $\boldsymbol{\theta}_s$ given by (GF) satisfies*

$$\mathcal{L}(\boldsymbol{\theta}_s) < e^{-\mathsf{p}_a(\|\boldsymbol{\theta}_s\|)}/n, \quad (7)$$

*where $\mathsf{p}_a$ is defined in (4) and $n$ is the number of samples.*

Our Assumption 2 is a natural extension of the separability condition (that is, $\mathcal{L}(\boldsymbol{\theta}_s) < 1/n$ for some $s$) used in the analysis of homogeneous networks (Lyu & Li, 2020; Ji & Telgarsky, 2020). Specifically, for homogeneous networks, we can set $\mathsf{p} = 0$, in which $\mathsf{p}_a = 0$ by its definition in (4). Then Assumption 2 reduces to the separability condition.

For non-homogeneous networks, Assumption 2 requires GF to attain an empirical risk that is sufficiently small compared to a function of the homogeneous error. Note that this condition can be satisfied by any non-degenerate near-homogeneous network that is able to separate the training data, which will be discussed further in Section 5.

In detail, Assumptions 1 and 2, and the choice of $\mathsf{p}_a$ together force the "leading term" in the network to be exactly $M$-homogeneous. Therefore, the near-homogeneity order in Definition 1 must be understood as the minimum possible one. This is precisely explained in the following lemma, whose proof is deferred to Appendix C.2.

**Lemma 3.1** (Near-homogeneity order). *Let $f$ be such that*

$$f(\boldsymbol{\theta}; \mathbf{x}) = \sum_{i=0}^{\infty} f^{(i)}(\boldsymbol{\theta}; \mathbf{x}),$$

*where $f^{(i)}(\boldsymbol{\theta}; \mathbf{x})$ is $i$-homogeneous with respect to $\boldsymbol{\theta}$. If $f$ satisfies Assumptions 1 and 2, then for every $j \in [n]$, we must have*

$$f^{(i)}(\cdot; \mathbf{x}_j) \begin{cases} = 0, & \text{if } i > M, \\ \neq 0, & \text{if } i = M. \end{cases}$$

*Furthermore, we have $f^{(M)}(\boldsymbol{\theta}_s; \mathbf{x}_j) > 0$ for all $j \in [n]$.*

**Margin Improvement.** We are ready to present our first main theorem on the margin improvement of GF. The proof is deferred to Appendix C.1.

**Theorem 3.2** (Risk convergence and margin improvement). *Suppose that Assumptions 1 and 2 hold. For $(\boldsymbol{\theta}_t)_{t>s}$ given by (GF), we have:*

- *For all $t > s$, the risk and the parameter norm satisfy*

$$\mathcal{L}(\boldsymbol{\theta}_t) < e^{-\mathsf{p}_a(\|\boldsymbol{\theta}_t\|)}/n.$$

*Furthermore, we have*

$$\mathcal{L}(\boldsymbol{\theta}_t) \asymp \frac{1}{t(\log t)^{2-2/M}}, \quad \|\boldsymbol{\theta}_t\| \asymp (\log t)^{\frac{1}{M}},$$

*where $\asymp$ hides constants that depend on $M$, $\gamma^{\mathsf{GF}}(\boldsymbol{\theta}_s)$, and coefficients of $\mathsf{q}$.*

- *The modified margin $\gamma^{\mathsf{GF}}(\boldsymbol{\theta}_t)$ is positive, increasing, and upper bounded. Moreover, $\gamma^{\mathsf{GF}}(\boldsymbol{\theta}_t)$ is an $\epsilon_t$-multiplicative approximation of $\gamma(\boldsymbol{\theta}_t)$, that is,*

$$\gamma^{\mathsf{GF}}(\boldsymbol{\theta}_t) \leq \gamma(\boldsymbol{\theta}_t) \leq \left(1 + \epsilon_t\right) \cdot \gamma^{\mathsf{GF}}(\boldsymbol{\theta}_t), \quad \text{for all } t > s,$$

*where*

$$\epsilon_t := \frac{\log n + \mathsf{p}_a(\|\boldsymbol{\theta}_t\|)}{\phi(\mathcal{L}(\boldsymbol{\theta}_t)) - \mathsf{p}_a(\|\boldsymbol{\theta}_t\|)} = \mathcal{O}\big((\log t)^{-1/M}\big) \to 0.$$

This result generalizes Theorem 4.1 in Lyu & Li (2020) from homogeneous networks to near-$M$-homogeneous networks for $M \geq 1$. In particular, we recover their results when the network is $M$-homogeneous, in which we set $\mathsf{p}_a(x) = 0$. When the network is non-homogeneous, our Assumption 2 is stronger than the separability condition in Lyu & Li (2020). This is one of the key conditions that enables our analysis for non-homogeneous networks. In fact, Assumption 2 is necessary for generic near-homogeneous model to exhibit implicit bias. We illustrate this point via the following example, whose proof is deferred to Appendix C.5.

**Example 3.3** (Necessity of Assumption 2). *Assume we only have one sample: $(x, y) = (1, 1) \in \mathbb{R}^2$ and our model is $f(\boldsymbol{\theta}) = \boldsymbol{\theta}^M + M|\boldsymbol{\theta}|^{M-1}$ for $\boldsymbol{\theta} \in \mathbb{R}$ and some odd integer $M \geq 3$. Then, $f$ satisfies Assumption 1 with $\mathsf{p}(\boldsymbol{\theta}) = |\boldsymbol{\theta}|^M$. If $f$ does not satisfy Assumption 2 at $t = s$, we have*

$$\boldsymbol{\theta}_t \leq 0, \quad y \cdot \boldsymbol{\theta}_t x \leq 0 \quad \text{and} \quad \mathcal{L}(\boldsymbol{\theta}_t) \geq 1,$$

*for all $t \geq s$ and $\boldsymbol{\theta}_t$ following (GF).*

**Directional Convergence.** Our next theorem establishes the directional convergence of GF for non-homogeneous networks. The proof is deferred to Appendix C.6.

**Theorem 3.4** (Directional convergence). *Under the setting of Theorem 3.2, let $\tilde{\boldsymbol{\theta}}_t := \boldsymbol{\theta}_t/\|\boldsymbol{\theta}_t\|$ be the direction of (GF). Then the curve swept by $\tilde{\boldsymbol{\theta}}_t$ has finite length. Therefore, the directional limit $\boldsymbol{\theta}_* := \lim_{t \to \infty} \tilde{\boldsymbol{\theta}}_t$ exists.*

Our Theorem 3.4 extends Theorem 3.1 in Ji & Telgarsky (2020) from $M$-homogeneous networks to near-$M$-homogeneous networks for $M \geq 1$. Our Assumptions 1 and 2 allow the application of tools (specifically the desingularizing functions) from Ji & Telgarsky (2020) for showing the direction convergence in the non-homogeneous cases.

**KKT Convergence.** Provided with the directional convergence of GF, our next step is to characterize the limiting direction. To this end, we need to understand the asymptotic behavior of the near-homogeneous network $f$ as $\|\boldsymbol{\theta}\| \to \infty$. Since $f$ is near-$M$-homogeneous, one can expect that $f$ is close to an $M$-homogeneous function for large $\boldsymbol{\theta}$. Motivated by this, we define the *homogenization* of $f$ as

$$f_{\mathsf{H}}(\boldsymbol{\theta}; \mathbf{x}) := \lim_{r \to +\infty} \frac{f(r\boldsymbol{\theta}; \mathbf{x})}{r^M}. \tag{8}$$

The well-definedness, continuity, and differentiability of $f_{\mathsf{H}}$ are guaranteed by the near-homogeneity of $f$ (see Proposition 5.1 in Section 5). We will show that the limiting direction of GF satisfies the KKT conditions of the following margin maximization problem:

$$\text{minimize } \|\boldsymbol{\theta}\|^2, \quad \text{s.t. } \min_{i \in [n]} y_i f_{\mathsf{H}}(\boldsymbol{\theta}; \mathbf{x}_i) \geq 1. \tag{P}$$

This generalizes the margin maximization problem in (Lyu & Li, 2020) from homogeneous to non-homogeneous cases. It is worth noting that when $f$ is already homogeneous, (P) reduces to the same optimization problem as that in (Lyu & Li, 2020).

To ensure the limiting direction satisfies the KKT conditions of (P), we need to compare the gradients of $f$ and $f_{\mathsf{H}}$. This comparison requires an additional regularity assumption.

**Assumption 3** (Weak-homogeneous gradient). *Assume that the network $f(\boldsymbol{\theta}; \mathbf{x})$ is continuously differentiable with respect to $\boldsymbol{\theta}$ for $\mathbf{x} \in (\mathbf{x}_i)_{i=1}^n$ and that $\lim_{r \to \infty} \nabla f(r\boldsymbol{\theta}; \mathbf{x})/r^{M-1}$ exists for all $\boldsymbol{\theta}$ and $\mathbf{x} \in (\mathbf{x}_i)_{i=1}^n$. Assume that the limit*

$$(\nabla f)_{\mathsf{H}}(\boldsymbol{\theta}; \mathbf{x}) := \lim_{r \to \infty} \frac{\nabla f(r\boldsymbol{\theta}; \mathbf{x})}{r^{M-1}}$$

*satisfies*

$$|\nabla f(\boldsymbol{\theta}; \mathbf{x}) - (\nabla f)_{\mathsf{H}}(\boldsymbol{\theta}; \mathbf{x})| \leq \mathsf{r}\left(\|\boldsymbol{\theta}\|\right),$$

*where $\mathsf{r} : \mathbb{R}_{\geq 0} \to \mathbb{R}_{\geq 0}$ is a function such that*

$$\lim_{x \to +\infty} \frac{\mathsf{r}(x)}{x^{M-1}} = 0.$$

It is worth noting that for $M \geq 2$, Assumptions 1 and 3 are satisfied as long as $\nabla_{\boldsymbol{\theta}} f(\boldsymbol{\theta}; \mathbf{x})$ is component-wise near-$(M-1)$-homogeneous. See Lemma D.3 for a precise statement. In comparison, our Assumption 3 is less restrictive and allows for $M = 1$. The next theorem shows that the limiting direction of GF satisfies the KKT conditions of (P), with its proof included in Appendix C.8.

**Theorem 3.5** (KKT convergence). *Under the same setting of Theorem 3.4, and additionally assume Assumption 3 hold. Then the rescaled limiting direction*

$$\boldsymbol{\theta}_* / \left( \min_{i \in [n]} y_i f(\boldsymbol{\theta}_*; \mathbf{x}_i) \right)^{1/M}$$

satisfies the KKT conditions of (P), where $\boldsymbol{\theta}_*$ is the directional limit in Theorem 3.4.

This theorem is a generalization of Theorem 4.4 in Lyu & Li (2020) from homogeneous networks to non-homogeneous networks. Note that our margin maximization problem (P) is defined for the homogenization of the non-homogeneous network. This is because, asymptotically, every near-homogeneous network $f$ can be approximated by its homogenization $f_{\mathsf{H}}$ (see Proposition 5.1 in Section 5). As an important implication, understanding the implicit bias of a near homogeneous network can be reduced to understanding that of its homogenization. It is worth noting that Ji & Telgarsky (2020) also established the asymptotic alignment between parameters and gradients, which we leave as future work.

We have presented our results on the implicit bias for non-homogeneous networks. In the next two sections, we discuss the satisfiability of Assumptions 1 and 2, respectively.

## 4. Near-Homogeneity Condition

In this section, we verify that a large class of building blocks used in deep learning are near-homogeneous, and, by a composition rule, networks constructed using these blocks are also near-homogeneous.

We denote a block by $s(\boldsymbol{\theta}; \mathbf{x})$, where $\boldsymbol{\theta}$ are the trainable parameters in this block and $\mathbf{x}$ is the input (and the output of the preceding block). We use $s_{\boldsymbol{\theta}}(\mathbf{x})$ as a shorthand for $s(\boldsymbol{\theta}; \mathbf{x})$. Then, a network is defined as

$$f(\boldsymbol{\theta}; \mathbf{x}) := s^1_{\boldsymbol{\theta}_1} \circ s^2_{\boldsymbol{\theta}_2} \circ \cdots \circ s^L_{\boldsymbol{\theta}_L}(\mathbf{x}), \qquad (9)$$

where $\boldsymbol{\theta} = (\boldsymbol{\theta}_i)_{i=1}^L$ and $s^i$ is the $i$-th block. Here and in sequel, we assume all the blocks are locally Lipschitz and definable with respect to some o-minimal structure.

To deal with the compositional structure of networks, we need to introduce the following generalized definition of near-homogeneity, which takes both trainable parameters and input into account.

**Definition 2** (Near-$(M, N)$-homogeneity). Let $M, N$ be two non-negative integers such that $M + N \geq 1$. A function $s(\boldsymbol{\theta}; \mathbf{x})$ is called *near-$(M, N)$-homogeneous*, if there exist $\mathsf{p}_s, \mathsf{q}_s, \mathsf{r}_s, \mathsf{t}_s \in \mathbb{R}_{\geq 0}[x]$ with $\deg \mathsf{p}_s, \deg \mathsf{q}_s \leq M$ and $\deg \mathsf{r}_s, \deg \mathsf{t}_s \leq N$ such that the following holds for every $\mathbf{x} \in (\mathbf{x}_i)_{i=1}^n$, $\boldsymbol{\theta}$, and $(\mathbf{h}_{\boldsymbol{\theta}}, \mathbf{h}_{\mathbf{x}}) \in \partial s(\boldsymbol{\theta}; \mathbf{x})$:

(B1). $|\langle \mathbf{h}_{\boldsymbol{\theta}}, \boldsymbol{\theta} \rangle - M s(\boldsymbol{\theta}; \mathbf{x})| \leq \mathsf{p}'_s(\|\boldsymbol{\theta}\|)\mathsf{r}_s(\|\mathbf{x}\|)$,
$\qquad |\langle \mathbf{h}_{\mathbf{x}}, \mathbf{x} \rangle - N s(\boldsymbol{\theta}; \mathbf{x})| \leq \mathsf{p}_s(\|\boldsymbol{\theta}\|)\mathsf{r}'_s(\|\mathbf{x}\|)$;

(B2). $\|\mathbf{h}_{\boldsymbol{\theta}}\| \leq \mathsf{q}'_s(\|\boldsymbol{\theta}\|)\mathsf{t}_s(\|\mathbf{x}\|), \|\mathbf{h}_{\mathbf{x}}\| \leq \mathsf{q}_s(\|\boldsymbol{\theta}\|)\mathsf{t}'_s(\|\mathbf{x}\|)$;

(B3). $\|s(\boldsymbol{\theta}; \mathbf{x})\| \leq \mathsf{q}_s(\|\boldsymbol{\theta}\|)\mathsf{t}_s(\|\mathbf{x}\|)$.

A block $s(\boldsymbol{\theta}; \mathbf{x}) : \mathbb{R}^{d_1} \times \mathbb{R}^{d_2} \to \mathbb{R}^{d_3}$ is called near-$(M, N)$-homogeneous if all of its components, $s(\boldsymbol{\theta}; \mathbf{x})_i$ for $i \in [d_3]$, are near-$(M, N)$-homogeneous with the same polynomials $\mathsf{p}_s, \mathsf{q}_s, \mathsf{r}_s, \mathsf{t}_s$.

From Definition 2, it is easily seen that a near-$(0, N)$-homogeneous block $s(\boldsymbol{\theta}; \mathbf{x})$ must be independent of $\boldsymbol{\theta}$ and near-$N$-homogeneous in $\mathbf{x}$, and vice versa. Below are a few examples of near-homogeneous blocks used in practice.

**Example 4.1.** *The following blocks are near-homogeneous:*

A. *For $\boldsymbol{\theta} = (A, b)$, the linear mapping $s(\boldsymbol{\theta}; \mathbf{x}) = A\mathbf{x} + b$ is near-$(1, 1)$-homogeneous.*

B. *Let $M \geq 1$ be an integer. Then for $\boldsymbol{\theta} = (A, b)$, the perceptron layer $s(\boldsymbol{\theta}; \mathbf{x}) = \phi^M(A\mathbf{x} + b)$ is near-$(M, M)$-homogeneous, where the activation function $\phi$ is one of the following: ReLU, Softplus, GELU, Swish, SiLU, and Leaky ReLU.*

C. *Max pooling layer, average pooling layer, convolution layer, and residual connection are near-$(0, 1)$-homogeneous.*

D. *The SwiGLU activation (Shazeer, 2020) is near-$(2, 2)$-homogeneous.*

E. *The linear self-attention (Zhang et al., 2024) is near-$(2, 3)$-homogeneous and the ReLU attention (Wortsman et al., 2023) is near-$(4, 3)$-homogeneous.*

The following lemma suggests that near-homogeneity is preserved under functional composition and tensor product. Note the residual connection mentioned in Example 4.1 is not a specific block mapping, but a way to enhance an existing block by adding the input from the previous block to the output of this block. Part C of Lemma 4.2 can thus be applied to compute the near-homogeneous order of networks with residual connections. The proof of Lemma 4.2 is deferred to Appendix D.2.

**Lemma 4.2** (Composition and multiplication rules). *Let the blocks $s^1(\boldsymbol{\theta}_1; \mathbf{x}) : \mathbb{R}^{d_1} \times \mathbb{R}^{d_2} \to \mathbb{R}^{d_3}$ and $s^2(\boldsymbol{\theta}_2; \mathbf{x}) : \mathbb{R}^{d_4} \times \mathbb{R}^{d_5} \to \mathbb{R}^{d_6}$ be near-$(M_1, M_2)$-homogeneous and near-$(M_3, M_4)$-homogeneous, respectively. Then, the followings hold:*

A. *Let $d_2 = d_6$, then $s^1_{\boldsymbol{\theta}_1} \circ s^2_{\boldsymbol{\theta}_2}(\mathbf{x})$ is near-$(M_1 + M_2 M_3, M_2 M_4)$-homogeneous.*

B. *The tensor product $s^1(\boldsymbol{\theta}_1; \mathbf{x}) \otimes s^2(\boldsymbol{\theta}_2; \mathbf{x})$ is near-$(M_1 + M_3, M_2 + M_4)$-homogeneous.*

C. *Let $(d_1, d_2, d_3) = (d_4, d_5, d_6)$, then $s^1(\boldsymbol{\theta}_1; \mathbf{x}) + s^2(\boldsymbol{\theta}_2; \mathbf{x})$ is near-$(\max(M_1, M_3), \max(M_2, M_4))$-homogeneous.*

D. *Let* $T : \mathbb{R}^{d_3} \to \mathbb{R}^{d_6}$ *be a linear mapping, then* $T(s^1(\boldsymbol{\theta}_1; \mathbf{x}))$ *is near-*$(M_1, M_2)$*-homogeneous.*

E. *Let* $f : \mathbb{R}^{d_4} \times \mathbb{R}^{d_5} \to \mathbb{R}^{d_2}$ *be near-*$M_5$*-homogeneous as a function of* $\boldsymbol{\theta}_3$*, then* $s^1_{\boldsymbol{\theta}_1} \circ f(\boldsymbol{\theta}_2; \mathbf{x})$ *is near-*$(M_1 + M_2 M_5)$*-homogeneous.*

The following corollary establishes the near-homogeneity of a network composed of near-homogeneous blocks. We defer its proof to Appendix D.3.

**Corollary 4.3** (Near-homogeneous networks)**.** *Consider a network defined as* (9)*. If the block* $s^i(\boldsymbol{\theta}_i; \mathbf{x})$ *is near-*$(M_1^i, M_2^i)$*-homogeneous for* $i \in [L]$*, then the network* $f$ *is near-*$(M_1, M_2)$*-homogeneous with*

$$M_1 := \sum_{j=1}^{L} M_1^j \cdot \prod_{i=1}^{j-1} M_2^i, \quad M_2 := \prod_{j=1}^{L} M_2^j. \quad (10)$$

*In particular,* $f$ *is near-*$M_1$*-homogeneous as a function of* $\boldsymbol{\theta}$ *for any fixed* $\mathbf{x}$*.*

Based on Example 4.1, Lemma 4.2 and Corollary 4.3, we can show that a broad class of commonly used networks are near-homogeneous, including the following examples:

**Example 4.4.** *The following networks are near-homogeneous:*

A. *An L-layer MLP with* $k$*-th power of ReLU activation is near-*$(k^L - 1)/(k - 1)$*-homogeneous, or near-*$L$*-homogeneous when* $k = 1$*. The same holds when ReLU is replaced by other activation functions in Example 4.1 (B).*

B. *VGG-L (*Simonyan & Zisserman, 2015*) is near-*$L$*-homogeneous where* $L \in \{11, 13, 16, 19\}$*.*

C. *Without batch normalization, ResNet-L (*He et al., 2016*) is near-*$L$*-homogeneous where* $L \in \{18, 34, 50, 101, 152\}$*, and DenseNet-L (*Huang et al., 2017*) is near-*$L$*-homogeneous where* $L \in \{121, 169, 201, 264\}$*.*

To conclude this section, we comment that normalization layers and softmax map violate our near-homogeneity definitions. Intuitively, these blocks should be "near-0-homogeneous" as their outputs are bounded. However, our near-homogeneity definitions are only non-trivial for $M \geq 1$. As a consequence, the softmax attention architecture also violates our definitions of near-homogeneity, thus requiring a different treatment. We believe that our notion of near-homogeneity can be generalized to include those components, which we leave as future work.

## 5. Strong Separability Condition

In this section, we discuss the satisfiability of Assumption 2. We first give an intuitive explanation of why Assumption 2 should be expected to hold. Note that Assumption 2 is equivalent to

$$\mathcal{L}(\boldsymbol{\theta}_s) < e^{-\mathsf{p}_a(\|\boldsymbol{\theta}_s\|)}/n \iff \frac{\log\left(1/(n\mathcal{L}(\boldsymbol{\theta}_s))\right)}{\mathsf{p}_a(\|\boldsymbol{\theta}_s\|)} > 1.$$

Recall that $\deg \mathsf{p}_a \leq M - 1$ by Assumption 1. Thus Assumption 2 can be understood as requiring GF to induce a "lower-order" smoothed margin which is at least 1. If the normalized margin (5) is asymptotically positive (as $\|\boldsymbol{\theta}_t\|$ grows), then the "lower-order" smoothed margin must diverge, hence Assumption 2 must hold at some point.

In what follows, we establish a sufficient condition under which a non-homogeneous network satisfies Assumption 2. To this end, we establish below several important properties of the *homogenization* of a non-homogeneous network.

**Homogenization.** The idea of homogenization appears in Section 3 for constructing the margin maximization problem, where its KKT conditions characterize the limiting direction of GF. Our next theorem rigorously controls the approximation error between a non-homogeneous network and its homogenization.

**Proposition 5.1** (Homogenization)**.** *Suppose that* $f$ *satisfies Assumption 1. Then for every* $\mathbf{x} \in (\mathbf{x}_i)_{i=1}^{n}$*, the homogenization of* $f(\boldsymbol{\theta}; \mathbf{x})$*:*

$$f_{\mathsf{H}}(\boldsymbol{\theta}; \mathbf{x}) := \lim_{r \to +\infty} \frac{f(r\boldsymbol{\theta}; \mathbf{x})}{r^M}$$

*exists and is well-defined. Moreover, as a function of* $\boldsymbol{\theta}$*,* $f_{\mathsf{H}}(\boldsymbol{\theta}; \mathbf{x})$ *is continuous, differentiable almost everywhere, and* $M$*-homogeneous. We also have*

$$\text{for every } \boldsymbol{\theta}, \quad |f(\boldsymbol{\theta}; \mathbf{x}) - f_{\mathsf{H}}(\boldsymbol{\theta}; \mathbf{x})| \leq \mathsf{p}_a(\|\boldsymbol{\theta}\|),$$

*where* $\mathsf{p}_a$ *is given by* (4)*.*

*If, in addition,* $f(\boldsymbol{\theta}; \mathbf{x})$ *satisfies Assumption 3, then* $f_{\mathsf{H}}(\boldsymbol{\theta}; \mathbf{x})$ *is continuously differentiable for every nonzero* $\boldsymbol{\theta}$*, and that* $(\nabla f)_{\mathsf{H}}(\boldsymbol{\theta}; \mathbf{x}) = \nabla f_{\mathsf{H}}(\boldsymbol{\theta}; \mathbf{x})$*.*

Proposition 5.1 ensures the well-definedness, continuity, and differentiability of $f_{\mathsf{H}}$. Consequently, the following theorem guarantees that the strong separability condition can be satisfied by a near-homogeneous network, as long as its homogenization satisfies the (weak) separability condition of (Lyu & Li, 2020).

**Proposition 5.2** (A sufficient condition)**.** *Suppose that* $f$ *satisfies Assumption 1. Then* $f$ *admits a homogenization, denoted by* $f_{\mathsf{H}}$*. Assume that* $f_{\mathsf{H}}$ *satisfies the weak separability*

*condition, that is,*

$$\sum_i \ell\big(-y_i f_{\mathsf{H}}(\boldsymbol{\theta}'; \mathbf{x}_i)\big) < 1 \text{ for some } \boldsymbol{\theta}'.$$

*Then, there exists a constant $c > 0$ such that $f$ with $\boldsymbol{\theta}_s := c\boldsymbol{\theta}'$ satisfies Assumption 2.*

To further elaborate the idea of homogenization, we provide the following compositional rule.

**Proposition 5.3** (Homogenization of networks). *Consider a network given by* (9), *where each block $s^i(\boldsymbol{\theta}_i; \mathbf{x})$ is near-$(M_1^i, M_2^i)$-homogeneous (see Definition 2) for $i = 1, \ldots, L$. Then each block $s^i(\boldsymbol{\theta}_i; \mathbf{x})$ admits a well-defined homogenization*

$$s_{\mathsf{H}}^i(\boldsymbol{\theta}_i; \mathbf{x}) := \lim_{r_1, r_2 \to \infty} \frac{s^i(r_1 \boldsymbol{\theta}_i; r_2 \mathbf{x})}{r_1^{M_1^i} r_2^{M_2^i}}.$$

*Moreover, the homogenization network* (9) *is well-defined and satisfies*

$$f_{\mathsf{H}}(\boldsymbol{\theta}; \mathbf{x}) = s_{\mathsf{H}, \boldsymbol{\theta}_1}^1 \circ s_{\mathsf{H}, \boldsymbol{\theta}_2}^2 \circ \cdots \circ s_{\mathsf{H}, \boldsymbol{\theta}_L}^L(\mathbf{x}).$$

**A Two-Layer Network.** We next provide a two-layer network example where GF with zero initialization can provably reach a point that satisfies Assumption 2. The two-layer network is defined as

$$f(\boldsymbol{\theta}; \mathbf{x}) := \mathbf{w}_1^\top \mathbf{x} + \mathbf{w}_2^\top \mathbf{x} + a_1 \varphi(\mathbf{w}_1^\top \mathbf{x}) - a_2 \varphi(-\mathbf{w}_2^\top \mathbf{x}),$$
(11)

where $\boldsymbol{\theta} := (\mathbf{w}_1, \mathbf{w}_2, a_1, a_2)$ are the trainable parameters, and $\varphi$ is the leaky ReLU activation function,

$$\varphi(x) := \max\{x, \alpha_L x\}, \quad 0 < \alpha_L < 1.$$

Our example (11) is motived by the two-layer network considered in Lyu et al. (2021). However, their network is homogeneous, while ours is non-homogeneous (but near-2-homogeneous) due to the residual connections. Similar to Lyu et al. (2021), we consider a symmetric and linearly separable dataset.

**Assumption 4** (Dataset conditions). *Assume that the dataset $(\mathbf{x}_i, y_i)_{i=1}^n$ satisfies $\max_i \|\mathbf{x}_i\| \leq 1$ and $\min_i y_i \mathbf{x}_i^\top \mathbf{w}_* \geq \gamma$ for some unit vector $\mathbf{w}_*$ and margin $\gamma > 0$; moreover, $n$ is even and $(y_{i+n/2}, \mathbf{x}_{i+n/2}) = -(y_i, \mathbf{x}_i)$ for $i = 1, \ldots, n/2$.*

Note that the network (11) is not Lipschitz continuous, thus (GF) is not well-defined. Instead, we study the small stepsize limit of gradient descent for (11), which is well-defined under our assumptions. In the next theorem, we establish that the small-stepsize limit of gradient descent with zero initialization for (11) can satisfy Assumption 2.

**Theorem 5.4** (A two-layer network example). *Consider the network $f$ given by* (11) *and a dataset satisfying Assumption 4. Let $(\boldsymbol{\theta}_t)_{t \geq 0}$ be the limit of GD iterates with initialization $\boldsymbol{\theta}_0 = \mathbf{0}$ as the stepsize tends to zero. Then there exists $s > 0$ such that $f$ with $\boldsymbol{\theta}_s$ satisfies Assumption 2.*

## 6. Implicit Bias of Gradient Descent

In this section, we extend our previous results for GF to gradient descent (GD), which is defined as

$$\boldsymbol{\theta}_{t+1} := \boldsymbol{\theta}_t - \eta \nabla \mathcal{L}(\boldsymbol{\theta}_t), \quad \text{(GD)}$$

where $\eta > 0$ is a fixed stepsize. Similarly, we will assume the network $f$ is near-$M$-homogeneous with $\mathsf{p}$ and $\mathsf{q}$ (see Assumption 1). For GD, the definition of $\mathsf{p}_a$ needs to be slightly modified. Let $\mathsf{p}(x) := \sum_{i=0}^M a_i x^i$. For $M \geq 2$, we define

$$\mathsf{p}_a(x) := \begin{cases} \displaystyle\sum_{i=1}^{M-1} \frac{(i+1)a_{i+1}}{M-i} x^i + \frac{a_1}{M-1/2}, & \text{if } x \geq 1, \\[3mm] \displaystyle\sum_{i=2}^{M-1} \frac{(i+1)a_{i+1}}{M-i} x^i \\[3mm] \displaystyle + \frac{2a_2}{M-1} \frac{x^2+1}{2} + \frac{a_1}{M-1/2}, & \text{if } 0 \leq x < 1. \end{cases}$$
(12)

For $M = 1$, we define $\mathsf{p}_a(x) := a_1/(M-1/2)$. Compared to (4), (12) replaces the linear term $x$ with a quadratic polynomial $(x^2+1)/2$. This is purely for circumventing a technical issue when $\|\boldsymbol{\theta}\|$ is small (see the proof of Lemma F.7 in Appendix F.1).

To handle the discretization error introduced by GD, we need a stronger version of the strong separability condition.

**Assumption 5** (Strong separability condition for GD). *Let $f(\boldsymbol{\theta}; \mathbf{x})$ be a network satisfying Assumption 1. Assume that $f$ is twicely differentiable for $\boldsymbol{\theta}$ and that for some constant $A > 0$, we have*

$$\|\nabla_{\boldsymbol{\theta}}^2 f(\boldsymbol{\theta}; \mathbf{x})\| \leq \begin{cases} A\left(\|\boldsymbol{\theta}\|^{M-2} + 1\right), & \text{if } M \geq 2 \\ A, & \text{if } M = 1 \end{cases}$$

*for every $\mathbf{x} \in (\mathbf{x}_i)_{i=1}^n$. For $\mathsf{p}_a$ given by* (12), *assume that $\deg \mathsf{p}_a \geq 1$ if $M \geq 2$, and that there exists a time $s > 0$ such that*

$$\mathcal{L}(\boldsymbol{\theta}_s) < \min\left\{\frac{1}{ne^2}, \frac{1}{B\eta}\right\} e^{-\mathsf{p}_a(\|\boldsymbol{\theta}_s\|)},$$

*where $B$ is a constant depending only on $(M, \mathsf{p}, \mathsf{q})$ and $A$.*

Note that in Assumption 5, the stepsize $\eta$ can be arbitrarily large as long as the empirical risk $\mathcal{L}(\boldsymbol{\theta}_s)$ is of the order of $\mathcal{O}(1/\eta)$. We note that Proposition 5.2 can also be adapted to guarantee the satisfiability of Assumption 5.

We consider the following modified margin for GD:

$$\gamma^{\mathsf{GD}}(\boldsymbol{\theta}) := \frac{\exp\left(\Phi\left(e^{\mathsf{p}_a(\|\boldsymbol{\theta}\|)} \mathcal{L}(\boldsymbol{\theta})\right)\right)}{\|\boldsymbol{\theta}\|^M},$$
(13)

where $\Phi(x) := \log(\log \frac{1}{nx}) + \frac{2}{\log(nx)}$.

In the following two theorems, we extend our results for GF to GD. The proofs are included in Appendix F.2.

**Theorem 6.1** (Risk convergence and margin improvement for GD). *Suppose that Assumptions 1 and 5 hold. For $(\boldsymbol{\theta}_t)_{t \geq 0}$ given by (GD) with any stepsize $\eta > 0$, we have:*

- *For all $t > s$, the risk and parameter norm satisfy*

$$\mathcal{L}(\boldsymbol{\theta}_t) < \min\left\{\frac{1}{ne^2}, \frac{1}{B\eta}\right\} \cdot e^{-\mathsf{p}_a(\|\boldsymbol{\theta}_t\|)}.$$

*Furthermore, as $t \to \infty$, we have*

$$\mathcal{L}(\boldsymbol{\theta}_t) \asymp \frac{1}{\eta t (\log \eta t)^{2 - 2/M}}, \quad \|\boldsymbol{\theta}_t\| \asymp (\log \eta t)^{\frac{1}{M}},$$

*where $\asymp$ hides constants that depend on $M$, $\gamma^{\mathsf{GD}}(\boldsymbol{\theta}_s)$, and coefficients of $\mathsf{q}$, but not $\eta$.*

- *The modified margin $\gamma^{\mathsf{GD}}(\boldsymbol{\theta}_t)$ in (13) is increasing and bounded. Moreover, $\gamma^{\mathsf{GD}}(\boldsymbol{\theta}_t)$ is an $\epsilon_t$-multiplicative approximation of $\gamma(\boldsymbol{\theta}_t)$, that is, for all $t > s$,*

$$\gamma^{\mathsf{GD}}(\boldsymbol{\theta}_t) \leq \gamma(\boldsymbol{\theta}_t) \leq (1 + \epsilon_t)\gamma^{\mathsf{GD}}(\boldsymbol{\theta}_t),$$

*where $\epsilon_t \to 0$ as $t \to \infty$.*

**Theorem 6.2** (Directional and KKT convergence for GD). *Under Assumptions 1 and 5, the same results in Theorem 3.4 hold for (GD) with any stepsize $\eta > 0$. Under Assumptions 3 and 5, the same results in Theorem 3.5 hold for (GD) with any stepsize $\eta > 0$.*

## 7. Conclusion

We show the implicit bias of gradient descent (GD) for training generic non-homogeneous deep networks under exponential loss. We show that the normalized margin induced by GD increases nearly monotonically. Moreover, GD converges in direction, with the limiting direction satisfying the KKT conditions of a margin maximization problem. Our results rely on a near-homogeneity condition and a strong separability condition, both of which are natural generalizations of the conditions used in prior implicit bias analysis for homogeneous networks. In particular, our results apply to networks with residual connections and non-homogeneous activation functions.

## Acknowledgements

We thank Fabian Pedregosa, Nathan Srebro, and Matus Telgarsky for their helpful comments and discussion. We gratefully acknowledge the support of the NSF for FODSI through grant DMS-2023505, of the Founder's Postdoctoral Fellowship in Statistics at Columbia University, of the Sloan Fellowship by the U.S. Department of Energy, Office of Science, Office of Advanced Scientific Computing Research's Applied Mathematics Competitive Portfolios program under Contract No. AC02-05CH11231, of the NSF and the Simons Foundation for the Collaboration on the Theoretical Foundations of Deep Learning through awards DMS-2031883 and #814639, DMS-2210827, CCF-2315725, and of the ONR through MURI award N000142112431 and N00014-24-S-B001.

## Impact Statement

This paper presents work whose goal is to advance the field of Machine Learning. There are many potential societal consequences of our work, none of which we feel must be specifically highlighted here.

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

# A. O-minimal Structure

In this section, we give a brief introduction to the o-minimal structure. We mainly follow the notation and definitions in Ji & Telgarsky (2020).

**Definition 3** (O-minimal structure). An O-minimal structure is a collection $\mathcal{S} = (\mathcal{S}_n)_{n=1}^\infty$ where $\mathcal{S}_n$ is a subset of $\mathbb{R}^n$ that satisfies the following properties:

1. $\mathcal{S}_1$ is the collection of all finite unions of open intervals and points.

2. $\mathcal{S}_n$ includes the zero sets of all polynomials (algebraic sets) on $\mathbb{R}^n$: if $\mathsf{p} \in \mathbb{R}[x_1, \ldots, x_n]$, then $\{x \in \mathbb{R}^n | \mathsf{p}(x) = 0\} \in \mathcal{S}_n$.

3. $\mathcal{S}_n$ is closed under finite union, finite intersection, and complement.

4. $\mathcal{S}$ is closed under Cartesian product: if $A \in \mathcal{S}_m$ and $B \in \mathcal{S}_n$, then $A \times B \in \mathcal{S}_{m+n}$.

5. $\mathcal{S}$ is closed under projection $\prod_n$ onto the first $n$ coordinates: if $A \in \mathcal{S}_{n+1}$, then $\prod_n(A) \in \mathcal{S}_n$.

Then we can define the definable functions.

**Definition 4** (Definable sets and functions). Given an o-minimal structure $\mathcal{S}$, a set $A \subset \mathbb{R}^n$ is *definable* if $A \in \mathcal{S}_n$. A function $f : D \to \mathbb{R}^m$ with $D \subset \mathbb{R}^n$ is *definable* if its graph is in $\mathcal{S}_{n+m}$.

The following properties of definable sets and functions can then be derived (check Coste (2000); Loi (2010); van den Dries & Miller (1996)).

**Proposition A.1** (Property of definable functions). *We fix an arbitrary o-minimal structure $\mathcal{S}$. Then the following properties hold:*

1. *Given any $\alpha, \beta \in \mathbb{R}$ and any definable functions $f, g : D \to \mathbb{R}$, we have $\alpha f + \beta g$ and $fg$ are definable. If $g \neq 0$ on $D$, then $f/g$ is definable. If $f \geq 0$ on $D$, then $f^{\frac{1}{\ell}}$ is definable for any positive integer $\ell$.*

2. *Given a function $f : D \to \mathbb{R}^m$, let $f_i$ denote the $i$-th coordinate of its output. Then $f$ is definable if and only if all $f_i$ are definable.*

3. *Any composition of definable functions is definable.*

4. *Any coordinate permutation of a definable set is definable. Consequently, if the inverse of a definable function exists, it's also definable.*

4. *The image and preimage of a defianble set by a definable function is definable. Particulalry, given any real-valued definable function $f$, all of $f^{-1}(0), f^{-1}(-\infty, 0)$ and $f^{-1}((0, \infty))$ are definable.*

5. *Any combination of finitely many definable functions with disjoint domains is definable. For example, the pointwise maximum and minimum of definable functions are definable.*

Once we have these properties, we can observe that almost all the network structures are definable.

**Lemma A.2** (Definable network structures, Lemma B.2 in Ji & Telgarsky (2020)). *Suppose there exist $k, d_0, d_1, \ldots, d_k > 0$ and $L$ definable functions $(g_1, \ldots, g_L)$ where $g_j : \mathbb{R}^{d_0} \times \cdots \times \mathbb{R}^{d_j-1} \times \mathbb{R}^k \to \mathbb{R}^{d_j}$. Let $h_1(x; \mathbf{w}) := g_1(x, \mathbf{w})$ and for $2 \leq j \leq L$,*

$$h_j(x, \mathbf{w}) := g_j(x, h_1(x, \mathbf{w}), \ldots, h_{j-1}(x, \mathbf{w}), \mathbf{w}),$$

*then all $h_j$ are definable. It suffices if each output coordinate of $g_j$ is the minimum or maximum over some finite set of polynomials, which allows for linear, convolutional, ReLU, max-pooling layers, and skip connections.*

One important corollary is that this allows us to composite and multiply the near-homogeous blocks.

**Corollary A.3** (Composition and multiplication of definable blocks). *Given two definable block mappings: $s_1(\boldsymbol{\theta}_1; \mathbf{x}) : \mathbb{R}^{d_1} \times \mathbb{R}^{d_2} \to \mathbb{R}^{d_3}$ and $s_2(\boldsymbol{\theta}_2; \mathbf{x}) : \mathbb{R}^{d_3} \times \mathbb{R}^{d_4} \to \mathbb{R}^{d_5}$, then,*

- *If $d_2 = d_5$, the composition $s(\boldsymbol{\theta}; \mathbf{x}) = s_1(\boldsymbol{\theta}_1; s_2(\boldsymbol{\theta}_2; \mathbf{x}))$ is definable.*

- *$s(\boldsymbol{\theta}; \mathbf{x}) = s_1(\boldsymbol{\theta}_1; \mathbf{x}) \otimes s_2(\boldsymbol{\theta}_2; \mathbf{x})$ is definable.*

- *If $d_1 = d_3$, $s(\boldsymbol{\theta}_1; \mathbf{x}) = s_1(\boldsymbol{\theta}_1; \mathbf{x}) \otimes s_2(\boldsymbol{\theta}_1; \mathbf{x})$ is definable.*

*Proof.* This is a direct consequence of Lemma A.2 and Proposition A.1. □

Definable functions have good properties, especially there is some stratification which makes them piecewise differentiable. However, the gradient, even the Clarke subdifferential, of a definable function, is not well-defined everywhere. To overcome this issue, we need the local Lipschitz continuity.

**Definition 5** (Local Lipschitz continuity). Given a function $f : D \to \mathbb{R}$ with an open domain $D$, we say $f$ is locally Lipschitz continuous if for any $x \in D$, there exists a neighborhood $U$ of $x$ such that $f$ is Lipschitz continuous on $U$.

Similar to the definability, the local Lipschitz continuity is preserved under composition and multiplication. It's worth noting that the Clarke subdifferential of a locally Lipschitz function is well-defined everywhere.

**Lemma A.4** (Clarke subdifferential of locally Lipschitz functions, Corollary 6.1.2 in Borwein et al. (2000)). *Given a locally Lipschitz function $f : D \to \mathbb{R}$ with an open domain $D$, the Clarke subdifferential $\partial f(x)$ is nonempty, compact, and convex for all $x \in D$.*

Now we will assume all the models and block mappings are definable and locally Lipschitz. Since block mappings are multivalue functions, we can use the Clarke general Jacobian for them. Similarly, for locally Lipschitz functions to $\mathbb{R}^n$, the general Jacobian is well-defined for all points.

**Corollary A.5** (Clarke Jacobian of locally Lipschitz functions, Proposition 2.6.2 in Clarke (1990)). *Given a locally Lipschitz function $f : D \to \mathbb{R}^n$ with an open domain $D \subseteq \mathbb{R}^m$, the Clarke Jacobian $\partial f(x)$ is nonempty, compact, and convex for all $x \in D$.*

Following the notation in Ji & Telgarsky (2020), we let $\bar{\partial} f(x)$ denote the unique minimum-norm subgradient:

$$\bar{\partial} f(x) := \arg \min_{x^* \in \partial f(x)} \|x^*\|. \tag{14}$$

Another important property is that for definable functions, it admits a chain rule almost everywhere.

**Lemma A.6** (Chain rule, Lemma B.9 in Ji & Telgarsky (2020)). *Given a locally Lipschitz definable $f : D \to \mathbb{R}$ with an open domain $D$, for any interval $I$ and any arc $z : I \to D$, it holds that for almost every $t \in I$ that*

$$\frac{\mathrm{d} f(z_t)}{\mathrm{d} t} = \left\langle z_t^*, \frac{\mathrm{d} z_t}{\mathrm{d} t} \right\rangle, \quad \text{for all } z_t^* \in \partial f(z_t).$$

*Moreover, for (GF), we have for almost every $t \geq 0$,*

$$\frac{\mathrm{d} \boldsymbol{\theta}_t}{\mathrm{d} t} = -\bar{\partial} \mathcal{L}_t, \quad \frac{\mathrm{d} \mathcal{L}_t}{\mathrm{d} t} = -\|\bar{\partial} \mathcal{L}_t\|^2$$

$$\frac{\mathrm{d} \rho_t^2}{\mathrm{d} t} = -2\langle \boldsymbol{\theta}_t, \bar{\partial} \mathcal{L}_t \rangle, \quad \frac{\mathrm{d} \gamma^{\mathsf{GF}}(\boldsymbol{\theta}_t)}{\mathrm{d} t} = -2\langle \bar{\partial} \gamma^{\mathsf{GF}}(\boldsymbol{\theta}_t), \bar{\partial} \mathcal{L}_t \rangle.$$

Clarke has the following lemma to characterize the subgradient of the composition of functions.

**Theorem A.7** (Clark chain rule, Theorems 2.3.9 and 2.3.0 in Clarke (1990)). *Let $z_1, \ldots, z_n : \mathbb{R}^d \to \mathbb{R}$ and $f : \mathbb{R}^n \to \mathbb{R}$ be locally Lipschitz functions. Let $(f \circ z)(x) = f(z_1(x), \ldots, z_n(x))$ be the composition of $f$ and $z$. Then,*

$$\partial (f \circ z)(x) \subseteq conv \left\{ \sum_{i=1}^n \alpha_i \mathbf{h}_i : \alpha \in \partial f(z_1(x), \ldots, z_n(x)), \mathbf{h}_i \in \partial z_i(x) \right\}.$$

Recall that:

$$\mathcal{L}(\boldsymbol{\theta}) = \frac{1}{n} \sum_{i=1}^n e^{-y_i f(\boldsymbol{\theta}; \mathbf{x}_i)}.$$

We have the following corollary which characterizes the min-norm subgradient of the exponential loss.

**Corollary A.8** (Subgradient of the exponential loss). *For the exponential loss $\mathcal{L}(\boldsymbol{\theta})$, there exists $\mathbf{h}_i \in \partial f(\boldsymbol{\theta}; \mathbf{x}_i)$ such that*

$$\bar{\partial}\mathcal{L}(\boldsymbol{\theta}) = \frac{1}{n}\sum_{i=1}^{n} -e^{-y_i f(\boldsymbol{\theta};\mathbf{x}_i)} y_i \mathbf{h}_i.$$

*Proof of Corollary A.8.* This is a direct consequence of Theorem A.7 and the definition of the exponential loss. □

Similarly, we can characterize the subgradient of the composition of block functions.

**Corollary A.9** (Subgradient of the composition of block functions). *Given two locally Lipschitz block functions $s_1(\boldsymbol{\theta}_1; \mathbf{x}) : \mathbb{R}^{d_1} \times \mathbb{R}^{d_2} \to \mathbb{R}^{d_3}$ and $s_2(\boldsymbol{\theta}_2; \mathbf{x}) : \mathbb{R}^{d_3} \times \mathbb{R}^{d_4} \to \mathbb{R}^{d_2}$, we have*

$$\partial_{\boldsymbol{\theta}} s_1(\boldsymbol{\theta}_1, s_2(\boldsymbol{\theta}_2; \mathbf{x})) \subset conv\{(\boldsymbol{\alpha}_1, \boldsymbol{\alpha}_2 \cdot \mathbf{h}_1) : (\boldsymbol{\alpha}_1, \boldsymbol{\alpha}_2) \in \partial_{\boldsymbol{\theta}_1, \mathbf{x}} s_1(\boldsymbol{\theta}_1; \mathbf{x}), \mathbf{h}_1 \in \partial_{\boldsymbol{\theta}_2} s_2(\boldsymbol{\theta}_2; \mathbf{x})\},$$

$$\partial_{\mathbf{x}} s_1(\boldsymbol{\theta}_1, s_2(\boldsymbol{\theta}_2; \mathbf{x})) \subset conv\{\boldsymbol{\alpha}_2 \cdot \mathbf{h}_2 : \boldsymbol{\alpha}_2 \in \partial_{\mathbf{x}} s_1(\boldsymbol{\theta}_1; \mathbf{x}), \mathbf{h}_2 \in \partial_{\mathbf{x}} s_2(\boldsymbol{\theta}_2; \mathbf{x})\}.$$

*Proof of Corollary A.9.* It follows from Theorem A.7 and the definition of the block functions. □

Here we use $\partial_{\boldsymbol{\theta}_1, \mathbf{x}} s_1(\boldsymbol{\theta}_1; \mathbf{x})$ to denote the corresponding Jacobian. Note that $\boldsymbol{\alpha}_1 \in \mathbb{R}^{d_3 \times d_1}, \boldsymbol{\alpha}_2 \in \mathbb{R}^{d_3 \times d_2}, \mathbf{h}_1 \in \mathbb{R}^{d_2 \times d_3}, \mathbf{h}_2 \in \mathbb{R}^{d_2 \times d_4}$. Once we have this, to verify some properties of Jacobian of $s_1 \circ s_2$, we can focus on the Jacobian of $s_1$ and $s_2$.

The o-minimal structure eliminates many bad geometry, allowing us to focus on the good part of the optimization landscape.

# B. Proof Overview

In the main text, we present several results on the implicit bias of GF/GD for near-homogeneous networks. Below we briefly describe the approaches we use to prove these results, with actual proofs deferred to the subsequent appendices.

**Margin Improvement and Convergence Rates.** We first sketch the proof of Theorem 3.2 (GF), and then highlight the major technical innovations in the proof of Theorem 6.1 (GD), as compared to the case of GF. The key ingredient for proving Theorem 3.2 is the following lemma, which establishes the monotonicity of the modified margin under the strong separability condition.

**Lemma B.1** (Restatement of Theorem C.5). *Denote $\mathcal{L}_t = \mathcal{L}(\boldsymbol{\theta}_t)$ and $\rho_t = \|\boldsymbol{\theta}_t\|$. Under Assumptions 1 and 2, we have $\mathcal{L}_t < e^{-\mathsf{p}_a(\rho_t)}/n$ for all $t \geq s$, and*

$$\frac{\mathrm{d}\log\gamma^{\mathsf{GF}}(\boldsymbol{\theta}_t)}{\mathrm{d}t} > \frac{\|\bar{\partial}\mathcal{L}_t\|^2 \rho_t^2 - \langle\bar{\partial}\mathcal{L}_t, \boldsymbol{\theta}_t\rangle^2}{\rho_t^2 \mathcal{L}_t(\phi(\mathcal{L}_t) - \mathsf{p}_a(\rho_t))} \geq 0. \tag{15}$$

The proof of Lemma B.1 is similar to the proof of Lyu & Li (2020, Lemma 5.1). Since $\gamma^{\mathsf{GF}}(\boldsymbol{\theta}_t)$ only depends on $\mathcal{L}_t$ and $\rho_t$, its growth rate can be attributed to two quantities: $\mathrm{d}\mathcal{L}_t/\mathrm{d}t$ and $\mathrm{d}\rho_t/\mathrm{d}t$. We use the same argument as that in Lyu & Li (2020, Proof sketch of Lemma 5.1) to estimate each of these two quantities, which finally leads to the lower bound (15) on $\mathrm{d}\gamma^{\mathsf{GF}}(\boldsymbol{\theta}_t)/\mathrm{d}t$. Further since

$$\phi(\mathcal{L}_t) - \mathsf{p}_a(\rho_t) > 0 \iff \mathcal{L}_t < \frac{1}{n}e^{-\mathsf{p}_a(\rho_t)},$$

we know that the strong separability condition is necessary for the lower bound (15) at $t = s$, and can be established for $t \geq s$ using continuous induction. It is noteworthy that while we follow a similar proof scheme as (Lyu & Li, 2020), the design of the modified margin $\gamma^{\mathsf{GF}}$ is completely novel and highly non-trivial.

From this lower bound, the "margin improvement" part of Theorem 3.2 directly follows. For the "convergence rates" part, we use the monotonicity of modified margin to upper and lower bound $-\mathrm{d}\mathcal{L}_t/\mathrm{d}t$, thus establishing convergence rates for $\mathcal{L}_t$. The claim for $\rho_t$ can be proved similarly.

For the proof of Theorem 6.1, we need to establish a lower bound on $\log\gamma^{\mathsf{GD}}(\boldsymbol{\theta}_{t+1}) - \log\gamma^{\mathsf{GD}}(\boldsymbol{\theta}_t)$ similar to (15). Due to the discrete nature of GD, the Hessian of $\log\gamma^{\mathsf{GD}}$ should also be taken into account when dealing with $\log\gamma^{\mathsf{GD}}(\boldsymbol{\theta}_t)$ as a function of $t$. To address this challenge, we analyze a modified loss $\mathcal{G}(\boldsymbol{\theta}) = \exp(\mathsf{p}_a(\|\boldsymbol{\theta}\|))\mathcal{L}(\boldsymbol{\theta})$ that is closely related to $\log\gamma^{\mathsf{GD}}$.

We establish tight upper bounds on the Hessian of $\mathcal{G}(\boldsymbol{\theta}_t)$ in terms of $\mathcal{L}_t$ and $\rho_t$, thus leading to a tight lower bound on $\log \gamma^{\mathsf{GD}}(\boldsymbol{\theta}_{t+1}) - \log \gamma^{\mathsf{GD}}(\boldsymbol{\theta}_t)$. Notably, our lower bound allows for arbitrarily large step size $\eta$, as long as $\mathcal{G}(\boldsymbol{\theta}_s) = O(1/\eta)$, which is guaranteed by Assumption 5. In contrast, Lyu & Li (2020) assumes that the step size is upper bounded by a function of the initial margin, preventing it to be large. The proof of other parts in Theorem 6.1 is completely analogous to the case of GF.

**Convergence to the KKT Direction.** The proofs of Theorems 3.4 and 6.2 largely rely on the techniques developed in Ji & Telgarsky (2020). As before, we will first sketch the proof for GF, and then explain how to adapt it to establish KKT convergence for GD. For GF, define the arc length swept by the direction of $\boldsymbol{\theta}_t$:

$$\zeta_t = \int_s^t \left\| \frac{\mathrm{d}\tilde{\boldsymbol{\theta}}_u}{\mathrm{d}u} \right\| \mathrm{d}u.$$

Similar to Ji & Telgarsky (2020), we construct a desingularizing function $\Psi$ that controls the growth rate of $\zeta_t$ using that of $\Psi(\gamma_* - \gamma^{\mathsf{GF}}(\boldsymbol{\theta}_t))$, where $\gamma_* := \lim_{t \to \infty} \gamma^{\mathsf{GF}}(\boldsymbol{\theta}_t)$:

**Lemma B.2** (Restatement of Lemma C.14). *There exist $R > 0, \nu > 0$ and a definable desingularizing function $\Psi$ on $[0, \nu)$, such that for a.e. large enough $t$ with $\|\boldsymbol{\theta}_t\| > R$ and $\gamma^{\mathsf{GF}}(\boldsymbol{\theta}_t) > \gamma_* - \nu$, it holds that*

$$\frac{\mathrm{d}\zeta_t}{\mathrm{d}t} \leq -c \frac{\mathrm{d}\Psi\left(\gamma_* - \gamma^{\mathsf{GF}}(\boldsymbol{\theta}_t)\right)}{\mathrm{d}t} \tag{16}$$

*for some constant $c > 0$.*

The proof of the above lemma is similar to that of Ji & Telgarsky (2020, Lemma 3.1). Integrating both sides of (16), we deduce that $\lim_{t \to \infty} \zeta_t < \infty$. Therefore, $\tilde{\boldsymbol{\theta}}_t$ must converge to some $\boldsymbol{\theta}_*$. This establishes directional convergence of GF path.

To go further and show that $\boldsymbol{\theta}_*$ satisfies the KKT conditions (P), the key ingredient is to show the asymptotic alignment between $\boldsymbol{\theta}_t$ and $\mathbf{h}_M(\boldsymbol{\theta}_t)$ for subsequence of $t$, i.e.,

$$\lim_{t_m \to \infty} \beta(t_m) = \lim_{t_m \to \infty} \frac{\langle \boldsymbol{\theta}_{t_m}, \mathbf{h}_M(\boldsymbol{\theta}_{t_m}) \rangle}{\|\boldsymbol{\theta}_{t_m}\| \cdot \|\mathbf{h}_M(\boldsymbol{\theta}_{t_m})\|} = 1,$$

where we define

$$\mathbf{h}_M(\boldsymbol{\theta}_t) := \frac{1}{n} \sum_{i=1}^n e^{-y_i f(\boldsymbol{\theta}_t; \mathbf{x}_i)} y_i \nabla f_{\mathsf{H}}(\boldsymbol{\theta}_t; \mathbf{x}_i)$$

as a proxy of $\nabla f(\boldsymbol{\theta}_t; \mathbf{x})$. We establish the following bound by looking further into the first inequality in (15).

**Lemma B.3** (Restatement of Corollary C.26). *For any $t_2 > t_1$ large enough, there exists $t_* \in (t_1, t_2)$ such that*

$$\frac{1 - p_1(t_*)}{\left(\beta(t_*) + p_2(t_*)\right)^2} - 1 \leq \frac{1}{M} \cdot \frac{\log \gamma^{\mathsf{GF}}(\boldsymbol{\theta}_{t_2}) - \log \gamma^{\mathsf{GF}}(\boldsymbol{\theta}_{t_1})}{\log \|\boldsymbol{\theta}_{t_2}\| - \log \|\boldsymbol{\theta}_{t_1}\|},$$

*where $p_1(t), p_2(t) \to 0$ as $t \to \infty$.*

As $\gamma^{\mathsf{GF}}(\boldsymbol{\theta}_t)$ converges and $\|\boldsymbol{\theta}_t\|$ diverges, the right-hand side of the above inequality converges to 0. Hence, there must exist a subsequence $\{\beta(t_m)\}$ converging to 1. This shows that $\boldsymbol{\theta}_t$ and $\mathbf{h}_M(\boldsymbol{\theta}_t)$ asymptotically aligns along this subsequence, and consequently verifies that $\boldsymbol{\theta}_*$ is a KKT point.

To establish directional convergence for GD, we need to show that the discrete arc length swept by $\tilde{\boldsymbol{\theta}}_t$ is finite, i.e.,

$$\sum_{t=s}^{\infty} \left\| \tilde{\boldsymbol{\theta}}_{t+1} - \tilde{\boldsymbol{\theta}}_t \right\| < \infty. \tag{17}$$

Similar to the proof of GF, we construct a desingularizing function $\Psi$, and show that there exists a constant $c > 0$, such that for all large enough $t$,

$$\left\| \tilde{\boldsymbol{\theta}}_{t+1} - \tilde{\boldsymbol{\theta}}_t \right\| \leq c \left( \Psi\left(\gamma_* - \gamma^{\mathsf{GD}}(\boldsymbol{\theta}_t)\right) - \Psi\left(\gamma_* - \gamma^{\mathsf{GD}}(\boldsymbol{\theta}_{t+1})\right) \right), \tag{18}$$

where $\gamma^{\mathrm{GD}}$ is the modified margin for GD and $\gamma_* = \lim_{t\to\infty} \gamma^{\mathrm{GD}}(\boldsymbol{\theta}_t)$. To this end, we construct an arc by connecting all GD iterates using line segments, and carefully estimate the spherical and the radial parts of this arc, addressing several new technical challenges that arise in the analysis of GD.

Finally, summing up both sides of (18) immediately leads to (17), thus proving directional convergence of $\boldsymbol{\theta}_t$ along the GD path. The proof of KKT convergence is completely similar to the case of GF.

## C. Proofs for Section 3

In this section, we will provide the proofs of the results for (GF) in Section 3. Recall that we have the following notation:

$$\phi(x) := \log \frac{1}{nx}, \quad \nabla := \nabla_{\boldsymbol{\theta}}, \quad \rho_t := \|\boldsymbol{\theta}_t\|, \quad \mathcal{L}_t := \mathcal{L}(\boldsymbol{\theta}_t),$$
$$f_i(\boldsymbol{\theta}) := f(\boldsymbol{\theta}; \mathbf{x}_i), \quad \bar{f}_i(\boldsymbol{\theta}) := y_i f(\boldsymbol{\theta}, \mathbf{x}_i), \quad \bar{f}_{\min}(\boldsymbol{\theta}) := \min_{i \in [n]} \bar{f}_i(\boldsymbol{\theta}).$$

For a function of the time, $h(t)$, we use $h'(t)$ as a shorthand of $\frac{\mathrm{d}}{\mathrm{d}t} h(t)$.

### C.1. Margin Improvement

In this section, we always assume $f(\boldsymbol{\theta}; \mathbf{x})$ satisfies Assumption 1 with $\mathsf{p}_a$ defined in (4). We can directly verify the following property of $\mathsf{p}$ and $\mathsf{p}_a$.

**Lemma C.1** (Property of $\mathsf{p}_a$)**.** *For* $\mathsf{p}$ *and* $\mathsf{p}_a$ *defined in* (4)*, we have*

$$\text{for every } x, \quad \mathsf{p}'_a(x)x + \mathsf{p}'(x) \le M\mathsf{p}_a(x).$$

*Proof of Lemma C.1.* Recall that $\mathsf{p}(x) = \sum_{i=0}^{M} a_i x^i$ and

$$\mathsf{p}_a(x) := \sum_{i=1}^{M-1} \frac{(i+1)a_{i+1}}{M-i} x^i + \frac{a_1}{M-1/2}.$$

Hence,

$$\mathsf{p}'_a(x) + \mathsf{p}'(x) - M\mathsf{p}_a(x) = \sum_{i=1}^{M-1} \left[ \frac{i(i+1)a_{i+1}}{M-i} x^i + (i+1)a_{i+1}x^i - \frac{Ma_{i+1}}{M-i} x^i \right] + a_1 - \frac{Ma_1}{M-1/2}$$

$$= -\frac{a_1/2}{M-1/2} \le 0.$$

This completes the proof of Lemma C.1. $\qquad\square$

The following bounds from Lyu & Li (2020) connect $\bar{f}_{\min}(\boldsymbol{\theta}_t)$ and $\mathcal{L}_t$.

**Lemma C.2** (property of $\bar{f}_{\min}(\boldsymbol{\theta})$)**.** *For the Loss function* $\mathcal{L}$ *defined in* (1)*, we have*

$$\text{for every } \boldsymbol{\theta}, \quad \log \frac{1}{n\mathcal{L}(\boldsymbol{\theta})} \le \bar{f}_{\min}(\boldsymbol{\theta}) \le \log \frac{1}{\mathcal{L}(\boldsymbol{\theta})}.$$

*Proof of Lemma C.2.* Recall that we consider the exponential loss

$$\mathcal{L}(\boldsymbol{\theta}) = \frac{1}{n} \sum_{i=1}^{n} e^{-\bar{f}_i(\boldsymbol{\theta})}.$$

Therefore, we can get that

$$\frac{1}{n}\mathcal{L}(\boldsymbol{\theta}) \le \frac{1}{n} e^{-\bar{f}_{\min}(\boldsymbol{\theta})} \le \mathcal{L}(\boldsymbol{\theta}) \implies \log \frac{1}{n\mathcal{L}(\boldsymbol{\theta})} \le \bar{f}_{\min}(\boldsymbol{\theta}) \le \log \frac{1}{\mathcal{L}(\boldsymbol{\theta})}.$$

This completes the proof of Lemma C.2. $\qquad\square$

We have the following fundamental lemma from Lyu & Li (2020) about the dynamics of the GF.

**Lemma C.3** (Decrease of the risk). *For* (GF)*, we have for a.e. $t \geq s$,*

$$-\frac{d\mathcal{L}_t}{dt} = \left(\frac{1}{2}\frac{d\rho_t^2}{dt}\right) \cdot \frac{d\log\rho_t}{dt} + \frac{\|\bar{\partial}\mathcal{L}_t\|^2\rho_t^2 - \langle\bar{\partial}\mathcal{L}_t, \boldsymbol{\theta}_t\rangle^2}{\rho_t^2}, \quad t \geq 0.$$

*Proof of Lemma C.3.* By lemma A.6, we have for a.e. $t \geq s$,

$$
\begin{aligned}
-\frac{d\mathcal{L}_t}{dt} &= \|\bar{\partial}\mathcal{L}_t\|^2 && \text{by lemma A.6}\\
&= \frac{\langle\bar{\partial}\mathcal{L}_t, \boldsymbol{\theta}_t\rangle^2}{\rho_t^2} + \frac{\|\bar{\partial}\mathcal{L}_t\|^2\rho_t^2 - \langle\bar{\partial}\mathcal{L}_t, \boldsymbol{\theta}_t\rangle^2}{\rho_t^2} && \text{Projecting } \boldsymbol{\theta} \text{ to spherical and radical parts}\\
&= \frac{1}{\rho_t^2} \cdot \left(\frac{1}{2}\frac{d\rho_t^2}{dt}\right)^2 + \frac{\|\bar{\partial}\mathcal{L}_t\|^2\rho_t^2 - \langle\bar{\partial}\mathcal{L}_t, \boldsymbol{\theta}_t\rangle^2}{\rho_t^2} && \text{Since } \frac{1}{2}\frac{d\rho_t^2}{dt} = -\langle\bar{\partial}\mathcal{L}_t, \boldsymbol{\theta}_t\rangle\\
&= \left(\frac{1}{2}\frac{d\rho_t^2}{dt}\right) \cdot \frac{d\log\rho_t}{dt} + \frac{\|\bar{\partial}\mathcal{L}_t\|^2\rho_t^2 - \langle\bar{\partial}\mathcal{L}_t, \boldsymbol{\theta}_t\rangle^2}{\rho_t^2}. && \text{Since } \frac{d\log\rho_t}{dt} = \frac{1}{\rho_t}\rho_t' = \frac{1}{2\rho_t^2}\frac{d\rho_t^2}{dt}
\end{aligned}
$$

We have completed the proof of Lemma C.3. $\qquad\square$

The next lemma shows the parameter norm is increasing under the strong separability condition.

**Lemma C.4** (Increase of the parameter norm). *Under Assumption 1, for almost every $t \geq s$, we have*

$$\mathcal{L}_t < e^{-\mathsf{p}_a(\rho_t)}/n \quad \Rightarrow \quad \frac{1}{2}\frac{d\rho_t^2}{dt} \geq \left(M\phi(\mathcal{L}_t) - \mathsf{p}'(\rho_t)\right)\mathcal{L}_t > 0.$$

*Proof of Lemma C.4.* By Lemma C.2, we have

$$\bar{f}_i(\boldsymbol{\theta}_t) \geq \bar{f}_{\min}(\boldsymbol{\theta}_t) \geq \log\frac{1}{n\mathcal{L}_t} = \phi(\mathcal{L}_t).$$

Then we have for almost every $t \geq s$,

$$
\begin{aligned}
\frac{1}{2}\frac{d\rho_t^2}{dt} &= \langle-\bar{\partial}\mathcal{L}_t, \boldsymbol{\theta}_t\rangle && \text{by Lemma A.6}\\
&= \frac{1}{n}\sum_{i=1}^n e^{-\bar{f}_i(\boldsymbol{\theta}_t)}y_i\langle\mathbf{h}_i, \boldsymbol{\theta}_t\rangle && \text{By Corollary A.8}\\
&\geq M\cdot\frac{1}{n}\sum_{i=1}^n e^{-\bar{f}_i(\boldsymbol{\theta}_t)}\bar{f}_i(\boldsymbol{\theta}_t) - \mathcal{L}_t\cdot\mathsf{p}'(\rho_t) && \text{By Assumption 1}\\
&\geq M\cdot\frac{1}{n}\sum_{i=1}^n e^{-\bar{f}_i(\boldsymbol{\theta}_t)}\phi(\mathcal{L}_t) - \mathcal{L}_t\cdot\mathsf{p}'(\rho_t) && \text{Since } \bar{f}_i(\boldsymbol{\theta}_t) \geq \phi(\mathcal{L}_t)\\
&= M\mathcal{L}_t\phi(\mathcal{L}_t) - \mathsf{p}'(\rho_t)\mathcal{L}_t.
\end{aligned}
$$

Note that

$$\mathcal{L}_t < e^{-\mathsf{p}_a(\rho_t)}/n \quad \Leftrightarrow \quad M\phi(\mathcal{L}_t) > M\mathsf{p}_a(\rho_t).$$

Additionally, Lemma C.1 implies $M\mathsf{p}_a(x) \geq \mathsf{p}_a'(x)x + \mathsf{p}'(x) \geq \mathsf{p}'(x)$. As a consequence of these two inequalities, we have

$$\frac{1}{2}\frac{d\rho_t^2}{dt} \geq M\phi(\mathcal{L}_t)\mathcal{L}_t - \mathsf{p}'(\rho_t)\mathcal{L}_t > M\mathsf{p}_a(\rho_t)\mathcal{L}_t - \mathsf{p}'(\rho_t)\mathcal{L}_t \geq 0.$$

We have completed the proof of Lemma C.4. $\qquad\square$

**Theorem C.5** (Monotonicity of the modified margin). *Under Assumptions 1 and 2, we have $\mathcal{L}_t < e^{-\mathsf{p}_a(\rho_t)}/n$ for $t \geq s$. Moreover, we have*

$$\frac{\mathrm{d}\log\gamma^{\mathsf{GF}}(\boldsymbol{\theta}_t)}{\mathrm{d}t} > \frac{\|\bar{\partial}\mathcal{L}_t\|^2\rho_t^2 - \langle\bar{\partial}\mathcal{L}_t, \boldsymbol{\theta}_t\rangle^2}{\rho_t^2\mathcal{L}_t(\phi(\mathcal{L}_t) - \mathsf{p}_a(\rho_t))} \geq 0, \quad \text{for almost every } t \geq s. \tag{19}$$

*As a consequence, the modified margin $\gamma^{\mathsf{GF}}(\boldsymbol{\theta}_t)$ is increasing for $t \geq s$.*

*Proof of Theorem C.5.* We first show that $\mathcal{L}_t < e^{-\mathsf{p}_a(\rho_t)}/n$ implies $\gamma^{\mathsf{GF}}(\boldsymbol{\theta}_t)$ is positive and increasing. Then we show $\mathcal{L}_t < e^{-\mathsf{p}_a(\rho_t)}/n$ for all $t \geq s$ by contradiction.

**Step 1: the condition that $\mathcal{L}_t < e^{-\mathsf{p}_a(\rho_t)}/n$ implies that $\gamma^{\mathsf{GF}}(\boldsymbol{\theta}_t)$ is positive and increasing.** Recall that

$$\mathcal{L}_t < e^{-\mathsf{p}_a(\rho_t)}/n \quad \Leftrightarrow \quad M\phi(\mathcal{L}_t) > M\mathsf{p}_a(\rho_t).$$

Under this condition, the modified margin $\gamma^{\mathsf{GF}}(\boldsymbol{\theta}_t)$ defined in (6) is positive. Moreover, we have for almost every $t \geq s$,

$$\frac{\mathrm{d}\log\gamma^{\mathsf{GF}}(\boldsymbol{\theta}_t)}{\mathrm{d}t} = \frac{\mathrm{d}\log(\phi(\mathcal{L}_t) - \mathsf{p}_a(\rho_t))}{\mathrm{d}t} - M\frac{\mathrm{d}\log\rho_t}{\mathrm{d}t}$$
$$= \frac{-\frac{\mathrm{d}\mathcal{L}_t}{\mathrm{d}t} - \mathcal{L}_t\mathsf{p}_a'(\rho_t)\rho_t}{\mathcal{L}_t(\phi(\mathcal{L}_t) - \mathsf{p}_a(\rho_t))} - M\frac{\mathrm{d}\log\rho_t}{\mathrm{d}t}.$$

By Lemma C.4 we have $\rho_t$ is increasing, which implies $\mathrm{d}\log\rho_t/\mathrm{d}t > 0$ almost everywhere. Then we have for almost every $t \geq s$,

$$-\frac{\mathrm{d}\mathcal{L}_t}{\mathrm{d}t} = \left(\frac{1}{2}\frac{\mathrm{d}\rho_t^2}{\mathrm{d}t}\right) \cdot \frac{\mathrm{d}\log\rho_t}{\mathrm{d}t} + \frac{\|\bar{\partial}\mathcal{L}_t\|^2\rho_t^2 - \langle\bar{\partial}\mathcal{L}_t, \boldsymbol{\theta}_t\rangle^2}{\rho_t^2} \qquad \text{By Lemma C.3}$$
$$> \mathcal{L}_t\left(\phi(\mathcal{L}_t) - \frac{\mathsf{p}'(\rho_t)}{M}\right) \cdot M\frac{\mathrm{d}\log\rho_t}{\mathrm{d}t} + \frac{\|\bar{\partial}\mathcal{L}_t\|^2\rho_t^2 - \langle\bar{\partial}\mathcal{L}_t, \boldsymbol{\theta}_t\rangle^2}{\rho_t^2}. \qquad \text{By Lemma C.4}$$

Furthermore,

$$\mathcal{L}_t\mathsf{p}_a'(\rho_t)\rho_t' = \mathcal{L}_t\frac{\mathsf{p}_a'(\rho_t)\rho_t}{M} \cdot M\frac{\mathrm{d}\log\rho_t}{\mathrm{d}t}. \qquad \text{By } \frac{\mathrm{d}\log\rho_t}{\mathrm{d}t} = \frac{\rho_t'}{\rho_t}$$

Applying the above two bounds in the derivative of $\log\gamma^{\mathsf{GF}}(\boldsymbol{\theta}_t)$, we get for almost every $t \geq s$,

$$\frac{\mathrm{d}\log\gamma^{\mathsf{GF}}(\boldsymbol{\theta}_t)}{\mathrm{d}t} = \frac{-\frac{\mathrm{d}\mathcal{L}_t}{\mathrm{d}t} - \mathcal{L}_t\mathsf{p}_a'(\rho_t)\rho_t'}{\mathcal{L}_t(\phi(\mathcal{L}_t) - \mathsf{p}_a(\rho_t))} - M\frac{\mathrm{d}\log\rho_t}{\mathrm{d}t}$$
$$> \frac{\mathcal{L}_t\left(\phi(\mathcal{L}_t) - \frac{\mathsf{p}_a'(\rho_t)\rho_t}{M} - \frac{\mathsf{p}'(\rho_t)}{M}\right)}{\mathcal{L}_t(\phi(\mathcal{L}_t) - \mathsf{p}_a(\rho_t))} \cdot M\frac{\mathrm{d}\log\rho_t}{\mathrm{d}t} + \frac{\|\bar{\partial}\mathcal{L}_t\|^2\rho_t^2 - \langle\bar{\partial}\mathcal{L}_t, \boldsymbol{\theta}_t\rangle^2}{\rho_t^2\mathcal{L}_t(\phi(\mathcal{L}_t) - \mathsf{p}_a(\rho_t))} - M\frac{\mathrm{d}\log\rho_t}{\mathrm{d}t}$$
$$\geq \frac{\mathcal{L}_t(\phi(\mathcal{L}_t) - \mathsf{p}_a(\rho_t))}{\mathcal{L}_t(\phi(\mathcal{L}_t) - \mathsf{p}_a(\rho_t))} \cdot M\frac{\mathrm{d}\log\rho_t}{\mathrm{d}t} - M\frac{\mathrm{d}\log\rho_t}{\mathrm{d}t} + \frac{\|\bar{\partial}\mathcal{L}_t\|^2\rho_t^2 - \langle\bar{\partial}\mathcal{L}_t, \boldsymbol{\theta}_t\rangle^2}{\rho_t^2\mathcal{L}_t(\phi(\mathcal{L}_t) - \mathsf{p}_a(\rho_t))} \qquad \text{By Lemma C.1}$$
$$= \frac{\|\bar{\partial}\mathcal{L}_t\|^2\rho_t^2 - \langle\bar{\partial}\mathcal{L}_t, \boldsymbol{\theta}_t\rangle^2}{\rho_t^2\mathcal{L}_t(\phi(\mathcal{L}_t) - \mathsf{p}_a(\rho_t))} \geq 0.$$

That is, $\gamma^{\mathsf{GF}}(\boldsymbol{\theta}_t)$ is increasing.

**Step 2: proving $\mathcal{L}_t < e^{-\mathsf{p}_a(\rho_t)}/n$ for all $t \geq s$ by contradiction.** For $t = s$, $\mathcal{L}_s$ satisfies the condition by Assumption 2. Let $s_1$ be the first time where $L_{s_1}$ violates the condition, that is,

$$s_1 := \sup\{s'|\mathcal{L}_t < e^{-\mathsf{p}_a(\rho_t)}/n, \text{ for all } s \leq t < s'\}.$$

We need to show $s_1 = +\infty$. If not, we have $s_1 < \infty$. Note that

$$\frac{e^{-\mathsf{p}_a(\rho_t)}}{n\mathcal{L}_t} = \exp\left(\log(1/n\mathcal{L}_t) - \mathsf{p}_a(\rho_t)\right) = \exp(\gamma^{\mathsf{GF}}(\boldsymbol{\theta}_t) \cdot \rho_t^M).$$

Since $\mathcal{L}_t < e^{-\mathsf{p}_a(\rho_t)}/n$ for $s \leq t < s_1$, by **Step 1** we have $\gamma^{\mathsf{GF}}(\boldsymbol{\theta}_t)$ and $\rho_t$ are both positive and increasing. Hence $\frac{e^{-\mathsf{p}_a(\rho_t)}}{n\mathcal{L}_t}$ is increasing for $s \leq t < s_1$. So we have

$$\frac{e^{-\mathsf{p}_a(\rho_t)}}{n\mathcal{L}_t} \geq \frac{e^{-\mathsf{p}_a(\rho_s)}}{n\mathcal{L}_s}, \quad s \leq t < s_1.$$

Since $\boldsymbol{\theta}_t$ forms an arc and the left-hand side is a continuous function of $\boldsymbol{\theta}_t$, it is continuous as a function of $t$. As a consequence, we have

$$\frac{e^{-\mathsf{p}_a(\rho_{s_1})}}{n\mathcal{L}_{s_1}} \geq \frac{e^{-\mathsf{p}_a(\rho_s)}}{n\mathcal{L}_s} > 1 \quad \Longrightarrow \quad \mathcal{L}_{s_1} < e^{-\mathsf{p}_a(\rho_{s_1})}/n.$$

Since $\mathcal{L}_t$ and $\mathsf{p}_a(\rho_t)$ are both continuous functions with respect to $t$ ($\boldsymbol{\theta}_t$ is an arc), this leads to that there exists $s_2 > s_1$ and for all $s \leq t \leq s_2$, $\mathcal{L}_t < e^{-\mathsf{p}_a(\rho_t)}/n$. This is a contradiction. So we have $\mathcal{L}_t < e^{-\mathsf{p}_a(\rho_t)}/n$ for all $t \geq s$. This completes the proof of Theorem C.5. $\qquad\square$

## C.2. Proof of Lemma 3.1

*Proof of Lemma 3.1.* Recall that we have the decomposition:

$$f(\boldsymbol{\theta}; \mathbf{x}) = \sum_{i=0}^{\infty} f^{(i)}(\boldsymbol{\theta}; \mathbf{x}), \quad \mathbf{x} \in (\mathbf{x}_i)_{i=1}^n,$$

where $f^{(i)}(\boldsymbol{\theta}; \mathbf{x})$ is $i$-homogeneous with respect to $\boldsymbol{\theta}$. Then Assumption 1 implies that $f^{(i)}(\boldsymbol{\theta}; \mathbf{x}) \equiv 0$ for all $i > M$. By Lemma C.1 we have

$$M\mathsf{p}_a(x) - \mathsf{p}'_a(x)x \geq \mathsf{p}'(x).$$

Let $g(\boldsymbol{\theta}; \mathbf{x}) := f(\boldsymbol{\theta}; \mathbf{x}) - \mathsf{p}_a(\|\boldsymbol{\theta}\|)$. Then we have

$$\begin{aligned}
&\langle \boldsymbol{\theta}, \nabla_{\boldsymbol{\theta}} g(\boldsymbol{\theta}; \mathbf{x}) \rangle - Mg(\boldsymbol{\theta}; \mathbf{x}) \\
&= \langle \boldsymbol{\theta}, \nabla_{\boldsymbol{\theta}} f(\boldsymbol{\theta}; \mathbf{x}) \rangle - Mf(\boldsymbol{\theta}; \mathbf{x}) - \langle \boldsymbol{\theta}, \nabla_{\boldsymbol{\theta}} \mathsf{p}_a(\|\boldsymbol{\theta}\|) \rangle + M\mathsf{p}_a(\|\boldsymbol{\theta}\|) \\
&= \langle \boldsymbol{\theta}, \nabla_{\boldsymbol{\theta}} f(\boldsymbol{\theta}; \mathbf{x}) \rangle - Mf(\boldsymbol{\theta}; \mathbf{x}) - \mathsf{p}'_a(\|\boldsymbol{\theta}\|)\|\boldsymbol{\theta}\| + M\mathsf{p}_a(\|\boldsymbol{\theta}\|) \\
&\geq -\mathsf{p}'(\|\boldsymbol{\theta}\|) - \mathsf{p}'_a(\|\boldsymbol{\theta}\|)\|\boldsymbol{\theta}\| + M\mathsf{p}_a(\|\boldsymbol{\theta}\|) \geq 0.
\end{aligned}$$

Let $h(\alpha; \mathbf{x}) := g(\alpha\boldsymbol{\theta}_s; \mathbf{x})/(\alpha\|\boldsymbol{\theta}_s\|)^M$. Then we have

$$\begin{aligned}
\frac{\mathrm{d}h(\alpha; \mathbf{x})}{\mathrm{d}\alpha} &= \frac{1}{(\alpha\|\boldsymbol{\theta}_s\|)^M} \frac{\mathrm{d}g(\alpha\boldsymbol{\theta}_s; \mathbf{x})}{\mathrm{d}\alpha} - M\frac{g(\alpha\boldsymbol{\theta}_s; \mathbf{x})\|\boldsymbol{\theta}_s\|}{(\alpha\|\boldsymbol{\theta}_s\|)^{M+1}} \\
&= \frac{\langle \alpha\boldsymbol{\theta}_s, \nabla g(\alpha\boldsymbol{\theta}_s; \mathbf{x}) \rangle}{\alpha^{M+1}\|\boldsymbol{\theta}_s\|^M} - M\frac{g(\alpha\boldsymbol{\theta}_s; \mathbf{x})}{\alpha^{M+1}\|\boldsymbol{\theta}_s\|^M} \\
&= \frac{\langle \alpha\boldsymbol{\theta}_s, \nabla g(\alpha\boldsymbol{\theta}_s; \mathbf{x}) \rangle - Mg(\alpha\boldsymbol{\theta}_s; \mathbf{x})}{\alpha^{M+1}\|\boldsymbol{\theta}_s\|^M} \\
&\geq 0.
\end{aligned}$$

Since $\mathcal{L}(\boldsymbol{\theta}_s) < e^{-\mathsf{p}_a(\|\boldsymbol{\theta}_s\|)}/n$, we have $h(1; \mathbf{x}) = g(\boldsymbol{\theta}_s; \mathbf{x})/(\alpha\|\boldsymbol{\theta}_s\|)^M > 0$. As $\alpha \to \infty$, $h(\alpha; \mathbf{x}) > 0$. Note that

$$\lim_{\alpha\to\infty} h(\alpha; \mathbf{x}) = \lim_{\alpha\to\infty} \frac{g(\alpha\boldsymbol{\theta}_s; \mathbf{x})}{(\alpha\|\boldsymbol{\theta}_s\|)^M} = \lim_{\alpha\to\infty} \frac{f(\alpha\boldsymbol{\theta}_s; \mathbf{x}) - \mathsf{p}_a(\alpha\|\boldsymbol{\theta}_s\|)}{(\alpha\|\boldsymbol{\theta}_s\|)^M} = \frac{f^{(M)}(\boldsymbol{\theta}_s; \mathbf{x})}{\|\boldsymbol{\theta}_s\|^M} > 0.$$

This leads to $f^{(M)}(\boldsymbol{\theta}; \mathbf{x}) \not\equiv 0$. This completes the proof of Lemma 3.1. $\qquad\square$

## C.3. Convergence Rates

We will show that $\gamma^{\mathsf{GF}}(\boldsymbol{\theta}_t)$ is a good approximation of $\gamma(\boldsymbol{\theta}_t)$. Before we proceed, we need one auxiliary margin.

$$\gamma^{\mathsf{S}}(\boldsymbol{\theta}_t) := \frac{\phi(\mathcal{L}_t)}{\rho_t^M}. \tag{20}$$

**Lemma C.6** (Modified margin is a good approximation). *Under Assumption 1 and $\mathcal{L}_t < e^{-\mathsf{p}_a(\rho_t)}/n$, for the (GF), we have*

$$\gamma^{\mathsf{GF}}(\boldsymbol{\theta}_t) \leq \gamma^{\mathsf{S}}(\boldsymbol{\theta}_t) \leq \gamma(\boldsymbol{\theta}_t) \leq \left(1 + \frac{\log n + \mathsf{p}_a(\rho_t)}{\phi(\mathcal{L}_t) - \mathsf{p}_a(\rho_t)}\right)\gamma^{\mathsf{GF}}(\boldsymbol{\theta}_t).$$

*Proof of Lemma C.6.* Note that we have $\bar{f}_{\min}(\boldsymbol{\theta}_t) \geq \phi(\mathcal{L}_t)$ from the proof of Lemma C.4. Then, we have

$$\gamma^{\mathsf{GF}}(\boldsymbol{\theta}_t) = \frac{\phi(\mathcal{L}_t) - \mathsf{p}_a(\rho_t)}{\rho_t^M} \leq \frac{\phi(\mathcal{L}_t)}{\rho_t^M} = \gamma^{\mathsf{S}}(\boldsymbol{\theta}_t) \leq \frac{\bar{f}_{\min}(\boldsymbol{\theta}_t)}{\rho_t^M} = \gamma(\boldsymbol{\theta}_t).$$

For the last inequality, note that $\bar{f}_{\min}(\boldsymbol{\theta}_t) \leq \log \frac{1}{\mathcal{L}_t}$ from Lemma C.2, then using the definition of $\phi(\cdot)$ we have

$$\frac{\gamma(\boldsymbol{\theta}_t)}{\gamma^{\mathsf{GF}}(\boldsymbol{\theta}_t)} = \frac{\bar{f}_{\min}(\boldsymbol{\theta}_t)}{\phi(\mathcal{L}_t) - \mathsf{p}_a(\rho_t)} \leq \frac{\log \frac{1}{\mathcal{L}_t}}{\phi(\mathcal{L}_t) - \mathsf{p}_a(\rho_t)} = 1 + \frac{\log n + \mathsf{p}_a(\rho_t)}{\phi(\mathcal{L}_t) - \mathsf{p}_a(\rho_t)}.$$

This completes the proof of Lemma C.6. □

We will later show that

$$\frac{\log n + \mathsf{p}_a(\rho_t)}{\phi(\mathcal{L}_t) - \mathsf{p}_a(\rho_t)} \approx \frac{\log n + \mathsf{p}_a(\rho_t)}{\log \frac{1}{\mathcal{L}_t}} \to 0.$$

This implies that $\gamma^{\mathsf{GF}}(\boldsymbol{\theta}_t)$ is a good multiplicative approximation of $\gamma(\boldsymbol{\theta}_t)$. We need to characterize the behaviors of $\mathcal{L}_t$ and $\rho_t$.

**Lemma C.7** (Upper bound of the risk). *Under Assumptions 1 and 2, for (GF), we have*

$$\mathcal{L}_t = \mathcal{O}\left(\frac{1}{t(\log t)^{2-2/M}}\right),$$

*where the hidden constant depends $\gamma^{\mathsf{GF}}(\boldsymbol{\theta}_s)^{\frac{2}{M}}$.*

*Proof of Lemma C.7.* Recall that in Lemma C.3, we have shown that

$$-\frac{\mathrm{d}\mathcal{L}_t}{\mathrm{d}t} \geq \frac{1}{\rho_t^2} \cdot \left(\frac{1}{2}\frac{\mathrm{d}\rho_t^2}{\mathrm{d}t}\right)^2.$$

By Theorem C.5, $\mathcal{L}_t < e^{-\mathsf{p}_a(\rho_t)}/n$ for all $t \geq s$. Note that when $\mathcal{L}_t < e^{-\mathsf{p}_a(\rho_t)}/n$, we have for $M \geq 2$,

$$(M - 1/2)\phi(\mathcal{L}_t) > (M - 1/2)\mathsf{p}_a(\rho_t)$$

$$= \sum_{i=1}^{M-1} \frac{(M - 1/2)(i + 1)a_{i+1}}{M - i}\rho_t^i + \frac{(M - 1/2)a_1}{M - 1/2}$$

$$\geq \sum_{i=1}^{M-1} (i + 1)a_{i+1}\rho_t^i + a_1 = \mathsf{p}'(\rho_t).$$

This leads to

$$M\phi(\mathcal{L}_t) - \mathsf{p}'(\rho_t) \geq \frac{1}{2}\phi(\mathcal{L}_t) > 0. \tag{21}$$

By Lemma C.4, we have

$$-\frac{\mathrm{d}\mathcal{L}_t}{\mathrm{d}t} \geq \frac{1}{\rho_t^2} \cdot \left(M\mathcal{L}_t\phi(\mathcal{L}_t) - p'(\rho_t)\mathcal{L}_t\right)^2$$

$$\geq \frac{1}{4\rho_t^2} \cdot \left(\mathcal{L}_t\phi(\mathcal{L}_t)\right)^2 \qquad\qquad \text{By the inequality above}$$

$$
= \frac{\gamma^{\mathsf{S}}(\boldsymbol{\theta}_t)^{\frac{2}{M}}}{4\big(\phi(\mathcal{L}_t)\big)^{\frac{2}{M}}} \cdot \big(\mathcal{L}_t\phi(\mathcal{L}_t)\big)^2 \qquad\qquad \text{By the definition of } \gamma^{\mathsf{S}}(\boldsymbol{\theta}_t)
$$

$$
\geq \frac{\gamma^{\mathsf{GF}}(\boldsymbol{\theta}_t)^{\frac{2}{M}}}{4} \cdot \mathcal{L}_t^2\big(\phi(\mathcal{L}_t)\big)^{2-2/M} \qquad\qquad \text{Since } \gamma^{\mathsf{S}}(\cdot) \geq \gamma^{\mathsf{GF}}(\cdot)
$$

$$
\geq \frac{\gamma^{\mathsf{GF}}(\boldsymbol{\theta}_s)^{\frac{2}{M}}}{4} \cdot \mathcal{L}_t^2\big(\phi(\mathcal{L}_t)\big)^{2-2/M} \qquad\qquad \text{By Theorem C.5}
$$

$$
= c\mathcal{L}_t^2\big(\phi(\mathcal{L}_t)\big)^{2-2/M},
$$

where $c = \frac{\gamma^{\mathsf{GF}}(\boldsymbol{\theta}_s)^{\frac{2}{M}}}{4}$. Equivalently, we have

$$
\frac{1}{\big(\phi(\mathcal{L}_t)\big)^{2-2/M}} \cdot \frac{\mathrm{d}}{\mathrm{d}t}\frac{1}{n\mathcal{L}_t} \geq \frac{c}{n}.
$$

Let $\mathcal{G}_t := \frac{1}{n\mathcal{L}_t}$ and $S(x) = \int_{\mathcal{G}_s}^x \frac{1}{(\log t)^{2-2/M}}\,dt$. Then we have

$$
\frac{\mathcal{G}_t'}{(\log \mathcal{G}_t)^{2-\frac{2}{L}}} \geq \frac{c}{n} \quad\Rightarrow\quad S(\mathcal{G}_t) \geq \frac{c}{n}(t-s).
$$

Note that

$$
\mathcal{G}_s = \frac{1}{n\mathcal{L}_s} \geq \frac{\exp\big(-\mathsf{p}_a(\rho_s)\big)}{n\mathcal{L}_s} > 1.
$$

We can invoke Lemma G.2 to get

$$
\mathcal{G}_t \geq S^{-1}\left(\frac{c}{n}(t-s)\right) = \Omega(t(\log t)^{2-2/M}) \implies \mathcal{L}_t = \mathcal{O}\left(\frac{1}{t(\log t)^{2-2/M}}\right).
$$

This completes the proof of Lemma C.7. □

Then we control the rate of the parameter norm with respect to the risk. The following lemma characterizes the M-near homogeneity.

**Lemma C.8** (Near homogeneity). *Let $f(\boldsymbol{\theta}; \mathbf{x}_i)$ be locally Lipschitz and $M$-near-homogeneous, then for $\|\boldsymbol{\theta}\| \geq r$, there exists a constant $B_r$ such that*

$$
|f(\boldsymbol{\theta}; \mathbf{x}_i)| \leq B_r \cdot \|\boldsymbol{\theta}_t\|^M \quad \text{for all } i \in [n].
$$

*Proof of Lemma C.8.* We can fix an index $i$ and let $c_1 = \max_{\|\boldsymbol{\theta}\|=r}|f(\boldsymbol{\theta}; \mathbf{x}_i)|/r^M$. For any $\|\boldsymbol{\theta}\| \geq r$, we let $\bar{\boldsymbol{\theta}}_t = r\boldsymbol{\theta}/\|\boldsymbol{\theta}\|$. Consider $g(t) = f(t\bar{\boldsymbol{\theta}}; \mathbf{x}_i)/t^M$. Since $f$ is locally Lipschitz, $g(t)$ is also Locally Lipschitz and its derivative exists almost everywhere. Then we have for almost every $t > 0$,

$$
g'(t) = \frac{\langle \nabla_\theta f(t\bar{\boldsymbol{\theta}}; \mathbf{x}_i), t\bar{\boldsymbol{\theta}}_t \rangle - Mf(t\bar{\boldsymbol{\theta}}; \mathbf{x}_i)}{t^{M+1}} \leq \frac{a\|\bar{\boldsymbol{\theta}}\|^{M-1}}{t^2} = \frac{ar^{M-1}}{t^2}.
$$

We can also show that $g'(t) \geq -ar^{M-1}/t^2$. Note that $|g(1)| \leq c_1 r^M$. Since $g(t)$ is locally Lipschitz, it's also absolutely continuous. Hence we can apply the fundamental theorem of calculus to get

$$
\frac{|f(\boldsymbol{\theta}; \mathbf{x}_i)|r^M}{\|\boldsymbol{\theta}\|^M} = g(\|\boldsymbol{\theta}\|/r)
$$

$$
= g(1) + \int_1^{\|\boldsymbol{\theta}\|/r} g'(t)dt
$$

$$
\leq c_1 r^M + \int_1^{\|\boldsymbol{\theta}\|/r} |g'(t)|dt
$$

$$
\leq c_1 r^M + ar^{M-1}.
$$

This is equivalent to

$$|f(\boldsymbol{\theta}; \mathbf{x}_i)| \leq (a/r + c_1) \cdot \|\boldsymbol{\theta}\|^M.$$

Let $B_r = a/r + c_1$. We complete the proof of Lemma C.8. □

With this Lemma, we show that $\gamma$ and $\gamma^{\mathsf{GF}}$ are both bounded.

**Lemma C.9** (Boundedness of margins). *Under Assumptions 1 and 2, we have*

$$\gamma^{\mathsf{GF}}(\boldsymbol{\theta}_t) \leq \gamma(\boldsymbol{\theta}_t) \leq B_{\|\rho_s\|}.$$

*Proof of Lemma C.9.* Note that $\rho_t$ is increasing. Therefore, $\rho_t \geq \rho_s$. Then we can apply Lemma C.8 to get

$$\gamma(\boldsymbol{\theta}_t) = \frac{\min y_i f(\boldsymbol{\theta}_t; \mathbf{x}_i)}{\rho_t^M} \leq B_{\|\rho_s\|}.$$

Therefore, $\gamma^{\mathsf{GF}}(\boldsymbol{\theta}_t) \leq \gamma(\boldsymbol{\theta}_t) \leq B_{\|\rho_s\|}.$ □

**Lemma C.10** (Rate of the parameter norm). *Under Assumptions 1 and 2, for (GF), we have*

$$\rho_t^M = \Theta\left(\log \frac{1}{\mathcal{L}_t}\right),$$

*where the hidden constant depends on $\gamma^{\mathsf{GF}}(\boldsymbol{\theta}_s)$ and $\mathsf{q}$.*

*Proof of Lemma C.10.* Recall that

$$\gamma^{\mathsf{S}}(\boldsymbol{\theta}_t) \geq \gamma^{\mathsf{GF}}(\boldsymbol{\theta}_t) \geq \gamma^{\mathsf{GF}}(\boldsymbol{\theta}_s).$$

Then we have

$$\rho_t^M \leq \frac{1}{\gamma^{\mathsf{GF}}(\boldsymbol{\theta}_s)} \cdot \phi(\mathcal{L}_t) = \mathcal{O}\left(\log \frac{1}{\mathcal{L}_t}\right).$$

On the other hand, by Lemmas C.2 and C.8 we have

$$B\rho_t^M \geq \bar{f}_{\min}(\boldsymbol{\theta}_t) \geq \log \frac{1}{n\mathcal{L}_t} \implies \rho_t^M = \Omega\left(\log \frac{1}{\mathcal{L}_t}\right).$$

Combining these two bounds, we have

$$\rho_t^M = \Theta\left(\log \frac{1}{\mathcal{L}_t}\right).$$

We complete the proof of Lemma C.10. □

**Lemma C.11** (Lower bound of the risk). *Under Assumptions 1 and 2, for (GF), we have*

$$\mathcal{L}_t = \Omega\left(\frac{1}{t(\log t)^{2-2/M}}\right),$$

*where the hidden constant depends on $\gamma^{\mathsf{GF}}(\boldsymbol{\theta}_s)$ and $\mathsf{q}(x)$.*

*Proof of Lemma C.11.* Note that by Lemma A.6 and corollary A.8,

$$\frac{\mathrm{d}\boldsymbol{\theta}_t}{\mathrm{d}t} = -\bar{\partial}\mathcal{L}(\boldsymbol{\theta}_t) = \frac{1}{n}\sum_{i=1}^{n} e^{-y_i f(\boldsymbol{\theta}; \mathbf{x}_i)} y_i \mathbf{h}_i,$$

where $\bar{\partial}$ (14) denotes the minimal norm subgradient and $\mathbf{h}_i \in \partial f(\boldsymbol{\theta}; \mathbf{x}_i)$. Therefore under Assumption 1, we have

$$\|\bar{\partial}\mathcal{L}_t\| \leq \frac{1}{n}\sum_{i=1}^{n} e^{-y_i f(\boldsymbol{\theta}; \mathbf{x}_i)}\|\mathbf{h}_i\| \leq \mathcal{L}_t \cdot \mathsf{q}'(\rho_t).$$

By Assumption 1 and increase of $\rho_t$, we can bound $\mathsf{q}'(\rho_t)$ by $b\|\rho_t\|^{M-1}$ for some constant $b$. Then we have

$$-\frac{\mathrm{d}\mathcal{L}_t}{\mathrm{d}t} = \|\bar{\partial}\mathcal{L}_t\|^2 \leq \mathcal{L}_t^2 b^2 \rho_t^{2M-2} \leq \mathcal{L}_t^2 \cdot \mathcal{O}\left(\left(\log\frac{1}{\mathcal{L}_t}\right)^{2-2/M}\right).$$

By the definition of $S(\cdot)$ in the proof of Lemma C.7, this implies that

$$\frac{\mathrm{d}}{\mathrm{d}t}S\left(\frac{1}{\mathcal{L}_t}\right) \leq c.$$

Therefore we have

$$\frac{1}{\mathcal{L}_t} \leq S^{-1}(c(t-s)) = \mathcal{O}(t(\log t)^{2-2/M}) \implies \mathcal{L}_t = \Omega\left(\frac{1}{t(\log t)^{2-2/M}}\right).$$

This completes the proof of Lemma C.11. $\qquad\square$

**Theorem C.12** (Rates of the risk and the parameter norm). *Under Assumptions 1 and 2, for* (GF), *we have*

$$\mathcal{L}_t = \Theta\left(\frac{1}{t(\log t)^{2-2/M}}\right), \quad \rho_t = \Theta\left((\log t)^{\frac{1}{M}}\right),$$

*where the hidden constant depends on* $\gamma^{\mathsf{GF}}(\boldsymbol{\theta}_s)$ *and* $\mathsf{q}(x)$.

*Proof of Theorem C.12.* By Lemmas C.7 and C.11, we have

$$\mathcal{L}_t = \Theta\left(\frac{1}{t(\log t)^{2-2/M}}\right).$$

By Lemma C.10, we have

$$\rho_t^M = \Theta\left(\log\frac{1}{\mathcal{L}_t}\right) = \Theta(\log t) \implies \rho_t = \Theta\left((\log t)^{\frac{1}{M}}\right).$$

We complete the proof of Theorem C.12. $\qquad\square$

**Lemma C.13** (Multiplicative error). *The multiplicative error of* $\gamma^{\mathsf{GF}}(\boldsymbol{\theta}_t)$ *satisfies*

$$\frac{\log n + \mathsf{p}_a(\rho_t)}{\phi(\mathcal{L}_t) - \mathsf{p}_a(\rho_t)} \to 0.$$

*Proof of Lemma C.13.* Applying Theorem C.12, we have

$$\log\frac{1}{\mathcal{L}_t} = \Theta(\log t), \quad \mathsf{p}_a(\rho_t) = \mathsf{p}_a(\rho_t) = \Theta(\rho_t^{M-1}) = \Theta((\log t)^{1-1/M}).$$

Therefore we have

$$\frac{\log n + \mathsf{p}_a(\rho_t)}{\log\frac{1}{\mathcal{L}_t} - \log n - \mathsf{p}_a(\rho_t)} = \Theta\left(\frac{(\log t)^{1-1/M}}{\log t}\right) = \Theta\left(\frac{1}{(\log t)^{\frac{1}{M}}}\right) \to 0.$$

This completes the proof of Lemma C.13. $\qquad\square$

### C.4. Proof of Theorem 3.2

*Proof of Theorem 3.2.* By Theorem C.5, we know that $\gamma^{\mathsf{GF}}(\boldsymbol{\theta}_t)$ is increasing and $\mathcal{L}_t < e^{-\mathsf{p}_a(\rho_t)}/n$ for all $t \geq s$. By Theorem C.12, we get the rates of $\rho$ and $\mathcal{L}_t$. By Lemma C.13, we know $\gamma^{\mathsf{GF}}$ is a good approximator of $\gamma$. This completes the proof of Theorem 3.2. $\qquad\square$

## C.5. Proof of Example 3.3

*Proof of Example 3.3.* Note that for this example we have $\mathsf{p}(\boldsymbol{\theta}) = |\boldsymbol{\theta}|^M$, $\mathsf{p}_a(\boldsymbol{\theta}) = M|\boldsymbol{\theta}|^{M-1}$, and $f(\boldsymbol{\theta}) = \boldsymbol{\theta}^M + \mathsf{p}_a(|\boldsymbol{\theta}|)$. This means Assumption 2 is equivalent to:

$$\mathcal{L}(\boldsymbol{\theta}_s) = \exp(-f(\boldsymbol{\theta}_s)) < \exp(-p_a(|\boldsymbol{\theta}_s|)) \iff \boldsymbol{\theta}_s^M > 0 \iff \boldsymbol{\theta}_s > 0.$$

If the above condition does not hold, then $\boldsymbol{\theta}_s \leq 0$. Note that $0$ is a stationary point for $\mathcal{L}(\boldsymbol{\theta})$. So if (GF) is initialized from $\boldsymbol{\theta}_s$, it cannot produce positive parameters in the future, that is, $\boldsymbol{\theta}_t \leq 0$ for every $t \geq s$. Hence, (GF) cannot minimize the loss or exhibit any implicit bias. This completes the proof of Example 3.3. $\square$

## C.6. Directional Convergence

The main idea of the proof is to show that the curve swept by $\tilde{\boldsymbol{\theta}}_t$ has a finite length, thus $\tilde{\boldsymbol{\theta}}_t$ converges. We use $\zeta_t$ to denote the curve swept by $\tilde{\boldsymbol{\theta}}_t$. Another important quantity in our proof is the modified margin $\gamma^{\mathsf{GF}}$ defined in (6).

In Theorem 3.2, we show that $\gamma^{\mathsf{GF}}(\boldsymbol{\theta}_t)$ is nondecreasing with some limit $\gamma_* \in (0, +\infty)$ and $\|\boldsymbol{\theta}_t\| \to \infty$.

Motivated by Ji & Telgarsky (2020), we will invoke a sophisticated but standard tool in the analysis of definable functions, the *desingularizing function*. A desingularizing function will witness the flow is well-behaved and the curve swept by $\tilde{\boldsymbol{\theta}}_t$ has a finite length.

**Definition 6** (Desingularizing function). A function $\Psi : [0, \nu) \to \mathbb{R}$ is called a desingularizing function when $\Psi$ is continuous on $[0, \nu)$ with $\Psi(0) = 0$, and continuously differentiable on $(0, \nu)$ with $\Psi' > 0$.

The following Lemma plays a key role in proving the directional convergence of params.

**Lemma C.14** (Existence of desingularizing function). *There exist $R > 0, \nu > 0$ and a definable desingularizing function $\Psi$ on $[0, \nu)$, such that for a.e. large enough $t$ with $\|\boldsymbol{\theta}_t\| > R$ and $\gamma^{\mathsf{GF}}(\boldsymbol{\theta}_t) > \gamma_* - \nu$, it holds that*

$$\frac{\mathrm{d}\zeta_t}{\mathrm{d}t} \leq -c\frac{\mathrm{d}\Psi\left(\gamma_* - \gamma^{\mathsf{GF}}(\boldsymbol{\theta}_t)\right)}{\mathrm{d}t}.$$

*for some constant $c > 0$.*

The central part of this subsection is to prove Lemma C.14, and then we will use this lemma to prove Theorem 3.4 in Appendix C.7. To prove Lemma C.14, we need many valuable tools. We present the tools first.

Following the notation in Ji & Telgarsky (2020), given any $f$ that is locally Lipschitz around a nonzero $\boldsymbol{\theta}$, let

$$\bar{\partial}_r f(\boldsymbol{\theta}) := \langle \bar{\partial} f(\boldsymbol{\theta}), \tilde{\boldsymbol{\theta}} \rangle \tilde{\boldsymbol{\theta}} \quad \text{and} \quad \bar{\partial}_\perp f(\boldsymbol{\theta}) := \bar{\partial} f(\boldsymbol{\theta}) - \bar{\partial}_r f(\boldsymbol{\theta})$$

be the radial and spherical parts of $\bar{\partial} f(\boldsymbol{\theta})$ respectively. We use

$$a_t := \phi(\mathcal{L}_t) - \mathsf{p}_a(\rho_t)$$

to denote the numerator of $\gamma^{\mathsf{GF}}(\boldsymbol{\theta}_t)$. Before we dive into the proof of Lemma C.14, we need to characterize the Clarke subdifferential of $\gamma^{\mathsf{GF}}(\boldsymbol{\theta}_t)$.

**Lemma C.15** (Subdifferential of $\gamma^{\mathsf{GF}}$). *For (GF), we have*

$$\partial a_t = \left\{ -\frac{\nabla\mathcal{L}_t}{\mathcal{L}_t} - \mathsf{p}'_a(\rho_t)\tilde{\boldsymbol{\theta}}_t \,\middle|\, \text{for any } \nabla\mathcal{L}_t \in \partial\mathcal{L}_t \right\},$$

$$\partial\gamma^{\mathsf{GF}}(\boldsymbol{\theta}_t) = \left\{ -\frac{\nabla\mathcal{L}_t}{\mathcal{L}_t\rho_t^M} - \frac{\mathsf{p}'_a(\rho_t)\tilde{\boldsymbol{\theta}}_t}{\rho_t^M} - \frac{Ma_t\tilde{\boldsymbol{\theta}}_t}{\rho_t^{M+1}} \,\middle|\, \text{for any } \nabla\mathcal{L}_t \in \partial\mathcal{L}_t \right\}.$$

*Proof of Lemma C.15.* The proof of these two results is a direct consequence of the definition of Clarke subdifferentials. Recall that

$$\partial f(x) := \text{conv}\left\{ \lim_{i\to\infty} \nabla f(x_i) : x = \lim_{i\to\infty} x_i, \text{where } x_i \in D \text{ and } \nabla f(x_i) \text{ exists} \right\}.$$

Recall that $a_t$ and $\gamma^{\mathsf{GF}}(\boldsymbol{\theta}_t)$ are differentiable functions of $\mathcal{L}_t$. Next, we prove that: for a locally Lipschitz function $a(x): \mathbb{R}^d \to \mathbb{R}$ and a differentiable function $b(\cdot): \mathbb{R} \to \mathbb{R}$, we have

$$\partial_x[b(a(x))] = \left\{ b'|_{a(x)} \cdot \mathbf{h} \mid \text{for any } \mathbf{h} \in \partial_x a(x) \right\}.$$

By Theorem A.7, we know that

$$\partial_x[b(a(x))] \subset \left\{ b'|_{a(x)} \cdot \mathbf{h} \mid \text{for any } \mathbf{h} \in \partial_x a(x) \right\}.$$

It remains to show the other direction. For every $\mathbf{h} \in \partial_x a(x)$, there exists $\lim_{i \to \infty} x_i = x$ such that $\mathbf{h} = \lim_{i \to \infty} \nabla a(x_i)$. Since $\lim_{i \to \infty} a(x_i) = a(x)$, we have

$$\lim_{i \to \infty} b'(a(x_i)) = b'(a(x)) \Rightarrow \lim_{i \to \infty} b'(a(x_i)) \nabla a(x_i) = b'(a(x)) \mathbf{h}.$$

Here, we use the facts that $b$ is differentiable at $a(x_i)$ and that $a$ is differentiable at $x_i$, which implies that $b(a(\cdot))$ is also differentiable at $x_i$. Hence we have

$$\left\{ b'|_{a(x)} \cdot \mathbf{h} \mid \text{for any } \mathbf{h} \in \partial_x a(x) \right\} \subset \partial_x[b(a(x))].$$

The above result completes the proof of Lemma C.15. $\qquad\square$

Consider the following two special elements within the set of subdifferentials,

$$\tilde{\partial} a_t := -\frac{\bar{\partial}\mathcal{L}_t}{\mathcal{L}_t} - \mathsf{p}'_a(\rho_t)\tilde{\boldsymbol{\theta}}_t, \tag{22}$$

$$\tilde{\partial}\gamma^{\mathsf{GF}}(\boldsymbol{\theta}_t) := -\frac{\bar{\partial}\mathcal{L}_t}{\mathcal{L}_t \rho_t^M} - \frac{\mathsf{p}'_a(\rho_t)\tilde{\boldsymbol{\theta}}_t}{\rho_t^M} - \frac{M a_t \tilde{\boldsymbol{\theta}}_t}{\rho_t^{M+1}}. \tag{23}$$

These two elements are crucial for our analysis, enabling us to deal with the non-homogeneity. Specifically, this together with Lemma A.6 leads to Lemma C.17. Another remarkable point is that Ji & Telgarsky (2020) did not consider these special elements, instead, they considered $\bar{\partial} a_t$ and $\bar{\partial}\gamma^{\mathsf{GF}}(\boldsymbol{\theta}_t)$. This is because their model is homogeneous so $\mathsf{p}_a = 0$ in their analysis. It is straightforward to check that $\bar{\partial} a_t$, and $-\bar{\partial}\mathcal{L}_t$ are all in the same direction for homogeneous models. However, for non-homogeneous models, we need to consider the above elements. Thanks to Lemma A.6, we can use any element of $\partial a_t$ and $\partial\gamma^{\mathsf{GF}}(\boldsymbol{\theta}_t)$ with the chain rule, that is, we have

$$\frac{\mathrm{d}a(\boldsymbol{\theta}_t)}{\mathrm{d}t} = \langle \tilde{\partial} a(\boldsymbol{\theta}_t), -\bar{\partial}\mathcal{L}_t \rangle, \quad \frac{\mathrm{d}\gamma^{\mathsf{GF}}(\boldsymbol{\theta}_t)}{\mathrm{d}t} = \langle \tilde{\partial}\gamma^{\mathsf{GF}}(\boldsymbol{\theta}_t), -\bar{\partial}\mathcal{L}_t \rangle.$$

This helps us to understand the increase of $\gamma^{\mathsf{GF}}(\boldsymbol{\theta}_t)$ in terms of the spherical and the radial change of $\bar{\partial}\mathcal{L}_t$. To further elaborate this, we define:

$$\tilde{\partial}_r a_t := \langle \tilde{\partial} a_t, \tilde{\boldsymbol{\theta}}_t \rangle \tilde{\boldsymbol{\theta}}_t, \qquad \tilde{\partial}_\perp a_t := \tilde{\partial} a_t - \tilde{\partial}_r a_t, \tag{24}$$

$$\tilde{\partial}_r \gamma^{\mathsf{GF}}(\boldsymbol{\theta}_t) := \langle \tilde{\partial}\gamma^{\mathsf{GF}}(\boldsymbol{\theta}_t), \tilde{\boldsymbol{\theta}}_t \rangle \tilde{\boldsymbol{\theta}}_t, \qquad \tilde{\partial}_\perp \gamma^{\mathsf{GF}}(\boldsymbol{\theta}_t) := \tilde{\partial}\gamma^{\mathsf{GF}}(\boldsymbol{\theta}_t) - \tilde{\partial}_r \gamma^{\mathsf{GF}}(\boldsymbol{\theta}_t). \tag{25}$$

With simple calculations, we get the following inequalities.

**Lemma C.16** (Decomposition of tilde subdifferential)**.** *Under Assumptions 1 and 2, for $\tilde{\partial}_r \gamma^{\mathsf{GF}}(\boldsymbol{\theta}_t)$ and $\tilde{\partial}_\perp \gamma^{\mathsf{GF}}(\boldsymbol{\theta}_t)$ in (24), we have*

$$\langle \tilde{\partial}_r a_t, \tilde{\boldsymbol{\theta}}_t \rangle \geq \frac{M a_t}{\rho_t} \geq 0, \quad \tilde{\partial}_\perp a_t = -\frac{\bar{\partial}_\perp \mathcal{L}_t}{\mathcal{L}_t}, \tag{26}$$

$$\langle \tilde{\partial}_r \gamma^{\mathsf{GF}}(\boldsymbol{\theta}_t), \tilde{\boldsymbol{\theta}}_t \rangle \geq 0, \quad \tilde{\partial}_\perp \gamma^{\mathsf{GF}}(\boldsymbol{\theta}_t) = -\frac{\bar{\partial}_\perp \mathcal{L}_t}{\mathcal{L}_t \rho_t^M}. \tag{27}$$

*Proof of Lemma C.16.* We prove the results for $a_t$ and $\gamma^{\mathsf{GF}}(\boldsymbol{\theta}_t)$ separately.

The first inequality for $a_t$. We have

$$
\begin{aligned}
\langle \tilde{\partial}_r a_t, \tilde{\boldsymbol{\theta}}_t \rangle &= -\frac{\langle \bar{\partial}\mathcal{L}_t, \tilde{\boldsymbol{\theta}}_t \rangle}{\mathcal{L}_t} - \mathsf{p}'_a(\rho_t) = \frac{\langle -\bar{\partial}\mathcal{L}_t, \boldsymbol{\theta}_t \rangle}{\mathcal{L}_t \rho_t} - \mathsf{p}'_a(\rho_t) \\
&= \frac{1}{n\mathcal{L}_t \rho_t} \sum_{i=1}^n e^{-\bar{f}_i(\boldsymbol{\theta}_t)} \langle \nabla \bar{f}_i(\boldsymbol{\theta}_t), \boldsymbol{\theta}_t \rangle - \mathsf{p}'_a(\rho_t) \\
&\geq \frac{1}{n\mathcal{L}_t \rho_t} \sum_{i=1}^n e^{-\bar{f}_i(\boldsymbol{\theta}_t)} (M \bar{f}_i(\boldsymbol{\theta}_t) - \mathsf{p}'(\rho_t)) - \mathsf{p}'_a(\rho_t) \\
&\geq \frac{1}{n\mathcal{L}_t \rho_t} \sum_{i=1}^n e^{-\bar{f}_i(\boldsymbol{\theta}_t)} (M \log(1/n\mathcal{L}_t) - \mathsf{p}'(\rho_t) - \mathsf{p}'_a(\rho_t)\rho_t) \\
&\geq \frac{1}{n\mathcal{L}_t \rho_t} \sum_{i=1}^n e^{-\bar{f}_i(\boldsymbol{\theta}_t)} M a_t = \frac{M a_t}{\rho_t} \geq 0.
\end{aligned}
$$

The second inequality for $a_t$. We have

$$
\begin{aligned}
\tilde{\partial}_\perp a_t &= -\frac{\bar{\partial}\mathcal{L}_t}{\mathcal{L}_t} - \mathsf{p}'_a(\rho_t)\tilde{\boldsymbol{\theta}}_t - \left\langle -\frac{\bar{\partial}\mathcal{L}_t}{\mathcal{L}_t} - \mathsf{p}'_a(\rho_t)\tilde{\boldsymbol{\theta}}_t, \tilde{\boldsymbol{\theta}}_t \right\rangle \tilde{\boldsymbol{\theta}}_t \\
&= -\frac{\bar{\partial}\mathcal{L}_t}{\mathcal{L}_t} + \frac{\bar{\partial}_r \mathcal{L}_t}{\mathcal{L}_t} = -\frac{\bar{\partial}_\perp \mathcal{L}_t}{\mathcal{L}_t}.
\end{aligned}
$$

The first inequality for $\gamma^{\mathsf{GF}}(\boldsymbol{\theta}_t)$. We have

$$
\begin{aligned}
\langle \tilde{\partial}_r \gamma^{\mathsf{GF}}(\boldsymbol{\theta}_t), \tilde{\boldsymbol{\theta}}_t \rangle &= -\frac{\langle \bar{\partial}\mathcal{L}_t, \tilde{\boldsymbol{\theta}}_t \rangle}{\mathcal{L}_t \rho_t^M} - \frac{\mathsf{p}'_a(\rho_t)}{\rho_t^M} - \frac{M a_t}{\rho_t^{M+1}} = \frac{\langle -\bar{\partial}\mathcal{L}_t, \boldsymbol{\theta}_t \rangle}{\mathcal{L}_t \rho_t^{M+1}} - \frac{\mathsf{p}'_a(\rho_t)}{\rho_t^M} - \frac{M a_t}{\rho_t^{M+1}} \\
&= \frac{1}{n\mathcal{L}_t \rho_t^{M+1}} \sum_{i=1}^n e^{-\bar{f}_i(\boldsymbol{\theta}_t)} \langle \nabla \bar{f}_i(\boldsymbol{\theta}_t), \boldsymbol{\theta}_t \rangle - \frac{\mathsf{p}'_a(\rho_t)}{\rho_t^M} - \frac{M a_t}{\rho_t^{M+1}} \\
&\geq \frac{1}{n\mathcal{L}_t \rho_t^{M+1}} \sum_{i=1}^n e^{-\bar{f}_i(\boldsymbol{\theta}_t)} (M \bar{f}_i(\boldsymbol{\theta}_t) - \mathsf{p}'(\rho_t)) - \frac{\mathsf{p}'_a(\rho_t)}{\rho_t^M} - \frac{M a_t}{\rho_t^{M+1}} \\
&\geq \frac{1}{n\mathcal{L}_t \rho_t^{M+1}} \sum_{i=1}^n e^{-\bar{f}_i(\boldsymbol{\theta}_t)} (M \log(1/n\mathcal{L}_t) - \mathsf{p}'(\rho_t) - \mathsf{p}'_a(\rho_t)\rho_t) - \frac{M a_t}{\rho_t^{M+1}} \\
&\geq \frac{1}{n\mathcal{L}_t \rho_t^{M+1}} \sum_{i=1}^n e^{-\bar{f}_i(\boldsymbol{\theta}_t)} M a_t - \frac{M a_t}{\rho_t^{M+1}} = 0.
\end{aligned}
$$

The second inequality for $\gamma^{\mathsf{GF}}(\boldsymbol{\theta}_t)$. We have

$$
\begin{aligned}
\tilde{\partial}_\perp \gamma^{\mathsf{GF}}(\boldsymbol{\theta}_t) &= -\frac{\bar{\partial}\mathcal{L}_t}{\mathcal{L}_t \rho_t^M} - \frac{\mathsf{p}'_a(\rho_t)\tilde{\boldsymbol{\theta}}_t}{\rho_t^M} - \frac{M a_t \tilde{\boldsymbol{\theta}}_t}{\rho_t^{M+1}} - \left\langle -\frac{\bar{\partial}\mathcal{L}_t}{\mathcal{L}_t \rho_t^M} - \frac{\mathsf{p}'_a(\rho_t)\tilde{\boldsymbol{\theta}}_t}{\rho_t^M} - \frac{M a_t \tilde{\boldsymbol{\theta}}_t}{\rho_t^{M+1}}, \tilde{\boldsymbol{\theta}}_t \right\rangle \tilde{\boldsymbol{\theta}}_t \\
&= -\frac{\bar{\partial}\mathcal{L}_t}{\mathcal{L}_t \rho_t^M} + \frac{\bar{\partial}_r \mathcal{L}_t}{\mathcal{L}_t \rho_t^M} = -\frac{\bar{\partial}_\perp \mathcal{L}_t}{\mathcal{L}_t \rho_t^M}.
\end{aligned}
$$

This completes the proof of Lemma C.16. $\qquad\square$

After a detailed characterization of $\tilde{\partial}\boldsymbol{\theta}_t$, we can give a decomposition of the increase of margin. This decomposition will help us construct the desired desingularizing function.

**Lemma C.17** (Decomposition of radial and spherical parts)**.** *Under Assumptions 1 and 2, for almost every $t \geq s$, we have*

$$
\frac{\mathrm{d}\gamma^{\mathsf{GF}}(\boldsymbol{\theta}_t)}{\mathrm{d}t} = \|\tilde{\partial}_r \gamma^{\mathsf{GF}}(\boldsymbol{\theta}_t)\| \|\bar{\partial}_r \mathcal{L}_t\| + \|\tilde{\partial}_\perp \gamma^{\mathsf{GF}}(\boldsymbol{\theta}_t)\| \|\bar{\partial}_\perp \mathcal{L}_t\|, \tag{28}
$$

*and*

$$\|\tilde{\partial}_\perp \gamma^{\mathsf{GF}}(\boldsymbol{\theta}_t)\| = \frac{\|\tilde{\partial}_\perp a_t\|}{\rho_t^M}, \quad \textit{and} \quad \frac{\mathrm{d}\zeta_t}{\mathrm{d}t} = \frac{\|\bar{\partial}_\perp \mathcal{L}_t\|}{\rho_t}. \tag{29}$$

*Proof of Lemma C.17.* We prove the two equations separately.

Proof of (28). By Lemma A.6, we have for almost every $t \geq s$,

$$\frac{\mathrm{d}\gamma^{\mathsf{GF}}(\boldsymbol{\theta}_t)}{\mathrm{d}t} = \langle \tilde{\partial}\gamma^{\mathsf{GF}}(\boldsymbol{\theta}_t), -\bar{\partial}\mathcal{L}_t \rangle = \langle \tilde{\partial}_r \gamma^{\mathsf{GF}}(\boldsymbol{\theta}_t), -\bar{\partial}_r \mathcal{L}_t \rangle + \langle \tilde{\partial}_\perp \gamma^{\mathsf{GF}}(\boldsymbol{\theta}_t), -\bar{\partial}_\perp \mathcal{L}_t \rangle.$$

To prove (28), we need to show two arguments:

- $\langle \tilde{\partial}_r \gamma^{\mathsf{GF}}(\boldsymbol{\theta}_t), \boldsymbol{\theta}_t \rangle$ and $\langle -\bar{\partial}_r \mathcal{L}_t, \boldsymbol{\theta}_t \rangle$ share the same sign.

- $\tilde{\partial}_\perp \gamma^{\mathsf{GF}}(\boldsymbol{\theta}_t)$ and $-\bar{\partial}_\perp \mathcal{L}_t$ point the same direction.

● **The first argument.** By Lemma C.16 we have that $\langle \tilde{\partial}_r \gamma^{\mathsf{GF}}(\boldsymbol{\theta}_t), \tilde{\boldsymbol{\theta}}_t \rangle \geq 0$. On the other hand, we have

$$
\begin{aligned}
\langle -\nabla \mathcal{L}_t, \boldsymbol{\theta}_t \rangle &= \frac{1}{n}\sum_{i=1}^n e^{-\bar{f}_i(\boldsymbol{\theta}_t)} \langle \nabla \bar{f}_i(\boldsymbol{\theta}_t), \boldsymbol{\theta}_t \rangle \\
&\geq \frac{1}{n}\sum_{i=1}^n e^{-\bar{f}_i(\boldsymbol{\theta}_t)} M \bar{f}_i(\boldsymbol{\theta}_t) - p'(\rho_t)\mathcal{L}_t \qquad &&\text{By Assumption 1} \\
&\geq M\mathcal{L}_t \phi(\mathcal{L}_t) - \mathcal{L}_t p'(\rho_t) \qquad &&\text{By Lemma C.2} \\
&\geq \mathcal{L}_t M a_t \qquad &&\text{By Lemma C.1} \\
&\geq 0. \qquad &&\text{By monotonicity of } \gamma^{\mathsf{GF}}(\boldsymbol{\theta}_t) \text{ in Theorem 3.2}
\end{aligned}
$$

Therefore $\langle \bar{\partial}_r \gamma^{\mathsf{GF}}(\boldsymbol{\theta}_t), \boldsymbol{\theta}_t \rangle$ and $\langle -\bar{\partial}_r \mathcal{L}_t, \boldsymbol{\theta}_t \rangle$ share the same sign. This completes the proof of the first argument.

● **The second argument.** This is a direct result of Lemma C.16.

Proof of (29). The first part is also a direction result of Lemma C.16. For the second part, since $\boldsymbol{\theta}_t$ is an arc and $\|\boldsymbol{\theta}_t\| \geq \|\boldsymbol{\theta}_s\| > 0$, it follows that $\tilde{\boldsymbol{\theta}}_t$ is also an arc. Moreover, for almost every $t \geq 0$,

$$\frac{\mathrm{d}\tilde{\boldsymbol{\theta}}_t}{\mathrm{d}t} = \frac{1}{\rho_t}\frac{\mathrm{d}\boldsymbol{\theta}_t}{\mathrm{d}t} - \frac{1}{\rho_t}\left\langle \frac{\mathrm{d}\boldsymbol{\theta}_t}{\mathrm{d}t}, \tilde{\boldsymbol{\theta}}_t \right\rangle \cdot \tilde{\boldsymbol{\theta}}_t = -\frac{\bar{\partial}_\perp \mathcal{L}_t}{\rho_t}.$$

Since $\tilde{\boldsymbol{\theta}}_t$ is an arc, $\mathrm{d}\tilde{\boldsymbol{\theta}}_t/\mathrm{d}t$ and $\|\mathrm{d}\tilde{\boldsymbol{\theta}}_t/\mathrm{d}t\|$ are both integrable. By definition of the curve length,

$$\zeta_t = \int_s^t \left\|\frac{\mathrm{d}\tilde{\boldsymbol{\theta}}_t}{\mathrm{d}t}\right\|\mathrm{d}t,$$

and for almost every $t \geq s$, we have

$$\frac{\mathrm{d}\zeta(t)}{\mathrm{d}t} = \left\|\frac{\mathrm{d}\tilde{\boldsymbol{\theta}}_t}{\mathrm{d}t}\right\| = \frac{\|\bar{\partial}_\perp \mathcal{L}_t\|}{\rho_t}.$$

This completes the proof of (29). Therefore, we have proved Lemma C.17. □

Then we need some inequalities to connect $(\bar{\partial}_r \mathcal{L}_t, \bar{\partial}_\perp \mathcal{L}_t)$ and $(\tilde{\partial}_r \gamma^{\mathsf{GF}}(\boldsymbol{\theta}_t), \tilde{\partial}_\perp \gamma^{\mathsf{GF}}(\boldsymbol{\theta}_t))$. The main idea is to show that their ratios are close, that is,

$$\frac{\|\bar{\partial}_\perp \mathcal{L}_t\|}{\|\bar{\partial}_r \mathcal{L}_t\|} \approx \frac{\|\tilde{\partial}_\perp \gamma^{\mathsf{GF}}(\boldsymbol{\theta}_t)\|}{\|\tilde{\partial}_r \gamma^{\mathsf{GF}}(\boldsymbol{\theta}_t)\|}.$$

The technique is to use $a_t$ to bridge these two ratios.

**Lemma C.18** (Inequalities between $a_t$ and $\gamma^{\mathsf{GF}}(\boldsymbol{\theta}_t)$). *Under Assumption 1 and the condition $\mathcal{L}_s \leq e^{-\mathsf{p}_a(\rho_s)}/n$, for almost every $t \geq s$, we have*

$$\|\tilde{\partial}_r a_t\| \geq M\gamma^{\mathsf{GF}}(\boldsymbol{\theta}_s)\rho_t^{M-1}, \tag{30}$$

*and*

$$\|\tilde{\partial}_r \gamma^{\mathsf{GF}}(\boldsymbol{\theta}_t)\| \leq \frac{M\log n + 2M\mathsf{p}_a(\rho_t)}{\rho_t^{M+1}}. \tag{31}$$

*Combining these two inequalities, there exists a threshold $s_1 > s > 0$, for almost every $t \geq s_1$, we have*

$$\|\tilde{\partial}_r a_t\| \geq \gamma^{\mathsf{GF}}(\boldsymbol{\theta}_s)\rho_t^{M+1/2}\|\tilde{\partial}_r \gamma^{\mathsf{GF}}(\boldsymbol{\theta}_t)\|. \tag{32}$$

*Proof of Lemma C.18.* We prove the three inequalities separately.

**Proof of** (30). We expand the formula. For almost every $t \geq s$, we have

$$
\begin{aligned}
\|\tilde{\partial}_r a_t\| = \langle \tilde{\partial} a_t, \tilde{\boldsymbol{\theta}}_t \rangle &\geq \frac{Ma_t}{\rho_t} && \text{By Lemma C.16} \\
&= M\gamma^{\mathsf{GF}}(\boldsymbol{\theta}_t)\rho_t^{M-1} && \text{By definition of } \gamma^{\mathsf{GF}}(\boldsymbol{\theta}_t) \\
&\geq M\gamma^{\mathsf{GF}}(\boldsymbol{\theta}_s)\rho_t^{M-1}. && \text{By monotonicity in Theorem 3.2}
\end{aligned}
$$

**Proof of** (31). For almost every $t \geq s$, we have

$$\|\tilde{\partial}_r \gamma^{\mathsf{GF}}(\boldsymbol{\theta}_t)\| = \langle \tilde{\partial}\gamma^{\mathsf{GF}}(\boldsymbol{\theta}_t), \tilde{\boldsymbol{\theta}}_t \rangle = -\frac{\langle \bar{\partial}\mathcal{L}_t, \tilde{\boldsymbol{\theta}}_t \rangle}{\mathcal{L}_t \rho_t^M} - \frac{\mathsf{p}_a'(\rho_t)}{\rho_t^M} - \frac{Ma_t}{\rho_t^{M+1}} = \frac{\langle \tilde{\partial}a_t, \boldsymbol{\theta}_t \rangle - Ma_t}{\rho_t^{M+1}}. \tag{33}$$

Now we derive an upper bound for $\langle \tilde{\partial}a_t, \boldsymbol{\theta}_t \rangle$. For almost every $t \geq s$, we have

$$
\begin{aligned}
\langle \tilde{\partial}a_t, \boldsymbol{\theta}_t \rangle &= \frac{1}{\mathcal{L}_t}\left[\frac{1}{n}\sum_{i=1}^n e^{-\bar{f}_i(\boldsymbol{\theta}_t)}\langle \nabla\bar{f}_i(\boldsymbol{\theta}_t), \boldsymbol{\theta}_t \rangle - \mathcal{L}_t \mathsf{p}_a'(\rho_t)\rho_t\right] \\
&\leq \frac{1}{\mathcal{L}_t}\left[\frac{1}{n}\sum_{i=1}^n e^{-\bar{f}_i(\boldsymbol{\theta}_t)}M\bar{f}_i(\boldsymbol{\theta}_t) + \mathcal{L}_t\rho'(\rho_t) - \mathcal{L}_t\mathsf{p}_a'(\rho_t)\rho_t\right] && \text{By Assumption 1} \\
&\leq \frac{1}{\mathcal{L}_t}\left[\frac{1}{n}\sum_{i=1}^n e^{-\bar{f}_i(\boldsymbol{\theta}_t)}M\bar{f}_i(\boldsymbol{\theta}_t) + \mathcal{L}_t\rho'(\rho_t) + \mathcal{L}_t\mathsf{p}_a'(\rho_t)\rho_t\right] \\
&= \frac{1}{n\mathcal{L}_t}\sum_{i=1}^n e^{-\bar{f}_i(\boldsymbol{\theta}_t)}M\bar{f}_i(\boldsymbol{\theta}_t) + M\mathsf{p}_a(\rho_t). && \text{By Lemma C.1 .}
\end{aligned}
$$

By Lemma G.1, we have $-\nabla\pi(v)v = \nabla\pi(v)(0-v) \leq \pi(0) - \pi(v)$. Applying this to the above inequality, we have

$$\langle \tilde{\partial}a_t, \boldsymbol{\theta}_t \rangle \leq M\log(1/\mathcal{L}_t) + M\mathsf{p}_a(\rho_t).$$

Therefore,

$$\langle \tilde{\partial}a_t, \boldsymbol{\theta}_t \rangle - Ma_t \leq M\log(n) + 2M\mathsf{p}_a(\rho_t).$$

Plugging this into (33), we have proved

$$\|\tilde{\partial}_r \gamma^{\mathsf{GF}}(\boldsymbol{\theta}_t)\| \leq \frac{M\log n + 2M\mathsf{p}_a(\rho_t)}{\rho_t^{M+1}}.$$

This completes the proof of of (31).

**Proof of** (32). To prove (32), we combine (30) and (31) to get

$$\|\tilde{\partial}_r a_t\| \geq M\gamma^{\mathsf{GF}}(\boldsymbol{\theta}_t)\rho_t^{M-1} \geq \frac{\gamma^{\mathsf{GF}}(\boldsymbol{\theta}_t)\rho_t^{2M}}{\log n + 2p_a(\rho_t)}\|\tilde{\partial}_r\gamma^{\mathsf{GF}}(\boldsymbol{\theta}_t)\|.$$

Since $\mathsf{p}_a(\rho_t)$ is a polynomial of $\rho_t$ with degree $(M-1)$ and $\rho_t$ is increasing with rate $O([\log(t)]^{\frac{1}{M}})$ (by Theorem 3.2), there exists a threshold $s_1 > s > 0$ such that for almost every $t \geq s_1$, we have

$$\log n + 2\mathsf{p}_a(\rho_t) \leq \rho_t^{M-\frac{1}{2}}.$$

Therefore for almost every $t \geq s_1$, we have

$$\|\tilde{\partial}_r a_t\| \geq \gamma^{\mathsf{GF}}(\boldsymbol{\theta}_s)\rho_t^{M+1/2}\|\tilde{\partial}_r \gamma^{\mathsf{GF}}(\boldsymbol{\theta}_t)\|.$$

This completes the proof of (32). We have proved Lemma C.18. $\qquad\square$

We proceed to prove Lemma C.14. The following two Kurdyka-Lojasiewicz inequalities in Ji & Telgarsky (2020) will help us construct the desired desingularizing function. We refer the readers to Appendix B in Ji & Telgarsky (2020) for the proofs of them.

**Lemma C.19** (Lemma 3.6 in Ji & Telgarsky (2020)). *Given a locally Lipschitz definable function $f$ with an open domain $D \subset \{x \mid \|x\| > 1\}$, for any $c, \eta > 0$, there exists $\nu > 0$ and a definable desingularizing function $\Psi$ on $[0, \nu)$ such that*

$$\Psi'(f(x))\|x\|\|\bar{\partial}f(x)\| \geq 1, \quad \text{if } f(x) \in (0, \nu) \text{ and } \left\|\bar{\partial}_\perp f(x)\right\| \geq c\|x\|^\eta \left\|\bar{\partial}_r f(x)\right\|.$$

**Lemma C.20** (Lemma 3.7 in Ji & Telgarsky (2020)). *Given a locally Lipschitz definable function $f$ with an open domain $D \subset \{x \mid \|x\| > 1\}$, for any $\lambda > 0$, there exists $\nu > 0$ and a definable desingularizing function $\Psi$ on $[0, \nu)$ such that*

$$\max\left\{1, \frac{2}{\lambda}\right\}\Psi'(f(x))\|x\|^{1+\lambda}\|\bar{\partial}f(x)\| \geq 1, \quad \text{if } f(x) \in (0, \nu).$$

*Proof of Lemma C.14.* Recall that we have the following decomposition by (28):

$$\frac{\mathrm{d}\gamma^{\mathsf{GF}}(\boldsymbol{\theta}_t)}{\mathrm{d}t} = \|\tilde{\partial}_r \gamma^{\mathsf{GF}}(\boldsymbol{\theta}_t)\|\|\bar{\partial}_r \mathcal{L}_t\| + \|\tilde{\partial}_\perp \gamma^{\mathsf{GF}}(\boldsymbol{\theta}_t)\|\|\bar{\partial}_\perp \mathcal{L}_t\|.$$

Two cases will be considered in this proof:

- Case 1: $\|\tilde{\partial}_r \gamma^{\mathsf{GF}}(\boldsymbol{\theta}_t)\|\|\bar{\partial}_r \mathcal{L}_t\|$ is larger, and we will apply Lemma C.20 for construction.

- Case 2: $\|\tilde{\partial}_\perp \gamma^{\mathsf{GF}}(\boldsymbol{\theta}_t)\|\|\bar{\partial}_\perp \mathcal{L}_t\|$ is larger, and we will apply Lemma C.19 for construction.

The two cases will be determined by the ratio of $\|\tilde{\partial}_\perp \gamma^{\mathsf{GF}}(\boldsymbol{\theta}_t)\|$ and $\|\tilde{\partial}_r \gamma^{\mathsf{GF}}(\boldsymbol{\theta}_t)\|$. Specifically, case 1 corresponds to

$$\|\tilde{\partial}_\perp \gamma^{\mathsf{GF}}(\boldsymbol{\theta}_t)\| \leq \rho_t^{\frac{1}{8}}\|\tilde{\partial}_r \gamma^{\mathsf{GF}}(\boldsymbol{\theta}_t)\|, \tag{34}$$

and case 2 corresponds to

$$\|\tilde{\partial}_\perp \gamma^{\mathsf{GF}}(\boldsymbol{\theta}_t)\| \geq \rho_t^{\frac{1}{8}}\|\tilde{\partial}_r \gamma^{\mathsf{GF}}(\boldsymbol{\theta}_t)\|. \tag{35}$$

**Case 1.** By (32), we have

$$\begin{aligned}
\|\tilde{\partial}_r a_t\| &\geq \gamma^{\mathsf{GF}}(\boldsymbol{\theta}_s)\rho_t^{M+\frac{1}{2}}\|\tilde{\partial}_r \gamma^{\mathsf{GF}}(\boldsymbol{\theta}_t)\| \\
&\geq \gamma^{\mathsf{GF}}(\boldsymbol{\theta}_s)\rho_t^{M+\frac{3}{8}}\|\tilde{\partial}_\perp \gamma^{\mathsf{GF}}(\boldsymbol{\theta}_t)\| &&\text{By (34)} \\
&= \gamma^{\mathsf{GF}}(\boldsymbol{\theta}_s)\rho_t^{\frac{3}{8}}\|\tilde{\partial}_\perp a_t\|. &&\text{By Lemma C.17} \tag{36}
\end{aligned}$$

Now we transfer this inequality to the ratio of $\|\bar{\partial}_\perp \mathcal{L}_t\|$ and $\|\bar{\partial}_r \mathcal{L}_t\|$. Note that

$$\|\tilde{\partial}_r a_t\| = \langle \tilde{\partial}a_t, \boldsymbol{\theta}_t \rangle = \frac{1}{\rho_t}\langle \tilde{\partial}a_t, \boldsymbol{\theta}_t \rangle = -\frac{\langle \bar{\partial}\mathcal{L}_t, \boldsymbol{\theta}_t \rangle}{\mathcal{L}_t \rho_t} - \mathsf{p}_a'(\rho_t)$$

$$\leq -\frac{\langle \bar{\partial}\mathcal{L}_t, \boldsymbol{\theta}_t \rangle}{\mathcal{L}_t \rho_t} = -\frac{\langle \bar{\partial}\mathcal{L}_t, \tilde{\boldsymbol{\theta}}_t \rangle}{\mathcal{L}_t} = \frac{1}{\mathcal{L}_t}\|\bar{\partial}_r \mathcal{L}_t\|. \tag{37}$$

On the other hand, we have

$$\|\tilde{\partial}_\perp a_t\| = \frac{1}{\mathcal{L}_t}\|\bar{\partial}_\perp \mathcal{L}_t\|. \tag{38}$$

Plugging (37) and (38) into (36), we have

$$\|\bar{\partial}_r \mathcal{L}_t\| \geq \gamma^{\mathsf{GF}}(\boldsymbol{\theta}_s)\rho_t^{\frac{3}{8}}\|\bar{\partial}_\perp \mathcal{L}_t\|. \tag{39}$$

On the other hand, we know that there exists $s_2 > s > 0$ such that for almost every $t \geq s_2$, we have $\rho_t > 1$. Hence we have

$$
\begin{aligned}
\|\tilde{\partial}\gamma^{\mathsf{GF}}(\boldsymbol{\theta}_t)\| &\leq \|\tilde{\partial}_r \gamma^{\mathsf{GF}}(\boldsymbol{\theta}_t)\| + \|\tilde{\partial}_\perp \gamma^{\mathsf{GF}}(\boldsymbol{\theta}_t)\| &\\
&\leq (1 + \rho_t^{\frac{1}{8}})\|\tilde{\partial}_r \gamma^{\mathsf{GF}}(\boldsymbol{\theta}_t)\| &\text{By (34)}\\
&\leq 2\rho_t^{\frac{1}{8}}\|\tilde{\partial}_r \gamma^{\mathsf{GF}}(\boldsymbol{\theta}_t)\|. &\text{By } \rho_t \geq 1
\end{aligned}
\tag{40}
$$

Therefore by (28), we have

$$
\begin{aligned}
\frac{\mathrm{d}\gamma^{\mathsf{GF}}(\boldsymbol{\theta}_t)}{\mathrm{d}t} &\geq \|\tilde{\partial}_r \gamma^{\mathsf{GF}}(\boldsymbol{\theta}_t)\|\|\bar{\partial}_r \mathcal{L}_t\| &\\
&\geq \frac{1}{2}\rho_t^{\frac{1}{4}}\gamma^{\mathsf{GF}}(\boldsymbol{\theta}_s)\|\tilde{\partial}\gamma^{\mathsf{GF}}(\boldsymbol{\theta}_t)\|\|\bar{\partial}_\perp \mathcal{L}_t\| &\text{By (39) and (40)}\\
&\geq \frac{1}{2}\rho_t^{\frac{1}{4}}\gamma^{\mathsf{GF}}(\boldsymbol{\theta}_s)\|\bar{\partial}\gamma^{\mathsf{GF}}(\boldsymbol{\theta}_t)\|\|\bar{\partial}_\perp \mathcal{L}_t\| &\text{By } \|\tilde{\partial}\gamma^{\mathsf{GF}}(\boldsymbol{\theta}_t)\| \geq \|\bar{\partial}\gamma^{\mathsf{GF}}(\boldsymbol{\theta}_t)\|\\
&\geq \rho_t^{\frac{5}{4}}\|\bar{\partial}\gamma^{\mathsf{GF}}(\boldsymbol{\theta}_t)\|\frac{\mathrm{d}\zeta_t}{\mathrm{d}t}. &\text{By Lemma C.17}
\end{aligned}
\tag{41}
$$

Now we invoke Lemma C.20 to construct the desingularizing function. We apply it to the definable function $\gamma_* - \gamma^{\mathsf{GF}}(\boldsymbol{\theta}_t)$ with $\lambda = \frac{1}{4}$. Then there exists $\nu_1 > 0$ and a definable desingularizing function $\Psi_1$ on $[0, \nu_1)$ such that

$$8\Psi_1'(\gamma_* - \gamma^{\mathsf{GF}}(\boldsymbol{\theta}_t))\rho_t^{5/4}\|\bar{\partial}\gamma^{\mathsf{GF}}(\boldsymbol{\theta}_t)\| \geq 1, \quad \text{if } \gamma^{\mathsf{GF}}(\boldsymbol{\theta}_t) \geq \gamma_* - \nu_1.$$

Plugging (41) into the above inequality, we have

$$8\Psi_1'(\gamma_* - \gamma^{\mathsf{GF}}(\boldsymbol{\theta}_t))\frac{\mathrm{d}\gamma^{\mathsf{GF}}(\boldsymbol{\theta}_t)}{\mathrm{d}t} \geq \frac{\mathrm{d}\zeta_t}{\mathrm{d}t}, \quad \text{if } \gamma^{\mathsf{GF}}(\boldsymbol{\theta}_t) \geq \gamma_* - \nu_1.$$

This completes the proof for case 1.

**Case 2.** By (28), we have

$$\frac{\mathrm{d}\gamma^{\mathsf{GF}}(\boldsymbol{\theta}_t)}{\mathrm{d}t} \geq \|\tilde{\partial}_\perp \gamma^{\mathsf{GF}}(\boldsymbol{\theta}_t)\|\|\bar{\partial}_\perp \mathcal{L}_t\| = \rho_t\|\tilde{\partial}_\perp \gamma^{\mathsf{GF}}(\boldsymbol{\theta}_t)\|\frac{\mathrm{d}\zeta_t}{\mathrm{d}t}. \tag{42}$$

For almost every $t \geq s_1 > s > 0$, we have $\rho_t \geq 1$ and

$$\|\tilde{\partial}_\perp \gamma^{\mathsf{GF}}(\boldsymbol{\theta}_t)\| \geq \rho_t^{\frac{1}{8}}\|\tilde{\partial}_r \gamma^{\mathsf{GF}}(\boldsymbol{\theta}_t)\| \geq \|\tilde{\partial}_r \gamma^{\mathsf{GF}}(\boldsymbol{\theta}_t)\|.$$

This leads to

$$\|\tilde{\partial}_\perp \gamma^{\mathsf{GF}}(\boldsymbol{\theta}_t)\| \geq \frac{1}{2}(\|\tilde{\partial}_\perp \gamma^{\mathsf{GF}}(\boldsymbol{\theta}_t)\| + \|\tilde{\partial}_r \gamma^{\mathsf{GF}}(\boldsymbol{\theta}_t)\|) \geq \frac{1}{2}\|\tilde{\partial}\gamma^{\mathsf{GF}}(\boldsymbol{\theta}_t)\|.$$

Plugging this into (42), we have

$$\frac{\mathrm{d}\gamma^{\mathsf{GF}}(\boldsymbol{\theta}_t)}{\mathrm{d}t} \geq \frac{\rho_t}{2}\|\tilde{\partial}\gamma^{\mathsf{GF}}(\boldsymbol{\theta}_t)\|\frac{\mathrm{d}\zeta_t}{\mathrm{d}t} \geq \frac{\rho_t}{2}\|\bar{\partial}\gamma^{\mathsf{GF}}(\boldsymbol{\theta}_t)\|\frac{\mathrm{d}\zeta_t}{\mathrm{d}t}. \tag{43}$$

We invoke Lemma C.19 to construct the desingularizing function. We apply it to the definable function $\gamma_* - \gamma^{\mathsf{GF}}(\boldsymbol{\theta}_t)$ with $c = 1$ and $\eta = \frac{1}{8}$. Then there exists $\nu_2 > 0$ and a definable desingularizing function $\Psi_2$ on $[0, \nu_2)$ such that

$$\Psi_2'(\gamma_* - \gamma^{\mathsf{GF}}(\boldsymbol{\theta}_t))\rho_t\|\bar{\partial}\gamma^{\mathsf{GF}}(\boldsymbol{\theta}_t)\| \geq 1, \quad \text{if } \gamma^{\mathsf{GF}}(\boldsymbol{\theta}_t) \geq \gamma_* - \nu_2.$$

Plugging (43) into the above inequality, we have

$$2\Psi_2'(\gamma_* - \gamma^{\mathsf{GF}}(\boldsymbol{\theta}_t))\frac{\mathrm{d}\gamma^{\mathsf{GF}}(\boldsymbol{\theta}_t)}{\mathrm{d}t} \geq \frac{\mathrm{d}\zeta_t}{\mathrm{d}t}, \quad \text{if } \gamma^{\mathsf{GF}}(\boldsymbol{\theta}_t) \geq \gamma_* - \nu_2.$$

This completes the proof for case 2.

**Combining the two cases.** Since $\Psi_1' - \Psi_2'$ is a definable function, it is nonnegative or nonpositive on some interval $(0, \nu)$. Let $\Psi = \max\{\Psi_1, \Psi_2\}$. Then we have for almost every large enough $t$ such that $\rho_t \geq 1$ and $\gamma^{\mathsf{GF}}(\boldsymbol{\theta}_t) \geq \gamma_* - \nu$, and $\log n + 2p_a(\rho_t) \leq \rho_t^{M-\frac{1}{2}}$, it holds that

$$\frac{\mathrm{d}\gamma^{\mathsf{GF}}(\boldsymbol{\theta}_t)}{\mathrm{d}t} \geq \frac{1}{c\Psi'(\gamma_* - \gamma^{\mathsf{GF}}(\boldsymbol{\theta}_t))}\frac{\mathrm{d}\zeta_t}{\mathrm{d}t},$$

for some constant $c > 0$. This completes the proof of Lemma C.14. $\qquad\square$

### C.7. Proof of Theorem 3.4

Now we are ready to prove Theorem 3.4 with the powerful tool, Lemma C.14.

*Proof of Theorem 3.4.* Let $t_0$ be large enough so that the condition in Lemma C.14 holds. Then we have

$$\lim_{t \to \infty} \zeta_t \leq \zeta_{t_0} + c\Psi(\gamma_* - \gamma^{\mathsf{GF}}(\boldsymbol{\theta}_{t_0})) \leq \infty.$$

This completes the proof of Theorem 3.4. $\qquad\square$

### C.8. KKT convergence

We now prove Theorem 3.5 by verifying the KKT conditions. Recall that the optimization problem (P) is defined as follows:

$$\min \|\boldsymbol{\theta}\|_2^2 \quad \text{s.t. } y_i f_{\mathsf{H}}(\boldsymbol{\theta}; \mathbf{x}_i) \geq 1 \text{ for all } i \in [n].$$

Recall that $\bar{f}_i(\boldsymbol{\theta}) = y_i f(\boldsymbol{\theta}; \mathbf{x}_i)$ and $\bar{f}_{\min}(\boldsymbol{\theta}) = \min_{i \in [n]} \bar{f}_i(\boldsymbol{\theta})$. We use $\bar{f}_{\mathsf{H},i}(\boldsymbol{\theta})$ to denote $y_i f_{\mathsf{H}}(\boldsymbol{\theta}; \mathbf{x}_i)$ and $\bar{f}_{\mathsf{H},\min}$ to denote $\min_{i \in [n]} \bar{f}_{\mathsf{H},i}(\boldsymbol{\theta})$. Recall that by Proposition 5.1, $\bar{f}_{\mathsf{H},i}$ is continuously differentiable on $\mathbb{R}^d/\{0\}$. We also assume that $f$ is continuous differentiable with respect to $\boldsymbol{\theta}$ on $\mathbb{R}^d/\{0\}$ in Assumption 3. For simplicity, we use $\mathbf{h}_i$ to denote $\nabla \bar{f}_i$ and $\mathbf{h}_{\mathsf{H},i}$ to denote $\nabla \bar{f}_{\mathsf{H},i}$. We define

$$\mathbf{h} = \frac{1}{n}\sum_{i=1}^{n} e^{-\bar{f}_i(\boldsymbol{\theta}_t)}\mathbf{h}_i, \quad \mathbf{h}_{\mathsf{H}} = \frac{1}{n}\sum_{i=1}^{n} e^{-\bar{f}_i(\boldsymbol{\theta}_t)}\mathbf{h}_{\mathsf{H},i}.$$

Notice that $\mathbf{h}$ and $\mathbf{h}_{\mathsf{H}}$ are weighted sum of $\mathbf{h}_i$'s and $\mathbf{h}_{\mathsf{H},i}$'s respectively, where the weights are the same. Actually, this $\mathbf{h}_{\mathsf{H}}$ is very tricky. Unlike $\mathbf{h} = -\nabla\mathcal{L}_t$, $\mathbf{h}_{\mathsf{H}}$ is not the negative version of gradient of

$$\mathcal{L}_{\mathsf{H}}(\boldsymbol{\theta}) := \frac{1}{n}\sum_{i=1}^{n} e^{-\bar{f}_{\mathsf{H},i}(\boldsymbol{\theta};\mathbf{x}_i)}.$$

It is just a weighted sum of $\nabla \bar{f}_{\mathsf{H},i}$. Next, we define the KKT conditions and approximate KKT conditions for the optimization problem (P).

**Definition 7** (KKT conditions). A feasible point $\boldsymbol{\theta}$ of (P) is a KKT point if there exist $\lambda_1, \ldots, \lambda_n \geq 0$ such that

1. $\boldsymbol{\theta} - \sum_{i=1}^{n} \lambda_i \mathbf{h}_{\mathsf{H},i} = 0$ for some $\mathbf{h}_{\mathsf{H},1}, \ldots, \mathbf{h}_{\mathsf{H},n}$ satisfying $\mathbf{h}_{\mathsf{H},i} = \nabla \bar{f}_{\mathsf{H},i}(\boldsymbol{\theta})$ for every $i \in [n]$;

2. For every $i \in [n]$, $\lambda_i(\bar{f}_{\mathsf{H},i}(\boldsymbol{\theta}) - 1) = 0$.

**Definition 8** (Approximate KKT conditions). A feasible point $\boldsymbol{\theta}$ of (P) is an $(\epsilon, \delta)$-KKT point if ther exists $\lambda_1, \ldots, \lambda_n \geq 0$ such that

1. $\|\boldsymbol{\theta} - \sum_{i=1}^{n} \lambda_i \mathbf{h}_{\mathsf{H},i}\| \leq \epsilon$ for some $\mathbf{h}_{\mathsf{H},1}, \ldots, \mathbf{h}_{\mathsf{H},n}$ satisfying $\mathbf{h}_{\mathsf{H},i} = \nabla \bar{f}_{\mathsf{H},i}(\boldsymbol{\theta})$ for every $i \in [n]$;

2. For every $i \in [n]$, $\lambda_i \left| \bar{f}_{\mathsf{H},i}(\boldsymbol{\theta}) - 1 \right| \leq \delta$.

The first step is to bound the difference between $\bar{f}_{\min}$ and $\bar{f}_{\mathsf{H},\min}$.

**Lemma C.21** (Bound between $\bar{f}_{\min}$ and $\bar{f}_{\mathsf{H},\min}$). *Under Assumptions 1 and 2, for any $\epsilon$, there exists $s_2 > s > 0$ such that*

$$\text{for almost every } t \geq s_2, \quad \left\| \frac{\boldsymbol{\theta}_t}{\left( \bar{f}_{\min}(\boldsymbol{\theta}_t) \right)^{1/M}} - \frac{\boldsymbol{\theta}_t}{\left( \bar{f}_{\mathsf{H},\min}(\boldsymbol{\theta}_t) \right)^{1/M}} \right\| \leq \epsilon.$$

*Proof of Lemma C.21.* By Theorem 3.4, we have $\tilde{\boldsymbol{\theta}}_t$ converges to $\boldsymbol{\theta}_*$. Then we are going to show that $\bar{f}_i(\boldsymbol{\theta}_t)/\rho_t^M$ converges for all $i$. Recall that in Proposition 5.1, we have

$$|f(\boldsymbol{\theta}; \mathbf{x}) - f_{\mathsf{H}}(\boldsymbol{\theta}_t; \mathbf{x})| \leq \mathsf{p}_a(\|\boldsymbol{\theta}_t\|).$$

Therefore we have

$$\left| \frac{\bar{f}_i(\boldsymbol{\theta}_t)}{\rho_t^M} - \frac{\bar{f}_{\mathsf{H},i}(\boldsymbol{\theta}_t)}{\rho_t^M} \right| \leq \frac{\mathsf{p}_a(\rho_t)}{\rho_t^M} \to 0, \quad \text{as } t \to \infty.$$

Hence we have

$$\lim_{t \to \infty} \frac{\bar{f}_i(\boldsymbol{\theta}_t)}{\rho_t^M} = \lim_{t \to \infty} \frac{\bar{f}_{\mathsf{H},i}(\boldsymbol{\theta}_t)}{\rho_t^M} = \bar{f}_{\mathsf{H},i}(\boldsymbol{\theta}_*).$$

As a consequence, we have

$$\lim_{t \to \infty} \frac{\rho_t}{\left( \bar{f}_i(\boldsymbol{\theta}_t) \right)^{1/M}} = \lim_{t \to \infty} \frac{\rho_t}{\left( \bar{f}_{\mathsf{H},i}(\boldsymbol{\theta}_t) \right)^{1/M}} = \frac{1}{\left( \bar{f}_{\mathsf{H},i}(\boldsymbol{\theta}_*) \right)^{1/M}}.$$

Because we only have a finite number of $i$, we must have

$$\lim_{t \to \infty} \frac{\rho_t}{\left( \bar{f}_{\min}(\boldsymbol{\theta}_t) \right)^{1/M}} = \lim_{t \to \infty} \frac{\rho_t}{\left( \bar{f}_{\mathsf{H},\min}(\boldsymbol{\theta}_t) \right)^{1/M}} = \frac{1}{\left( \bar{f}_{\mathsf{H},\min}(\boldsymbol{\theta}_*) \right)^{1/M}}.$$

Therefore, for every $\epsilon$, there exists $s_2 > s > 0$ such that for almost every $t \geq s_2$, we have

$$\left\| \frac{\boldsymbol{\theta}_t}{\left( \bar{f}_{\min}(\boldsymbol{\theta}_t) \right)^{1/M}} - \frac{\boldsymbol{\theta}_t}{\left( \bar{f}_{\mathsf{H},\min}(\boldsymbol{\theta}_t) \right)^{1/M}} \right\| \leq \epsilon.$$

This completes the proof of Lemma C.21. $\square$

Due to the above lemma, we can focus on

$$\hat{\boldsymbol{\theta}}_t := \boldsymbol{\theta}_t / \left( \bar{f}_{\mathsf{H},\min}(\boldsymbol{\theta}_t) \right)^{1/M}$$

and verify it satisfies the $(\epsilon, \delta)$-KKT conditions. The key is to construct the coefficients $\lambda_i$:

$$\lambda_i(\boldsymbol{\theta}_t) = \frac{\bar{f}_{\mathsf{H},\min}^{1-2/M}(\boldsymbol{\theta}_t) \rho_t e^{-\bar{f}_i(\boldsymbol{\theta}_t)}}{n \|\mathbf{h}_{\mathsf{H}}\|}. \tag{44}$$

We define the following key quantity for our proof,

$$\beta(t) = \frac{\langle \boldsymbol{\theta}_t, \mathbf{h}_{\mathsf{H}}(\boldsymbol{\theta}_t) \rangle}{\|\boldsymbol{\theta}_t\| \cdot \|\mathbf{h}_{\mathsf{H}}\|}. \tag{45}$$

**Lemma C.22** (Lower bound of the homogeneous margin). *Under Assumptions 1 and 2, we have*

$$\frac{f_{\mathsf{H},\min}(\boldsymbol{\theta}_t)}{\|\boldsymbol{\theta}_t\|^M} \geq \gamma^{\mathsf{GF}}(\boldsymbol{\theta}_s) > 0.$$

*Specifically, there exists $B$, such that for all $t$,*

$$\bar{f}_{\mathsf{H},i}^{-1/M}(\boldsymbol{\theta}_t) \cdot \|\boldsymbol{\theta}_t\| \leq B. \tag{46}$$

*Proof.* Note that

$$f_{\mathsf{H},\min}(\boldsymbol{\theta}_t) \geq f_{\min}(\boldsymbol{\theta}_t) - p_a(\rho_t) \qquad \text{by Proposition 5.1}$$

$$\geq \log \frac{1}{\mathcal{L}_t n} - p_a(\rho_t). \qquad \text{by Lemma C.2}$$

Hence,

$$\frac{f_{\mathsf{H},\min}(\boldsymbol{\theta}_t)}{\|\boldsymbol{\theta}_t\|^M} \geq \gamma^{\mathsf{GF}}(\boldsymbol{\theta}_t) \geq \gamma^{\mathsf{GF}}(\boldsymbol{\theta}_s) > 0.$$

And,

$$\bar{f}_{\mathsf{H},i}^{-1/M}(\boldsymbol{\theta}_t) \cdot \|\boldsymbol{\theta}_t\| \leq \left(\gamma^{\mathsf{GF}}(\boldsymbol{\theta}_s)\right)^{-\frac{1}{M}} =: B.$$

This completes the proof of Lemma C.22. □

**Lemma C.23** ($\tilde{\boldsymbol{\theta}}$ satisfies the $(\epsilon, \delta)$-KKT conditions)**.** *Under Assumptions 1 and 2, there exists $s_2 > s > 0$ such that for a.e. $t \geq s_2$, $\hat{\boldsymbol{\theta}}_t$ is an $(\epsilon, \delta)$-KKT point of* (P)*, where*

$$\epsilon = \sqrt{2} B \sqrt{1 - \beta(t)} \quad \text{and} \quad \delta = nB^2 \frac{1 + 2p_a(\rho_t)}{M \bar{f}_{\mathsf{H},\min}(\boldsymbol{\theta}_t)}.$$

*Proof of Lemma C.23.* We need to verify:

1. $\|\hat{\boldsymbol{\theta}} - \sum_{i=1}^n \lambda_i \mathbf{h}_{\mathsf{H},i}\| \leq \epsilon$ for some $\mathbf{h}_{\mathsf{H},1}, \ldots, \mathbf{h}_{\mathsf{H},n}$ satisfying $\mathbf{h}_{\mathsf{H},i} = \nabla \bar{f}_{\mathsf{H},i}(\hat{\boldsymbol{\theta}})$ for all $i \in [n]$.

2. For any $i \in [n]$, $\lambda_i |\bar{f}_{\mathsf{H},i}(\hat{\boldsymbol{\theta}}) - 1| \leq \delta$.

Note that we only verify one side of condition 2, since by the definition of $\hat{\boldsymbol{\theta}}$, we have $\bar{f}_{\mathsf{H},i}(\hat{\boldsymbol{\theta}}) \geq 1$ for all $i$.

**Condition 1.** We expand the left side of the inequality:

$$\hat{\boldsymbol{\theta}}_t - \sum_{i=1}^n \lambda_i \mathbf{h}_{\mathsf{H},i} = \bar{f}_{\mathsf{H},\min}^{-1/M}(\boldsymbol{\theta}_t) \rho_t \cdot \tilde{\boldsymbol{\theta}}_t - \bar{f}_{\mathsf{H},\min}^{-1/M}(\boldsymbol{\theta}_t) \cdot \frac{1}{n} \sum_{i=1}^n \frac{\bar{f}_{\mathsf{H},\min}^{1-1/M}(\boldsymbol{\theta}_t) \rho_t e^{-\bar{f}_i(\boldsymbol{\theta}_t)}}{\|\mathbf{h}_{\mathsf{H}}\|} \nabla \bar{f}_{\mathsf{H},i}(\hat{\boldsymbol{\theta}}_t)$$

$$= \bar{f}_{\mathsf{H},\min}^{-1/M}(\boldsymbol{\theta}_t) \rho_t \cdot \tilde{\boldsymbol{\theta}}_t - \bar{f}_{\mathsf{H},\min}^{-1/M}(\boldsymbol{\theta}_t) \rho_t \cdot \frac{1}{n} \sum_{i=1}^n \frac{e^{-\bar{f}_i(\boldsymbol{\theta}_t)}}{\|\mathbf{h}_{\mathsf{H}}\|} \nabla \bar{f}_{\mathsf{H},i}(\boldsymbol{\theta}_t)$$

$$= \bar{f}_{\mathsf{H},\min}^{-1/M}(\boldsymbol{\theta}_t) \rho_t \cdot \left(\tilde{\boldsymbol{\theta}}_t - \frac{\mathbf{h}_{\mathsf{H}}}{\|\mathbf{h}_{\mathsf{H}}\|}\right),$$

where the second equation is by the definition of $\hat{\boldsymbol{\theta}}$ and the homogeneity of $\bar{f}_{\mathsf{H},i}$, and the last equation is by the definition of $\mathbf{h}_{\mathsf{H}}$. Hence we have

$$\left\|\hat{\boldsymbol{\theta}}_t - \sum_{i=1}^n \lambda_i \mathbf{h}_{\mathsf{H},i}\right\|^2 \leq B^2 \left\|\tilde{\boldsymbol{\theta}}_t - \frac{\mathbf{h}_{\mathsf{H}}}{\|\mathbf{h}_{\mathsf{H}}\|}\right\|^2 = 2B^2(1 - \beta(t)).$$

This completes the proof of Condition 1.

**Condition 2.** For condition 2, we have shown that $\mathcal{L}_t \leq e^{-p_a(\rho_t)}/n$. This leads to

$$\bar{f}_i(\boldsymbol{\theta}_t) > p_a(\rho_t) \quad \Rightarrow \quad \bar{f}_{\mathsf{H},i}(\boldsymbol{\theta}_t) > \bar{f}_i(\boldsymbol{\theta}_t) - p_a(\rho_t) \geq 0.$$

Therefore we have $\bar{f}_{\mathsf{H},i}(\hat{\boldsymbol{\theta}}_t) \geq 1$ since $\hat{\boldsymbol{\theta}}_t := \boldsymbol{\theta}_t / \left(\bar{f}_{\mathsf{H},\min}(\boldsymbol{\theta}_t)\right)^{1/M}$. It suffices to show $\sum \lambda_i \left(\bar{f}_{\mathsf{H},i}(\hat{\boldsymbol{\theta}}_t) - 1\right) \leq \delta$. This is because

$$\sum_{i=1}^n \lambda_i \left(\bar{f}_{\mathsf{H},i}(\hat{\boldsymbol{\theta}}_t) - 1\right) = \frac{\rho_t \bar{f}_{\mathsf{H},\min}^{-2/M}(\boldsymbol{\theta}_t)}{\|\mathbf{h}_{\mathsf{H}}\|} \cdot \frac{1}{n} \sum_{i=1}^n e^{-\bar{f}_i(\boldsymbol{\theta}_t)} \left(\bar{f}_{\mathsf{H},i}(\boldsymbol{\theta}_t) - \bar{f}_{\mathsf{H},\min}(\boldsymbol{\theta}_t)\right)$$

$$
= \frac{\rho_t \bar{f}_{\mathsf{H},\min}^{-2/M}(\boldsymbol{\theta}_t)}{\|\mathbf{h}_{\mathsf{H}}\| e^{\bar{f}_{\min}(\boldsymbol{\theta}_t)}} \cdot \frac{1}{n} \sum_{i=1}^{n} e^{-\left(\bar{f}_i(\boldsymbol{\theta}_t) - \bar{f}_{\min}(\boldsymbol{\theta}_t)\right)} \left(\bar{f}_{\mathsf{H},i}(\boldsymbol{\theta}_t) - \bar{f}_{\mathsf{H},\min}(\boldsymbol{\theta}_t)\right)
$$

$$
\leq \frac{\rho_t \bar{f}_{\mathsf{H},\min}^{-2/M}(\boldsymbol{\theta}_t)}{\|\mathbf{h}_{\mathsf{H}}\| e^{\bar{f}_{\min}(\boldsymbol{\theta}_t)}} \cdot \frac{1}{n} \sum_{i=1}^{n} e^{-\left(\bar{f}_i(\boldsymbol{\theta}_t) - \bar{f}_{\min}(\boldsymbol{\theta}_t)\right)} \left(\bar{f}_i(\boldsymbol{\theta}_t) - \bar{f}_{\min}(\boldsymbol{\theta}_t) + 2\mathsf{p}_a(\rho_t)\right)
$$

$$
\text{By } |\bar{f}_{\mathsf{H},i}(\boldsymbol{\theta}_t) - \bar{f}_i(\boldsymbol{\theta}_t)| \leq \mathsf{p}_a(\rho_t) \text{ in Proposition 5.1}
$$

$$
\leq \frac{\rho_t \bar{f}_{\mathsf{H},\min}^{-2/M}(\boldsymbol{\theta}_t)}{\|\mathbf{h}_{\mathsf{H}}\| e^{\bar{f}_{\min}(\boldsymbol{\theta}_t)}} \cdot \left(1 + 2\mathsf{p}_a(\rho_t)\right). \tag{47}
$$

$$
\text{By } \bar{f}_i(\boldsymbol{\theta}_t) - \bar{f}_{\min}(\boldsymbol{\theta}_t) \geq 0 \text{ and } e^{-x} x \leq 1 \text{ when } x > 0
$$

The remaining part is to lower bound $\|\mathbf{h}_{\mathsf{H}}\|$. We have

$$
\|\mathbf{h}_{\mathsf{H}}\| \geq \frac{\langle \mathbf{h}_{\mathsf{H}}, \boldsymbol{\theta}_t \rangle}{\|\boldsymbol{\theta}_t\|}
$$

$$
= \frac{M}{n \|\boldsymbol{\theta}_t\|} \sum_{i=1}^{n} e^{-\bar{f}_i(\boldsymbol{\theta}_t)} \bar{f}_{\mathsf{H},i}(\boldsymbol{\theta}_t)
$$

$$
\geq \frac{M e^{-\bar{f}_{\min}(\boldsymbol{\theta}_t)} \bar{f}_{\mathsf{H},\min}(\boldsymbol{\theta}_t)}{n \rho_t}, \qquad\qquad \text{by } \bar{f}_{\mathsf{H},i}(\boldsymbol{\theta}) \geq 0 \text{ and Lemma C.21}
$$

Plugging this into (47), we get

$$
\sum_{i=1}^{n} \lambda_i \left(f_{\mathsf{H},i}(\tilde{\boldsymbol{\theta}}_t) - 1\right) \leq \frac{n}{M} \cdot \rho_t^2 \bar{f}_{\mathsf{H},\min}^{-2/M}(\boldsymbol{\theta}_t) \cdot \frac{1 + 2\mathsf{p}_a(\rho_t)}{\bar{f}_{\mathsf{H},\min}(\boldsymbol{\theta}_t)} \qquad \text{by (46)}
$$

$$
\leq \frac{nB^2}{M} \cdot \frac{1 + 2\mathsf{p}_a(\rho_t)}{\bar{f}_{\mathsf{H},\min}(\boldsymbol{\theta}_t)}.
$$

This completes the proof of Lemma C.23. $\qquad\qquad\qquad\qquad\qquad\qquad\qquad\qquad\qquad\qquad$ □

It remains to show that $\epsilon$ and $\delta$ are both close to zero. The second is easy to verify because $f_{\mathsf{H},\min}$ is M-homogeneous.

**Lemma C.24** ($\delta$ goes to 0)**.** *Under Assumptions 1 and 2, we have*

$$
\delta = nB^2 \frac{1 + 2p_a(\rho_t)}{M \bar{f}_{\mathsf{H},\min}(\boldsymbol{\theta}_t)} \to 0, \quad as \ t \to \infty.
$$

*Proof of Lemma C.24.* Recall that

$$
\lim_{t \to \infty} \frac{\bar{f}_{\mathsf{H},i}(\boldsymbol{\theta}_t)}{\rho_t^M} = \bar{f}_{\mathsf{H},i}(\boldsymbol{\theta}_*).
$$

Denote the limit as $\gamma_*$. Then there exists $s_3 > s > 0$ such that for almost every $t \geq s_3$, we have

$$
\frac{\bar{f}_{\mathsf{H},i}(\boldsymbol{\theta}_t)}{\rho_t^M} \geq \gamma_*/2 \implies \bar{f}_{\mathsf{H},i}(\boldsymbol{\theta}_t) \geq \gamma_* \rho_t^M/2.
$$

Then we have

$$
\lim_{t \to \infty} \delta \leq \lim_{\rho_t \to \infty} nB^2 \frac{1 + 2\mathsf{p}_a(\rho_t)}{M \gamma_* \rho_t^M/2} = 0.
$$

This completes the proof of Lemma C.24. $\qquad\qquad\qquad\qquad\qquad\qquad\qquad\qquad\qquad\qquad$ □

We next show $\epsilon = \mathcal{O}(\sqrt{1 - \beta(t)})$ is close to zero. However, $\beta(t)$ might not converge as it is given by

$$
\beta(t) = \frac{\langle \boldsymbol{\theta}_t, \mathbf{h}_{\mathsf{H}} \rangle}{\|\boldsymbol{\theta}_t\| \cdot \|\mathbf{h}_{\mathsf{H}}\|}, \quad \mathbf{h}_{\mathsf{H}} = \frac{1}{n} \sum_{i=1}^{n} e^{-\bar{f}_i(\boldsymbol{\theta}_t)} \nabla \bar{f}_{\mathsf{H},i}(\boldsymbol{\theta}_t),
$$

where $\mathbf{h}_{\mathsf{H}}$ might not converge in direction. To address this issue, our strategy is to show that there exists a subsequence $(t_k)_{k \geq 0}$ such that $\beta(t_k) \to 1$. To this end, we need to characterize $\beta(t)$ using the stronger Assumption 3. By Assumption 3 and Proposition 5.1, there is a function $\mathsf{r}(x) = o(x^{M-1})$ as $x \to \infty$, such that for almost every $\boldsymbol{\theta}_t$ and any $i \in [n]$, we have

$$\|\nabla \bar{f}_i(\boldsymbol{\theta}_t) - \nabla \bar{f}_{\mathsf{H},i}(\boldsymbol{\theta}_t)\| \leq \mathsf{r}(\|\boldsymbol{\theta}_t\|) = \mathsf{r}(\rho_t).$$

**Lemma C.25** (Characterization of $\beta(t)$). *Under Assumptions 2 and 3 , there exists $s_5 > s > 0$ such that for almost every $t_2 > t_1 \geq s_5$, we have*

$$\int_{t_1}^{t_2} \left( \frac{1 - p_1(t)}{\left(\beta(t) + p_2(t)\right)^2} - 1 \right) \cdot \frac{\mathrm{d}}{\mathrm{d}\tau} \log \rho(\tau) \cdot \mathrm{d}\tau \leq \frac{1}{M} \log \frac{\gamma^{\mathsf{GF}}(\boldsymbol{\theta}_{t_2})}{\gamma^{\mathsf{GF}}(\boldsymbol{\theta}_{t_1})},$$

*where*

$$p_1(t) := \frac{2\mathsf{r}(\rho_t)}{M\gamma^{\mathsf{GF}}(\boldsymbol{\theta}_s)\rho_t^{M-1}}, \quad p_2(t) := \frac{2\mathsf{p}_a(\rho_t)}{\gamma^{\mathsf{GF}}(\boldsymbol{\theta}_s)\rho_t^M}.$$

*Proof of Lemma C.25.* Recall that we have (19), that is,

$$
\begin{aligned}
\frac{\mathrm{d} \log \gamma^{\mathsf{GF}}(\boldsymbol{\theta}_t)}{\mathrm{d}t} &\geq \frac{\|\boldsymbol{\theta}_t'\|^2 \rho_t^2 - \langle \boldsymbol{\theta}_t', \boldsymbol{\theta}_t \rangle^2}{\rho_t^2 \mathcal{L}_t(\phi(\mathcal{L}_t) - \mathsf{p}_a(\rho_t))} \\
&\geq \frac{\|\mathbf{h}\|^2 \rho_t^2 - \langle \mathbf{h}, \boldsymbol{\theta}_t \rangle^2}{\rho_t^2 \mathcal{L}_t(\phi(\mathcal{L}_t) - \mathsf{p}_a(\rho_t))} \\
&= \frac{2\|\mathbf{h}\|^2 - 2\langle \mathbf{h}, \tilde{\boldsymbol{\theta}}_t \rangle^2}{\mathcal{L}_t a_t \cdot \frac{\mathrm{d}\rho_t^2}{\mathrm{d}t}/\rho_t^2} \cdot \frac{\mathrm{d} \log \rho_t}{\mathrm{d}t}. \qquad \text{By } \frac{\mathrm{d} \log \rho_t}{\mathrm{d}t} = \frac{\mathrm{d}\rho_t^2}{2\rho_t^2 \mathrm{d}t}
\end{aligned}
$$

Note that for almost every $t \geq s$,

$$
\begin{aligned}
\frac{1}{2}\frac{\mathrm{d}\rho_t^2}{\mathrm{d}t} &= \frac{1}{n} \sum_{i=1}^{n} e^{-\bar{f}_i(\boldsymbol{\theta}_t)} \langle \nabla \bar{f}_i(\boldsymbol{\theta}_t), \boldsymbol{\theta}_t \rangle \\
&\leq \frac{1}{n} \sum_{i=1}^{n} e^{-\bar{f}_i(\boldsymbol{\theta}_t)} \left[ M\bar{f}_i(\boldsymbol{\theta}_t) + \mathsf{p}'(\|\boldsymbol{\theta}\|) \right] \\
&\qquad \text{By Assumption 1} \\
&\leq \frac{1}{n} \sum_{i=1}^{n} e^{-\bar{f}_i(\boldsymbol{\theta}_t)} \left[ M\bar{f}_{\mathsf{H},i}(\boldsymbol{\theta}_t) + M\mathsf{p}_a(\|\boldsymbol{\theta}_t\|) + \mathsf{p}'(\|\boldsymbol{\theta}\|) \right] \\
&\qquad \text{By } |f(\boldsymbol{\theta};\mathbf{x}) - f_{\mathsf{H}}(\boldsymbol{\theta}_t;\mathbf{x})| \leq \mathsf{p}_a(\|\boldsymbol{\theta}_t\|) \\
&\leq \frac{1}{n} \sum_{i=1}^{n} e^{-\bar{f}_i(\boldsymbol{\theta}_t)} \langle \nabla \bar{f}_{\mathsf{H},i}(\boldsymbol{\theta}_t), \boldsymbol{\theta}_t \rangle + 2M\mathcal{L}_t \mathsf{p}_a(\rho_t) \\
&\qquad \text{By } M\mathsf{p}_a(x) \geq \mathsf{p}'(x) \text{ for } x > 0 \\
&= \langle \mathbf{h}_{\mathsf{H}}, \boldsymbol{\theta}_t \rangle + 2M\mathcal{L}_t \mathsf{p}_a(\rho_t). \qquad\qquad (48)
\end{aligned}
$$

Similarly, we have

$$
\begin{aligned}
\mathcal{L}_t a_t &= \mathcal{L}_t \phi(\mathcal{L}_t) - \mathsf{p}_a(\rho_t)\mathcal{L}_t \\
&\leq \mathcal{L}_t \phi(\mathcal{L}_t) - \mathsf{p}'(\rho_t)\mathcal{L}_t/M \\
&\qquad \text{By } M\mathsf{p}_a(x) \geq \mathsf{p}'(x) \text{ for } x > 0 \\
&\leq \frac{1}{2M}\frac{\mathrm{d}\rho_t^2}{\mathrm{d}t} \\
&\qquad \text{By Lemma C.4} \\
&\leq \frac{1}{M} \langle \mathbf{h}_{\mathsf{H}}, \boldsymbol{\theta}_t \rangle + 2\mathcal{L}_t \mathsf{p}_a(\rho_t). \qquad\qquad (49)
\end{aligned}
$$

The remaining part is to bound the term $\|\mathbf{h}\|^2 = \|\boldsymbol{\theta}_t'\|^2$ in the numerator. Recall that

$$\|\nabla f_i(\boldsymbol{\theta}_t) - \nabla f_{\mathsf{H},i}(\boldsymbol{\theta}_t)\| \leq \mathsf{r}(\|\boldsymbol{\theta}_t\|) \quad \text{for all } i \in [n].$$

Hence we have

$$\|\mathbf{h} - \mathbf{h}_{\mathsf{H}}\| \leq \frac{1}{n} \sum_{i=1}^{n} e^{-\bar{f}_i(\boldsymbol{\theta}_t)} \|\mathbf{h}_i - \mathbf{h}_{\mathsf{H},i}\| \leq \mathcal{L}_t \mathsf{r}(\|\boldsymbol{\theta}_t\|).$$

We get

$$\|\mathbf{h}\| \geq \|\mathbf{h}_{\mathsf{H}}\| - \|\mathbf{h} - \mathbf{h}_{\mathsf{H}}\| \geq \|\mathbf{h}_{\mathsf{H}}\| - \mathcal{L}_t \mathsf{r}(\|\boldsymbol{\theta}_t\|).$$

Since $\|\boldsymbol{\theta}_t\| \to \infty$, for all sufficiently large $t$, we have

$$
\begin{align}
\|\mathbf{h}_{\mathsf{H}}\| &\geq \frac{\langle \mathbf{h}_{\mathsf{H}}, \boldsymbol{\theta}_t \rangle}{\rho_t} \tag{50} \\
&= \frac{M \sum_{i=1}^{n} e^{-\bar{f}_i(\boldsymbol{\theta}_t)} \bar{f}_{\mathsf{H},i}(\boldsymbol{\theta}_t)}{n\rho_t} \\
&\geq \frac{M \sum_{i=1}^{n} e^{-\bar{f}_i(\boldsymbol{\theta}_t)} \big(\bar{f}_i(\boldsymbol{\theta}_t) - \mathsf{p}_a(\rho_t)\big)}{n\rho_t} \quad &&\text{by Proposition 5.1} \\
&\geq \frac{M\mathcal{L}_t \Big(\log \frac{1}{n\mathcal{L}_t} - \mathsf{p}_a(\rho_t)\Big)}{\rho_t} \quad &&\text{by Lemma C.2} \tag{51} \\
&= M\mathcal{L}_t \gamma^{\mathsf{GF}}(\boldsymbol{\theta}_t)\rho_t^{M-1} \quad &&\text{By the definition of } \gamma^{\mathsf{GF}} \\
&\geq M\mathcal{L}_t \gamma^{\mathsf{GF}}(\boldsymbol{\theta}_s)\rho_t^{M-1} \quad &&\text{by Theorem 3.2} \tag{52} \\
&\geq 3M\mathcal{L}_t \mathsf{r}(\rho). \quad &&\text{Since the degree of } r \text{ is } M-2
\end{align}
$$

Therefore, there exists $s_5 > s > 0$ such that for almost every $t \geq s_5$, we have

$$
\begin{align}
\|\mathbf{h}\|_2^2 &\geq \|\mathbf{h}_{\mathsf{H}}\|_2^2 - 2\|\mathbf{h}_{\mathsf{H}}\|\mathcal{L}_t \mathsf{r}(\|\boldsymbol{\theta}_t\|) + \mathcal{L}_t^2 \mathsf{r}(\|\boldsymbol{\theta}_t\|)^2 \\
&\geq \|\mathbf{h}_{\mathsf{H}}\|_2^2 - 2\|\mathbf{h}_{\mathsf{H}}\|\mathcal{L}_t \mathsf{r}(\rho_t) \tag{53} \\
&\geq \|\mathbf{h}_{\mathsf{H}}\|(3M-2)\mathcal{L}_t \mathsf{r}(\rho_t) \geq 0. \quad &&\text{by (52)}
\end{align}
$$

Moreover, we have

$$\langle \mathbf{h}, \boldsymbol{\theta}_t \rangle = \frac{1}{\rho_t} \frac{1}{2} \frac{\mathrm{d}\rho_t^2}{\mathrm{d}t} \leq \langle \mathbf{h}_{\mathsf{H}}, \tilde{\boldsymbol{\theta}}_t \rangle + 2M\mathcal{L}_t \mathsf{p}_a(\rho_t)/\rho_t. \quad \text{by (49)} \tag{54}$$

Combining (48), (49), (53) and (54), when $t \geq s_5$, we have

$$
\begin{align}
\frac{2\|\mathbf{h}\|^2 - 2\langle \mathbf{h}, \tilde{\boldsymbol{\theta}}_t \rangle^2}{\mathcal{L}_t a_t \cdot \frac{\mathrm{d}\rho_t^2}{\mathrm{d}t}/\rho_t^2} &\geq M \frac{\|\mathbf{h}_{\mathsf{H}}\|_2^2 - 2\|\mathbf{h}_{\mathsf{H}}\|\mathcal{L}_t \mathsf{r}(\rho_t) - \big(\langle \mathbf{h}_{\mathsf{H}}, \tilde{\boldsymbol{\theta}}_t \rangle + 2M\mathcal{L}_t \mathsf{p}_a(\rho_t)/\rho_t\big)^2}{\big(\langle \mathbf{h}_{\mathsf{H}}, \tilde{\boldsymbol{\theta}}_t \rangle + 2M\mathcal{L}_t \mathsf{p}_a(\rho_t)/\rho_t\big)^2} \\
&= M \cdot \left( \frac{1 - 2\|\mathbf{h}_{\mathsf{H}}\|^{-1}\mathcal{L}_t \mathsf{r}(\rho_t)}{\big(\beta(t) + 2M\mathcal{L}_t \mathsf{p}_a(\rho_t)/(\rho_t\|\mathbf{h}_{\mathsf{H}}\|)\big)^2} - 1 \right) \\
&\geq M \cdot \left( \frac{1 - 2\mathsf{r}(\rho_t)/\big(M\gamma^{\mathsf{GF}}(\boldsymbol{\theta}_s)\rho_t^{M-1}\big)}{\big(\beta(t) + 2\mathsf{p}_a(\rho_t)/\big(\gamma^{\mathsf{GF}}(\boldsymbol{\theta}_s)\rho_t^M\big)\big)^2} - 1 \right) \\
&= M \cdot \left( \frac{1 - p_1(t)}{\big(\beta(t) + p_2(t)\big)^2} - 1 \right),
\end{align}
$$

where

$$p_1(t) := \frac{2\mathsf{r}(\rho_t)}{M\gamma^{\mathsf{GF}}(\boldsymbol{\theta}_s)\rho_t^{M-1}}, \quad p_2(t) := \frac{2\mathsf{p}_a(\rho_t)}{\gamma^{\mathsf{GF}}(\boldsymbol{\theta}_s)\rho_t^M}.$$

This completes the proof of Lemma C.25. $\qquad\square$

**Corollary C.26** ($\beta$ bound). *Under Assumptions 1 and 2, there exists $s_5 > s > 0$ such that for almost every $t_2 > t_1 \geq s_5$, there exists $t_* \in (t_1, t_2)$ satisfying*

$$\frac{1 - p_1(t_*)}{\big(\beta(t_*) + p_2(t_*)\big)^2} - 1 \leq \frac{1}{M} \cdot \frac{\log \gamma^{\mathsf{GF}}(\boldsymbol{\theta}_{t_2}) - \log \gamma^{\mathsf{GF}}(\boldsymbol{\theta}_{t_1})}{\log \rho_{t_2} - \log \rho_{t_1}}.$$

*Proof of Corollary C.26.* Denote the RHS as $C$. Assume to the contrary that

$$\frac{1 - p_1(t_*)}{\big(\beta(t_*) + p_2(t_*)\big)^2} - 1 > C \quad \text{for all } t_* \in (t_1, t_2).$$

By Lemma C.25, we have

$$\frac{1}{M} \log \frac{\gamma^{\mathsf{GF}}(\boldsymbol{\theta}_{t_2})}{\gamma^{\mathsf{GF}}(\boldsymbol{\theta}_{t_1})} > \int_{t_1}^{t_2} C \cdot \frac{\mathrm{d}}{\mathrm{d}\tau} \log \rho(\tau) \cdot \mathrm{d}\tau = C(\log \rho_{t_2} - \log \rho_{t_1}) = \frac{1}{M} \log \frac{\gamma^{\mathsf{GF}}(\boldsymbol{\theta}_{t_2})}{\gamma^{\mathsf{GF}}(\boldsymbol{\theta}_{t_1})},$$

which leads to a contradiction. This completes the proof of Corollary C.26. $\qquad \square$

**Lemma C.27** ($\beta$ converges to 1). *Under Assumptions 1 and 2 and $\nabla f$ is near-$(M-1)$-homogeneous, there exists a sequence $t_k$ such that $\lim_{k \to \infty} \beta_{t_k} \to 1$.*

*Proof of Lemma C.27.* By Corollary C.26, there exists $s_5 > s > 0$ such that for almost every $t_2 > t_1 \geq s_5$, there exists $t_* \in (t_1, t_2)$ satisfying

$$\beta(t_*) \geq \sqrt{\frac{1 - p_1(t_*)}{1 + \frac{1}{M} \cdot \frac{\log \gamma^{\mathsf{GF}}(\boldsymbol{\theta}_{t_2}) - \log \gamma^{\mathsf{GF}}(\boldsymbol{\theta}_{t_1})}{\log \rho_{t_2} - \log \rho_{t_1}}}} - p_2(t_*).$$

We know that as $t_* \to \infty$, $p_1(t_*), p_2(t_*) \to 0$. Besides, we know that $\log \rho_t \to \infty$ and $\log \gamma^{\mathsf{GF}}(\boldsymbol{\theta}_t) \to \gamma_*$. For any $\epsilon_m$, there exists $t_2 > t_2 > s_5$ such that

$$p_1(t_*) \leq \epsilon_m/2, \log(\rho(t_2) - \log(\rho(t_1)) \geq \frac{1}{M}, \log \gamma^{\mathsf{GF}}(\boldsymbol{\theta}_{t_2}) - \log \gamma^{\mathsf{GF}}(\boldsymbol{\theta}_{t_1}) \leq \epsilon_m/2, \text{ and } p_2(t_*) \leq \epsilon_m/2.$$

Hence we have

$$\beta(t_*) \geq \sqrt{\frac{1 - \epsilon_m/2}{1 + \epsilon_m/2}} - \epsilon_m/2 \geq 1 - \epsilon_m.$$

Hence, for each $\epsilon_m$ we can find a corresponding $t_m$ such that $\beta(t_m) \geq 1 - \epsilon_m$. This completes the proof of Lemma C.27. $\quad \square$

It's worthy noting that Lemma C.27 shows that a subsequence of $(\mathbf{h}_{\mathsf{H}}(\boldsymbol{\theta}_t), \boldsymbol{\theta}_t)$ aligns, which matches the result in Lyu & Li (2020). However, Ji & Telgarsky (2020) showed that $(\mathbf{h}_{\mathsf{H}}(\boldsymbol{\theta}_t), \boldsymbol{\theta}_t)$ aligns assuming that $f$ is homogeneous. We leave investigating gradient alignment for near-homogeneous functions as a future direction.

**Lemma C.28** (approximate KKT point). *Under Assumptions 1 and 2 and $\nabla f$ is $(M-1)$ near-homogeneous, there exists a sequence $t_k$ such that $\hat{\boldsymbol{\theta}}_{t_k}$ is an $(\epsilon_k, \delta_k)$-KKT point of (P) for all $k \in \mathbb{N}$, where $\epsilon_k \to 0$ and $\delta_k \to 0$ as $k \to \infty$.*

*Proof of Lemma C.28.* We just apply Lemma C.23, Lemma C.24, and Lemma C.27. This completes the proof of Lemma C.28. $\qquad \square$

Now we can prove the main theorem.

## C.9. Proof of Theorem 3.5

*Proof of Theorem 3.5.* Appplying Lemma C.28, we have a sequence $\{t_k\}$ such that $\hat{\boldsymbol{\theta}}_{t_k}$ is an $(\epsilon_k, \delta_k)$-KKT point of (P) for all $k \in \mathbb{N}$, where $\epsilon_k \to 0$ and $\delta_k \to 0$ as $k \to \infty$. Since $\tilde{\boldsymbol{\theta}}_t$ and $\tilde{\boldsymbol{\theta}}$ share the same direction, $\hat{\boldsymbol{\theta}}_{t_k}$ will converge to the same direction as one of the KKT points of (P). By Theorem 3.4, we know $\tilde{\boldsymbol{\theta}}_{t_k}$ also converges to the limit $\boldsymbol{\theta}_*$. Hence, $\boldsymbol{\theta}_* / \big(\bar{f}_{\min}(\boldsymbol{\theta}_*)\big)^{1/M}$ is a KKT point of (P). This completes the proof of Theorem 3.5. $\quad \square$

# D. Proofs for Section 4

The proofs in this section are divided into two parts. In the first part, we provide some basic results about near-homogeneous block functions. In the second part, we verify that many commonly used neural network architectures are near-homogeneous block functions.

## D.1. Near-Homogeneous Networks

The first thing we want to show in this section is that if we want to verify a model (block map) is near-homogeneous (dual-homogeneous), we only need to verify the first two conditions. This is because the frist condition will imply the the last condition.

**Lemma D.1** (M-near homogeneity bound). *Given a locally Lipschitz function $s(\boldsymbol{\theta})$, $M \in \mathbb{Z}_+$ and $p_s(x) \in \mathbb{R}_+[x]$ with $\deg p_s = M$, we assume that for all $\boldsymbol{\theta}$,*

$$\|\langle \nabla s(\boldsymbol{\theta}), \boldsymbol{\theta} \rangle - M \cdot s(\boldsymbol{\theta}) \| \leq p_s'(\|\boldsymbol{\theta}\|).$$

*Then, there exists $p_s^+ \in \mathbb{R}_+[x]$ such that $\deg p_s^+ = M$ and*

$$\|s(\boldsymbol{\theta})\| \leq p_s^+(\|\boldsymbol{\theta}\|).$$

We skip the proof here as it is almost the same as the proof of Lemma C.8. Indeed, note that in the proof of Lemma C.8 we only use Assumption (A1). However, this lemma indicates that the near homogeneity will lead to the boundedness of the block function. In other words, we have

$$(A1) \implies (A3), \quad (B1) \implies (B3).$$

In detail, we have the following corollary.

**Corollary D.2** (From near homogeneity to boundedness). *Given a model $f(\boldsymbol{\theta}; \mathbf{x})$ satisfying (A1) and (A2) in Definition 1, then $f(\boldsymbol{\theta}; \mathbf{x})$ satisfies (A1), (A2) and (A3) in Definition 1. Similarly, given a block function $s(\boldsymbol{\theta}; \mathbf{x})$ satisfying (B1) and (B2) in Definition 2, then it satisfies (B1), (B2) and (B3) in Definition 2.*

Once we have this corollary, we must only verify the first two conditions in the dual homogeneity definition. The first result we want to verify is that near-homogeneous gradients lead to near-homogeneous functions.

**Lemma D.3** (Near-homogeneous gradient leads to near-homogeneous function). *Assume that $f(\boldsymbol{\theta}; \mathbf{x})$ is continuously differentiable and $\nabla f$ is locally Lipschitz and satisfies Assumption 3 with parameter $M - 1$ for some $M \geq 2$. Then, $f$ is near-M-homogeneous.*

*Proof.* Since $\nabla f$ is locally Lipschitz, $\nabla^2 f$ exists almost everywhere. Here, we focus on showing (A1) in Assumption 1 since (A2) is straightforward. For simplicity, we omit the $\mathbf{x}$ in the function. Note (A1) is to show there exists $p$ s.t. $\deg p \leq M$ and

$$|\langle \nabla f(\boldsymbol{\theta}), \boldsymbol{\theta} \rangle - M f(\boldsymbol{\theta}) | \leq p'(\|\boldsymbol{\theta}\|).$$

Here we assume there is a $p_1 \in \mathbb{R}_+[x]$ such that $\deg p_1 \leq M - 1$ and

$$|\langle \nabla [\nabla f(\boldsymbol{\theta})]_i, \boldsymbol{\theta} \rangle - (M - 1)[\nabla f(\boldsymbol{\theta})]_i | \leq p_1'(\|\boldsymbol{\theta}\|).$$

Since $f$ is a continuous function, we can assume for all $\boldsymbol{\theta} \in \overline{B(0,1)}$,

$$|\langle \nabla f(\boldsymbol{\theta}), \boldsymbol{\theta} \rangle - M f(\boldsymbol{\theta})| \leq C.$$

For general $\boldsymbol{\theta}$, let $\tilde{\boldsymbol{\theta}} := \boldsymbol{\theta}/\|\boldsymbol{\theta}\|$ and

$$g(r) := \langle \nabla f(r\tilde{\boldsymbol{\theta}}), r\tilde{\boldsymbol{\theta}} \rangle - M f(r\tilde{\boldsymbol{\theta}}).$$

We aim to show that $|g(r) - g(1)|$ is bounded by some polynomial in $r$. Note that

$$|g'(r)| = \left| \langle \tilde{\boldsymbol{\theta}}^\top \nabla^2 f(r\tilde{\boldsymbol{\theta}}), r\tilde{\boldsymbol{\theta}} \rangle + \nabla f(r\tilde{\boldsymbol{\theta}})^\top \tilde{\boldsymbol{\theta}} - M \nabla f(r\tilde{\boldsymbol{\theta}})^\top \tilde{\boldsymbol{\theta}} \right|$$

$$
\begin{aligned}
&= \left| \langle \tilde{\boldsymbol{\theta}}^\top \nabla^2 f(r\tilde{\boldsymbol{\theta}}), r\tilde{\boldsymbol{\theta}} \rangle - (M-1)\nabla f(r\tilde{\boldsymbol{\theta}})^\top \tilde{\boldsymbol{\theta}} \right| \\
&= \left| \tilde{\boldsymbol{\theta}}^\top \left[ \langle \tilde{\nabla}^2 f(r\tilde{\boldsymbol{\theta}}), r\tilde{\boldsymbol{\theta}} \rangle - (M-1)\nabla f(r\tilde{\boldsymbol{\theta}}) \right] \right| \\
&\leq \left\| \langle \tilde{\nabla}^2 f(r\tilde{\boldsymbol{\theta}}), r\tilde{\boldsymbol{\theta}} \rangle - (M-1)\nabla f(r\tilde{\boldsymbol{\theta}}) \right\| \\
&\leq \mathsf{p}_1'(r).
\end{aligned}
$$

Hence, we have

$$
g(r) \leq |g(1)| + |g(r) - g(1)| \leq C + \int_1^r \mathsf{p}_1'(s)\,\mathrm{d}s \leq \mathsf{p}_1(r) + C - \mathsf{p}_1(1) \leq \mathsf{p}_1(r) + C.
$$

Therefore, we can set $\mathsf{p} = \int(\mathsf{p}_1 + C)$. This verifies (A1). Hence, $f$ is near-M-homogeneous. This completes the proof. $\qquad\square$

Now we can prove Lemma 4.2.

### D.2. Proof of Proposition 4.2

*Proof of Proposition 4.2.* For this proposition, we will only present proofs for argument 1 and 2. For argument 3, 4 and 5, they are direct consequences of argument 1 and 2.

**Argument 1 for $s(\boldsymbol{\theta}, \mathbf{x}) = s^1_{\boldsymbol{\theta}_1} \circ s^2_{\boldsymbol{\theta}_2}(\mathbf{x})$.**

By Corollary A.5, we know the existence of the Clarke Jacobian of $s^1(\boldsymbol{\theta}_1; s^2(\boldsymbol{\theta}_2; \mathbf{x}))$, since it's locally Lipschitz. Right now, for $s(\boldsymbol{\theta}; \mathbf{x}) = s^1(\boldsymbol{\theta}_1; s^2(\boldsymbol{\theta}_2; \mathbf{x}))$, we want to verify $(B1)$ and $(B2)$. Since they are some convex constraints, we only need to verify the constraints at the boundary.

To prove this, we need to invoke Corollary A.9. Recall that the statement in Corollary A.9 is:

---

Given two locally Lipschitz block functions $s^1(\boldsymbol{\theta}_1; \mathbf{x}) : \mathbb{R}^{d_1} \times \mathbb{R}^{d_2} \to \mathbb{R}^{d_3}$ and $s^2(\boldsymbol{\theta}_2; \mathbf{x}) : \mathbb{R}^{d_3} \times \mathbb{R}^{d_4} \to \mathbb{R}^{d_2}$, we have

$$
\begin{aligned}
\partial_{\boldsymbol{\theta}} s^1(\boldsymbol{\theta}_1, s^2(\boldsymbol{\theta}_2; \mathbf{x})) &\subset \mathrm{conv}\big\{ (\boldsymbol{\alpha}_1, \boldsymbol{\alpha}_2 \cdot \mathbf{h}_1) : (\boldsymbol{\alpha}_1, \boldsymbol{\alpha}_2) \in \partial_{\boldsymbol{\theta}_1, \mathbf{x}} s^1(\boldsymbol{\theta}_1; \mathbf{x}), \mathbf{h}_1 \in \partial_{\boldsymbol{\theta}_2} s^2(\boldsymbol{\theta}_2; \mathbf{x}) \big\}, \\
\partial_{\mathbf{x}} s^1(\boldsymbol{\theta}_1, s^2(\boldsymbol{\theta}_2; \mathbf{x})) &\subset \mathrm{conv}\big\{ \boldsymbol{\alpha}_2 \cdot \mathbf{h}_2 : \boldsymbol{\alpha}_2 \in \partial_{\mathbf{x}} s^1(\boldsymbol{\theta}_1; \mathbf{x}), \mathbf{h}_2 \in \partial_{\mathbf{x}} s^2(\boldsymbol{\theta}_2; \mathbf{x}) \big\}.
\end{aligned}
$$

---

This indicates if suffices to verify $(B1)$ and $(B2)$ for all combinations of $(\boldsymbol{\alpha}_1, \boldsymbol{\alpha}_2, \mathbf{h}_1, \mathbf{h}_2)$, where $(\boldsymbol{\alpha}_1, \boldsymbol{\alpha}_2) \in \partial_{\boldsymbol{\theta}_1, \mathbf{x}} s^1(\boldsymbol{\theta}_1; \mathbf{x})$ and $(\mathbf{h}_1, \mathbf{h}_2) \in \partial_{\boldsymbol{\theta}_2, \mathbf{x}} s^2(\boldsymbol{\theta}_2; \mathbf{x})$.

Recall that we already know $(\boldsymbol{\alpha}_1, \boldsymbol{\alpha}_2, \mathbf{h}_1, \mathbf{h}_2)$ satisfy the following conditions:

$$
\begin{aligned}
\|\langle \boldsymbol{\alpha}_1, \boldsymbol{\theta} \rangle - M_1 \cdot s^1(\boldsymbol{\theta}; \mathbf{x})\| &\leq \mathsf{p}'_{s^1}(\|\boldsymbol{\theta}\|) \mathsf{r}_{s^1}(\|\mathbf{x}\|), \\
\|\langle \boldsymbol{\alpha}_2, \mathbf{x} \rangle - M_2 \cdot s^1(\boldsymbol{\theta}; \mathbf{x})\| &\leq \mathsf{p}_{s^1}(\|\boldsymbol{\theta}\|) \mathsf{r}'_{s^1}(\|\mathbf{x}\|), \\
\|\boldsymbol{\alpha}_1\| &\leq \mathsf{q}'_{s^1}(\|\boldsymbol{\theta}\|) \mathsf{t}_{s^1}(\|\mathbf{x}\|), \\
\|\boldsymbol{\alpha}_2\| &\leq \mathsf{q}_{s^1}(\|\boldsymbol{\theta}\|) \mathsf{t}'_{s^1}(\|\mathbf{x}\|), \\
\|s^1(\boldsymbol{\theta}; \mathbf{x})\| &\leq \mathsf{q}_{s^1}(\|\boldsymbol{\theta}\|) \mathsf{t}_{s^1}(\|\mathbf{x}\|).
\end{aligned}
$$

For $s^2$, we have

$$
\begin{aligned}
\|\langle \mathbf{h}_1, \boldsymbol{\theta} \rangle - M_3 \cdot s^2(\boldsymbol{\theta}; \mathbf{x})\| &\leq \mathsf{p}'_{s^2}(\|\boldsymbol{\theta}\|) \mathsf{r}_{s^2}(\|\mathbf{x}\|), \\
\|\langle \mathbf{h}_2, \mathbf{x} \rangle - M_4 \cdot s^2(\boldsymbol{\theta}; \mathbf{x})\| &\leq \mathsf{p}_{s^2}(\|\boldsymbol{\theta}\|) \mathsf{r}'_{s^2}(\|\mathbf{x}\|), \\
\|\mathbf{h}_1\| &\leq \mathsf{q}'_{s^2}(\|\boldsymbol{\theta}\|) \mathsf{t}_{s^2}(\|\mathbf{x}\|), \\
\|\mathbf{h}_2\| &\leq \mathsf{q}_{s^2}(\|\boldsymbol{\theta}\|) \mathsf{t}'_{s^2}(\|\mathbf{x}\|), \\
\|s^2(\boldsymbol{\theta}; \mathbf{x})\| &\leq \mathsf{q}_{s^2}(\|\boldsymbol{\theta}\|) \mathsf{t}_{s^2}(\|\mathbf{x}\|).
\end{aligned}
$$

Then, we will verify the two conditions (B1) and (B2).

Verify (B1) in Definition 2.

As for (B1), we have

$$
\langle \boldsymbol{\alpha}_1, \boldsymbol{\theta}_1 \rangle + \langle \boldsymbol{\alpha}_2 \cdot \mathbf{h}_1, \boldsymbol{\theta}_2 \rangle \\
= \langle \boldsymbol{\alpha}_1, \boldsymbol{\theta}_1 \rangle + \langle \boldsymbol{\alpha}_2, M_3 s_{\boldsymbol{\theta}_2}^2(\mathbf{x}) \rangle + \langle \boldsymbol{\alpha}_2, \langle \mathbf{h}_1, \boldsymbol{\theta}_2 \rangle - M_3 s_{\boldsymbol{\theta}_2}^2(\mathbf{x}) \rangle.
$$

Therefore,

$$
\sup_{\mathbf{g}_1 \in \partial_{\boldsymbol{\theta}} s(\boldsymbol{\theta};\mathbf{x})} \| \langle \mathbf{g}_1, \boldsymbol{\theta} \rangle - (M_1 + M_2 M_3) s(\boldsymbol{\theta};\mathbf{x}) \|
$$
$$
\leq \sup_{(\boldsymbol{\alpha}_1,\boldsymbol{\alpha}_2) \in \partial s^1, (\mathbf{h}_1,\mathbf{h}_2) \in \partial s^2} \| \langle \boldsymbol{\alpha}_1, \boldsymbol{\theta}_1 \rangle - M_1 s(\boldsymbol{\theta};\mathbf{x}) \| + M_3 \| \langle \boldsymbol{\alpha}_2, s_{\boldsymbol{\theta}_2}^2(\mathbf{x}) \rangle - M_2 s(\boldsymbol{\theta};\mathbf{x}) \|
$$
$$
+ \| \langle \boldsymbol{\alpha}_2, \langle \mathbf{h}_1, \boldsymbol{\theta}_2 \rangle - M_3 s_{\boldsymbol{\theta}_2}^2(\mathbf{x}) \rangle \|
$$
$$
\leq \mathsf{p}'_{s^1}(\|\boldsymbol{\theta}_1\|) \mathsf{r}_{s^1}(\|s_{\boldsymbol{\theta}_2}^2(\mathbf{x})\|) + M_3 \mathsf{p}_{s^1}(\|\boldsymbol{\theta}_1\|) \mathsf{r}'_{s^1}(\|s_{\boldsymbol{\theta}_2}^2(\mathbf{x})\|) + \mathsf{p}_{s^1}(\|\boldsymbol{\theta}_1\|) \mathsf{r}'_{s^1}(\|s_{\boldsymbol{\theta}_2}^2(\mathbf{x})\|) \mathsf{p}'_{s^2}(\|\boldsymbol{\theta}_2\|) \mathsf{r}_{s^2}(\|\mathbf{x}\|)
$$
$$
\leq \mathsf{p}'_{s^1}(\|\boldsymbol{\theta}_1\|) \cdot \mathsf{r}_{s^1} \circ \mathsf{q}_{s^2}(\|\boldsymbol{\theta}_2\|) \cdot \mathsf{r}_{s^1} \circ \mathsf{t}_{s^2}(\|\mathbf{x}\|) + M_3 \mathsf{p}_{s^1}(\|\boldsymbol{\theta}_1\|) \cdot \mathsf{r}'_{s^1} \circ \mathsf{q}_{s^2}(\|\boldsymbol{\theta}_2\|) \cdot \mathsf{r}'_{s^1} \circ \mathsf{t}_{s^2}(\|\mathbf{x}\|)
$$
$$
+ \mathsf{p}_{s^1}(\|\boldsymbol{\theta}_1\|) \mathsf{r}'_{s^1} \circ \mathsf{q}_{s^2}(\|\boldsymbol{\theta}_2\|) \cdot \mathsf{r}'_{s^1} \circ \mathsf{t}_{s^2}(\|\mathbf{x}\|) \cdot \mathsf{p}'_{s^2}(\|\boldsymbol{\theta}_2\|) \cdot \mathsf{r}_{s^2}(\|\mathbf{x}\|)
$$
$$
\leq \underbrace{\left[ \mathsf{p}'_{s^1}(\|\boldsymbol{\theta}_1\|) \mathsf{r}_{s^1} \circ \mathsf{q}_{s^2}(\|\boldsymbol{\theta}_2\|) + M_3 \mathsf{p}_{s^1}(\|\boldsymbol{\theta}_1\|) \mathsf{r}'_{s^1} \circ \mathsf{q}_{s^2}(\|\boldsymbol{\theta}_2\|) + \mathsf{p}_{s^1}(\|\boldsymbol{\theta}_1\|) \cdot \mathsf{r}'_{s^1} \circ \mathsf{q}_{s^2}(\|\boldsymbol{\theta}_2\|) \mathsf{p}'_{s^2}(\|\boldsymbol{\theta}_2\|) \right]}_{\text{of order } \leq M_1 + M_2 M_3 - 1}
$$
$$
\times \underbrace{\left[ \mathsf{r}_{s^1} \circ \mathsf{t}_{s^2}(\|\mathbf{x}\|) + M_3 \mathsf{r}'_{s^1} \circ \mathsf{t}_{s^2}(\|\mathbf{x}\|) + \mathsf{r}'_{s^1} \circ \mathsf{t}_{s^2}(\|\mathbf{x}\|) \cdot \mathsf{r}_{s^2}(\|\mathbf{x}\|) \right]}_{\text{of order } \leq M_2 M_4}.
$$

We can prove something similar for $\langle \mathbf{g}_2, \mathbf{x} \rangle - M_2 M_4 s(\boldsymbol{\theta};\mathbf{x})$, where $\mathbf{g}_2 \in \partial_{\mathbf{x}} s(\boldsymbol{\theta};\mathbf{x})$. Then the orders for $\langle \mathbf{g}_2, \mathbf{x} \rangle - M_2 M_4 s(\boldsymbol{\theta};\mathbf{x})$ will be bound by $(M_1 + M_2 M_3, M_2 M_4 - 1)$. Combining them, there exist $\mathsf{p}_s, \mathsf{r}_s \in \mathbb{R}_+[x]$ and $\deg \mathsf{p}_s \leq M_1 + M_2 M_3, \deg \mathsf{r}_s \leq M_2 M_4$ such that

$$
\sup_{\mathbf{g}_1 \in \partial_{\boldsymbol{\theta}} s(\boldsymbol{\theta};\mathbf{x})} \| \langle \mathbf{g}_1, \boldsymbol{\theta} \rangle - (M_1 + M_2 M_3) s(\boldsymbol{\theta};\mathbf{x}) \| \leq \mathsf{p}'_s(\|\boldsymbol{\theta}\|) \mathsf{r}_s(\|\mathbf{x}\|),
$$
$$
\sup_{\mathbf{g}_2 \in \partial_{\mathbf{x}} s(\boldsymbol{\theta};\mathbf{x})} \| \langle \mathbf{g}_2, \mathbf{x} \rangle - (M_2 M_4) s(\boldsymbol{\theta};\mathbf{x}) \| \leq \mathsf{p}_s(\|\boldsymbol{\theta}\|) \mathsf{r}'_s(\|\mathbf{x}\|).
$$

Verify (B2) in Definition 2.

Before we proceed, we have the following two inequalities:

$$
\mathsf{t}_{s^1}(\|s_{\boldsymbol{\theta}_2}^2(\mathbf{x})\|) \leq \mathsf{t}_{s^1}\left( \mathsf{q}_{s^2}(\|\boldsymbol{\theta}_2\|) \mathsf{t}_{s^2}(\|\mathbf{x}\|) \right) \leq [\mathsf{t}_{s^1} \circ \mathsf{q}_{s^2}(\|\boldsymbol{\theta}_2\|)] \cdot [\mathsf{t}_{s^1} \circ \mathsf{t}_{s^2}(\|\mathbf{x}\|)],
$$
$$
\mathsf{t}'_{s^1}(\|s_{\boldsymbol{\theta}_2}^2(\mathbf{x})\|) \leq \mathsf{t}'_{s^1}\left( \mathsf{q}_{s^2}(\|\boldsymbol{\theta}_2\|) \mathsf{t}_{s^2}(\|\mathbf{x}\|) \right) \leq [\mathsf{t}'_{s^1} \circ \mathsf{q}_{s^2}(\|\boldsymbol{\theta}_2\|)] \cdot [\mathsf{t}'_{s^1} \circ \mathsf{t}_{s^2}(\|\mathbf{x}\|)].
$$

Note that

$$
\sup_{\mathbf{g}_1 \in \partial_{\boldsymbol{\theta}} s} \|\mathbf{g}_1\|^2 \leq \sup_{(\boldsymbol{\alpha}_1,\boldsymbol{\alpha}_2) \in \partial s^1, (\mathbf{h}_1,\mathbf{h}_2) \in \partial s^2} \|\boldsymbol{\alpha}_1\|^2 + \|\boldsymbol{\alpha}_2 \cdot \mathbf{h}_1\|^2
$$
$$
\leq \left[ \mathsf{q}'_{s^1}(\|\boldsymbol{\theta}_1\|) \mathsf{t}_{s^1}(\|s_{\boldsymbol{\theta}_2}^2(\mathbf{x})\|) \right]^2 + \left[ \mathsf{q}_{s^1}(\|\boldsymbol{\theta}_1\|) \mathsf{t}'_{s^1}(\|s_{\boldsymbol{\theta}_2}^2(\mathbf{x})\|) \mathsf{q}'_{s^2}(\|\boldsymbol{\theta}_2\|) \mathsf{t}_{s^2}(\|\mathbf{x}\|) \right]^2
$$
By (B2) and Lemma D.1
$$
\leq \left[ \mathsf{q}'_{s^1}(\|\boldsymbol{\theta}_1\|) \cdot \mathsf{t}_{s^1} \circ \mathsf{q}_{s^2}(\|\boldsymbol{\theta}_2\|) \cdot \mathsf{t}_{s^1} \circ \mathsf{t}_{s^2}(\|\mathbf{x}\|) \right]^2
$$
$$
+ \left[ \mathsf{q}_{s^1}(\|\boldsymbol{\theta}_1\|) \cdot \mathsf{t}'_{s^1} \circ \mathsf{q}_{s^2}(\|\boldsymbol{\theta}_2\|) \cdot \mathsf{t}'_{s^1} \circ \mathsf{t}_{s^2}(\|\mathbf{x}\|) \cdot \mathsf{q}'_{s^2}(\|\boldsymbol{\theta}_2\|) \cdot \mathsf{t}_{s^2}(\|\mathbf{x}\|) \right]^2
$$
By two inequalities above

$$\leq \Big[ \underbrace{\mathsf{q}'_{s^1}(\|\boldsymbol{\theta}_1\|) \cdot \mathsf{t}_{s^1} \circ \mathsf{q}_{s^2}(\|\boldsymbol{\theta}_2\|) + \mathsf{q}_{s^1}(\|\boldsymbol{\theta}_1\|) \cdot \mathsf{t}'_{s^1} \circ \mathsf{q}_{s^2}(\|\boldsymbol{\theta}_2\|) \cdot \mathsf{q}'_{s^2}(\|\boldsymbol{\theta}_2\|)}_{\text{of order } \leq M_1 + M_2 M_3 - 1} \Big]^2$$

$$\times \Big[ \underbrace{\mathsf{t}_{s^1} \circ \mathsf{t}_{s^2}(\|\mathbf{x}\|) + \mathsf{t}'_{s^1} \circ \mathsf{t}_{s^2}(\|\mathbf{x}\|) \cdot \mathsf{t}_{s^2}(\|\mathbf{x}\|)}_{\text{of order } \leq M_2 M_4} \Big]^2.$$

Therefore, there exist $\mathsf{q}_s, \mathsf{t}_s \in \mathbb{R}_+[x]$ and $\deg \mathsf{q}_s = M_1 + M_2 M_3, \deg \mathsf{t}_s = M_2 M_4$ such that

$$\sup_{\mathbf{g}_1 \in \partial_{\boldsymbol{\theta}} s} \|\mathbf{g}_1\|^2 \leq \mathsf{q}'_s(\|\boldsymbol{\theta}\|) \mathsf{t}_s(\|\mathbf{x}\|).$$

On the other hand,

$$\sup_{\mathbf{g}_2 \in \partial_{\mathbf{x}} s} \|\mathbf{g}_2\| \leq \sup_{(\boldsymbol{\alpha}_1, \boldsymbol{\alpha}_2) \in \partial s^1, (\mathbf{h}_1, \mathbf{h}_2) \in \partial s^2} \|\boldsymbol{\alpha}_2 \cdot \mathbf{h}_2\|$$

$$\leq \mathsf{q}_{s^1}(\|\boldsymbol{\theta}_1\|) \mathsf{t}'_{s^1}(\|s^2_{\boldsymbol{\theta}_2}(\mathbf{x})\|) \mathsf{q}_{s^2}(\|\boldsymbol{\theta}_2\|) \mathsf{t}'_{s^2}(\|\mathbf{x}\|)$$

$$\leq \mathsf{q}_{s^1}(\|\boldsymbol{\theta}_1\|) \cdot \mathsf{t}'_{s^1} \circ \mathsf{q}_{s^2}(\|\boldsymbol{\theta}_2\|) \cdot \mathsf{t}'_{s^1} \circ \mathsf{t}_{s^2}(\|\mathbf{x}\|) \cdot \mathsf{q}_{s^2}(\|\boldsymbol{\theta}_2\|) \cdot \mathsf{t}'_{s^2}(\|\mathbf{x}\|)$$

$$= \underbrace{\big[ \mathsf{q}_{s^1}(\|\boldsymbol{\theta}_1\|) \cdot \mathsf{t}'_{s^1} \circ \mathsf{q}_{s^2}(\|\boldsymbol{\theta}_2\|) \cdot \mathsf{q}_{s^2}(\|\boldsymbol{\theta}_2\|) \big]}_{\text{of order } \leq M_1 + M_2 M_3} \times \underbrace{\big[ \mathsf{t}'_{s^1} \circ \mathsf{t}_{s^2}(\|\mathbf{x}\|) \cdot \mathsf{t}'_{s^2}(\|\mathbf{x}\|) \big]}_{\text{of order } \leq M_2 M_4 - 1}.$$

Therefore, there exist $\mathsf{q}_s, \mathsf{t}_s \in \mathbb{R}_+[x]$ and $\deg \mathsf{q}_s \leq M_1 + M_2 M_3, \deg \mathsf{t}_s \leq M_2 M_4$ such that

$$\sup_{\mathbf{g}_1 \in \partial_{\boldsymbol{\theta}} s} \|\mathbf{g}_1\| \leq \mathsf{q}'_s(\|\boldsymbol{\theta}\|) \mathsf{t}_s(\|\mathbf{x}\|),$$

$$\sup_{\mathbf{g}_2 \in \partial_{\mathbf{x}} s} \|\mathbf{g}_2\| \leq \mathsf{q}_s(\|\boldsymbol{\theta}\|) \mathsf{t}'_s(\|\mathbf{x}\|).$$

We have shown that $s$ satisfies (B1) and (B2) with parameter $(M_1 + M_2 M_3, M_2 M_4)$. Since (B1) and (B2) lead to (B3), This indicates that $s(\boldsymbol{\theta}; \mathbf{x})$ is near-$(M_1 + M_2 M_3, M_2 M_4)$-homogeneous.

**Argument 2 for** $s(\boldsymbol{\theta}, \mathbf{x}) = s^1_{\boldsymbol{\theta}_1} \otimes s^2_{\boldsymbol{\theta}_2}(\mathbf{x})$**.** Without loss of generality, we can focus on just one entry of the output like:

$$\big[ s(\boldsymbol{\theta}, \mathbf{x}) \big]_{i,j} = \big[ s^1_{\boldsymbol{\theta}_1}(\mathbf{x}) \big]_i \cdot \big[ s^2_{\boldsymbol{\theta}_2}(\mathbf{x}) \big]_j.$$

To simplify the notation, we can assume $s^1, s^2$ are both scalar functions. Then, by Lemma A.6, we have

$$\partial(s^1 \cdot s^2) \subseteq \text{conv}\{(s^2 \boldsymbol{\alpha}_1, s^1 \mathbf{h}_1, s^2 \boldsymbol{\alpha}_2 + s^1 \mathbf{h}_2) : (\boldsymbol{\alpha}_1, \boldsymbol{\alpha}_2) \in \partial s^1, (\mathbf{h}_1, \mathbf{h}_2) \in \partial s^2\}.$$

We will also use $(\mathbf{g}_1, \mathbf{g}_2)$ to denote the Jacobian of $s$. Then, we can verify $(B1)$ and $(B2)$.

Verify (B1) in Definition 2. As for $\mathbf{g}_1 \in \partial_{\boldsymbol{\theta}} s$, we have

$$\sup_{\mathbf{g}_1 \in \partial_{\boldsymbol{\theta}} s} \big\| \langle \mathbf{g}_1, \boldsymbol{\theta} \rangle - (M_1 + M_3) s \big\|$$

$$\leq \sup_{\boldsymbol{\alpha}_1 \in \partial_{\boldsymbol{\theta}} s^1, \mathbf{h}_1 \in \partial_{\boldsymbol{\theta}} s^2} \big\| \langle s^2 \boldsymbol{\alpha}_1, \boldsymbol{\theta}_1 \rangle + \langle s^1 \mathbf{h}_1, \boldsymbol{\theta}_2 \rangle - (M_1 + M_3) s \big\|$$

$$\leq \sup_{\boldsymbol{\alpha}_1 \in \partial_{\boldsymbol{\theta}} s^1} \|s^2\| \cdot \|\langle \boldsymbol{\alpha}_1, \boldsymbol{\theta}_1 \rangle - M_1 s^1\| + \sup_{\mathbf{h}_1 \in \partial_{\boldsymbol{\theta}} s^2} \|s^1\| \cdot \|\langle \mathbf{h}_1, \boldsymbol{\theta}_2 \rangle - M_3 s^2\|$$

$$\leq \mathsf{q}_{s^2}(\|\boldsymbol{\theta}\|) \mathsf{t}_{s^2}(\|\mathbf{x}\|) \mathsf{p}'_{s^1}(\|\boldsymbol{\theta}\|) \mathsf{r}_{s^1}(\|\mathbf{x}\|) + \mathsf{q}_{s^1}(\|\boldsymbol{\theta}\|) \mathsf{t}_{s^1}(\|\mathbf{x}\|) \mathsf{p}'_{s^2}(\|\boldsymbol{\theta}\|) \mathsf{r}_{s^2}(\|\mathbf{x}\|)$$

$$= \underbrace{\mathsf{q}_{s^2}(\|\boldsymbol{\theta}\|) \mathsf{p}'_{s^1}(\|\boldsymbol{\theta}\|)}_{\text{of order } (M_1 + M_3 - 1)} \underbrace{\mathsf{t}_{s^2}(\|\mathbf{x}\|) \mathsf{r}_{s^1}(\|\mathbf{x}\|)}_{\text{of order } \leq (M_2 + M_4)} + \underbrace{\mathsf{q}_{s^1}(\|\boldsymbol{\theta}\|) \mathsf{p}'_{s^2}(\|\boldsymbol{\theta}\|)}_{\text{of order } \leq (M_1 + M_3 - 1)} \underbrace{\mathsf{t}_{s^1}(\|\mathbf{x}\|) \mathsf{r}_{s^2}(\|\mathbf{x}\|)}_{\text{of order } \leq (M_2 + M_4)}.$$

Similarly, we can get that

$$\sup_{\mathbf{g}_2 \in \partial_{\mathbf{x}} s} \big\| \langle \mathbf{g}_2, \mathbf{x} \rangle - (M_2 + M_4) s \big\|$$

$$\leq \sup_{\boldsymbol{\alpha}_2\in\partial_{\mathbf{x}}s^1,\mathbf{h}_2\in\partial_{\mathbf{x}}s^2} \|\langle s^2\boldsymbol{\alpha}_2,\mathbf{x}\rangle + \langle s^1\mathbf{h}_2,\mathbf{x}\rangle - (M_1+M_3)s\|$$

$$\leq \sup_{\boldsymbol{\alpha}_2\in\partial_{\mathbf{x}}s^1} \|s^2\|\cdot\|\langle\boldsymbol{\alpha}_2,\mathbf{x}\rangle - M_1 s^1\| + \sup_{\mathbf{h}_2\in\partial_{\mathbf{x}}s^2} \|s^1\|\cdot\|\langle\mathbf{h}_2,\mathbf{x}\rangle - M_3 s^2\|$$

$$\leq \mathsf{q}_{s^2}(\|\boldsymbol{\theta}\|)\mathsf{t}_{s^2}(\|\mathbf{x}\|)\mathsf{p}_{s^1}(\|\boldsymbol{\theta}\|)\mathsf{r}'_{s^1}(\|\mathbf{x}\|) + \mathsf{q}_{s^1}(\|\boldsymbol{\theta}\|)\mathsf{t}_{s^1}(\|\mathbf{x}\|)\mathsf{p}_{s^2}(\|\boldsymbol{\theta}\|)\mathsf{r}'_{s^2}(\|\mathbf{x}\|)$$

$$= \underbrace{\mathsf{q}_{s^2}(\|\boldsymbol{\theta}\|)\mathsf{p}_{s^1}(\|\boldsymbol{\theta}\|)}_{\text{of order }(M_1+M_3)}\underbrace{\mathsf{t}_{s^2}(\|\mathbf{x}\|)\mathsf{r}'_{s^1}(\|\mathbf{x}\|)}_{\text{of order }\leq(M_2+M_4-1)} + \underbrace{\mathsf{q}_{s^1}(\|\boldsymbol{\theta}\|)\mathsf{p}_{s^2}(\|\boldsymbol{\theta}\|)}_{\text{of order }\leq(M_1+M_3)}\underbrace{\mathsf{t}_{s^1}(\|\mathbf{x}\|)\mathsf{r}'_{s^2}(\|\mathbf{x}\|)}_{\text{of order }\leq(M_2+M_4-1)}.$$

Combining these two inequalities, we can see that there exists $\mathsf{q}_s,\mathsf{r}_s\in\mathbb{R}_+[x]$ such that $\deg\mathsf{q}_s\leq(M_1+M_3)$, $\deg\mathsf{r}_s\leq(M_2+M_4)$, and

$$\sup_{\mathbf{g}_1\in\partial_{\boldsymbol{\theta}}s(\boldsymbol{\theta};\mathbf{x})} \|\langle\mathbf{g}_1,\boldsymbol{\theta}\rangle - (M_1+M_3)s(\boldsymbol{\theta};\mathbf{x})\| \leq \mathsf{p}'_s(\|\boldsymbol{\theta}\|)\mathsf{r}_s(\|\mathbf{x}\|),$$

$$\sup_{\mathbf{g}_2\in\partial_{\mathbf{x}}s(\boldsymbol{\theta};\mathbf{x})} \|\langle\mathbf{g}_2,\mathbf{x}\rangle - (M_2+M_4)s(\boldsymbol{\theta};\mathbf{x})\| \leq \mathsf{p}_s(\|\boldsymbol{\theta}\|)\mathsf{r}'_s(\|\mathbf{x}\|).$$

Verify (B2) in Definition 2.

As for $\mathbf{g}_1$, we have

$$\sup_{\mathbf{g}_1\in\partial_{\boldsymbol{\theta}}s} \|\mathbf{g}_1\|^2 \leq \sup_{\boldsymbol{\alpha}_1\in\partial_{\boldsymbol{\theta}}s^1,\mathbf{h}_1\in\partial_{\boldsymbol{\theta}}s^2} \|s^2\boldsymbol{\alpha}_1\|^2 + \|s^1\mathbf{h}_1\|^2$$

$$\leq \left[\mathsf{q}_{s^2}(\|\boldsymbol{\theta}\|)\mathsf{t}_{s^2}(\|\mathbf{x}\|)\mathsf{q}'_{s^1}(\|\boldsymbol{\theta}\|)\mathsf{t}_{s^1}(\|\mathbf{x}\|)\right]^2 + \left[\mathsf{q}_{s^1}(\|\boldsymbol{\theta}\|)\mathsf{t}_{s^1}(\|\mathbf{x}\|)\mathsf{q}'_{s^2}(\|\boldsymbol{\theta}\|)\mathsf{t}_{s^2}(\|\mathbf{x}\|)\right]^2$$

$$= \left[\underbrace{\mathsf{q}_{s^2}(\|\boldsymbol{\theta}\|)\mathsf{q}'_{s^1}(\|\boldsymbol{\theta}\|)}_{\text{of order }\leq(M_1+M_3-1)}\cdot\underbrace{\mathsf{t}_{s^1}(\|\mathbf{x}\|)\mathsf{t}_{s^2}(\|\mathbf{x}\|)}_{\text{of order }\leq(M_2+M_4)}\right]^2 + \left[\underbrace{\mathsf{q}_{s^1}(\|\boldsymbol{\theta}\|)\mathsf{q}'_{s^2}(\|\boldsymbol{\theta}\|)}_{\text{of order }\leq(M_1+M_3-1)}\cdot\underbrace{\mathsf{t}_{s^2}(\|\mathbf{x}\|)\mathsf{t}_{s^1}(\|\mathbf{x}\|)}_{\text{of order }\leq(M_2+M_4)}\right]^2.$$

Similarly, we have

$$\sup_{\mathbf{g}_2\in\partial_{\mathbf{x}}s} \|\mathbf{g}_2\|^2 \leq \sup_{\boldsymbol{\alpha}_2\in\partial_{\mathbf{x}}s^1,\mathbf{h}_2\in\partial_{\mathbf{x}}s^2} \|s^2\boldsymbol{\alpha}_2 + s^1\mathbf{h}_2\|^2$$

$$\leq \sup_{\boldsymbol{\alpha}_2\in\partial_{\mathbf{x}}s^1,\mathbf{h}_2\in\partial_{\mathbf{x}}s^2} 2\|s^2\boldsymbol{\alpha}_2\|^2 + 2\|s^1\mathbf{h}_2\|^2$$

$$\leq \left[\mathsf{q}_{s^2}(\|\boldsymbol{\theta}\|)\mathsf{t}_{s^2}(\|\mathbf{x}\|)\mathsf{q}_{s^1}(\|\boldsymbol{\theta}\|)\mathsf{t}'_{s^1}(\|\mathbf{x}\|)\right]^2 + \left[\mathsf{q}_{s^1}(\|\boldsymbol{\theta}\|)\mathsf{t}_{s^1}(\|\mathbf{x}\|)\mathsf{q}_{s^2}(\|\boldsymbol{\theta}\|)\mathsf{t}'_{s^2}(\|\mathbf{x}\|)\right]^2$$

$$= \left[\underbrace{\mathsf{q}_{s^2}(\|\boldsymbol{\theta}\|)\mathsf{q}_{s^1}(\|\boldsymbol{\theta}\|)}_{\text{of order }\leq(M_1+M_3)}\cdot\underbrace{\mathsf{t}'_{s^1}(\|\mathbf{x}\|)\mathsf{t}_{s^2}(\|\mathbf{x}\|)}_{\text{of order }\leq(M_2+M_4-1)}\right]^2 + \left[\underbrace{\mathsf{q}_{s^1}(\|\boldsymbol{\theta}\|)\mathsf{q}_{s^2}(\|\boldsymbol{\theta}\|)}_{\text{of order }\leq(M_1+M_3)}\cdot\underbrace{\mathsf{t}'_{s^2}(\|\mathbf{x}\|)\mathsf{t}_{s^1}(\|\mathbf{x}\|)}_{\text{of order }\leq(M_2+M_4-1)}\right]^2.$$

Therefore, there exist $\mathsf{q}_s,\mathsf{t}_s\in\mathbb{R}_+[x]$ and $\deg\mathsf{q}_s\leq M_1+M_2M_3$, $\deg\mathsf{t}_s\leq M_2M_4$ such that

$$\sup_{\mathbf{g}_1\in\partial_{\boldsymbol{\theta}}s} \|\mathbf{g}_1\| \leq \mathsf{q}'_s(\|\boldsymbol{\theta}\|)\mathsf{t}_s(\|\mathbf{x}\|),$$

$$\sup_{\mathbf{g}_2\in\partial_{\mathbf{x}}s} \|\mathbf{g}_2\| \leq \mathsf{q}_s(\|\boldsymbol{\theta}\|)\mathsf{t}'_s(\|\mathbf{x}\|).$$

We have shown that $s$ satisfies (B1) and (B2) with parameter $(M_1+M_3,M_2+M_4)$. Since (B1) and (B2) lead to (B3), this leads to that $s(\boldsymbol{\theta};\mathbf{x})$ is near-$(M_1+M_3,M_2+M_4)$-homogeneous.

This completes the proof of Lemma 4.2. □

### D.3. Proof of Corollary 4.3

*Proof of Corollary 4.3.* This is a direct consequence of Lemma 4.2 and induction on the number of layers. To prove this, we need a stronger result. For

$$f(\boldsymbol{\theta};\mathbf{x}) = s^1_{\boldsymbol{\theta}_1}\circ s^2_{\boldsymbol{\theta}_2}\circ\cdots\circ s^L_{\boldsymbol{\theta}_L}(\mathbf{x}),$$

it's

$$\left( \sum_{j=1}^{L} M_1^j \prod_{i=1}^{j-1} M_2^i, \ \prod_{i=1}^{L} M_2^i \right)$$

dual-homogeneous. When $L = 1$, the result is trivial. We assume it holds for $L - 1$ layers. Then we have $f^{L-1}(\boldsymbol{\theta}; \mathbf{x}) = s_{\boldsymbol{\theta}_1}^1 \circ s_{\boldsymbol{\theta}_2}^2 \circ \cdots \circ s_{\boldsymbol{\theta}_{L-1}}^{L-1}(\mathbf{x})$ is dual-homogeneous with parameters

$$\left( \sum_{j=1}^{L-1} M_1^j \prod_{i=1}^{j-1} M_2^i, \ \prod_{i=1}^{L-1} M_2^i \right).$$

Then we apply Lemma 4.2 to $f(\boldsymbol{\theta}; \mathbf{x}) = f^{L-1}(\boldsymbol{\theta}; \mathbf{x}) \circ s_{\boldsymbol{\theta}_L}^L(\mathbf{x})$, we have $f$ is dual-homogeneous with parameters

$$\left( \sum_{j=1}^{L-1} M_1^j \prod_{i=1}^{j-1} M_2^i + M_1^L \prod_{i=1}^{L-1} M_2^i, \ \prod_{i=1}^{L} M_2^i \right) = \left( \sum_{j=1}^{L} M_1^j \prod_{i=1}^{j-1} M_2^i, \ \prod_{i=1}^{L} M_2^i \right).$$

This completes the proof of Corollary 4.3. $\qquad\qquad\square$

### D.4. Dual-Homogeneous Examples

In this subsection, we will try to provide some examples of dual-homogeneous blocks. The frist example we want to verify is the linear layers in neural networks.

**Example D.4** (Linear transform)**.** *Let the linear transform block function $s(\boldsymbol{\theta}; \mathbf{x}) = A\mathbf{x} + b$. Then, $s$ is near-homogeneous of parameter $(1, 1)$. We assume that $\mathbf{x} \in \mathbb{R}^{d_1}, A \in \mathbb{R}^{d_2 \times d_1}$ and $b \in \mathbb{R}^{d_2}$.*

*Proof.* Note that

$$\nabla_A s = I_{d_2 \times d_2} \otimes \mathbf{x}, \quad \nabla_b s = I_{d_2 \times d_2}, \quad \nabla_{\mathbf{x}} s = A.$$

Therefore, we have

$$\langle \nabla_{\mathbf{x}} s, \mathbf{x} \rangle - s = b, \quad \langle \nabla_{\boldsymbol{\theta}} s, \boldsymbol{\theta} \rangle - s = 0.$$

Furthermore, if we define the norm on $A$ as the $\|\text{vec}(A)\|_2$, then we have

$$\|\nabla_{\boldsymbol{\theta}} s\|_2 \le \sqrt{d_2} \|\mathbf{x}\|_2 + \sqrt{d_1}, \quad \|\nabla_{\mathbf{x}} s\|_2 \le \|A\|_2.$$

Then we can choose

$$\begin{cases} \mathsf{p}(x) = x \\ \mathsf{r}(x) = x \\ \mathsf{q}(x) = x \\ \mathsf{t}(x) = \sqrt{d_2} x + \sqrt{d_1}. \end{cases}$$

Therefore, $s$ satisfies (B1), (B2) and (B3) in Definition 2 with parameter $(1, 1)$. $\qquad\qquad\square$

The second kind of block functions is the activation functions $\phi(\mathbf{x})$. Note that the activation function doesn't have parameter $\boldsymbol{\theta}$. Therefore, we can reduce the dual homogeneity assumptions (B1), (B2) and (B3) to:

**Definition 9** (Near-homogeneous activation function)**.** M-homogeneous activation assumptions. Given a definable and locally Lipschitz activation function $\phi(\mathbf{x})$, we assume that there exists $M \in \mathbb{Z}_+$ and $\mathsf{p}, \mathsf{q} \in \mathbb{R}_+[x]$ with $\deg \mathsf{r}, \deg \mathsf{t} \le M$, such that for all $\mathbf{x} \in (\mathbf{x})_{i=1}^n$ and all $\mathbf{h} \in \partial \phi(\mathbf{x})$:

(C1) $M$**-Near homogeneity.**

$$\|\langle \mathbf{h}, \mathbf{x} \rangle - M\phi(\mathbf{x})\| \le \mathsf{r}'(\|\mathbf{x}\|).$$

(C2) $M$**-bounded gradient.**

$$\|\mathbf{h}\| \le \mathsf{t}'(\|\mathbf{x}\|).$$

(C3) $M$**-bounded value.**

$$\|\phi(\mathbf{x})\| \leq \mathsf{t}(\|\mathbf{x}\|).$$

In this case, we say that $\phi(\mathbf{x})$ satisfies (C1), (C2) and (C3) with parameter $M$.

A direct property of this definition is that:

**Corollary D.5** (Near-homo activation as dual homo block). *Given a near-homogeneous activation function $\phi(x)$ with parameter $M$, then it's dual homogeneous with parameters $(0, M)$.*

We can easily verify the following results:

**Example D.6** (Activation functions). *The following activation functions are near-homogenneous activations with parameter 1: ReLU, Leaky ReLU, Softplus, Huberized ReLU, Swish, GELU, SiLU.*

Another important property of the activation function is that:

**Lemma D.7** (Power of activation functions). *Given an activation function $s$ of parameter $(0, M)$, $s^k$ is of parameter $(0, kM)$.*

*Proof.* Similar to Lemma D.1, we can show that (C1) implies (C3). Therefore, it suffices to prove (C1) and (C2) for $s^k$ with $M$ replaced by $kM$. By the definition of Clarke subdifferential, we have

$$\partial s^k(\mathbf{x}) = \{s^{k-1} \cdot \boldsymbol{\alpha} : \boldsymbol{\alpha} \in \partial s(\mathbf{x})\}.$$

By direct calculation, we obtain for any $\mathbf{h} \in \partial s^k(x)$,

$$\|\langle \mathbf{h}, \mathbf{x}\rangle - kM s^k(\mathbf{x})\| = |ks^{k-1}(\mathbf{x})| \|\langle \nabla s(\mathbf{x}), \mathbf{x}\rangle - M s(\mathbf{x})\|$$
$$\leq k \underbrace{\mathsf{t}^{k-1}(\|\mathbf{x}\|)\mathsf{r}'(\|\mathbf{x}\|)}_{\text{of order } \leq (kM-1)},$$

which implies (C1). For (C2), note that

$$\|\mathbf{h}\| = |ks^{k-1}(\mathbf{x})| \|\boldsymbol{\alpha}\| \leq \underbrace{k\mathsf{t}^{k-1}(\|\mathbf{x}\|)\mathsf{t}'(\|\mathbf{x}\|)}_{\text{of order } \leq (kM-1)},$$

completing the proof. □

With this lemma, we can understand the near homogeneity of the polynomial activation functions. Then we have the following result for a layer of the neural network, i.e.,

$$s(\boldsymbol{\theta}; \mathbf{x}) = \phi(A\mathbf{x} + b).$$

**Corollary D.8** (Near homogeneity layer). *Given a layer of the neural network $s(\boldsymbol{\theta}; \mathbf{x}) = \phi(A\mathbf{x} + b)$, where $\phi$ is a near-homogeneous activation with parameter $M$, then $s$ is dual-homogeneous with parameter $(1, M)$.*

*Proof.* Apply Lemma 4.2 to $s^1_{\boldsymbol{\theta}_1}(\mathbf{x}) = \phi(\mathbf{x})$ and $s^2_{\boldsymbol{\theta}_2}(\mathbf{x}) = A\mathbf{x} + b$. □

With all the tools we build here, we can now understand the near homogeneity of many commonly used network architectures.

### D.5. Proof of Example 4.1

*Proof of Example 4.1.* There are multiple arguments in this example. We will prove them one by one.

Argument 1. This is the result of Example D.4.

Argument 2. This is the result of Corollary D.8.

Argument 3. Max pooling layer, convolution layer, max layer, average layer are 1-homogeneous operations. Hence, they are dual-homogeneous with parameter $(0, 1)$. For the residual connection, we just keep the input as the output. Therefore, it's dual-homogeneous with parameter $(0, 1)$.

Argument 4. Recall that the SwiGLU is defined as:

$$\text{SwiGLU}(x, W, V, b, c, \beta) = \text{Swish}_\beta(xW + b) \otimes (xV + c),$$

where $\beta$ is one hyperparameter and $W, V, b, c$ are trainable. We can see that SwiGLU is a tensor product of two near-homogeneous functions. In terms of Corollary D.8, $\text{Swish}_\beta(xW + b)$ is dual-homogeneous with parameter $(1, 1)$. $(xV + c)$ is also dual-homogeneous with parameter $(1, 1)$. Then by Lemma 4.2, we have SwiGLU is dual-homogeneous with parameter $(2, 2)$.

Argument 5. The self-linear attention (Zhang et al., 2024) is defined as:

$$f(\boldsymbol{\theta}; \mathbf{H}) = \mathbf{H} + \mathbf{W}^{PV}\mathbf{H} \cdot \frac{\mathbf{H}^\top \mathbf{W}^{KQ} \mathbf{H}}{\rho},$$

where $\mathbf{H}$ is the input token matrix and $\boldsymbol{\theta} = (\mathbf{W}^{PV}, \mathbf{W}^{KQ})$. We can observe the dominating term in the linear attention layer is $\mathbf{W}^{PV}\mathbf{H} \cdot \frac{\mathbf{H}^\top \mathbf{W}^{KQ} \mathbf{H}}{\rho}$. Therefore, the self-linear attention layer is dual-homogeneous with parameter $(2, 3)$.

The ReLU attention (Wortsman et al., 2023) is defined as:

$$f(\boldsymbol{\theta}; \mathbf{H}) = \mathbf{H} + \mathbf{W}^P \mathbf{W}^V \mathbf{H} \cdot \text{ReLU}\left(\frac{\mathbf{H}^\top \mathbf{W}^K \mathbf{W}^Q \mathbf{H}}{\sqrt{dL}}\right),$$

where $\mathbf{H}$ is the input token matrix, $L$ is the contextual length, and $\boldsymbol{\theta} = (\mathbf{W}^P, \mathbf{W}^V, \mathbf{W}^K, \mathbf{W}^Q)$. The dominating term here is

$$\mathbf{W}^P \mathbf{W}^V \mathbf{H} \cdot \text{ReLU}\left(\frac{\mathbf{H}^\top \mathbf{W}^K \mathbf{W}^Q \mathbf{H}}{\sqrt{dL}}\right),$$

which is $(4, 3)$-dual-homogeneous. Hence, the ReLU attention layer is dual-homogeneous with parameter $(4, 3)$. This completes the proof of Example 4.1. □

### D.6. Proof of Example 4.4

*Proof of Example 4.4.* We will prove the following arguments one by one.

Argument 1.

This is a direct result of Corollary 4.3 and Example 4.1. Now we have

$$f(\boldsymbol{\theta}; \mathbf{x}) = s_{\boldsymbol{\theta}_1}^1 \circ s_{\boldsymbol{\theta}_2}^2 \circ \cdots \circ s_{\boldsymbol{\theta}_L}^L(\mathbf{x}),$$

and

$$(M_1^i, M_2^i) = (1, 1), \quad \forall i \in [L].$$

Hence, by Corollary 4.3, we have $f(\boldsymbol{\theta}; \mathbf{x})$ is near-homogeneous with parameter

$$M = \sum_{j=1}^L M_1^j \cdot \left[\prod_{i=1}^{j-1} M_2^i\right] = L.$$

Argument 2.

For this case, we have

$$(M_1^i, M_2^i) = (1, k), \quad \forall i \in [L].$$

Hence, by Corollary 4.3, we have $f(\boldsymbol{\theta}; \mathbf{x})$ is near-homogeneous with parameter

$$M = \sum_{j=1}^{L} M_1^j \cdot \left[\prod_{i=1}^{j-1} M_2^i\right] = \sum_{j=1}^{L} k^{j-1} = \frac{k^L - 1}{k - 1}.$$

Argument 3.

For VGG networks, all the layers are average pooling, max pooling, and convolutional layers followed by a ReLU activation. Since convolutional layers are $(1,1)$-dual-homogeneous and ReLU activation is $(0,1)$-dual-homogeneous, we can apply Corollary 4.3 to VGG networks and get VGG-i is near-homogeneous with parameter $i$.

Argument 4.

For ResNet and DenseNet without batch normalization, all the layers are average pooling, max pooling, convolutional layers followed by a ReLU, and residual connections. By argument C in Lemma 4.2, residual connections won't influence the homogeneous degree of the network. Since convolutional layers are $(1,1)$-dual-homogeneous and ReLU activation is $(0,1)$-dual-homogeneous, we can apply Corollary 4.3 to ResNet and DenseNet and get ResNet-$L$ and DenseNet-$L$ are near-homogeneous with parameter $L$. $\qquad\square$

# E. Proofs for Section 5

The proofs in this section also has two parts. In Part I, we introduce the homogeneization of models and show the existence of parmeter which satisfies the initial condition. In Part II, we focus on a toy example and show that the initial condition can be reached in the GF.

## E.1. Homogenization

For those Homogenization results, the most important idea is to analysis:

$$g(r) := \frac{f(r\boldsymbol{\theta}; \mathbf{x})}{r^M}.$$

Since $f$ is locally Lipschitz and definable, we know that $g$ is also locally Lipschitz and definable in any open subset of $\mathbb{R}\backslash\{0\}$. Hence, $g$ is differentiable almost everywhere. Actually, the best thing is that $g$ is $M$-homogeneous.

## E.2. Proof of Proposition 5.1

*Proof of Proposition 5.1.* By direct calculation, we get (see also the proof of Lemma C.8) that for a.e. $r > 0$:

$$g'(r) = \frac{r^M \langle \boldsymbol{\theta}, \nabla_{\boldsymbol{\theta}} f(r\boldsymbol{\theta}; \mathbf{x}) \rangle - Mr^{M-1} f(r\boldsymbol{\theta}; \mathbf{x})}{r^{2M}} = \frac{\langle r\boldsymbol{\theta}, \nabla_{\boldsymbol{\theta}} f(r\boldsymbol{\theta}; \mathbf{x}) \rangle - M f(r\boldsymbol{\theta}; \mathbf{x})}{r^{M+1}},$$

which leads to the following estimate

$$|g'(r)| \leq \frac{\mathsf{p}'(r\|\boldsymbol{\theta}\|)}{r^{M+1}} = \sum_{i=0}^{M-1} a_{i+1}(i+1) r^{-(M+1-i)} \|\boldsymbol{\theta}\|^i,$$

where the equality follows that $\mathsf{p}(x) = \sum_{i=0}^{M} a_i x^i$. This further implies that for any $0 < r_1 < r_2$:

$$|g(r_1) - g(r_2)| \leq \int_{r_1}^{r_2} |g'(r)| \mathrm{d}r \leq \sum_{i=0}^{M-1} \frac{a_{i+1}(i+1)}{M-i} r_1^{-(M-i)} \|\boldsymbol{\theta}\|^i \leq \frac{\mathsf{p}_a(r_1\|\boldsymbol{\theta}\|)}{r_1^M}.$$

Note that $\deg \mathsf{p}_a = M - 1$, and this leads to $\mathsf{p}_a(r_1\|\boldsymbol{\theta}\|)/r_1^M \to 0$ as $r_1 \to \infty$. As a consequence, for any sequence $\{r_k\}$ such that $r_k \to +\infty$ as $k \to \infty$, $\{g(r_k)\}$ is a Cauchy sequence. Therefore, $\lim_{r \to +\infty} g(r)$ exists, and $f_{\mathsf{H}}(\boldsymbol{\theta}; \mathbf{x})$ is well-defined. From its definition, it is easily seen that $f_{\mathsf{H}}(\boldsymbol{\theta}; \mathbf{x})$ is $M$-homogeneous.

We next show that $f_{\mathsf{H}}(\boldsymbol{\theta};\mathbf{x})$ is continuous and a.e. differentiable in $\boldsymbol{\theta}$ for fixed $\mathbf{x}$. To this end, it suffices to show that $f_{\mathsf{H}}(\boldsymbol{\theta};\mathbf{x})|_{\boldsymbol{\theta}\in\mathbb{S}^{d-1}}$ is Lipschitz continuous. For any $r > 0$ and $\boldsymbol{\theta}_1, \boldsymbol{\theta}_2 \in \mathbb{S}^{d-1}$, we have the following estimate:

$$\left| \frac{f(r\boldsymbol{\theta}_1;\mathbf{x})}{r^M} - \frac{f(r\boldsymbol{\theta}_2;\mathbf{x})}{r^M} \right| = \frac{1}{r^M} \left| f(r\boldsymbol{\theta}_1;\mathbf{x}) - f(r\boldsymbol{\theta}_2;\mathbf{x}) \right|$$

$$\stackrel{(i)}{\leq} \frac{\mathsf{p}'(r)}{r^M} \|r\boldsymbol{\theta}_1 - r\boldsymbol{\theta}_2\| \leq C_{M,\mathsf{p}} \|\boldsymbol{\theta}_1 - \boldsymbol{\theta}_2\|,$$

where $(i)$ follows from Assumption (A2) in Definition 1, and $C_{M,\mathsf{p}}$ is a constant depending on $M$ and $\mathsf{p}$. Taking $r \to \infty$ yields

$$|f_{\mathsf{H}}(\boldsymbol{\theta}_1;\mathbf{x}) - f_{\mathsf{H}}(\boldsymbol{\theta}_2;\mathbf{x})| \leq C_{M,\mathsf{p}} \|\boldsymbol{\theta}_1 - \boldsymbol{\theta}_2\|,$$

i.e., $f_{\mathsf{H}}(\boldsymbol{\theta};\mathbf{x})$ is locally Lipschitz continuous. The error bound follows directly from our previous calculation:

$$|f(\boldsymbol{\theta};\mathbf{x}) - f_{\mathsf{H}}(\boldsymbol{\theta};\mathbf{x})| = |g(1) - g(+\infty)| \leq \mathsf{p}_a(\|\boldsymbol{\theta}\|).$$

At last, we show that if $f$ is differentiable and $\nabla f$ satisfies Assumption 3, $f_{\mathsf{H}}$ is differentiable. This is equivalent to showing that the following limit exists for any $\theta$ and $j \in [d]$:

$$\lim_{t\to 0} \frac{f_{\mathsf{H}}(\boldsymbol{\theta} + te_j;\mathbf{x}) - f_{\mathsf{H}}(\boldsymbol{\theta};\mathbf{x})}{t}.$$

For simplicity, we omit the $\mathbf{x}$ in the function. Note that

$$\lim_{t\to 0} \frac{f_{\mathsf{H}}(\boldsymbol{\theta} + te_j) - f_{\mathsf{H}}(\boldsymbol{\theta})}{t} = \lim_{t\to 0}\lim_{r_n\to\infty} \frac{f(r_n\boldsymbol{\theta} + r_n te_j) - f(r_n\boldsymbol{\theta})}{r_n^M t},$$

due to the definition of $f_{\mathsf{H}}$. We let

$$\tilde{f}_n(t) := \frac{f(r_n\boldsymbol{\theta} + r_n te_j) - f(r_n\boldsymbol{\theta})}{r_n^M t}, \quad \tilde{f}_{\mathsf{H}}(t) := \frac{f_{\mathsf{H}}(\boldsymbol{\theta} + te_j) - f_{\mathsf{H}}(\boldsymbol{\theta})}{t}.$$

And we want to invoke Lemma G.5 to finish this. To invoke Lemma G.5, we need to verify two conditions.

- Condition 1: There exists an open interval $(-a, a)$ such that $(\tilde{f}_n(t))_{n=1}^{\infty}$ converges uniformly to $\tilde{f}_{\mathsf{H}}(t)$.

- Condition 2: Dor any fixed $n$, as $t \to 0$, $\tilde{f}_n(t)$ converges to $[\nabla f]_j(r_n\boldsymbol{\theta})/r_n^{M-1}$.

Once these two conditions are verified, we can exchange the limit and get

$$\lim_{n\to\infty} \frac{[\nabla f]_j(r_n\boldsymbol{\theta})}{r_n^{M-1}} = \lim_{n\to\infty}\lim_{t\to 0}\tilde{f}_n(t) = \lim_{t\to 0}\lim_{n\to\infty}\tilde{f}_n(t) = \lim_{t\to 0}\tilde{f}_{\mathsf{H}}(t).$$

By Assumption 3, this will lead to:

$$[(\nabla f)_{\mathsf{H}}]_j(\boldsymbol{\theta}) = \lim_{t\to 0}\tilde{f}_{\mathsf{H}}(t) = \lim_{t\to 0} \frac{f_{\mathsf{H}}(\boldsymbol{\theta} + te_j) - f_{\mathsf{H}}(\boldsymbol{\theta})}{t} = [\nabla(f_{\mathsf{H}})]_j(\boldsymbol{\theta}).$$

This indicates that $\nabla(f_{\mathsf{H}})$ exists and is equal to $(\nabla f)_{\mathsf{H}}$. Now we focus on verifying the two conditions.

Verify condition 1. Here we fix some $\boldsymbol{\theta} \in \mathbb{R}^d/\{0\}$ and pick some special $a$ such that

$$\arg\max_{t\in[-a,a],j\in[d]} \|\boldsymbol{\theta} + te_j\| \leq \bar{r}_*, \quad \arg\min_{t\in[-a,a],j\in[d]} \|\boldsymbol{\theta} + te_j\| \geq \underline{r}_*,$$

for some $\bar{r}_*, \underline{r}_* > 0$. Then we will show that for any $t \in [a, -a]$ $\tilde{f}_n(t)$ converges uniformly to

$$\int_0^1 \langle (\nabla f)_{\mathsf{H}}(\boldsymbol{\theta} + s \cdot te_j), te_j \rangle \, ds.$$

The first thing to do is to show this intergral is well-defined. Note that $\nabla f$ is measurable. Hence, the limit $(\nabla f)_{\mathsf{H}}$ is also measurable. Furthermore, we have $\|\nabla f\| \leq \mathsf{q}'(x)$ where $\deg \mathsf{q} \leq M$. We must have

$$(\nabla f)_{\mathsf{H}}(\boldsymbol{\theta}) := \lim_{r \to \infty} \frac{\nabla f(r\boldsymbol{\theta})}{r^{M-1}} \leq \lim_{r \to \infty} \frac{\mathsf{q}'(r\|\boldsymbol{\theta}\|)}{r^{M-1}} \leq C\|\boldsymbol{\theta}\|^{M-1},$$

Since $\arg\max_{s \in [0,1], t \in [-a,a]} \|\boldsymbol{\theta} + s \cdot te_j\|$ is bounded, the integral is well-defined.

Now, we will show the uniform convergence. By Assumption 3, we know for any $\epsilon$:

$$\lim_{x \to \infty} \frac{\mathsf{r}(x)}{x^{M-1}} = 0 \implies \exists M, \text{ if } x \geq M, \mathsf{r}(x) \leq \frac{\epsilon}{(\bar{r}_*)^{M-1}} \cdot x^{M-1}.$$

Furthermore, since $r_n \to \infty$, there exists an $M'$ such that for any $n > M'$:

$$r_n \geq M/\underline{r}_* \implies \arg\min_{t \in [-a,a], j \in [d]} r_n\|\boldsymbol{\theta} + te_j\| \geq r_n\underline{r}_* \geq M.$$

Therefore, for any $t \in [-a,a]$ and $n > M'$,

$$\tilde{f}_n(t) - \int_0^1 \langle (\nabla f)_{\mathsf{H}}(\boldsymbol{\theta} + s \cdot te_j), te_j \rangle \, \mathrm{d}s$$

$$= \frac{f(r_n\boldsymbol{\theta} + r_nte_j) - f(r_n\boldsymbol{\theta})}{r_n^M t} - \int_0^1 \langle (\nabla f)_{\mathsf{H}}(\boldsymbol{\theta} + s \cdot te_j), te_j \rangle \, \mathrm{d}s$$

$$= \frac{1}{r_n^M t} \int_0^1 \langle \nabla f(r_n\boldsymbol{\theta} + s \cdot r_nte_j), r_nte_j \rangle \, \mathrm{d}s - \int_0^1 \langle (\nabla f)_{\mathsf{H}}(\boldsymbol{\theta} + s \cdot te_j), te_j \rangle \, \mathrm{d}s$$

$$= \int_0^1 \left\langle \frac{\nabla f(r_n\boldsymbol{\theta} + s \cdot r_nte_j)}{r_n^{M-1}} - (\nabla f)_{\mathsf{H}}(\boldsymbol{\theta} + s \cdot te_j), e_j \right\rangle \mathrm{d}s$$

$$= \int_0^1 \left\langle \frac{\nabla f(r_n\boldsymbol{\theta} + s \cdot r_nte_j) - (\nabla f)_{\mathsf{H}}(r_n\boldsymbol{\theta} + s \cdot r_nte_j)}{r_n^{M-1}}, e_j \right\rangle \mathrm{d}s$$

$$\leq \int_0^1 \frac{\mathsf{r}(\|r_n\boldsymbol{\theta} + s \cdot r_nte_j\|)}{r_n^{M-1}} \, \mathrm{d}s$$

$$\leq \int_0^1 \frac{\epsilon}{(\bar{r}_*)^{M-1}} \cdot \frac{\|r_n\boldsymbol{\theta} + s \cdot r_nte_j\|^{M-1}}{r_n^{M-1}} \, \mathrm{d}s$$

$$\quad\quad \text{By} \quad \arg\min_{t \in [-a,a], j \in [d]} r_n\|\boldsymbol{\theta} + te_j\| \geq M$$

$$= \int_0^1 \frac{\epsilon}{(\bar{r}_*)^{M-1}} \|\boldsymbol{\theta} + s \cdot te_j\|^{M-1} \, \mathrm{d}s$$

$$\leq \frac{\epsilon}{(\bar{r}_*)^{M-1}} (\bar{r}_*)^{M-1} = \epsilon.$$

At last, since we already know that $\tilde{f}_n(t)$ pointwise converges to $\tilde{f}_H(t)$ and $\tilde{f}_n(t)$ uniformly converges to $\int_0^1 \langle (\nabla f)_{\mathsf{H}}(\boldsymbol{\theta} + s \cdot te_j), te_j \rangle \, \mathrm{d}s$, we must have $\tilde{f}_h(t)$ uniformly converges to $\tilde{f}_H(t)$ and

$$\tilde{f}_H(t) = \int_0^1 \langle (\nabla f)_{\mathsf{H}}(\boldsymbol{\theta} + s \cdot te_j), te_j \rangle \, \mathrm{d}s.$$

Verify condition 2. Recall that $f$ is differentiable. Hence, for any fixed $n$, as $t \to 0$, we have

$$\lim_{t \to 0} \tilde{f}_n(t) = \lim_{t \to 0} \frac{f(r_n\boldsymbol{\theta} + r_nte_j) - f(r_n\boldsymbol{\theta})}{r_n^{M-1} \cdot (r_nt)} = \frac{1}{r_n^{M-1}} [\nabla f]_j(r_n\boldsymbol{\theta}).$$

Therefore, $f_{\mathsf{H}}$ is continuously differentiable on $\mathbb{R}^d / \{0\}$. Note that this further shows that

$$\nabla(f_{\mathsf{H}}) = (\nabla f)_{\mathsf{H}}.$$

This completes the proof of Proposition 5.1. $\qquad\qquad\qquad\qquad\qquad\qquad\qquad\qquad\qquad\qquad\square$

### E.3. Proof of Proposition 5.3

*Proof of Proposition 5.3.* This proof mainly has two parts.

- Part I: We show that

$$\lim_{r_1, r_2 \to \infty} \frac{f(r_1 \boldsymbol{\theta}; r_2 \mathbf{x})}{r_1^{M_1} r_2^{M_2}}$$

is well-defined, continuous, almost everywhere differentiable and $(M_1, M_2)$-homogeneous.

- Part II: We show that

$$f_{\mathsf{H}}(\boldsymbol{\theta}; \mathbf{x}) = s_{\mathsf{H}, \boldsymbol{\theta}_1}^1 \circ s_{\mathsf{H}, \boldsymbol{\theta}_2}^2 \circ \cdots \circ s_{\mathsf{H}, \boldsymbol{\theta}_L}^L(\mathbf{x}). \tag{55}$$

**Part I.** Similar to the proof of Proposition 5.1, we define for fixed $\boldsymbol{\theta}$ and $\mathbf{x}$:

$$g(r_1, r_2) = \frac{s(r_1 \boldsymbol{\theta}, r_2 \mathbf{x})}{r_1^{M_1} r_2^{M_2}}.$$

By Assumption (B1), we obtain the following estimates on the partial derivatives of $g$:

$$\partial_{r_1} g(r_1, r_2) = \frac{1}{r_1^{M_1+1} r_2^{M_2}} \|\nabla_{\boldsymbol{\theta}} s(r_1 \boldsymbol{\theta}; r_2 \mathbf{x}) r_1 \boldsymbol{\theta} - M_1 s(r_1 \boldsymbol{\theta}; r_2 \mathbf{x})\| \le \frac{\mathsf{p}_s'(r_1 \|\boldsymbol{\theta}\|) \mathsf{r}_s(r_2 \|\mathbf{x}\|)}{r_1^{M_1+1} r_2^{M_2}},$$

$$\partial_{r_2} g(r_1, r_2) = \frac{1}{r_1^{M_1} r_2^{M_2+1}} \|\nabla_{\mathbf{x}} s(r_1 \boldsymbol{\theta}; r_2 \mathbf{x}) r_2 \mathbf{x} - M_2 s(r_1 \boldsymbol{\theta}; r_2 \mathbf{x})\| \le \frac{\mathsf{p}_s(r_1 \|\boldsymbol{\theta}\|) \mathsf{r}_s'(r_2 \|\mathbf{x}\|)}{r_1^{M_1} r_2^{M_2+1}}.$$

As a consequence, for $r_3 > r_1$ and $r_4 > r_2$, we have

$$|g(r_3, r_4) - g(r_1, r_2)| \le |g(r_3, r_4) - g(r_3, r_2)| + |g(r_3, r_2) - g(r_1, r_2)|$$

$$\le \frac{\mathsf{p}_s(r_3 \|\boldsymbol{\theta}\|)}{r_3^{M_1}} \int_{r_2}^{r_4} \frac{\mathsf{r}_s'(r \|\mathbf{x}\|)}{r^{M_2+1}} dr + \frac{\mathsf{r}_s(r_2 \|\mathbf{x}\|)}{r_2^{M_2}} \int_{r_1}^{r_3} \frac{\mathsf{p}_s'(r \|\boldsymbol{\theta}\|)}{r^{M_1+1}} dr$$

$$\overset{(i)}{\le} \frac{\mathsf{p}_s(r_1 \|\boldsymbol{\theta}\|)}{r_1^{M_1}} \int_{r_2}^{+\infty} \frac{\mathsf{r}_s'(r \|\mathbf{x}\|)}{r^{M_2+1}} dr + \frac{\mathsf{r}_s(r_2 \|\mathbf{x}\|)}{r_2^{M_2}} \int_{r_1}^{+\infty} \frac{\mathsf{p}_s'(r \|\boldsymbol{\theta}\|)}{r^{M_1+1}} dr,$$

where $(i)$ follows from the fact that the coefficients of $\mathsf{p}_s$ (resp. $\mathsf{r}_s$) are all non-negative, so $r \mapsto \mathsf{p}_s(r \|\boldsymbol{\theta}\|)/r^{M_1}$ (resp. $r \mapsto \mathsf{r}_s(r \|\boldsymbol{\theta}\|)/r^{M_2}$) is non-increasing. Further, we can check that the right hand side goes to 0 as $r_1, r_2 \to \infty$. Using a similar argument as in the proof of Proposition 5.1, we deduce that $s_{\mathsf{H}}(\boldsymbol{\theta}; \mathbf{x}) = \lim_{r_1, r_2 \to \infty} g(r_1, r_2)$ exists, and is $M_1$-homogeneous in $\boldsymbol{\theta}$ and $M_2$-homogeneous in $\mathbf{x}$. This completes the proof of well-definedness. The proof of local Lipschitz is completely the same as that in the proof of Proposition 5.1.

**Part II.** We prove via induction on $L$. For $L = 1$, (55) holds automatically since $f(\boldsymbol{\theta}; \mathbf{x}) = s^1(\boldsymbol{\theta}_1; \mathbf{x})$ and $\boldsymbol{\theta} = \boldsymbol{\theta}_1$. Now assume the conclusion holds for $L - 1$, then we can write

$$f(\boldsymbol{\theta}; \mathbf{x}) = s^1(\boldsymbol{\theta}_1; f^{L-1}(\boldsymbol{\theta}^{L-1}; \mathbf{x})),$$

where $\boldsymbol{\theta}^{L-1} = (\boldsymbol{\theta}_2, \cdots, \boldsymbol{\theta}_L)$, and

$$f^{L-1}(\boldsymbol{\theta}^{L-1}; \mathbf{x}) = s_{\boldsymbol{\theta}_2}^2 \circ \cdots \circ s_{\boldsymbol{\theta}_L}^L(\mathbf{x}).$$

Therefore,

$$f_{\mathsf{H}}(\boldsymbol{\theta}; \mathbf{x}) = \lim_{r_1, r_2 \to \infty} \frac{f(r_1 \boldsymbol{\theta}; r_2 \mathbf{x})}{r_1^{M_1} r_2^{M_2}} = \lim_{r_1, r_2 \to \infty} \frac{s^1(r_1 \boldsymbol{\theta}_1; f^{L-1}(r_1 \boldsymbol{\theta}^{L-1}; r_2 \mathbf{x}))}{r_1^{M_1} r_2^{M_2}}.$$

Define $M_1' = \sum_{j=2}^L M_1^j \cdot \prod_{i=2}^{j-1} M_2^i$ and $M_2' = \prod_{j=2}^L M_2^j$. By our induction assumption,

$$f^{L-1}(r_1 \boldsymbol{\theta}^{L-1}; r_2 \mathbf{x}) = r_1^{M_1'} r_2^{M_2'} \left( f_{\mathsf{H}}^{L-1}(\boldsymbol{\theta}^{L-1}; \mathbf{x}) + o(1) \right)$$

$$= r_1^{M_1'} r_2^{M_2'} \left( s_{\mathsf{H},\boldsymbol{\theta}_2}^2 \circ \cdots \circ s_{\mathsf{H},\boldsymbol{\theta}_L}^L(\mathbf{x}) + o(1) \right).$$

Use a similar estimate as in the proof of Proposition 5.1, we obtain that

$$\left| s^1(r_1\boldsymbol{\theta}_1; f^{L-1}(r_1\boldsymbol{\theta}^{L-1}; r_2\mathbf{x})) - s^1\left( r_1\boldsymbol{\theta}_1; r_1^{M_1'} r_2^{M_2'} f_{\mathsf{H}}^{L-1}(\boldsymbol{\theta}^{L-1}; \mathbf{x}) \right) \right|$$

$$\overset{(i)}{\le} C(p,q,\boldsymbol{\theta},\mathbf{x}) r_1^{M_1^1} (r_1^{M_1'} r_2^{M_2'})^{M_2^1 - 1} o(r_1^{M_1'} r_2^{M_2'}) \overset{(ii)}{=} o(r_1^{M_1} r_2^{M_2}),$$

where $(i)$ follows from Assumption (B2), and the constant $C(p,q,\boldsymbol{\theta},\mathbf{x})$ might depend on the polynomials $(p_s, \mathsf{r}_s)$ associated with $s_{\boldsymbol{\theta}_i}^i$ and $(\boldsymbol{\theta},\mathbf{x})$, but not $r_1$ and $r_2$, and in $(ii)$ we use the identity $M_1 = M_1^1 + M_1' M_2^1$, $M_2 = M_2' M_2^1$. This implies that

$$
\begin{aligned}
f_{\mathsf{H}}(\boldsymbol{\theta}; \mathbf{x}) &= \lim_{r_1, r_2 \to \infty} \frac{s^1\left( r_1\boldsymbol{\theta}_1; r_1^{M_1'} r_2^{M_2'} f_{\mathsf{H}}^{L-1}(\boldsymbol{\theta}^{L-1}; \mathbf{x}) \right)}{r_1^{M_1} r_2^{M_2}} \\
&= \lim_{r_1, r_2 \to \infty} \frac{s^1\left( r_1\boldsymbol{\theta}_1; r_1^{M_1'} r_2^{M_2'} f_{\mathsf{H}}^{L-1}(\boldsymbol{\theta}^{L-1}; \mathbf{x}) \right)}{r_1^{M_1^1} \cdot (r_1^{M_1'} r_2^{M_2'})^{M_2^1}} \\
&= s_M^1\left( \boldsymbol{\theta}_1; f_{\mathsf{H}}^{L-1}(\boldsymbol{\theta}^{L-1}; \mathbf{x}) \right) \\
&= s_{\mathsf{H},\boldsymbol{\theta}_1}^1 \circ s_{\mathsf{H},\boldsymbol{\theta}_2}^2 \circ \cdots \circ s_{\mathsf{H},\boldsymbol{\theta}_L}^L(\mathbf{x}),
\end{aligned}
$$

completing the proof of Proposition 5.3. $\qquad\square$

### E.4. Proof of Proposition 5.2

*Proof of Proposition 5.2.* First, we let

$$\gamma = \arg\min_{i \in [n]} y_i f_{\mathsf{H}}(\boldsymbol{\theta}'; \mathbf{x}_i) > 0.$$

If we replace $\boldsymbol{\theta}$ with $c\boldsymbol{\theta}'$, then we have

$$\arg\min_{i \in [n]} y_i f_{\mathsf{H}}(c\boldsymbol{\theta}'; \mathbf{x}_i) = \gamma c^M.$$

Therefore,

$$\arg\min_{i \in [n]} y_i f(c\boldsymbol{\theta}'; \mathbf{x}_i) \ge \arg\min_{i \in [n]} y_i f_{\mathsf{H}}(c\boldsymbol{\theta}'; \mathbf{x}_i) - \mathsf{p}_a(c\|\boldsymbol{\theta}'\|) = \gamma c^M - \mathsf{p}_a(c\|\boldsymbol{\theta}'\|).$$

Hence,

$$\mathcal{L}(c\boldsymbol{\theta}') \le \exp(-\gamma c^M + \mathsf{p}_a(c\|\boldsymbol{\theta}'\|)).$$

To satisfy the initial condition in Assumption 2, we need to find $c$ such that

$$-\gamma c^M + \mathsf{p}_a(c\|\boldsymbol{\theta}'\|) < -\mathsf{p}_a(c\|\boldsymbol{\theta}'\|) - \log n \Leftrightarrow \gamma c^M > 2\mathsf{p}_a(c\|\boldsymbol{\theta}'\|) + \log n.$$

Since $\deg p_a \le M - 1$, we can find $c$ such that the above inequality holds. Similar argument can be applied to the case in Assumption 5. This completes the proof of Proposition 5.2. $\qquad\square$

### E.5. Two-Layer Example

The following subsections are devoted to the proof of the two-layer example in Section 5. We first show that the GD dynamics is symmetric and the limit solution of the GD dynamics is the solution to the ODE (56). Then, we show that the loss is of rate $O(1/t)$ and the parameter norm $\|\boldsymbol{\theta}_t\|$ is of rate $O(\sqrt{\log t})$. Combining these two rates, we have

$$\frac{\mathcal{L}(\boldsymbol{\theta}_t)}{\|\boldsymbol{\theta}_t\|} = \Omega(\sqrt{\log t}) \to \infty.$$

The first thing we are going to show here is that due to symmetry, the limit of GD can be reduced to a GF with the following dynamics.

**Lemma E.1** (Symmetry of the parameters). *We assume the dataset satisfies Assumption 4, and the model is (11) with initial condition $\mathbf{w}_{1,0} = \mathbf{w}_{2,0} = \mathbf{0}$ and $a_{1,0} = a_{2,0} = 0$. Then, $\mathbf{w}_{1,t} = \mathbf{w}_{2,t} = \mathbf{w}_t$ and $a_{1,t} = a_{2,t} = a_t$ is a solution to the gradient flow dynamics (GF), where $\mathbf{w}_t$ and $a_t$ satifies the following ODE:*

$$\begin{cases} \dot{\mathbf{w}}_t = \frac{1}{n} \sum_{i=1}^n e^{-y_i f(\boldsymbol{\theta}_t; \mathbf{x}_i)} (1 + c_L a) y_i \mathbf{x}_i \\ \dot{a}_t = \frac{1}{n} \sum_{i=1}^n e^{-y_i f(\boldsymbol{\theta}_t; \mathbf{x}_i)} c_L y_i \mathbf{x}_i^\top \mathbf{w}_t \end{cases} \tag{56}$$

*initialized at $(\mathbf{w}_0, a_0) = (\mathbf{0}, 0)$, with $c_L := (1 + \alpha_L)/2$. Furthermore, $(\mathbf{w}_t, a_t)$ is the limit of the GD dynamics (GD) with respect to (11), as the step size goes to $0$:*

$$(\mathbf{w}_t, a_t) = \lim_{\eta \to 0} (\mathbf{w}_{\eta t}, a_{\eta t}).$$

*Proof of Lemma E.1.* **Step 1.** We show that the GD dynamics (GD) is symmetric.

We can prove this by induction. We use the following notation for GD: $\boldsymbol{\theta}_t = (a_{1,t}, a_{2,t}, \mathbf{w}_{1,t}, \mathbf{w}_{2,t})$. For $t = 0$, we have $\mathbf{w}_{1,0} = \mathbf{w}_{2,0} = \vec{0}$ and $a_{1,0} = a_{2,0} = 0$. Assume that $\mathbf{w}_{1,t} = \mathbf{w}_{2,t}$ and $a_{1,t} = a_{2,t}$ for some $t$. Under these induction assumptions,

$$\begin{aligned} y_i f(\boldsymbol{\theta}_t; \mathbf{x}_i) &= y_i \mathbf{w}_{1,t}^\top \mathbf{x}_i + y_i \mathbf{w}_{2,t}^\top \mathbf{x}_i + y_i a_{1,t} \varphi(\mathbf{w}_{1,t}^\top \mathbf{x}_i) - y_i a_{2,t} \varphi(-\mathbf{w}_{2,t}^\top \mathbf{x}_i) \\ &= 2 y_i \mathbf{w}_{1,t}^\top \mathbf{x}_i + y_i (1 + \alpha_L) a_{1,t} \varphi(\mathbf{w}_{1,t}^\top \mathbf{x}_i) \qquad \text{Since } \varphi(x) - \varphi(-x) = (1 + \alpha_L) x \\ &= y_{i+n/2} f(\boldsymbol{\theta}_t; \mathbf{x}_{i+n/2}), \qquad \text{By assumption 4} \end{aligned}$$

for $i = 1, \ldots, n/2$. Then, we have

$$\begin{aligned} \nabla_{\mathbf{w}_1} \mathcal{L}(\boldsymbol{\theta}_t) &= \frac{1}{n} \sum_{i=1}^n e^{-y_i f_i(\boldsymbol{\theta}_t)} (1 + c_L a_{1,t} \varphi'(\mathbf{w}_{1,t}^\top \mathbf{x}_i)) y_i \mathbf{x}_i \\ &= \frac{1}{n} \sum_{i=1}^n e^{-f_i(\boldsymbol{\theta}_{t+n/2})} (1 + c_L a_{2,t} \varphi'(-\mathbf{w}_{2,t}^\top \mathbf{x}_{i+n/2})) y_{i+n/2} \mathbf{x}_{i+n/2} \\ &\qquad \text{By the symmetry of the dataset and the paremeter} \\ &= \nabla_{\mathbf{w}_2} \mathcal{L}(\boldsymbol{\theta}_t). \end{aligned}$$

Therefore, we have $\mathbf{w}_{1,t+1} = \mathbf{w}_{2,t+1}$ for all $t$. Similarly, we can show that $a_{1,t+1} = a_{2,t+1}$ for all $t$.

**Step 2.** We show that the limit solution of the GD dynamics is the solution to (56).

Since the GD dynamics is symmetric, we can treat $(a_{1,t}, \mathbf{w}_{1,t})$ as the result of Euler method with time step $\eta$ on the ODE (56). Since the ODE is locally Lipschitz continuous, there exists a unique solution $(a_t, \mathbf{w}_t)$ to the ODE (56), and the limit of results of Euler method as $\eta$ goes to zero is the solution. At last, it's easy to verify that $(a_t, a_t, \mathbf{w}_t, \mathbf{w}_t)$ satifies the (GF) for the model in (11) with respect to the zero initialization.

This completes the proof of Lemma E.1. $\qquad\square$

### E.6. Loss Convergence

Therefore, we can focus on the ODE (56). We will show that the loss is of rate $O(1/t)$ and the parameter norm $\|\boldsymbol{\theta}_t\|$ is of rate $O(\sqrt{\log t})$. The main trick we use here is the balancing of the parameters.

**Lemma E.2** (Parameter balancing). *Consider the dynamic of the parameters in (56) with initialization $\mathbf{w}_0 = \vec{0}$ and $a_0 = 0$. Then we have*

$$\left(a_t + \frac{1}{c_L}\right)^2 = \|\mathbf{w}_t\|^2 + \frac{1}{c_L^2}. \tag{57}$$

*Proof.* Note that

$$\frac{d}{dt}\left(a_t + \frac{1}{c_L}\right)^2 = 2\dot{a}_t\left(a_t + \frac{1}{c_L}\right)$$

$$= \frac{1}{n} \sum_{i=1}^{n} e^{-y_i f(\boldsymbol{\theta}_t; \mathbf{x}_i)} y_i \mathbf{x}_i^\top \mathbf{w}_t \left( 2c_L a_t + 2 \right)$$

$$= 2\langle \dot{\mathbf{w}}_t, \mathbf{w}_t \rangle = \frac{d}{dt} \|\mathbf{w}_t\|_2^2.$$

Therefore, we have

$$\left( a_t + \frac{1}{c_L} \right)^2 - \|\mathbf{w}_t\|^2 = \left( a_0 + \frac{1}{c_L} \right)^2 - \|\mathbf{w}_0\|^2 = \frac{1}{c_L^2}.$$

This completes the proof of Lemma E.2. □

Then, we can show that $a_t > 0$ for all $t$.

**Lemma E.3** (Positiveness of $a_t$). *Consider the dynamic of the parameters in* (56) *with initialization* $\mathbf{w}_0 = \vec{0}$ *and* $a_0 = 0$. *We have* $a_t \geq 0$ *for all* $t$.

*Proof.* Note that

$$\left( a_t + \frac{1}{c_L} \right)^2 = \|\mathbf{w}_t\|^2 + \frac{1}{c_L^2} \geq \frac{1}{c_L^2} \implies a_t \leq -\frac{2}{c_L} \text{ or } a_t \geq 0.$$

if there exits $T$ such that $a_t < -2/c_L$. Due to the continuity of $a_t$ and $a_0 = 0$, there exists $0 < T' < T$, such that $a(T') = -1/c_L$. Then, it contradicts with the condition that $a_t \leq -2/c_L$ or $a_t \geq 0$. This completes the proof of Lemma E.3. □

Now we can already write $a_t$ as a function of $\mathbf{w}_t$, i.e.,

$$a_t = \sqrt{\|\mathbf{w}_t\|^2 + \frac{1}{c_L^2}} - \frac{1}{c_L}.$$

We will show that the the loss is of rate $O(1/t)$.

**Lemma E.4** (Loss upper bound). *Given the dataset satisfying Assumption* 4 *and the model in* (11) *with initial condition* $\mathbf{w}_{1,0} = \mathbf{w}_{2,0} = \vec{0}$ *and* $a_{1,0} = a_{2,0} = 0$. *Then we must have*

$$\mathcal{L}(\boldsymbol{\theta}_T) \leq \frac{1 + \log^2(T)/4\gamma_*^2}{T}.$$

*Proof.* Let $\mathbf{w}^c = \beta \mathbf{w}_*$. Note that

$$\frac{\mathrm{d}}{\mathrm{d}t} \|\mathbf{w}_t - \mathbf{w}^c\|^2 = 2\langle \dot{\mathbf{w}}_t, \mathbf{w}_t - \mathbf{w}^c \rangle$$

$$= \frac{2}{n} \sum_{i=1}^{n} e^{-y_i f(\boldsymbol{\theta}_t; \mathbf{x}_i)} (1 + c_L a) y_i \mathbf{x}_i^\top (\mathbf{w}_t - \mathbf{w}^c)$$

$$= \frac{2}{n} \sum_{i=1}^{n} e^{-y_i f(\boldsymbol{\theta}_t; \mathbf{x}_i)} \left( \frac{1}{2} y_i f(\boldsymbol{\theta}; \mathbf{x}_i) - (1 + c_L a) \beta y_i \mathbf{x}_i^\top \mathbf{w}_* \right) \qquad \text{By } f(\boldsymbol{\theta}; \mathbf{x}) = (2 + 2c_L a) \mathbf{w}^\top \mathbf{x}$$

$$= \frac{1}{n} \sum_{i=1}^{n} e^{-y_i f(\boldsymbol{\theta}_t; \mathbf{x}_i)} \left( y_i f(\boldsymbol{\theta}; \mathbf{x}_i) - 2(1 + c_L a) \beta y_i \mathbf{x}_i^\top \mathbf{w}_* \right)$$

$$\leq \frac{1}{n} \sum_{i=1}^{n} e^{-y_i f(\boldsymbol{\theta}_t; \mathbf{x}_i)} \left( y_i f(\boldsymbol{\theta}; \mathbf{x}_i) - 2\beta \gamma_* \right) \qquad \text{by } y_i \mathbf{x}_i^\top \mathbf{w}_* \geq \gamma_* \text{ and } a \geq 0$$

$$= \frac{1}{n} \sum_{i=1}^{n} -e^{-y_i f(\boldsymbol{\theta}_t; \mathbf{x}_i)} \left( 2\beta \gamma_* - y_i f(\boldsymbol{\theta}; \mathbf{x}_i) \right)$$

$$\leq e^{-2\beta\gamma_*} - \mathcal{L}(\boldsymbol{\theta}_t). \qquad \text{By convexity of } e^{-x}$$

This implies that

$$\|\mathbf{w}_T - \mathbf{w}^c\|^2 + \int_0^T \mathcal{L}(\boldsymbol{\theta}_t)\, \mathrm{d}t \le T e^{-2\beta\gamma_*} + \|\mathbf{w}_0 - \mathbf{w}^c\|^2.$$

Note that $\mathcal{L}(\boldsymbol{\theta}_t)$ is decreasing, $\mathbf{w}_0 = \vec{0}$ and we can set $\beta = \log(T)/(2\gamma^*)$. We have

$$T\mathcal{L}(\boldsymbol{\theta}_t) \le 1 + \log^2(T)/4\gamma_*^2.$$

This completes the proof of Lemma E.4. □

### E.7. Parameter Norm Bounds

Now we can use the loss bound to achieve a bound for the parameter norm.

**Lemma E.5** (Parameter norm bound). *Under the same assumptions of Lemma E.4, when*

$$t \ge \max\{16, (4/\gamma_* \log(4/\gamma_*))^4\},$$

*we have*

$$\|\mathbf{w}_t\|^2 c_L^2 + 1 \ge \frac{\log t}{16}. \tag{58}$$

*Proof.* Applying the balancing equation (57), we have

$$\begin{aligned}
\mathcal{L}(\boldsymbol{\theta}_t) &= \frac{1}{n}\sum_{i=1}^n e^{-y_i f(\boldsymbol{\theta}_t; \mathbf{x}_i)} \\
&= \frac{1}{n}\sum_{i=1}^n e^{-y_i \mathbf{x}_i^\top \mathbf{w}_t \cdot 2\sqrt{\|\mathbf{w}_t\|^2 c_L^2 + 1}} \\
&\ge e^{-2\|\mathbf{w}_t\|\sqrt{\|\mathbf{w}_t\|^2 c_L^2 + 1}} \qquad\qquad \text{By } \|\mathbf{x}_i\| \le 1\,.
\end{aligned}$$

Combining this with the Loss upper bound in Lemma E.4, we have

$$\begin{aligned}
2\|\mathbf{w}_t\|\sqrt{\|\mathbf{w}_t\|^2 c_L^2 + 1} &\ge -\log(\mathcal{L}(\boldsymbol{\theta}_t)) \\
&\ge \log(t) - \log\left(1 + \frac{\log^2 t}{4\gamma_*^2}\right). \tag{59}
\end{aligned}$$

Since $t \ge \max\{16, (4/\gamma_* \log(4/\gamma_*))^4\}$, we have $1 \le \log^2 t/4\gamma_*^2$. Therefore, we have

$$\begin{aligned}
\log\left(1 + \frac{\log^2 t}{4\gamma_*^2}\right) &\le \log\left(\frac{2\log^2 t}{4\gamma_*^2}\right) \le \log 2 + 2\log\left(\frac{\log t}{2\gamma_*}\right) \\
&\le \log 2 + 2\log(t^{\frac{1}{4}}) \qquad\qquad \text{By Lemma G.4} \\
&= \log 2 + \frac{1}{2}\log t \le \frac{3}{4}\log t. \qquad\qquad \text{By } t \ge 16
\end{aligned}$$

Plugging in this into (59), we have

$$\frac{1}{4}\log t \le 2\|\mathbf{w}_t\|\sqrt{\|\mathbf{w}_t\|^2 c_L^2 + 1} \le \frac{2}{c_L}\left(\|\mathbf{w}_t\|^2 c_L^2 + 1\right).$$

At last, we use that $c_L > 1/2$ and we have

$$\|\mathbf{w}_t\|^2 c_L^2 + 1 \ge \frac{1}{16}\log t.$$

This completes the proof of Lemma E.5. □

Now we turn back to the analysis of the loss and try to extract the upper bound of the norm. Now we use $t_0$ to denote $\max\{16, (4/\gamma_* \log(4/\gamma_*))^4\}$.

**Lemma E.6** (Parameter norm upper bound)**.** *Under the same assumptions of Lemma E.4, let* $t_0 :=$ $\max\{16, (4/\gamma_* \log(4/\gamma_*))^4\}$. *Then we have*

$$\|\mathbf{w}_t\|_2 \leq \left(\frac{4}{\gamma_*} + 1\right)\sqrt{\log t} + \|\mathbf{w}_{t_0}\|_2, \quad \forall t \geq t_0. \tag{60}$$

*Proof.* Just like the proof of Lemma E.4, we let $\mathbf{w}^c = \beta \mathbf{w}_*$. Note that

$$
\begin{aligned}
\frac{d}{dt}\|\mathbf{w}_t - \mathbf{w}^c\|^2 &= 2\langle \dot{\mathbf{w}}_t, \mathbf{w}_t - \mathbf{w}^c \rangle \\
&= \frac{2}{n}\sum_{i=1}^{n} e^{-y_i f(\boldsymbol{\theta}_t; \mathbf{x}_i)}(1 + c_L a)y_i \mathbf{x}_i^\top(\mathbf{w}_t - \mathbf{w}^c) \\
&= \frac{2}{n}\sum_{i=1}^{n} e^{-y_i f(\boldsymbol{\theta}_t; \mathbf{x}_i)}\left(\frac{1}{2}y_i f(\boldsymbol{\theta}; \mathbf{x}_i) - (1 + c_L a)\beta y_i \mathbf{x}_i^\top \mathbf{w}_*\right) && \text{By } f(\boldsymbol{\theta}; \mathbf{x}) = (2 + 2c_L a)\mathbf{w}^\top \mathbf{x} \\
&= \frac{1}{n}\sum_{i=1}^{n} e^{-y_i f(\boldsymbol{\theta}_t; \mathbf{x}_i)}\left(y_i f(\boldsymbol{\theta}; \mathbf{x}_i) - 2\sqrt{\|\mathbf{w}_t\|^2 c_L^2 + 1}\beta y_i \mathbf{x}_i^\top \mathbf{w}_*\right) && \text{By (57)} \\
&\leq \frac{1}{n}\sum_{i=1}^{n} e^{-y_i f(\boldsymbol{\theta}_t; \mathbf{x}_i)}\left(y_i f(\boldsymbol{\theta}; \mathbf{x}_i) - \frac{1}{2}\sqrt{\log t}\beta\gamma_*\right) && \text{By (58)} \\
&\leq e^{-\beta\gamma_*\sqrt{\log t}/2} - \mathcal{L}(\boldsymbol{\theta}_t). && \text{By convexity of } e^{-x} \\
&\leq e^{-\beta\gamma_*\sqrt{\log t}/2}.
\end{aligned}
$$

We integral the above inequality from $t_0$ to $t$ and we have

$$\|\mathbf{w}_T - \mathbf{w}^c\|^2 \leq \|\mathbf{w}_{t_0} - \mathbf{w}^c\|^2 + \int_{t_0}^{t} e^{-\beta\gamma_*\sqrt{\log t}/2}dt.$$

Here we set $\beta = 2\sqrt{\log T}/\gamma_*$, then we have

$$
\begin{aligned}
\|\mathbf{w}_T - \mathbf{w}^c\|^2 &\leq \|\mathbf{w}_{t_0} - \mathbf{w}^c\|^2 + \int_{t_0}^{T} e^{-\sqrt{\log T \log t}}dt \\
&\leq \|\mathbf{w}_{t_0} - \mathbf{w}^c\|^2 + \int_{t_0}^{T} \frac{1}{t}dt && \text{By } \log T \log t \geq \log^2 t \\
&\leq \|\mathbf{w}_{t_0} - \mathbf{w}^c\|^2 + \log T.
\end{aligned}
$$

Hence,

$$
\begin{aligned}
\|\mathbf{w}_t\| &\leq \|\mathbf{w}_T - \mathbf{w}^c\| + \|\mathbf{w}^c\| \\
&\leq \|\mathbf{w}_{t_0} - \mathbf{w}^c\| + \sqrt{\log T} + \|\mathbf{w}^c\| \\
&\leq \|\mathbf{w}_{t_0}\| + 2\|\mathbf{w}^c\| + \sqrt{\log T} \\
&= \|\mathbf{w}_{t_0}\| + \frac{4}{\gamma_*}\sqrt{\log T} + \sqrt{\log T}.
\end{aligned}
$$

This completes the proof of Lemma E.6. □

Now we can give a theorem about the margin.

### E.8. Proof of Theorem 5.4

*Proof of Theorem 5.4.* By Lemma E.4 and Lemma E.6, we have

$$\frac{\log 1/\mathcal{L}(\boldsymbol{\theta}_t)}{\|\boldsymbol{\theta}_t\|_2} \geq \frac{\log t - \log\left(1 + \frac{\log^2 t}{4\gamma_*^2}\right)}{\|\mathbf{w}_{t_0}\| + (1 + 4/\gamma_*)\sqrt{\log t}} \to \infty, \text{ as } t \to \infty.$$

Therefore, there exists $T$ such that the initial bound is achieved. Now we complete the proof of Theorem 5.4. □

# F. Proofs for Section 6

In this section, we will provide the proofs omitted in Section 6. Recall the gradient descent dynamics:

$$\boldsymbol{\theta}_{t+1} = \boldsymbol{\theta}_t - \eta \nabla \mathcal{L}(\boldsymbol{\theta}_t), \quad \text{for } t \geq 0. \tag{61}$$

In order to analyze the convergence of the GD, we need to focus on the modified loss. We define

$$\mathcal{G}(\boldsymbol{\theta}_t) := e^{\mathsf{p}_a(\|\boldsymbol{\theta}_t\|)} \mathcal{L}(\boldsymbol{\theta}_t), \tag{62}$$

where we recall the definition of $\mathsf{p}_a$ from Eq. (12). Note that we assume $\deg(\mathsf{p}_a) \geq 1$ if $M \geq 2$. For notational simplicity, we denote

$$\mathcal{L}_t := \mathcal{L}(\boldsymbol{\theta}_t), \nabla \mathcal{L}_t := \nabla_{\boldsymbol{\theta}} \mathcal{L}(\boldsymbol{\theta}_t), \quad \mathcal{G}_t := \mathcal{G}(\boldsymbol{\theta}_t), \nabla \mathcal{G}_t := \nabla_{\boldsymbol{\theta}} \mathcal{G}(\boldsymbol{\theta}_t).$$

We recall the definitions of the link functions

$$\phi(x) = \log \frac{1}{nx}, \quad \Phi(x) = \log \phi(x) - \frac{2}{\phi(x)}, \tag{63}$$

and the modified margin:

$$\gamma^{\mathsf{GD}}(\boldsymbol{\theta}) := \frac{e^{\Phi(\mathcal{G}(\boldsymbol{\theta}))}}{\|\boldsymbol{\theta}\|^M}. \tag{64}$$

We also recall $\gamma^{\mathsf{GF}}$, the modified margin defined for gradient flow:

$$\gamma^{\mathsf{GF}}(\boldsymbol{\theta}) := \frac{1}{\|\boldsymbol{\theta}\|^M} \left( \log \frac{1}{n\mathcal{L}(\boldsymbol{\theta})} - \mathsf{p}_a(\|\boldsymbol{\theta}\|) \right) = \frac{\phi(\mathcal{G}(\boldsymbol{\theta}))}{\|\boldsymbol{\theta}\|^M}. \tag{65}$$

Finally, unless otherwise stated, the constants $B$ and $B_i$'s in this section only depend on $(M, \mathsf{p}, \mathsf{q}, A)$ and $(\mathsf{r}, \mathsf{s})$ in Assumptions 3 and 5.

## F.1. Preliminary Results

We collect below some important properties of $\mathsf{p}_a$ that will be used in this section.

**Lemma F.1.** *For all $M \geq 1$, the function $\mathsf{p}_a : [0, +\infty) \to [0, +\infty)$ is increasing, and satisfies the followings:*

*(i)* $x\mathsf{p}_a'(x) + \mathsf{p}'(x) \leq M\mathsf{p}_a(x)$.

*(ii) If $M \geq 2$, there exists a constant $B_1$, such that for all $x \geq 0$:*

$$\mathsf{p}_a(x) \leq B_1 \left( x^{M-1} + 1 \right), \quad \mathsf{p}_a'(x) \leq B_1 \cdot \left( xI\{x \leq 1\} + x^{M-2}I\{x > 1\} \right), \tag{66}$$

$$\mathsf{p}_a''(x) \leq B_1 \cdot \left( I\{x \leq 1\} + x^{M-3}I\{x > 1\} \right). \tag{67}$$

*Proof.* The conclusion of part $(i)$ for $M = 1$ can be directly verified, since $\mathsf{p}_a(x) = a_1/(M - 1/2)$. From now on, we assume $M \geq 2$. For part $(i)$, recall that $\mathsf{p}(x) = \sum_{i=0}^M a_i x^i$ and

$$\mathsf{p}_a(x) := \begin{cases} \sum_{i=1}^{M-1} \frac{(i+1)a_{i+1}}{M-i} x^i + \frac{a_1}{M-1/2}, & x \geq 1, \\ \sum_{i=2}^{M-1} \frac{(i+1)a_{i+1}}{M-i} x^i + \frac{2a_2}{M-1} \frac{x^2+1}{2} + \frac{a_1}{M-1/2}, & 0 \leq x < 1. \end{cases}$$

Hence,

$$\mathsf{p}_a'(x) + \mathsf{p}'(x) - M\mathsf{p}_a(x)$$

$$= \begin{cases} \sum_{i=1}^{M-1} \left[ \frac{i(i+1)a_{i+1}}{M-i} x^i + (i+1)a_{i+1}x^i - \frac{Ma_{i+1}}{M-i} x^i \right] + a_1 - \frac{Ma_1}{M-1/2}, & x \geq 1 \\ \sum_{i=2}^{M-1} \left[ \frac{i(i+1)a_{i+1}}{M-i} x^i + (i+1)a_{i+1}x^i - \frac{Ma_{i+1}}{M-i} x^i \right] \\ \quad + \frac{2a_2}{M-1} x^2 + 2a_2 x - \frac{2Ma_2}{M-1} \frac{x^2+1}{2} + a_1 - \frac{Ma_1}{M-1/2}, & 0 \leq x < 1 \end{cases}$$

$$= \begin{cases} -\dfrac{a_1}{M-1/2}, & x \geq 1 \\ -2a_2\dfrac{(x-1)^2}{2} - \dfrac{a_1}{M-1/2}, & 0 \leq x < 1 \end{cases}$$

$$\leq -\dfrac{a_1}{M-1/2} \leq 0.$$

Part $(ii)$ can be verified via direct calculation. □

The next lemma characterizes the derivatives of functions related to $\phi(x)$.

**Lemma F.2** (Derivatives of functions related to $\phi(x)$). *Given the function $\phi(x) = \log(1/(nx))$, we have*

$$\left( \log\phi(x) \right)' = -\frac{1}{x\phi(x)}, \quad \left( \frac{1}{\phi(x)} \right)' = \frac{1}{\phi(x)^2 x}, \quad \text{and} \quad \left( \frac{1}{\phi(x)^2} \right)' = \frac{2}{\phi(x)^3 x}.$$

**Lemma F.3** (Convexity of $\Phi$). *For $x \in (0, 1/ne^2)$, $\Phi(x)$ is convex.*

*Proof.* By direct calculation, we get

$$\Phi'(x) = \left( \frac{1}{\phi(x)} + \frac{2}{\phi(x)^2} \right) \phi'(x),$$

and

$$\begin{aligned} \Phi''(x) &= \left( \frac{1}{\phi(x)} + \frac{2}{\phi(x)^2} \right) \phi''(x) + \left( -\frac{1}{\phi(x)^2} - \frac{4}{\phi(x)^3} \right) \phi'(x)^2 \\ &\overset{(i)}{=} \frac{1}{x^2} \left( \frac{1}{\phi(x)} + \frac{1}{\phi(x)^2} - \frac{4}{\phi(x)^3} \right) \\ &\overset{(ii)}{=} \frac{1}{x^2} \left( \frac{1}{\phi(x)} - \frac{1}{\phi(x)^2} + 2 \left( \frac{1}{\phi(x)^2} - \frac{2}{\phi(x)^3} \right) \right) > 0, \end{aligned}$$

where $(i)$ follows from the fact that $\phi''(x) = 1/x^2 = \phi'(x)^2$, $(ii)$ is due to $\phi(x) > 2$ whenever $x < 1/ne^2$. This completes the proof. □

**Lemma F.4.** *For any $\boldsymbol{\theta}$ satisfying $\mathcal{G}(\boldsymbol{\theta}) < 1/n$, we have $\gamma^{\mathsf{GF}}(\boldsymbol{\theta}) > \gamma^{\mathsf{GD}}(\boldsymbol{\theta})$.*

*Proof.* Recall that

$$\gamma^{\mathsf{GD}}(\boldsymbol{\theta}) = \frac{\exp(\Phi(\mathcal{G}(\boldsymbol{\theta})))}{\|\boldsymbol{\theta}\|^M}, \quad \gamma^{\mathsf{GF}}(\boldsymbol{\theta}) = \frac{\phi(\mathcal{G}(\boldsymbol{\theta}))}{\|\boldsymbol{\theta}\|^M}.$$

The result follows directly from the fact that $\Phi < \log\phi$ when $n\mathcal{G}(\boldsymbol{\theta}) < 1$. □

We next give some a priori estimates on $f$ and $\mathsf{p}_a$.

**Lemma F.5** (Homogeneous constant). *Under Assumption 5, there exists some constant $B_2$, such that for any $\boldsymbol{\theta}$:*

$$\|\nabla f(\boldsymbol{\theta})\| \leq B_2 \left( \|\boldsymbol{\theta}\|^{M-1} + 1 \right), \quad \|\nabla^2 f(\boldsymbol{\theta})\| \leq B_2 \left( \|\boldsymbol{\theta}\|^{(M-2)+} + 1 \right). \tag{68}$$

**Lemma F.6** (Gradient and Hessian bound for $\mathcal{L}$). *Under Assumption 5, there exists some constant $B_3$:*

$$\|\nabla \mathcal{L}(\boldsymbol{\theta})\| \leq B_3 \mathcal{L}(\boldsymbol{\theta}) \left( \|\boldsymbol{\theta}\|^{M-1} + 1 \right), \quad \|\nabla^2 \mathcal{L}(\boldsymbol{\theta})\| \leq B_3 \mathcal{L}(\boldsymbol{\theta}) \left( \|\boldsymbol{\theta}\|^{2M-2} + 1 \right). \tag{69}$$

*Proof.* According to Lemma F.5, we obtain that

$$\|\nabla \mathcal{L}(\boldsymbol{\theta})\| = \left\| \frac{1}{n} \sum_{i=1}^{n} e^{-\bar{f}_i(\boldsymbol{\theta})} \nabla \bar{f}_i(\boldsymbol{\theta}) \right\| \leq B_2 \mathcal{L}(\boldsymbol{\theta}) \left( \|\boldsymbol{\theta}\|^{M-1} + 1 \right).$$

$$\|\nabla^2 \mathcal{L}(\boldsymbol{\theta})\| = \left\| \frac{1}{n} \sum_{i=1}^{n} e^{-\bar{f}_i(\boldsymbol{\theta})} (\nabla \bar{f}_i(\boldsymbol{\theta}) \nabla \bar{f}_i(\boldsymbol{\theta})^\top - \nabla^2 \bar{f}_i(\boldsymbol{\theta}) \right\|$$

$$\leq \mathcal{L}(\boldsymbol{\theta}) \left( B_2^2 (\|\boldsymbol{\theta}\|^{M-1} + 1)^2 + B_2(\|\boldsymbol{\theta}\|^{(M-2)_+} + 1) \right).$$

Using AM-GM inequality, the desired result follows immediately. □

**Lemma F.7** (Gradient and Hessian bound for $\mathcal{G}$). *Under Assumption 5, there exists a constant $B_4$ such that for any $\boldsymbol{\theta}$:*

$$\|\nabla \mathcal{G}(\boldsymbol{\theta})\| \leq B_4 \mathcal{G}(\boldsymbol{\theta}) \left( \|\boldsymbol{\theta}\|^{M-1} + 1 \right), \quad \|\nabla^2 \mathcal{G}(\boldsymbol{\theta})\| \leq B_4 \mathcal{G}(\boldsymbol{\theta}) \left( \|\boldsymbol{\theta}\|^{2M-2} + 1 \right). \tag{70}$$

*Proof.* We first upper bound $\|\nabla \mathcal{G}(\boldsymbol{\theta})\|$. By Lemma F.5 and Lemma F.6, we have

$$\|\nabla \mathcal{G}(\boldsymbol{\theta})\| = \left\| e^{\mathsf{p}_a(\|\boldsymbol{\theta}\|)} \nabla \mathcal{L}(\boldsymbol{\theta}) + \mathcal{L}(\boldsymbol{\theta}) e^{\mathsf{p}_a(\|\boldsymbol{\theta}\|)} \mathsf{p}_a'(\|\boldsymbol{\theta}\|) \tilde{\boldsymbol{\theta}} \right\|$$

$$\leq e^{\mathsf{p}_a(\|\boldsymbol{\theta}\|)} \|\nabla \mathcal{L}(\boldsymbol{\theta})\| + \mathcal{G}(\boldsymbol{\theta}) \mathsf{p}_a'(\|\boldsymbol{\theta}\|)$$

$$\leq B_3 \mathcal{G}(\boldsymbol{\theta}) \left( \|\boldsymbol{\theta}\|^{M-1} + 1 \right) + B_1 \mathcal{G}(\boldsymbol{\theta}) \left( \|\boldsymbol{\theta}\|^{(M-2)_+} + 1 \right)$$

$$\leq B_4 \mathcal{G}(\boldsymbol{\theta}) \left( \|\boldsymbol{\theta}\|^{M-1} + 1 \right),$$

where $B_4$ only depends on $B_3$ and $B_1$. For $\|\nabla^2 \mathcal{G}(\boldsymbol{\theta})\|$, note that

$$\|\nabla^2 \mathcal{G}(\boldsymbol{\theta})\| = \left\| e^{\mathsf{p}_a(\|\boldsymbol{\theta}\|)} \nabla^2 \mathcal{L}(\boldsymbol{\theta}) + 2 e^{\mathsf{p}_a(\|\boldsymbol{\theta}\|)} \mathsf{p}_a'(\|\boldsymbol{\theta}\|) \nabla \mathcal{L}(\boldsymbol{\theta}) \tilde{\boldsymbol{\theta}}^\top \right.$$

$$+ e^{\mathsf{p}_a(\|\boldsymbol{\theta}\|)} \mathcal{L}(\boldsymbol{\theta}) \left( \mathsf{p}_a''(\|\boldsymbol{\theta}\|) + \mathsf{p}_a'(\|\boldsymbol{\theta}\|)^2 \right) \tilde{\boldsymbol{\theta}} \tilde{\boldsymbol{\theta}}^\top$$

$$+ \left. e^{\mathsf{p}_a(\|\boldsymbol{\theta}\|)} \mathcal{L}(\boldsymbol{\theta}) \mathsf{p}_a'(\|\boldsymbol{\theta}\|) \left( I - \tilde{\boldsymbol{\theta}} \tilde{\boldsymbol{\theta}}^\top \right) / \|\boldsymbol{\theta}\| \right\|$$

$$\leq e^{\mathsf{p}_a(\|\boldsymbol{\theta}\|)} \left\| \nabla^2 \mathcal{L}(\boldsymbol{\theta}) \right\| + 2 e^{\mathsf{p}_a(\|\boldsymbol{\theta}\|)} \mathsf{p}_a'(\|\boldsymbol{\theta}\|) \left\| \nabla \mathcal{L}(\boldsymbol{\theta}) \right\|$$

$$+ \mathcal{G}(\boldsymbol{\theta}) \left( \mathsf{p}_a''(\|\boldsymbol{\theta}\|) + \mathsf{p}_a'(\|\boldsymbol{\theta}\|)^2 \right) + \mathcal{G}(\boldsymbol{\theta}) \frac{\mathsf{p}_a'(\|\boldsymbol{\theta}\|)}{\|\boldsymbol{\theta}\|}$$

$$\leq B_3 \mathcal{G}(\boldsymbol{\theta}) \left( \|\boldsymbol{\theta}\|^{2M-2} + 1 \right) + 2 B_1 B_3 \mathcal{G}(\boldsymbol{\theta}) \left( \|\boldsymbol{\theta}\|^{M-1} + 1 \right) \left( \|\boldsymbol{\theta}\|^{(M-2)_+} + 1 \right)$$

$$+ \mathcal{G}(\boldsymbol{\theta}) \left( B_1 \left( \|\boldsymbol{\theta}\|^{(M-3)_+} + 1 \right) + B_1^2 \left( \|\boldsymbol{\theta}\|^{(M-2)_+} + 1 \right)^2 \right) + B_1 \mathcal{G}(\boldsymbol{\theta}) \left( \|\boldsymbol{\theta}\|^{(M-3)_+} + 1 \right).$$

Hence, there exists a constant $B_4$ only depending on $B_1$ and $B_3$, such that

$$\|\nabla^2 \mathcal{G}(\boldsymbol{\theta})\| \leq B_4 \mathcal{G}(\boldsymbol{\theta}) \left( \|\boldsymbol{\theta}\|^{2M-2} + 1 \right). \tag{71}$$

This completes the proof. □

We introduce a crucial quantity, $v_t$, defined as the inner product of the gradient and the negative weight vector:

$$v_t := \langle \nabla \mathcal{L}(\boldsymbol{\theta}_t), -\boldsymbol{\theta}_t \rangle. \tag{72}$$

We have the following bound for $v_t$, recall that $\rho_t = \|\boldsymbol{\theta}_t\|$.

**Lemma F.8** (Bound of $v_t$). *Under Assumption 5, for any $\boldsymbol{\theta}_t$ satisfying $\mathcal{L}(\boldsymbol{\theta}_t) < (1/n) e^{-\mathsf{p}_a(\rho_t)}$, we have*

$$M\mathcal{L}(\boldsymbol{\theta}_t) \log \frac{1}{\mathcal{L}(\boldsymbol{\theta}_t)} + \mathsf{p}'(\rho_t) \mathcal{L}(\boldsymbol{\theta}_t) \geq v_t \geq M\mathcal{L}(\boldsymbol{\theta}_t) \log \frac{1}{n\mathcal{L}(\boldsymbol{\theta}_t)} - \mathsf{p}'(\rho_t) \mathcal{L}(\boldsymbol{\theta}_t) > 0.$$

*Proof.* We first prove the first "$\geq$". Note that

$$v_t = \langle -\nabla \mathcal{L}(\boldsymbol{\theta}_t), \boldsymbol{\theta}_t \rangle = \frac{1}{n} \sum_{i=1}^{n} e^{-\bar{f}_i(\boldsymbol{\theta}_t)} \langle \nabla \bar{f}_i(\boldsymbol{\theta}_t), \boldsymbol{\theta}_t \rangle \qquad \text{Since } -\nabla \mathcal{L}_t = \sum_{i=1}^{n} e^{-y_i f_i(t)} y_i \nabla f_i(t)$$

$$\leq \frac{1}{n} \sum_{i=1}^{n} e^{-\bar{f}_i(\boldsymbol{\theta}_t)} \left( M \bar{f}_i(\boldsymbol{\theta}_t) + \mathsf{p}'(\rho_t) \right) \qquad \text{By Assumption 1}$$

$$= M \frac{1}{n} \sum_{i=1}^{n} e^{-\bar{f}_i(\boldsymbol{\theta}_t)} \bar{f}_i(\boldsymbol{\theta}_t) + \mathsf{p}'(\rho_t)\mathcal{L}(\boldsymbol{\theta}_t)$$

$$\leq - M\mathcal{L}(\boldsymbol{\theta}_t) \log \mathcal{L}(\boldsymbol{\theta}_t) + \mathsf{p}'(\rho_t)\mathcal{L}(\boldsymbol{\theta}_t). \qquad \text{By Jensen's inequality}$$

For the second "$\geq$", by Lemma C.2, we have

$$\bar{f}_i(\boldsymbol{\theta}_t) \geq \bar{f}_{\min}(\boldsymbol{\theta}_t) \geq \log \frac{1}{n\mathcal{L}(\boldsymbol{\theta}_t)}. \tag{73}$$

Furthermore,

$$v_t = \langle -\nabla \mathcal{L}(\boldsymbol{\theta}_t), \boldsymbol{\theta}_t \rangle = \frac{1}{n} \sum_{i=1}^{n} e^{-\bar{f}_i(\boldsymbol{\theta}_t)} y_i \langle \nabla f_i(\boldsymbol{\theta}_t), \boldsymbol{\theta}_t \rangle \qquad \text{Since } -\nabla \mathcal{L}_t = \sum_{i=1}^{n} e^{-y_i f_i(t)} y_i \nabla f_i(t)$$

$$\geq M \cdot \frac{1}{n} \sum_{i=1}^{n} e^{-\bar{f}_i(\boldsymbol{\theta}_t)} \bar{f}_i(\boldsymbol{\theta}_t) - \frac{1}{n} \sum_{i=1}^{n} e^{-\bar{f}_i(\boldsymbol{\theta}_t)} \mathsf{p}'(\rho_t) \qquad \text{By Assumption 1}$$

$$= M \cdot \frac{1}{n} \sum_{i=1}^{n} e^{-\bar{f}_i(\boldsymbol{\theta}_t)} \bar{f}_i(\boldsymbol{\theta}_t) - \mathcal{L}(\boldsymbol{\theta}_t)\mathsf{p}'(\rho_t)$$

$$\geq M \cdot \frac{1}{n} \sum_{i=1}^{n} e^{-\bar{f}_i(\boldsymbol{\theta}_t)} \log \frac{1}{n\mathcal{L}(\boldsymbol{\theta}_t)} - \mathcal{L}(\boldsymbol{\theta}_t)\mathsf{p}'(\rho_t) \qquad \text{By Eq. (73)}$$

$$= M\mathcal{L}(\boldsymbol{\theta}_t) \log \frac{1}{n\mathcal{L}(\boldsymbol{\theta}_t)} - \mathsf{p}'(\rho_t)\mathcal{L}(\boldsymbol{\theta}_t).$$

Note that since $\rho_t \geq 0$, we have

$$\mathcal{L}(\boldsymbol{\theta}_t) < e^{-\mathsf{p}_a(\rho_t)}/n \iff M \log \frac{1}{n\mathcal{L}(\boldsymbol{\theta}_t)} > M\mathsf{p}_a(\rho_t).$$

By Lemma F.1, we have
$$M\mathsf{p}_a(x) - \mathsf{p}'(x) \geq x\mathsf{p}'_a(x) > 0.$$

As a consequence,
$$M \log \frac{1}{n\mathcal{L}(\boldsymbol{\theta}_t)} > M\mathsf{p}_a(\rho_t) \geq \mathsf{p}'(\rho_t) \implies v_t > 0.$$

This completes the proof of Lemma F.8. $\qquad\square$

## F.2. Margin Improvement: Proof of Theorem 6.1

This section is devoted to the proof of the margin improvement part of Theorem 6.1. We first establish the desired result under another set of assumptions in the following theorem, and then show that Assumption 5 implies this set of assumptions.

**Theorem F.9.** *Let $f(\boldsymbol{\theta}; \mathbf{x})$ be a twice-differentiable network (with respect to $\boldsymbol{\theta}$) satisfying Assumption 1 with $(M, \mathsf{p}, \mathsf{q})$, and that*

$$\left\| \nabla_{\boldsymbol{\theta}}^2 f(\boldsymbol{\theta}; \mathbf{x}) \right\| \leq A \left( \|\boldsymbol{\theta}\|^{(M-2)+} + 1 \right)$$

*for some constant $A > 0$. Further, assume that*

$$\mathcal{G}_s < \frac{1}{ne^2} \iff \mathcal{L}_s < \frac{1}{ne^2} \exp\left(-\mathsf{p}_a(\rho_s)\right). \tag{74}$$

*Recall $B_3$ and $B_4$ from Lemma F.6 and Lemma F.7, we define for $\eta, r > 0$:*

$$B_5(\eta, r) := \left( r + \eta\mathcal{G}_s B_3 e^{-\mathsf{p}_a(r)} \left( r^{M-1} + 1 \right) \right)^{2M-2} + 1,$$

*and for $t \geq s$:*

$$R_1(t) := B_3^2 \mathcal{L}_t (\rho_t^{M-1} + 1)^2, \quad R_2(t) := B_4 B_5(\eta, \rho_t) \mathcal{L}_t \phi(\mathcal{L}_t).$$

*Further, assume that*

$$\eta \sup_{\rho_t \geq \rho_s, \mathcal{G}_t \leq \mathcal{G}_s} R_1(t) \leq \frac{1}{2}, \quad \eta \sup_{\rho_t \geq \rho_s, \mathcal{G}_t \leq \mathcal{G}_s} R_2(t) \leq \frac{1}{2}.$$

*For all $t \geq s$, $t \in \mathbb{N}$, we interpolate between $\boldsymbol{\theta}_t$ and $\boldsymbol{\theta}_{t+1}$ by defining*

$$\boldsymbol{\theta}_{t+\alpha} = \boldsymbol{\theta}_t + \alpha \left( \boldsymbol{\theta}_{t+1} - \boldsymbol{\theta}_t \right) = \boldsymbol{\theta}_t - \alpha \eta \nabla \mathcal{L}(\boldsymbol{\theta}_t)$$

*for $\alpha \in [0, 1]$. Then, we have for all $t \geq s$ and $\alpha \in [0, 1]$, $v_t > 0$, and*

(1) $\gamma^{\mathsf{GF}}(\boldsymbol{\theta}_{t+\alpha}) > \gamma^{\mathsf{GD}}(\boldsymbol{\theta}_s)$.

(2) $2\eta\alpha v_t \leq \rho_{t+\alpha}^2 - \rho_t^2 \leq 2\eta\alpha v_t \left( 1 + \frac{\eta\alpha R_1(t)}{2M\phi(\mathcal{G}_t)} \right)$.

(3) $\mathcal{G}_{t+\alpha} - \mathcal{G}_t \leq -\alpha\eta \left( 1 - \alpha\eta R_2(t) \right) M \mathcal{G}_t \phi(\mathcal{G}_t) v_t^{-1} \|\nabla \mathcal{L}_t\|^2$.

(4) $\log \gamma^{\mathsf{GD}}(\boldsymbol{\theta}_{t+\alpha}) - \log \gamma^{\mathsf{GD}}(\boldsymbol{\theta}_t) \geq \frac{M \rho_t^2 \|\partial_\perp \mathcal{L}_t\|_2^2}{v_t^2} \log \frac{\rho_{t+\alpha}}{\rho_t}$.

To prove Theorem F.9, we use a similar strategy as the proof of Lemma E.8 in (Lyu & Li, 2020). To this end, it suffices to prove the following lemma (analogous to Lemma E.9 in (Lyu & Li, 2020)):

**Lemma F.10.** *Fix an integer $T \geq s$. Suppose that (1), (2), (3), (4) hold for any $t + \alpha \leq T$. Then if (1) holds for $(t, \alpha) \in \{T\} \times [0, A)$ for some $A \in (0, 1]$, then all of (1), (2), (3), (4) hold for $(t, \alpha) \in \{T\} \times [0, A]$.*

*Proof.* By our assumption, we know that $\mathcal{G}_t \leq \mathcal{G}_s < 1/ne^2$ for all $t \leq T$, hence Lemma F.8 implies $v_t > 0$. Further, we have $\rho_t \geq \rho_s$ and $\gamma^{\mathsf{GF}}(\boldsymbol{\theta}_t) \geq \gamma^{\mathsf{GF}}(\boldsymbol{\theta}_s)$ for all $t \leq T$.

Now we fix $t = T$. Since $\gamma^{\mathsf{GF}}(\boldsymbol{\theta}_{t+\alpha}) > \gamma^{\mathsf{GD}}(\boldsymbol{\theta}_s)$ for all $\alpha \in [0, A)$, by continuity of $\gamma^{\mathsf{GF}}$ we know that $\gamma^{\mathsf{GF}}(\boldsymbol{\theta}_{t+\alpha}) \geq \gamma^{\mathsf{GD}}(\boldsymbol{\theta}_s)$ for $\alpha = A$.

**Proof of (2) for $\alpha = A$.** By definition, we have

$$\rho_{t+\alpha}^2 - \rho_t^2 = 2\eta\alpha\langle \nabla \mathcal{L}_t, -\boldsymbol{\theta}_t \rangle + \eta^2 \alpha^2 \|\nabla \mathcal{L}_t\|^2$$
$$= 2\eta\alpha v_t + \eta^2 \alpha^2 \|\nabla \mathcal{L}_t\|^2 \geq 2\eta\alpha v_t > 0,$$

where the last inequality is due to Lemma F.8. According to Lemma F.6, we have

$$\|\nabla \mathcal{L}(\boldsymbol{\theta})\| \leq \mathcal{L}(\boldsymbol{\theta}) \cdot B_3 \left( \|\boldsymbol{\theta}\|^{M-1} + 1 \right),$$

namely

$$\|\nabla \mathcal{L}_t\| \leq B_3 \mathcal{L}_t \left( \rho_t^{M-1} + 1 \right).$$

Further, applying Lemma F.8 leads to

$$v_t \geq M \mathcal{L}_t \phi(\mathcal{G}_t).$$

We finally obtain that

$$\rho_{t+\alpha}^2 - \rho_t^2 = 2\eta\alpha v_t \left( 1 + \frac{\eta\alpha \|\nabla \mathcal{L}_t\|^2}{2v_t} \right)$$
$$\leq 2\eta\alpha v_t \left( 1 + \frac{\eta\alpha B_3^2 \mathcal{L}_t^2 (\rho_t^{M-1} + 1)^2}{2M \mathcal{L}_t \phi(\mathcal{G}_t)} \right)$$
$$= 2\eta\alpha v_t \left( 1 + \frac{\eta\alpha B_3^2 \mathcal{L}_t (\rho_t^{M-1} + 1)^2}{2M \phi(\mathcal{G}_t)} \right)$$

$$= 2\eta\alpha v_t\left(1 + \frac{\eta\alpha R_1(t)}{2M\phi(\mathcal{G}_t)}\right),$$

where we denote

$$R_1(t) = B_3^2\mathcal{L}_t(\rho_t^{M-1} + 1)^2.$$

This completes the proof of (2) for $\alpha = A$.

**Proof of (3) for $\alpha = A$.** Using Taylor expansion, we know that there exists $\epsilon \in (0, \alpha)$ such that

$$\mathcal{G}_{t+\alpha} = \mathcal{G}_t + \nabla\mathcal{G}_t^\top\left(\boldsymbol{\theta}_{t+\alpha} - \boldsymbol{\theta}_t\right) + \frac{1}{2}\left(\boldsymbol{\theta}_{t+\alpha} - \boldsymbol{\theta}_t\right)^\top\nabla^2\mathcal{G}_{t+\epsilon}\left(\boldsymbol{\theta}_{t+\alpha} - \boldsymbol{\theta}_t\right)$$

$$\leq \mathcal{G}_t - \alpha\eta e^{\mathsf{p}_a(\rho_t)}\|\nabla\mathcal{L}_t\|^2 + \eta\alpha\mathcal{G}_t\mathsf{p}_a'(\rho_t)\frac{\langle\boldsymbol{\theta}_t, -\nabla\mathcal{L}_t\rangle}{\rho_t} + \frac{\alpha^2\eta^2}{2}\|\nabla^2\mathcal{G}_{t+\epsilon}\| \cdot \|\nabla\mathcal{L}_t\|^2$$

$$= \mathcal{G}_t - \alpha\eta e^{\mathsf{p}_a(\rho_t)}\|\nabla\mathcal{L}_t\|^2 + \eta\alpha\mathcal{G}_t\mathsf{p}_a'(\rho_t)\frac{v_t}{\rho_t} + \frac{\alpha^2\eta^2}{2}\|\nabla^2\mathcal{G}_{t+\epsilon}\| \cdot \|\nabla\mathcal{L}_t\|^2$$

$$\leq \mathcal{G}_t - \alpha\eta e^{\mathsf{p}_a(\rho_t)}\|\nabla\mathcal{L}_t\|^2 + \eta\alpha\mathcal{G}_t\mathsf{p}_a'(\rho_t)\frac{\rho_t}{v_t}\|\nabla\mathcal{L}_t\|^2$$

$$+ \frac{\alpha^2\eta^2 B_4}{2}\mathcal{G}_{t+\epsilon}(\rho_{t+\epsilon}^{2M-2} + 1) \cdot \|\nabla\mathcal{L}_t\|^2$$

By Cauchy-Schwarz and Lemma F.7

$$= \mathcal{G}_t - \alpha\eta\left[e^{\mathsf{p}_a(\rho_t)}v_t - \mathcal{G}_t\mathsf{p}_a'(\rho_t)\rho_t - \frac{\alpha\eta}{2}B_4\mathcal{G}_{t+\epsilon}(\rho_{t+\epsilon}^{2M-2} + 1)v_t\right]\frac{\|\nabla\mathcal{L}_t\|^2}{v_t}.$$

To estimate the right hand side of the above equation, we first note that

$$\rho_{t+\epsilon}^{2M-2} + 1 \leq (\rho_t + \eta\|\nabla\mathcal{L}_t\|)^{2M-2} + 1 \leq \left(\rho_t + \eta B_3\mathcal{L}_t\left(\rho_t^{M-1} + 1\right)\right)^{2M-2} + 1$$

$$\leq \left(\rho_t + \eta\mathcal{G}_t B_3 e^{-\mathsf{p}_a(\rho_t)}\left(\rho_t^{M-1} + 1\right)\right)^{2M-2} + 1$$

$$\leq \left(\rho_t + \eta\mathcal{G}_s B_3 e^{-\mathsf{p}_a(\rho_t)}\left(\rho_t^{M-1} + 1\right)\right)^{2M-2} + 1 := B_5(\eta, \rho_t).$$

Next we show that $\mathcal{G}_{t+\epsilon} < \mathcal{G}_t$ for all $\epsilon \in (0, \alpha]$. If this is true, then we get

$$\mathcal{G}_{t+\epsilon}\left(\rho_{t+\epsilon}^{2M-2} + 1\right) \leq \mathcal{G}_t B_5(\eta, \rho_t),$$

and consequently

$$\mathcal{G}_{t+\alpha} \leq \mathcal{G}_t - \alpha\eta\left[e^{\mathsf{p}_a(\rho_t)}v_t - \mathcal{G}_t\mathsf{p}_a'(\rho_t)\rho_t - \frac{\alpha\eta}{2}B_4\mathcal{G}_t B_5(\eta, \rho_t)v_t\right]\frac{\|\nabla\mathcal{L}_t\|^2}{v_t}.$$

By Lemma F.8, we have

$$e^{\mathsf{p}_a(\rho_t)}v_t - \mathcal{G}_t\mathsf{p}_a'(\rho_t)\rho_t - \alpha\eta B_4\mathcal{G}_t B_5(\eta, \rho_t)v_t/2$$

$$\geq \left(e^{\mathsf{p}_a(\rho_t)} - \alpha\eta B_4\mathcal{G}_t B_5(\eta, \rho_t)/2\right)\left(M\mathcal{L}_t\phi(\mathcal{L}_t) - \mathsf{p}'(\rho_t)\mathcal{L}_t\right) - \mathcal{G}_t\mathsf{p}_a'(\rho_t)\rho_t$$

$$\geq \left(e^{\mathsf{p}_a(\rho_t)} - \alpha\eta B_4\mathcal{G}_t B_5(\eta, \rho_t)/2\right)M\mathcal{L}_t\phi(\mathcal{L}_t) - \mathcal{G}_t\left(\mathsf{p}'(\rho_t) + \mathsf{p}_a'(\rho_t)\rho_t\right)$$

$$= \left(e^{\mathsf{p}_a(\rho_t)} - \alpha\eta B_4\mathcal{G}_t B_5(\eta, \rho_t)/2\right)M\mathcal{L}_t\phi(\mathcal{L}_t) - M\mathcal{G}_t\mathsf{p}_a(\rho_t)$$

$$= M\mathcal{G}_t\phi(\mathcal{G}_t) - M\mathcal{G}_t \cdot \frac{\alpha\eta}{2}B_4\mathcal{L}_t B_5(\eta, \rho_t)\phi(\mathcal{L}_t)$$

$$\overset{(i)}{=} M\mathcal{G}_t\phi(\mathcal{G}_t) - M\mathcal{G}_t \cdot \frac{\alpha\eta}{2}R_2(t) \overset{(ii)}{\geq} M\mathcal{G}_t\phi(\mathcal{G}_t)\left(1 - \alpha\eta R_2(t)\right),$$

where in $(i)$ we define

$$R_2(t) = B_4 B_5(\eta, \rho_t)\mathcal{L}_t\phi(\mathcal{L}_t),$$

and $(ii)$ is because $\mathcal{G}_t \le \mathcal{G}_s < 1/ne^2$ implies $\phi(\mathcal{G}_t) \ge 2$. It finally follows that

$$\mathcal{G}_{t+\alpha} - \mathcal{G}_t \le -\alpha\eta\left(1 - \alpha\eta R_2(t)\right) M\mathcal{G}_t\phi(\mathcal{G}_t)v_t^{-1}\|\nabla\mathcal{L}_t\|^2.$$

Now it suffices to show that $\mathcal{G}_{t+\epsilon} < \mathcal{G}_t$ for all $\epsilon \in (0, \alpha]$. To this end, assume that $\epsilon_0 = \inf\{\epsilon \in (0, \alpha] : \mathcal{G}_{t+\epsilon} \ge \mathcal{G}_t\}$ exists. Note that

$$\begin{aligned}
\left.\frac{d\mathcal{G}_{t+\epsilon}}{d\epsilon}\right|_{\epsilon=0} &= \nabla G_t^\top(-\eta\nabla\mathcal{L}_t) \\
&= -\eta e^{\mathsf{p}_a(\rho_t)}\|\nabla\mathcal{L}_t\|^2 + \eta\mathcal{G}_t\mathsf{p}_a'(\rho_t)v_t/\rho_t \\
&\le -\eta e^{\mathsf{p}_a(\rho_t)}v_t^2/\rho_t^2 + \eta\mathcal{G}_t\mathsf{p}_a'(\rho_t)v_t/\rho_t \\
&= -v_t/\rho_t^2(\eta e^{\mathsf{p}_a(\rho_t)}v_t - \eta\mathcal{G}_t\mathsf{p}_a'(\rho_t)\rho_t) \\
&\le -(v_t/\rho_t^2)\eta M\mathcal{G}_t\phi(\mathcal{G}_t) < 0,
\end{aligned}$$

which implies $\epsilon_0 > 0$. Therefore, $\mathcal{G}_{t+\epsilon} < \mathcal{G}_t$ for all $\epsilon \in (0, \epsilon_0)$, thus leading to

$$\mathcal{G}_{t+\epsilon_0} \le \mathcal{G}_t - \epsilon_0\eta\left(1 - \epsilon_0\eta R_2(t)\right) M\mathcal{G}_t\phi(\mathcal{G}_t)v_t^{-1}\|\nabla\mathcal{L}_t\|^2 < \mathcal{G}_t,$$

which contradicts $\mathcal{G}_{t+\epsilon_0} \ge \mathcal{G}_t$. Hence, we must have $\mathcal{G}_{t+\epsilon} < \mathcal{G}_t$ for all $\epsilon \in (0, \alpha]$. This concludes the proof of (3).

**Proof of (4) for $\alpha = A$.**  We only need to show that (2) and (3) together imply (4). Note that

$$\mathcal{G}_{t+\alpha} - \mathcal{G}_t \le -\alpha\eta\left(1 - \alpha\eta R_2(t)\right) M\mathcal{G}_t\phi(\mathcal{G}_t)v_t^{-1}\|\nabla\mathcal{L}_t\|^2.$$

Hence,

$$\frac{\mathcal{G}_{t+\alpha} - \mathcal{G}_t}{(1 - \alpha\eta R_2(t))\mathcal{G}_t\phi(\mathcal{G}_t)} \le -\eta\alpha Mv_t^{-1}\|\nabla\mathcal{L}_t\|^2.$$

Multiplying $1 + \frac{\eta\alpha R_1(t)}{2M\phi(\mathcal{G}_t)}$ on both sides, we have

$$\begin{aligned}
&\frac{1 + \eta\alpha R_1(t)/2M\phi(\mathcal{G}_t)}{(1 - \alpha\eta R_2(t))\mathcal{G}_t\phi(\mathcal{G}_t)}(\mathcal{G}_{t+\alpha} - \mathcal{G}_t) \\
&\le -\eta\alpha Mv_t^{-1}\|\nabla\mathcal{L}_t\|^2\left(1 + \frac{\eta\alpha R_1(t)}{2M\phi(\mathcal{G}_t)}\right) \\
&\overset{(i)}{\le} -Mv_t^{-2}\|\nabla\mathcal{L}_t\|^2(\rho_{t+\alpha}^2 - \rho_t^2)/2 \\
&\overset{(ii)}{=} -\frac{1}{\rho_t^2}(\rho_{t+\alpha}^2 - \rho_t^2)\left(\frac{M}{2} + \frac{M\rho_t^2\|\bar{\partial}_\perp\mathcal{L}_t\|_2^2}{2v_t^2}\right),
\end{aligned}$$

where $(i)$ follows from part (2), and $(ii)$ is because we know that:

$$\|\nabla\mathcal{L}_t\|^2 = \|\bar{\partial}_r\mathcal{L}_t\|^2 + \|\bar{\partial}_\perp\mathcal{L}_t\|^2 = \frac{v_t^2}{\rho_t^2} + \|\bar{\partial}_\perp\mathcal{L}_t\|^2.$$

Note that since $\rho_t \ge \rho_s$, our assumption implies that

$$\alpha\eta R_1(t) \le \eta R_1(t) \le \frac{1}{2}, \quad \alpha\eta R_2(t) \le \eta R_2(t) \le \frac{1}{2}.$$

Therefore,

$$\begin{aligned}
\frac{1 + \eta\alpha R_1(t)/2M\phi(\mathcal{G}_t)}{(1 - \alpha\eta R_2(t))\mathcal{G}_t\phi(\mathcal{G}_t)} &\le \frac{1 + \eta R_1(t)/2M\phi(\mathcal{G}_t)}{(1 - \eta R_2(t))\mathcal{G}_t\phi(\mathcal{G}_t)} \\
&\le \frac{1 + \eta R_1(t)/2M\phi(\mathcal{G}_t) + 2\eta R_2(t)}{\mathcal{G}_t\phi(\mathcal{G}_t)}
\end{aligned}$$

$$\leq \frac{1}{\mathcal{G}_t \phi(\mathcal{G}_t)} \left( 1 + \frac{1}{M\phi(\mathcal{G}_t)} + \frac{1}{\phi(\mathcal{G}_t)} \right)$$

$$\leq \frac{1}{\mathcal{G}_t \phi(\mathcal{G}_t)} + \frac{2}{\mathcal{G}_t \phi(\mathcal{G}_t)^2} = -\Phi'(\mathcal{G}_t),$$

where we recall that

$$\Phi(x) = \log \phi(x) - \frac{2}{\phi(x)}.$$

By convexity of $\Phi$, we get that

$$\Phi(\mathcal{G}_{t+\alpha}) - \Phi(\mathcal{G}_t) - (\log \rho_{t+\alpha} - \log \rho_t) \left( M + \frac{M\rho_t^2 \|\bar{\partial}_\perp \mathcal{L}_t\|_2^2}{v_t^2} \right)$$

$$\geq \Phi'(\mathcal{G}_t)(\mathcal{G}_{t+\alpha} - \mathcal{G}_t) - \frac{1}{\rho_t^2}(\rho_{t+\alpha}^2 - \rho_t^2)\left( \frac{M}{2} + \frac{M\rho_t^2 \|\bar{\partial}_\perp \mathcal{L}_t\|_2^2}{2v_t^2} \right) \geq 0.$$

This leads to

$$\log \gamma^{\mathsf{GD}}(\boldsymbol{\theta}_{t+\alpha}) - \log \gamma^{\mathsf{GD}}(\boldsymbol{\theta}_t) = \Phi(\mathcal{G}_{t+\alpha}) - \Phi(\mathcal{G}_t) - M(\log \rho_{t+\alpha} - \log \rho_t)$$

$$\geq \frac{M\rho_t^2 \|\bar{\partial}_\perp \mathcal{L}_t\|_2^2}{v_t^2} \log \frac{\rho_{t+\alpha}}{\rho_t},$$

completing the proof of (4).

**Proof of (1) for $\alpha = A$.** By Lemma F.4 and (4), we have

$$\gamma^{\mathsf{GF}}(\boldsymbol{\theta}_{t+\alpha}) > \gamma^{\mathsf{GD}}(\boldsymbol{\theta}_{t+\alpha}) \geq \gamma^{\mathsf{GD}}(\boldsymbol{\theta}_t) \geq \gamma^{\mathsf{GD}}(\boldsymbol{\theta}_s),$$

which proves (1) for $\alpha = A$. This completes the proof of Theorem F.9. $\qquad\square$

We next show that Assumption 5 verifies the conditions in Theorem F.9.

**Lemma F.11.** *There exists a constant $B$ that only depends on $(M, \mathsf{p}, \mathsf{q}, A)$ and $(\mathsf{r}, \mathsf{s})$ in Assumptions 3 and 5, such that for any step size $\eta > 0$,*

$$\mathcal{G}_s < \min\left\{ \frac{1}{ne^2}, \frac{1}{B\eta} \right\}$$

*implies*

$$\eta \sup_{\rho_t \geq \rho_s, \mathcal{G}_t \leq \mathcal{G}_s} R_1(t) \leq \frac{1}{2}, \quad \eta \sup_{\rho_t \geq \rho_s, \mathcal{G}_t \leq \mathcal{G}_s} R_2(t) \leq \frac{1}{2}.$$

*As a consequence, the conclusions of Theorem F.9 hold.*

*Proof.* In this proof, we will use $B$ to denote a generic constant that depends on $(M, \mathsf{p}, \mathsf{q}, \mathsf{r}, \mathsf{s})$, while the meaning of $B$ can change from line to line. We begin with estimating $R_1(t)$. Since $\deg(\mathsf{p}_a) \geq 1$ when $M \geq 2$, we know that there exists a constant $B$ such that

$$r^{M-1} + 1 \leq B\exp\left( \frac{1}{2}\mathsf{p}_a(r) \right), \quad \forall r \geq 0.$$

Therefore, if $\rho_t \geq \rho_s, \mathcal{G}_t \leq \mathcal{G}_s$, we have

$$\eta R_1(t) = \eta B_3^2 \mathcal{L}_t (\rho_t^{M-1} + 1)^2 \leq \eta B_3^2 B^2 \mathcal{L}_t \exp(\mathsf{p}_a(\rho_t)) = \eta B\mathcal{G}_t \leq \eta B\mathcal{G}_s.$$

For estimating $R_2(t)$, we first note that $\forall r > 0$:

$$B_5(\eta, r) = \left( r + \eta \mathcal{G}_s B_3 e^{-\mathsf{p}_a(r)} \left( r^{M-1} + 1 \right) \right)^{2M-2} + 1$$

$$\leq \left( r + \eta \mathcal{G}_s B_3 B e^{-\mathsf{p}_a(r)/2} \right)^{2M-2} + 1$$

$$\leq 2^{2M-3} \left( r^{2M-2} + \eta^{2M-2} \mathcal{G}_s^{2M-2} B^{2M-2} e^{-(2M-2)\mathsf{p}_a(r)/2} \right) + 1$$

$$\leq B \left( r^{2M-2} + \eta^{2M-2} \mathcal{G}_s^{2M-2} e^{-\mathsf{p}_a(r)} + 1 \right).$$

Second, since $\mathcal{G}_t \leq \mathcal{G}_s$, we have

$$\mathcal{L}_t \phi(\mathcal{L}_t) = \exp(-\mathsf{p}_a(\rho_t)) \mathcal{G}_t \left( \phi(\mathcal{G}_t) + \mathsf{p}_a(\rho_t) \right)$$

$$\overset{(i)}{\leq} \exp(-\mathsf{p}_a(\rho_t)) \mathcal{G}_s \phi(\mathcal{G}_s) + \mathsf{p}_a(\rho_t) \exp(-\mathsf{p}_a(\rho_t)) \mathcal{G}_s$$

$$\overset{(ii)}{\leq} B \exp\left( -\frac{1}{2} \mathsf{p}_a(\rho_t) \right) \mathcal{G}_s \phi(\mathcal{G}_s),$$

where $(i)$ follows from the fact that $x \mapsto x\phi(x)$ is increasing, and $(ii)$ is because that there exists a constant $B$ such that $\mathsf{p}_a(\rho_t) \leq B \exp(\mathsf{p}_a(\rho_t)/2)$ and $\phi(\mathcal{G}_s) \geq 2$. We thus obtain that

$$R_2(t) = B_4 B_5(\eta, \rho_t) \mathcal{L}_t \phi(\mathcal{L}_t) \leq B \left( \rho_t^{2M-2} + \eta^{2M-2} \mathcal{G}_s^{2M-2} e^{-\mathsf{p}_a(\rho_t)} + 1 \right) \mathcal{L}_t \phi(\mathcal{L}_t)$$

$$\leq B \exp\left( -\frac{1}{2} \mathsf{p}_a(\rho_t) \right) \left( \rho_t^{2M-2} + \eta^{2M-2} \mathcal{G}_s^{2M-2} e^{-\mathsf{p}_a(\rho_t)} + 1 \right) \mathcal{G}_s \phi(\mathcal{G}_s)$$

$$\leq B \exp\left( -\frac{1}{4} \mathsf{p}_a(\rho_t) \right) \left( \eta^{2M-2} \mathcal{G}_s^{2M-2} + 1 \right) \mathcal{G}_s \phi(\mathcal{G}_s).$$

To further estimate this term, we use the following naive bound on $\mathcal{G}_s$:

$$\mathcal{G}_s \geq \exp\left( \mathsf{p}_a(\rho_s) - \mathsf{q}(\rho_s) \right),$$

which further implies

$$\phi(\mathcal{G}_s) = -\log n - \log \mathcal{G}_s \leq \mathsf{q}(\rho_s) \leq B \exp\left( \frac{1}{4} \mathsf{p}_a(\rho_s) \right) \leq B \exp\left( \frac{1}{4} \mathsf{p}_a(\rho_t) \right)$$

for some constant $B$, since the degree of $\mathsf{q}$ is at most $M - 1$. In the above display, the last step is due to $\rho_s \leq \rho_t$. It finally follows that

$$\eta R_2(t) \leq B \eta \mathcal{G}_s \left( \eta^{2M-2} \mathcal{G}_s^{2M-2} + 1 \right).$$

Therefore, the requirements $\eta R_1(t) \leq 1/2$ and $\eta R_2(t) \leq 1/2$ are satisfied as long as

$$\eta \mathcal{G}_s \left( \eta^{2M-2} \mathcal{G}_s^{2M-2} + 1 \right) \leq \frac{1}{2B},$$

which is equivalent to

$$\eta \mathcal{G}_s \leq \frac{1}{B}$$

for some constant $B$ only depending on $(M, \mathsf{p}, \mathsf{q}, \mathsf{r}, \mathsf{s})$. This completes the proof. $\qquad \square$

We have completed the proof of margin improvement. The proof of

$$\gamma^{\mathrm{GD}}(\boldsymbol{\theta}_t) \leq \gamma(\boldsymbol{\theta}_t) \leq (1 + \epsilon_t) \gamma^{\mathrm{GD}}(\boldsymbol{\theta}_t), \quad \epsilon_t \to 0$$

is the same as the gradient flow case, so we omit it here for simplicity.

### F.3. Convergence Rates: Proof of Theorem 6.1

In this section, we give refined bounds on the decrease rate of $\mathcal{L}_t = \mathcal{L}(\boldsymbol{\theta}_t)$ and the increase rate of $\rho_t = \|\boldsymbol{\theta}_t\|$, completing the proof of the convergence rates part of Theorem 6.1. In a similar spirit to Lemma F.11, we can show that the conditions of the following Lemma F.12 are satisfied as long as

$$\mathcal{G}_s < \min\left\{ \frac{1}{ne^2}, \frac{1}{B\eta} \right\},$$

which holds under Assumption 5.

**Lemma F.12** (Decrease of the loss function)**.** *Under the same assumptions as Theorem F.9, and further assume that*

$$\eta \sup_{\rho_t \geq \rho_s, \mathcal{G}_t \leq \mathcal{G}_s} \left\{ B_3 \mathcal{L}_t \left( \rho_t^{2M-2} + 1 \right) \right\} \leq \frac{1}{2}.$$

*Then, we have for all $t \geq s$:*

$$-C_2 \eta \mathcal{L}_t^2 \left( \phi(\mathcal{L}_t) \right)^{2-2/M} \leq \mathcal{L}_{t+1} - \mathcal{L}_t \leq -C_1 \eta \mathcal{L}_t^2 \left( \phi(\mathcal{L}_t) \right)^{2-2/M}, \tag{75}$$

*where $C_1, C_2$ are some constants depending on $\gamma^{\mathsf{GD}}(\boldsymbol{\theta}_s)$, $\rho_s$, $A$ and $(M, \mathsf{p}, \mathsf{q}, \mathsf{r}, \mathsf{s})$.*

*Proof.* Using Taylor expansion, we get that for some $\epsilon \in [0, 1]$:

$$\mathcal{L}_{t+1} - \mathcal{L}_t = -\eta \|\nabla \mathcal{L}_t\|^2 + \frac{\eta^2}{2} \nabla \mathcal{L}_t^\top \nabla^2 \mathcal{L}_{t+\epsilon} \nabla \mathcal{L}_t.$$

Note that by our assumption,

$$\frac{\eta^2}{2} \nabla \mathcal{L}_t^\top \nabla^2 \mathcal{L}_{t+\epsilon} \nabla \mathcal{L}_t \leq \frac{\eta^2}{2} \left\| \nabla^2 \mathcal{L}_{t+\epsilon} \right\| \left\| \nabla \mathcal{L}_t \right\|^2$$

$$\overset{(i)}{\leq} \frac{\eta^2}{2} \left\| \nabla \mathcal{L}_t \right\|^2 \cdot B_3 \mathcal{L}_{t+\epsilon} \left( \rho_{t+\epsilon}^{2M-2} + 1 \right)$$

$$\overset{(ii)}{\leq} \frac{\eta}{4} \left\| \nabla \mathcal{L}_t \right\|^2,$$

where $(i)$ follows from Lemma F.6, and $(ii)$ is due to our assumption. This implies that

$$-\frac{5}{4} \eta \|\nabla \mathcal{L}_t\|^2 \leq \mathcal{L}_{t+1} - \mathcal{L}_t \leq -\frac{3}{4} \eta \|\nabla \mathcal{L}_t\|^2.$$

We now establish lower and upper bounds on $\|\nabla \mathcal{L}_t\|$. On the one hand, we have

$$\|\nabla \mathcal{L}_t\| \leq B_3 \mathcal{L}_t \left( \rho_t^{M-1} + 1 \right) \leq C_2 \mathcal{L}_t \rho_t^{M-1} \leq C_2 \mathcal{L}_t \left( \frac{\phi(\mathcal{G}_t)}{\gamma^{\mathsf{GF}}(\boldsymbol{\theta}_t)} \right)^{1-1/M}$$

$$\leq C_2 \mathcal{L}_t \left( \frac{\phi(\mathcal{L}_t)}{\gamma^{\mathsf{GD}}(\boldsymbol{\theta}_s)} \right)^{1-1/M} \leq C_2 \mathcal{L}_t \phi(\mathcal{L}_t)^{1-1/M},$$

where $C_2$ depends on $\gamma^{\mathsf{GD}}(\boldsymbol{\theta}_s)$, $\rho_s$, and $(M, \mathsf{p}, \mathsf{q}, \mathsf{r}, \mathsf{s})$. On the other hand,

$$\|\nabla \mathcal{L}_t\| \geq \frac{v_t}{\rho_t} \geq \frac{M \mathcal{L}_t \phi(\mathcal{L}_t) - \mathsf{p}'(\rho_t) \mathcal{L}_t}{\rho_t} \qquad \text{By Lemma F.8}$$

$$\geq \frac{\gamma^{\mathsf{GF}}(\boldsymbol{\theta}_t)^{1/M} \mathcal{L}_t (M \phi(\mathcal{L}_t) - \mathsf{p}'(\rho_t))}{(\phi(\mathcal{L}_t) - \mathsf{p}_a(\rho_t))^{1/M}} \qquad \text{By definition of } \gamma^{\mathsf{GF}}$$

$$\geq M^{1/M} \frac{\gamma^{\mathsf{GF}}(\boldsymbol{\theta}_t)^{1/M} \mathcal{L}_t (M \phi(\mathcal{L}_t) - \mathsf{p}'(\rho_t))}{(M \phi(\mathcal{L}_t) - \mathsf{p}'(\rho_t))^{1/M}} \qquad \text{By Lemma C.1}$$

$$= M^{1/M} \gamma^{\mathsf{GF}}(\boldsymbol{\theta}_t)^{1/M} \mathcal{L}_t (M \phi(\mathcal{L}_t) - \mathsf{p}'(\rho_t))^{1-1/M}$$

$$\geq M^{1/M} \gamma^{\mathsf{GF}}(\boldsymbol{\theta}_t)^{1/M} \mathcal{L}_t \left( \phi(\mathcal{L}_t) \right)^{1-1/M} \qquad \text{By Eq. (21)}$$

$$\geq M^{1/M} \gamma^{\mathsf{GD}}(\boldsymbol{\theta}_s)^{1/M} \mathcal{L}_t \left( \phi(\mathcal{L}_t) \right)^{1-1/M} \qquad \text{By monotonicity of } \gamma^{\mathsf{GF}}$$

$$:= C_1 \mathcal{L}_t \left( \phi(\mathcal{L}_t) \right)^{1-1/M}.$$

This completes the proof of (75). $\qquad \square$

We first show that $\mathcal{L}_t$ converges to zero and derive its convergence rate.

**Lemma F.13** (Loss convergence rate for GD). *Under the same assumptions as Lemma F.12, we have*

$$\mathcal{L}_t = \Theta\left(\frac{1}{\eta t(\log \eta t)^{2-2/M}}\right) \to 0, \tag{76}$$

*as $t \to \infty$. Here, the constants hided by $\Theta$ only depend on $\gamma^{\mathsf{GD}}(\boldsymbol{\theta}_s)$, $\rho_s$, and $(M, \mathsf{p}, \mathsf{q}, \mathsf{r}, \mathsf{s})$.*

*Proof.* With Lemma F.12, we know that

$$\frac{\mathcal{L}_{t+1} - \mathcal{L}_t}{\mathcal{L}_t^2 \left(\phi(\mathcal{L}_t)\right)^{2-2/M}} \le -C_1 \eta.$$

Now we consider the function,

$$S(x) := \int_x^{\mathcal{L}_s} \frac{1}{u^2 \left(\log \frac{1}{nu}\right)^{2-2/M}} \, du, \quad S'(x) = -\frac{1}{x^2 \left(\log \frac{1}{nx}\right)^{2-2/M}}.$$

When $x \le \mathcal{L}_s < \frac{1}{ne^2}$, we have $S'(x)$ is increasing and $S(x)$ is convex. Hence,

$$S(\mathcal{L}_{t+1}) - S(\mathcal{L}_t) \ge S'(\mathcal{L}_t)(\mathcal{L}_{t+1} - \mathcal{L}_t) \ge C_1 \eta.$$

This leads to

$$S(\mathcal{L}_t) - S(\mathcal{L}_s) \ge C_1 \eta(t - s) \Rightarrow S(\mathcal{L}_t) = \Omega(\eta t).$$

Hence, we must have $S(\mathcal{L}_t) \to +\infty$ as $t \to \infty$. This leads to $\mathcal{L}_t \to 0$. Hence, there exists $t > s$ such that $\mathcal{L}_t < \mathcal{L}_s/2$. To get an upper bound on $S(\mathcal{L}_t)$, we show that $\mathcal{L}_{t+1} \ge \mathcal{L}_t/2$ for $t \ge s$. Note that

$$|\mathcal{L}_{t+1} - \mathcal{L}_t| \le \|\nabla \mathcal{L}_{t+\epsilon}\| \|\boldsymbol{\theta}_{t+1} - \boldsymbol{\theta}_t\| = \eta \|\nabla \mathcal{L}_t\| \|\nabla \mathcal{L}_{t+\epsilon}\|$$

$$\overset{(i)}{\le} \eta B_3^2 \mathcal{L}_t \mathcal{L}_{t+\epsilon} \left(\rho_t^{M-1} + 1\right) \left(\rho_{t+\epsilon}^{M-1} + 1\right)$$

$$\overset{(ii)}{\le} \eta \mathcal{L}_t B_3^2 \mathcal{L}_{t+\epsilon} \left(\rho_{t+\epsilon}^{M-1} + 1\right)^2 = \eta R_1(t+\epsilon)\mathcal{L}_t \le \frac{1}{2}\mathcal{L}_t,$$

where $(i)$ follows from Lemma F.6, $(ii)$ is because of $\rho_{t+\epsilon} \ge \rho_t$, and the last inequality is due to our assumption. Hence, we have (note that $\mathcal{L}_{t+1} \le \mathcal{L}_t$)

$$\mathcal{L}_t^2 \left(\phi(\mathcal{L}_t)\right)^{2-2/M} \le 4\mathcal{L}_{t+1}^2 \left(\log \frac{1}{n\mathcal{L}_{t+1}}\right)^{2-2/M}.$$

Therefore,

$$\frac{\mathcal{L}_{t+1} - \mathcal{L}_t}{\mathcal{L}_{t+1}^2 \left(\log \frac{1}{n\mathcal{L}_{t+1}}\right)^{2-2/M}} \ge -4C_2 \eta.$$

Now since $\mathcal{L}_{t+1} \le \mathcal{L}_{t+\epsilon} \le \mathcal{L}_t$, we have

$$S(\mathcal{L}_{t+1}) - S(\mathcal{L}_t) = S'(\mathcal{L}_{t+\epsilon})(\mathcal{L}_{t+1} - \mathcal{L}_t) \le S'(\mathcal{L}_{t+1})(\mathcal{L}_{t+1} - \mathcal{L}_t) \le 4C_2 \eta.$$

This leads to

$$S(\mathcal{L}_t) - S(\mathcal{L}_s) \le 4C_2 \eta(t - s) \Rightarrow S(\mathcal{L}_t) = O(\eta t).$$

Therefore, $S(\mathcal{L}_t) = \Theta(\eta t)$, where $\Theta$ only hides a constant depending on $C_1$ and $C_2$. By Lemma G.3, we know that

$$S^{-1}(x) = \Theta\left(\frac{1}{x(\log(nx))^{2-2/M}}\right).$$

Hence,

$$\mathcal{L}_t = \Theta\left(\frac{1}{\eta t(\log \eta t)^{2-2/M}}\right) \to 0.$$

This completes the proof of Lemma F.13. $\qquad\square$

**Lemma F.14** (Parameter convergence rate for GD). *Under the same assumptions as Lemma F.12, we have*

$$\rho_t = \Theta(\log \eta t)^{\frac{1}{M}}, \tag{77}$$

*as $t \to \infty$. Here, the constants hided by $\Theta$ only depend on $\gamma^{\mathsf{GD}}(\boldsymbol{\theta}_s)$, $\rho_s$, and $(M, \mathsf{p}, \mathsf{q}, \mathsf{r}, \mathsf{s})$.*

*Proof.* Note that by definition

$$\rho_t^M = \frac{\phi(\mathcal{G}_t)}{\gamma^{\mathsf{GF}}(\boldsymbol{\theta}_t)} \leq \frac{\phi(\mathcal{L}_t)}{\gamma^{\mathsf{GD}}(\boldsymbol{\theta}_s)} = O(\log \eta t).$$

On the other hand, by the near-homogeneity of $f$, we have

$$B\rho_t^M \geq \bar{f}_{\min}(\boldsymbol{\theta}_t) \geq \log \frac{1}{n\mathcal{L}_t} \Rightarrow \rho_t^M = \Omega(\log \eta t).$$

This completes the proof of Lemma F.14. $\qquad\qquad\square$

Combining Lemmas F.13 and F.14 concludes the proof of convergence rates in Theorem 6.1.

## F.4. Directional Convergence: Proof of Theorem 6.2 (Part 1)

Similar to the proof in Appendix C.6, the main idea is to construct the desingularizing function $\Phi$. For GD, we use $\gamma^{\mathsf{GD}}(\boldsymbol{\theta}_t)$ to denote $\gamma^{\mathsf{GD}}(\tilde{\boldsymbol{\theta}}_t)$ and $\zeta_t$ to denote $\|\tilde{\boldsymbol{\theta}}_{t+1} - \tilde{\boldsymbol{\theta}}_t\|$. Our goal is to show that $\sum_{t \geq s} \zeta_t < \infty$. This leads to that the curve swept by $\tilde{\boldsymbol{\theta}}_t$ has finite length, and that $\tilde{\boldsymbol{\theta}}_t$ converges to a limit, thus proving Theorem 6.2.

The rest of this section will be devoted to the proof of $\sum_{t \geq s} \zeta_t < \infty$. Since $\{\gamma^{\mathsf{GD}}(\boldsymbol{\theta}_t)\}_{t \geq s}$ is increasing and bounded from above, we know that

$$\gamma_* = \lim_{t \to \infty} \gamma^{\mathsf{GD}}(\boldsymbol{\theta}_t)$$

exists. To show that $\sum_{t \geq s} \zeta_t < \infty$, we prove the following lemma.

**Lemma F.15** (Desingularizing function for GD). *There exist $R > 0, \nu > 0$ and a definable desingularizing function $\Psi$ on $[0, \nu)$, such that for all large enough $t$ with $\|\boldsymbol{\theta}_t\| > R$ and $\gamma^{\mathsf{GD}}(\boldsymbol{\theta}_t) > \gamma_* - \nu$, and all $\alpha \in [0, 1]$, it holds that*

$$\zeta_t \leq c\Psi'\left(\gamma_* - \gamma^{\mathsf{GD}}(\boldsymbol{\theta}_{t+\alpha})\right)\left(\gamma^{\mathsf{GD}}(\boldsymbol{\theta}_{t+1}) - \gamma^{\mathsf{GD}}(\boldsymbol{\theta}_t)\right).$$

*for some constant $c > 0$. As a consequence, we have*

$$\zeta_t \leq c\left(\Psi\left(\gamma_* - \gamma^{\mathsf{GD}}(\boldsymbol{\theta}_t)\right) - \Psi\left(\gamma_* - \gamma^{\mathsf{GD}}(\boldsymbol{\theta}_{t+1})\right)\right).$$

Once we have this lemma, we immediately know that

$$\sum_{t \geq s} \zeta_t \leq c \sum_{t \geq s} \left(\Psi\left(\gamma_* - \gamma^{\mathsf{GD}}(\boldsymbol{\theta}_t)\right) - \Psi\left(\gamma_* - \gamma^{\mathsf{GD}}(\boldsymbol{\theta}_{t+1})\right)\right) < +\infty.$$

The remainder of this section will be devoted to the proof of Lemma F.15.

Recall that we defined the spherical parts and radial parts of $\partial f(\boldsymbol{\theta}) = \nabla f(\boldsymbol{\theta})$ for a differentiable function $f$. In this section, we also use $a_t$ to denote $e^{\Phi(\mathcal{G}_t)}$, where we recall that

$$\Phi(x) = \log \phi(x) - \frac{2}{\phi(x)}, \quad \Phi'(x) = -\frac{1}{x\phi(x)} - \frac{2}{x\phi(x)^2}.$$

We will first show that

**Lemma F.16** (Decomposition of radial and spherical parts for GD). *Under Assumption 5, we have for all sufficiently large $t \geq s$,*

$$\gamma^{\mathsf{GD}}(\boldsymbol{\theta}_{t+1}) - \gamma^{\mathsf{GD}}(\boldsymbol{\theta}_t) \geq c\eta\left(\|\partial_r \gamma^{\mathsf{GD}}(\boldsymbol{\theta}_t)\|\|\partial_r \mathcal{L}_t\| + \|\partial_\perp \gamma^{\mathsf{GD}}(\boldsymbol{\theta}_t)\|\|\partial_\perp \mathcal{L}_t\|\right). \tag{78}$$

*and*

$$\|\partial_\perp \gamma^{\mathsf{GD}}(\boldsymbol{\theta}_t)\| = \frac{\|\partial_\perp a_t\|}{\rho_t^M}, \quad \text{and} \quad \zeta_t \leq \eta\frac{\|\partial_\perp \mathcal{L}_t\|}{\rho_t}. \tag{79}$$

*Proof.* Recall from our definition that

$$\boldsymbol{\theta}_{t+\alpha} = \boldsymbol{\theta}_t - \alpha\eta\nabla\mathcal{L}_t, \quad \mathcal{G}_{t+\alpha} = \mathcal{G}(\boldsymbol{\theta}_{t+\alpha}),$$

$$a_{t+\alpha} = \exp(\Phi(\mathcal{G}_{t+\alpha})), \quad \rho_{t+\alpha} = \|\boldsymbol{\theta}_{t+\alpha}\|, \quad \gamma^{\mathsf{GD}}(\boldsymbol{\theta}_{t+\alpha}) = \frac{a_{t+\alpha}}{\rho_{t+\alpha}^M}.$$

We first show that

$$\frac{\mathrm{d}\gamma^{\mathsf{GD}}(\boldsymbol{\theta}_{t+\alpha})}{\mathrm{d}\alpha}\bigg|_{\alpha=0} = \frac{\mathrm{d}\gamma^{\mathsf{GD}}(\boldsymbol{\theta}_t)}{\mathrm{d}t} = \eta\Big(\|\partial_r\gamma^{\mathsf{GD}}(\boldsymbol{\theta}_t)\|\|\partial_r\mathcal{L}_t\| + \|\partial_\perp\gamma^{\mathsf{GD}}(\boldsymbol{\theta}_t)\|\|\partial_\perp\mathcal{L}_t\|\Big).$$

To this end, we follow the proof scheme of Lemma C.17. Note that

$$\frac{\mathrm{d}\gamma^{\mathsf{GD}}(\boldsymbol{\theta}_t)}{\mathrm{d}t} = \langle\partial\gamma^{\mathsf{GD}}(\boldsymbol{\theta}_t), -\eta\nabla\mathcal{L}_t\rangle = \eta\left(\langle\partial_r\gamma^{\mathsf{GD}}(\boldsymbol{\theta}_t), -\partial_r\mathcal{L}_t\rangle + \langle\partial_\perp\gamma^{\mathsf{GD}}(\boldsymbol{\theta}_t), -\partial_\perp\mathcal{L}_t\rangle\right),$$

it suffices to prove the following two statements:

(a) $\langle\partial_r\gamma^{\mathsf{GD}}(\boldsymbol{\theta}_t), \boldsymbol{\theta}_t\rangle$ and $\langle-\partial_r\mathcal{L}_t, \boldsymbol{\theta}_t\rangle$ have the same sign.

(b) $\partial_\perp\gamma^{\mathsf{GD}}(\boldsymbol{\theta}_t)$ and $-\partial_\perp\mathcal{L}_t$ point to the same direction.

**Proof of (a).** It suffices to show that $\langle\nabla\gamma^{\mathsf{GD}}(\boldsymbol{\theta}_t), \boldsymbol{\theta}_t\rangle$ and $-\langle\nabla\mathcal{L}_t, \boldsymbol{\theta}_t\rangle$ have the same sign. By Lemma F.8, we have $-\langle\nabla\mathcal{L}_t, \boldsymbol{\theta}_t\rangle = v_t \geq 0$. Next, we show that $\langle\nabla\gamma^{\mathsf{GD}}(\boldsymbol{\theta}_t), \boldsymbol{\theta}_t\rangle \geq 0$. Note that

$$\nabla\gamma^{\mathsf{GD}}(\boldsymbol{\theta}_t) = \frac{\nabla a_t}{\rho_t^M} - \frac{Ma_t\tilde{\boldsymbol{\theta}}_t}{\rho_t^{M+1}},$$

which implies

$$\langle\nabla\gamma^{\mathsf{GD}}(\boldsymbol{\theta}_t), \boldsymbol{\theta}_t\rangle = \frac{\langle\nabla a_t, \boldsymbol{\theta}_t\rangle}{\rho_t^M} - \frac{Ma_t}{\rho_t^M}.$$

Through direct calculation, we obtain that

$$\langle\nabla a_t, \boldsymbol{\theta}_t\rangle = a_t\Phi'(\mathcal{G}_t)\langle\nabla\mathcal{G}_t, \boldsymbol{\theta}_t\rangle$$

$$= a_t\Phi'(\mathcal{G}_t)\left\langle e^{\mathsf{p}_a(\rho_t)}\nabla\mathcal{L}_t + \mathcal{G}_t\mathsf{p}'_a(\rho_t)\tilde{\boldsymbol{\theta}}_t, \boldsymbol{\theta}_t\right\rangle$$

$$= a_t\Phi'(\mathcal{G}_t)\left(-e^{\mathsf{p}_a(\rho_t)}v_t + \mathcal{G}_t\mathsf{p}'_a(\rho_t)\rho_t\right).$$

From the proof of Lemma F.3, we know that

$$\Phi'(x) < -\frac{1}{x\phi(x)} < 0, \quad 0 < x < \frac{1}{ne^2}.$$

Further since (see, e.g., the proof of Theorem F.9)

$$e^{\mathsf{p}_a(\rho_t)}v_t - \mathcal{G}_t\mathsf{p}'_a(\rho_t)\rho_t \geq M\mathcal{G}_t\phi(\mathcal{G}_t),$$

we obtain that

$$\langle\nabla a_t, \boldsymbol{\theta}_t\rangle = a_t\Phi'(\mathcal{G}_t)\left(-e^{\mathsf{p}_a(\rho_t)}v_t + \mathcal{G}_t\mathsf{p}'_a(\rho_t)\rho_t\right)$$

$$\geq a_t\frac{1}{\mathcal{G}_t\phi(\mathcal{G}_t)} \cdot M\mathcal{G}_t\phi(\mathcal{G}_t) = Ma_t,$$

which leads to $\langle\nabla\gamma^{\mathsf{GD}}(\boldsymbol{\theta}_t), \boldsymbol{\theta}_t\rangle \geq 0$. This proves part (a).

**Proof of (b).** Straightforward calculation leads to

$$\partial_\perp \gamma^{\mathsf{GD}}(\boldsymbol{\theta}_t) = \nabla \gamma^{\mathsf{GD}}(\boldsymbol{\theta}_t) - \langle \nabla \gamma^{\mathsf{GD}}(\boldsymbol{\theta}_t), \tilde{\boldsymbol{\theta}}_t \rangle \tilde{\boldsymbol{\theta}}_t = \frac{\nabla a_t - \langle \nabla a_t, \tilde{\boldsymbol{\theta}}_t \rangle \tilde{\boldsymbol{\theta}}_t}{\rho_t^M}$$

$$= a_t \Phi'(\mathcal{G}_t) e^{\mathsf{p}_a(\rho_t)} \left( \nabla \mathcal{L}_t - \langle \nabla \mathcal{L}_t, \tilde{\boldsymbol{\theta}}_t \rangle \tilde{\boldsymbol{\theta}}_t \right)$$

$$= a_t \Phi'(\mathcal{G}_t) e^{\mathsf{p}_a(\rho_t)} \partial_\perp \mathcal{L}_t,$$

which points to the same direction as $-\partial_\perp \mathcal{L}_t$ since $\Phi'(\mathcal{G}_t) < 0$.

To complete the proof, we only need to show that

$$\gamma^{\mathsf{GD}}(\boldsymbol{\theta}_{t+1}) - \gamma^{\mathsf{GD}}(\boldsymbol{\theta}_t) \geq c \frac{\mathrm{d}\gamma^{\mathsf{GD}}(\boldsymbol{\theta}_t)}{\mathrm{d}t}$$

for some constant $c$. We first compute

$$\frac{\mathrm{d}\gamma^{\mathsf{GD}}(\boldsymbol{\theta}_t)}{\mathrm{d}t} = \frac{a_t'}{\rho_t^M} - \frac{M a_t \rho_t'}{\rho_t^{M+1}}.$$

Further since $x \mapsto e^{\Phi(x)}$ and $\alpha \mapsto \rho_{t+\alpha}^{-M} = \|\boldsymbol{\theta}_t - \alpha \nabla \mathcal{L}_t\|^{-M}$ are all convex, it follows that

$$\gamma^{\mathsf{GD}}(\boldsymbol{\theta}_{t+1}) - \gamma^{\mathsf{GD}}(\boldsymbol{\theta}_t) = \frac{a_{t+1}}{\rho_{t+1}^M} - \frac{a_t}{\rho_t^M}$$

$$= \frac{a_{t+1} - a_t}{\rho_{t+1}^M} + a_t \left( \rho_{t+1}^{-M} - \rho_t^{-M} \right)$$

$$\geq \frac{a_t \Phi'(\mathcal{G}_t)(\mathcal{G}_{t+1} - \mathcal{G}_t)}{\rho_{t+1}^M} - \frac{M a_t \rho_t'}{\rho_t^{M+1}}$$

$$\overset{(i)}{\geq} \frac{a_t \Phi'(\mathcal{G}_t)\mathcal{G}_t'}{\rho_{t+1}^M} - \frac{M a_t \rho_t'}{\rho_t^{M+1}} - \frac{a_t |\Phi'(\mathcal{G}_t)||\mathcal{G}_{t+\epsilon}''|}{2\rho_{t+1}^M}$$

$$\overset{(ii)}{\geq} \frac{a_t \Phi'(\mathcal{G}_t)\mathcal{G}_t'}{\rho_t^M} - \frac{M a_t \rho_t'}{\rho_t^{M+1}} - \frac{M a_t \Phi'(\mathcal{G}_t)\mathcal{G}_t'\rho_t'}{\rho_t^{M+1}} - \frac{a_t |\Phi'(\mathcal{G}_t)||\mathcal{G}_{t+\epsilon}''|}{2\rho_{t+1}^M}$$

$$= \frac{\mathrm{d}\gamma^{\mathsf{GD}}(\boldsymbol{\theta}_t)}{\mathrm{d}t} - \frac{M a_t \Phi'(\mathcal{G}_t)\mathcal{G}_t'\rho_t'}{\rho_t^{M+1}} - \frac{a_t |\Phi'(\mathcal{G}_t)||\mathcal{G}_{t+\epsilon}''|}{2\rho_{t+1}^M},$$

where $(i)$ follows from Taylor expansion: $\epsilon \in [0, 1]$, and $(ii)$ follows again from the fact that $\alpha \mapsto \rho_{t+\alpha}^{-M}$ is convex. To proceed, we first give a lower bound on $\frac{\mathrm{d}\gamma^{\mathsf{GD}}(\boldsymbol{\theta}_t)}{\mathrm{d}t}$. By definition,

$$\frac{\mathrm{d}\gamma^{\mathsf{GD}}(\boldsymbol{\theta}_t)}{\mathrm{d}t} = \frac{1}{\rho_t^{M+1}} \left( a_t'\rho_t - M a_t \rho_t' \right) = \frac{\eta a_t}{\rho_t^M} \left( \Phi'(\mathcal{G}_t) \langle \nabla \mathcal{G}_t, -\nabla \mathcal{L}_t \rangle - M \frac{v_t}{\rho_t^2} \right)$$

$$= \frac{\eta a_t}{\rho_t^M} \left( -\Phi'(\mathcal{G}_t) \left( e^{\mathsf{p}_a(\rho_t)} \|\nabla \mathcal{L}_t\|^2 - \mathcal{G}_t \mathsf{p}_a'(\rho_t) \frac{v_t}{\rho_t} \right) - M \frac{v_t}{\rho_t^2} \right).$$

To deal with this term, we note that

$$- \Phi'(\mathcal{G}_t) \left( e^{\mathsf{p}_a(\rho_t)} \|\nabla \mathcal{L}_t\|^2 - \mathcal{G}_t \mathsf{p}_a'(\rho_t) \frac{v_t}{\rho_t} \right) - M \frac{v_t}{\rho_t^2}$$

$$\overset{(i)}{=} \frac{1}{\mathcal{G}_t \phi(\mathcal{G}_t)} \left( e^{\mathsf{p}_a(\rho_t)} \|\nabla \mathcal{L}_t\|^2 - \mathcal{G}_t \mathsf{p}_a'(\rho_t) \frac{v_t}{\rho_t} \right) - M \frac{v_t}{\rho_t^2}$$

$$+ \frac{2}{\mathcal{G}_t \phi(\mathcal{G}_t)^2} \left( e^{\mathsf{p}_a(\rho_t)} \|\nabla \mathcal{L}_t\|^2 - \mathcal{G}_t \mathsf{p}_a'(\rho_t) \frac{v_t}{\rho_t} \right)$$

$$\overset{(ii)}{\geq} \frac{2}{\mathcal{G}_t \phi(\mathcal{G}_t)^2} \left( e^{\mathsf{p}_a(\rho_t)} \|\nabla \mathcal{L}_t\|^2 - \mathcal{G}_t \mathsf{p}_a'(\rho_t) \frac{v_t}{\rho_t} \right),$$

where $(i)$ is due to $\Phi'(x) = -1/x\phi(x) - 2/x\phi(x)^2$, and $(ii)$ follows from the fact that

$$
\begin{aligned}
e^{\mathsf{p}_a(\rho_t)}\|\nabla\mathcal{L}_t\|^2 - \mathcal{G}_t \mathsf{p}_a'(\rho_t)\frac{v_t}{\rho_t} &\geq e^{\mathsf{p}_a(\rho_t)}\frac{v_t^2}{\rho_t^2} - \mathcal{G}_t \mathsf{p}_a'(\rho_t)\frac{v_t}{\rho_t} &&\text{By Cauchy-Schwarz}\\
&= \frac{v_t}{\rho_t^2}\left(e^{\mathsf{p}_a(\rho_t)}v_t - \mathcal{G}_t \mathsf{p}_a'(\rho_t)\rho_t\right)\\
&\geq \frac{v_t}{\rho_t^2}\left(M\mathcal{G}_t\phi(\mathcal{L}_t) - \mathsf{p}'(\rho_t)\mathcal{G}_t - \mathcal{G}_t \mathsf{p}_a'(\rho_t)\rho_t\right) &&\text{By Lemma F.8}\\
&\geq \frac{v_t}{\rho_t^2}\left(M\mathcal{G}_t\phi(\mathcal{L}_t) - M\mathsf{p}_a(\rho_t)\mathcal{G}_t\right) &&\text{By Lemma F.1}\\
&= M\mathcal{G}_t\phi(\mathcal{G}_t)\frac{v_t}{\rho_t^2}.
\end{aligned}
$$

We thus obtain that

$$
\begin{aligned}
\frac{\mathrm{d}\gamma^{\mathsf{GD}}(\boldsymbol{\theta}_t)}{\mathrm{d}t} &\geq \frac{\eta a_t}{\rho_t^M}\frac{2}{\mathcal{G}_t\phi(\mathcal{G}_t)^2}\left(e^{\mathsf{p}_a(\rho_t)}\|\nabla\mathcal{L}_t\|^2 - \mathcal{G}_t \mathsf{p}_a'(\rho_t)\frac{v_t}{\rho_t}\right) &&(80)\\
&\geq \frac{\eta a_t}{\rho_t^M}\frac{2}{\mathcal{G}_t\phi(\mathcal{G}_t)^2}\left(e^{\mathsf{p}_a(\rho_t)}\|\nabla\mathcal{L}_t\|^2 - \mathcal{G}_t \mathsf{p}_a'(\rho_t)\rho_t\frac{\|\nabla\mathcal{L}_t\|^2}{v_t}\right) &&\text{By Cauchy-Schwarz}\\
&= \frac{\eta a_t}{\rho_t^M}\frac{2}{\mathcal{G}_t\phi(\mathcal{G}_t)^2}\frac{\|\nabla\mathcal{L}_t\|^2}{v_t}\left(e^{\mathsf{p}_a(\rho_t)}v_t - \mathcal{G}_t \mathsf{p}_a'(\rho_t)\rho_t\right)\\
&\geq \frac{\eta a_t}{\rho_t^M}\frac{2}{\mathcal{G}_t\phi(\mathcal{G}_t)^2}\frac{\|\nabla\mathcal{L}_t\|^2}{v_t}M\mathcal{G}_t\phi(\mathcal{G}_t) &&\text{Similar to the proof of }(ii)\text{ above}\\
&= \frac{\eta M a_t}{\rho_t^M}\frac{2}{\phi(\mathcal{G}_t)}\frac{\|\nabla\mathcal{L}_t\|^2}{v_t}.
\end{aligned}
$$

We next prove two useful claims:

(c) There exists a constant $c_1 < 1/2$, such that

$$
\frac{a_t|\Phi'(\mathcal{G}_t)||\mathcal{G}_{t+\epsilon}''|}{2\rho_{t+1}^M} \leq c_1\frac{\eta M a_t}{\rho_t^M}\frac{2}{\phi(\mathcal{G}_t)}\frac{\|\nabla\mathcal{L}_t\|^2}{v_t} \leq c_1\frac{\mathrm{d}\gamma^{\mathsf{GD}}(\boldsymbol{\theta}_t)}{\mathrm{d}t}.
$$

(d) There exists a constant $c_2 < 1/2$, such that

$$
\frac{M a_t\Phi'(\mathcal{G}_t)\mathcal{G}_t'\rho_t'}{\rho_t^{M+1}} \leq c_2\frac{a_t|\mathcal{G}_t'|}{\rho_t^M}\frac{2}{\mathcal{G}_t\phi(\mathcal{G}_t)^2} \leq c_2\frac{\mathrm{d}\gamma^{\mathsf{GD}}(\boldsymbol{\theta}_t)}{\mathrm{d}t}.
$$

**Proof of (c).** According to Lemma F.6 and Lemma F.7, we have

$$
|\mathcal{G}_{t+\epsilon}''| \leq \eta^2\|\nabla^2\mathcal{G}_{t+\epsilon}\|\|\nabla\mathcal{L}_t\|^2 \leq B_4\eta^2\mathcal{G}_{t+\epsilon}(\rho_{t+\epsilon}^{2M-2} + 1)\|\nabla\mathcal{L}_t\|^2
$$

for sufficiently large $t$. This implies

$$
\begin{aligned}
\frac{a_t|\Phi'(\mathcal{G}_t)||\mathcal{G}_{t+\epsilon}''|}{2\rho_{t+1}^M} &\leq B_4\eta^2\|\nabla\mathcal{L}_t\|^2\frac{a_t|\Phi'(\mathcal{G}_t)|\mathcal{G}_t(\rho_{t+1}^{M-2} + 1)}{2}\\
&\leq C\eta^2\|\nabla\mathcal{L}_t\|^2 a_t|\Phi'(\mathcal{G}_t)|\mathcal{G}_t(\rho_t^{M-2} + 1)\\
&\leq C\eta^2\|\nabla\mathcal{L}_t\|^2 a_t\frac{1}{\phi(\mathcal{G}_t)}\rho_t^{M-2},
\end{aligned}
$$

where the last line follows from the fact

$$
|\Phi'(\mathcal{G}_t)| = \frac{1}{\mathcal{G}_t\phi(\mathcal{G}_t)} + \frac{2}{\mathcal{G}_t\phi(\mathcal{G}_t)^2} \leq \frac{C}{\mathcal{G}_t\phi(\mathcal{G}_t)}
$$

when $\mathcal{G}_t < 1/ne^2$. Hence, it suffices to show that

$$C\eta^2\|\nabla\mathcal{L}_t\|^2 a_t \frac{1}{\phi(\mathcal{G}_t)}\rho_t^{M-2} \le c_1 \frac{\eta M a_t}{\rho_t^M}\frac{2}{\phi(\mathcal{G}_t)}\frac{\|\nabla\mathcal{L}_t\|^2}{v_t} \iff C\eta\rho_t^{2M-2}v_t \le c_1,$$

where $C$ depends on $\mathcal{G}_s$ and $M$. Using Lemma F.6, we know that for sufficiently large $t$,

$$\|\nabla\mathcal{L}_t\| \le B_3\mathcal{L}_t(\rho_t^{M-1}+1) \le C\mathcal{L}_t\rho_t^{M-1},$$

thus leading to

$$C\eta\rho_t^{2M-2}v_t \le C\eta\rho_t^{2M-1}\|\nabla\mathcal{L}_t\| \le C\eta\rho_t^{3M-2}\mathcal{L}_t,$$

which can be arbitrarily small as $t \to \infty$ due to our result on convergence rates of $\rho_t$ and $\mathcal{L}_t$. This proves part (c). Note that our proof works for any $\eta$ and sufficiently large $t$, and the constant $c_1$ can be arbitrarily small.

**Proof of (d).** In Eq. (80), we already proved that

$$\frac{\mathrm{d}\gamma^{\mathrm{GD}}(\boldsymbol{\theta}_t)}{\mathrm{d}t} \ge \frac{\eta a_t}{\rho_t^M}\frac{2}{\mathcal{G}_t\phi(\mathcal{G}_t)^2}\left(e^{\mathsf{p}_a(\rho_t)}\|\nabla\mathcal{L}_t\|^2 - \mathcal{G}_t\mathsf{p}'_a(\rho_t)\frac{v_t}{\rho_t}\right)$$

$$= \frac{a_t|\mathcal{G}'_t|}{\rho_t^M}\frac{2}{\mathcal{G}_t\phi(\mathcal{G}_t)^2}.$$

Therefore, it suffices to show that

$$\frac{Ma_t|\Phi'(\mathcal{G}_t)||\mathcal{G}'_t|\rho'_t}{\rho_t^{M+1}} \le c_2\frac{a_t|\mathcal{G}'_t|}{\rho_t^M}\frac{2}{\mathcal{G}_t\phi(\mathcal{G}_t)^2},$$

which reduces to

$$\frac{M|\Phi'(\mathcal{G}_t)|\rho'_t}{\rho_t} \le c_2\frac{2}{\mathcal{G}_t\phi(\mathcal{G}_t)^2}.$$

Since

$$|\Phi'(\mathcal{G}_t)| \le \frac{C}{\mathcal{G}_t\phi(\mathcal{G}_t)}, \quad \rho'_t = \eta\frac{v_t}{\rho_t},$$

we deduce that

$$\frac{M|\Phi'(\mathcal{G}_t)|\rho'_t}{\rho_t} \le \eta\frac{CM}{\mathcal{G}_t\phi(\mathcal{G}_t)}\frac{v_t}{\rho_t^2}.$$

It remains to show

$$C\eta v_t \le c_2\frac{\rho_t^2}{\phi(\mathcal{G}_t)}.$$

Using our estimates on $v_t$ from the proof of part (c), we know that

$$C\eta v_t \le C\eta\rho_t^M\mathcal{L}_t.$$

To complete the proof, we need an upper bound on $\phi(\mathcal{G}_t)$. By definition,

$$\phi(\mathcal{G}_t) = \log\frac{1}{ne^{\mathsf{p}_a(\rho_t)}\mathcal{L}_t} \le \log\frac{1}{\mathcal{L}_t}.$$

We thus obtain that

$$C\eta v_t\phi(\mathcal{G}_t) \le C\eta\rho_t^M\mathcal{L}_t\log\frac{1}{\mathcal{L}_t} \ll \rho_t^2$$

as $t \to \infty$, which is easily seen from the converges rates of $\mathcal{L}_t$ and $\rho_t$. Similarly, $c_2$ can be arbitrarily small and our proof works for any $\eta > 0$.

Combining parts (c) and (d), it follows immediately that

$$\gamma^{\mathrm{GD}}(\boldsymbol{\theta}_{t+1}) - \gamma^{\mathrm{GD}}(\boldsymbol{\theta}_t) \ge (1 - c_1 - c_2)(\gamma^{\mathrm{GD}})'(t),$$

completing the proof of (78). The first part of (79) can be shown by direct calculation. For the second part, we have

$$\zeta_t = \|\tilde{\boldsymbol{\theta}}_{t+1} - \tilde{\boldsymbol{\theta}}_t\| = \left\| \int_0^1 \frac{\mathrm{d}\tilde{\boldsymbol{\theta}}_{t+z}}{\mathrm{d}z} \mathrm{d}z \right\| \le \int_0^1 \left\| \frac{\mathrm{d}\tilde{\boldsymbol{\theta}}_{t+z}}{\mathrm{d}z} \right\| \mathrm{d}z.$$

We will show that $\|\dot{\tilde{\boldsymbol{\theta}}}_{t+z}\| := \left\| \frac{\mathrm{d}\tilde{\boldsymbol{\theta}}_{t+z}}{\mathrm{d}z} \right\|$ is a non-increasing function of $z$, so that

$$\zeta_t \le \|\dot{\tilde{\boldsymbol{\theta}}}_t\| = \eta \frac{\|\partial_\perp \mathcal{L}_t\|}{\rho_t},$$

where the last equality follows from the proof of lemma C.17. Note that by direct calculation,

$$\begin{aligned}
\left\| \dot{\tilde{\boldsymbol{\theta}}}_{t+z} \right\| &= \left\| \frac{1}{\rho_{t+z}} \frac{\mathrm{d}\boldsymbol{\theta}_{t+z}}{\mathrm{d}z} - \frac{1}{\rho_{t+z}} \left\langle \frac{\mathrm{d}\boldsymbol{\theta}_{t+z}}{\mathrm{d}z}, \tilde{\boldsymbol{\theta}}_{t+z} \right\rangle \cdot \tilde{\boldsymbol{\theta}}_{t+z} \right\| \\
&\le \frac{\eta}{\rho_t} \left\| \nabla \mathcal{L}(\boldsymbol{\theta}_t) - \left\langle \nabla \mathcal{L}(\boldsymbol{\theta}_t), \tilde{\boldsymbol{\theta}}_{t+z} \right\rangle \cdot \tilde{\boldsymbol{\theta}}_{t+z} \right\| \\
&\le \frac{\eta}{\rho_t} \left\| \nabla \mathcal{L}(\boldsymbol{\theta}_t) - \left\langle \nabla \mathcal{L}(\boldsymbol{\theta}_t), \tilde{\boldsymbol{\theta}}_t \right\rangle \cdot \tilde{\boldsymbol{\theta}}_t \right\| = \left\| \dot{\tilde{\boldsymbol{\theta}}}_t \right\|,
\end{aligned}$$

since $\rho_{t+z} \ge \rho_t$ and $\langle -\nabla\mathcal{L}(\boldsymbol{\theta}_t), \tilde{\boldsymbol{\theta}}_t \rangle \ge 0$. This proves (79). □

Then, we need some inequalities to connect $(\partial_r \mathcal{L}_t, \partial_\perp \mathcal{L}_t)$ and $(\partial_r \gamma^{\mathsf{GD}}(\boldsymbol{\theta}_t), \partial_\perp \gamma^{\mathsf{GD}}(\boldsymbol{\theta}_t))$. The main idea is to show that their ratios are close. i.e.,

$$\frac{\|\partial_\perp \mathcal{L}_t\|}{\|\partial_r \mathcal{L}_t\|} \approx \frac{\|\partial_\perp \gamma^{\mathsf{GD}}(\boldsymbol{\theta}_t)\|}{\|\partial_r \gamma^{\mathsf{GD}}(\boldsymbol{\theta}_t)\|}.$$

The technique is to use $a_t$ to bridge these two ratios. To this end, we need the following auxiliary result:

**Lemma F.17.** *There exists a constant $C > 0$ such that for all sufficiently large $t \ge s$ and $\alpha \in [0,1]$,*

$$\|\boldsymbol{\theta}_{t+\alpha}\| \le C\|\boldsymbol{\theta}_t\|, \quad \|\partial\gamma^{\mathsf{GD}}(\boldsymbol{\theta}_{t+\alpha})\| \le C\|\partial\gamma^{\mathsf{GD}}(\boldsymbol{\theta}_t)\|. \tag{81}$$

*Proof.* We first prove $\|\boldsymbol{\theta}_{t+\alpha}\| \le C\|\boldsymbol{\theta}_t\|$. By definition,

$$\|\boldsymbol{\theta}_{t+\alpha}\| \le \|\boldsymbol{\theta}_t\| + \eta\|\nabla\mathcal{L}(\boldsymbol{\theta}_t)\|.$$

From the proof of Lemma F.16, we get the following estimate:

$$\|\nabla\mathcal{L}(\boldsymbol{\theta}_t)\| \le B_1 \mathcal{L}_t \rho_t^{M-1} \le \frac{B_1}{n} \exp(-\mathsf{p}_a(\rho_t)) \rho_t^{M-1} \le \rho_t$$

for large enough $t$. Hence, $\|\boldsymbol{\theta}_{t+\alpha}\| \le C\|\boldsymbol{\theta}_t\|$. For the second part of (81), we have for any $\boldsymbol{\theta}$,

$$\partial\gamma^{\mathsf{GD}}(\boldsymbol{\theta}) = \frac{1}{\|\boldsymbol{\theta}\|^M} \left( \nabla a(\boldsymbol{\theta}) - M a(\boldsymbol{\theta}) \frac{\boldsymbol{\theta}}{\|\boldsymbol{\theta}\|^2} \right),$$

where $a(\boldsymbol{\theta}) = \exp(\Phi(\mathcal{G}(\boldsymbol{\theta})))$. It suffices to show that

$$\left\| \nabla a(\boldsymbol{\theta}_{t+\alpha}) - M a(\boldsymbol{\theta}_{t+\alpha}) \frac{\boldsymbol{\theta}_{t+\alpha}}{\|\boldsymbol{\theta}_{t+\alpha}\|^2} \right\| \le C \left\| \nabla a(\boldsymbol{\theta}_t) - M a(\boldsymbol{\theta}_t) \frac{\boldsymbol{\theta}_t}{\|\boldsymbol{\theta}_t\|^2} \right\|.$$

To this end, we first give a lower bound on the right hand side. By Cauchy-Schwarz inequality,

$$\begin{aligned}
\left\| \nabla a(\boldsymbol{\theta}_t) - M a(\boldsymbol{\theta}_t) \frac{\boldsymbol{\theta}_t}{\|\boldsymbol{\theta}_t\|^2} \right\| &\ge \frac{1}{\rho_t} |\langle \nabla a_t, \boldsymbol{\theta}_t \rangle - M a_t| \\
&\ge \frac{1}{\rho_t} \left| a_t \Phi'(\mathcal{G}_t) \left( -e^{\mathsf{p}_a(\rho_t)} v_t + \mathcal{G}_t \mathsf{p}'_a(\rho_t) \rho_t \right) - M a_t \right| \\
&\overset{(i)}{\ge} \frac{1}{\rho_t} a_t \frac{2}{\mathcal{G}_t \phi(\mathcal{G}_t)^2} \cdot M \mathcal{G}_t \phi(\mathcal{G}_t) = \frac{M a_t}{\rho_t} \cdot \frac{2}{\phi(\mathcal{G}_t)},
\end{aligned} \tag{82}$$

where $(i)$ follows similarly as in the proof of Lemma F.16. Denoting

$$A(\boldsymbol{\theta}) = \nabla a(\boldsymbol{\theta}) - M a(\boldsymbol{\theta}) \frac{\boldsymbol{\theta}}{\|\boldsymbol{\theta}\|^2},$$

then we have for some $\epsilon \in [0, \alpha]$,

$$\|A(\boldsymbol{\theta}_{t+\alpha}) - A(\boldsymbol{\theta}_t)\| \leq \|\nabla A(\boldsymbol{\theta}_{t+\epsilon})\| \|\boldsymbol{\theta}_{t+\alpha} - \boldsymbol{\theta}_t\|$$
$$= \eta \|\nabla \mathcal{L}(\boldsymbol{\theta}_t)\| \|\nabla A(\boldsymbol{\theta}_{t+\epsilon})\|.$$

We now upper bound $\|\nabla A(\boldsymbol{\theta}_{t+\epsilon})\|$ in the above display. For any $\boldsymbol{\theta}$,

$$\|\nabla A(\boldsymbol{\theta})\| = \left\| \nabla^2 a(\boldsymbol{\theta}) - M \frac{\boldsymbol{\theta} \nabla a(\boldsymbol{\theta})^\top}{\|\boldsymbol{\theta}\|^2} - M a(\boldsymbol{\theta}) \left( \frac{\|\boldsymbol{\theta}\|^2 I - \boldsymbol{\theta}\boldsymbol{\theta}^\top}{\|\boldsymbol{\theta}\|^4} \right) \right\|$$
$$\leq \|\nabla^2 a(\boldsymbol{\theta})\| + M \frac{\|\nabla a(\boldsymbol{\theta})\|}{\|\boldsymbol{\theta}\|} + M a(\boldsymbol{\theta}) \frac{1}{\|\boldsymbol{\theta}\|^2}.$$

We estimate these terms respectively. For $\|\boldsymbol{\theta}\|$ large enough:

$$\|\nabla a(\boldsymbol{\theta})\| = a(\boldsymbol{\theta})|\Phi'(\mathcal{G}(\boldsymbol{\theta}))| \|\nabla \mathcal{G}(\boldsymbol{\theta})\| \leq C_1 a(\boldsymbol{\theta}) |\Phi'(\mathcal{G}(\boldsymbol{\theta}))| \mathcal{G}(\boldsymbol{\theta}) \left( \|\boldsymbol{\theta}\|^{M-1} + 1 \right)$$
$$\leq C_1 a(\boldsymbol{\theta}) \frac{\|\boldsymbol{\theta}\|^{M-1} + 1}{\phi(\mathcal{G}(\boldsymbol{\theta}))} \leq C a(\boldsymbol{\theta}) \frac{\|\boldsymbol{\theta}\|^{M-1}}{\phi(\mathcal{G}(\boldsymbol{\theta}))},$$
$$\|\nabla^2 a(\boldsymbol{\theta})\| \leq |\Phi'(\mathcal{G}(\boldsymbol{\theta}))| \|\nabla \mathcal{G}(\boldsymbol{\theta})\| \|\nabla a(\boldsymbol{\theta})\| + a(\boldsymbol{\theta})|\Phi''(\mathcal{G}(\boldsymbol{\theta}))| \|\nabla \mathcal{G}(\boldsymbol{\theta})\|^2 + a(\boldsymbol{\theta})|\Phi'(\mathcal{G}(\boldsymbol{\theta}))| \|\nabla^2 \mathcal{G}(\boldsymbol{\theta})\|$$
$$\leq C_1^2 a(\boldsymbol{\theta}) \frac{\left(\|\boldsymbol{\theta}\|^{M-1} + 1\right)^2}{\phi(\mathcal{G}(\boldsymbol{\theta}))^2} + C_1 a(\boldsymbol{\theta}) \frac{\left(\|\boldsymbol{\theta}\|^{M-1} + 1\right)^2}{\phi(\mathcal{G}(\boldsymbol{\theta}))} + C_2 a(\boldsymbol{\theta}) \frac{\|\boldsymbol{\theta}\|^{2M-2} + 1}{\phi(\mathcal{G}(\boldsymbol{\theta}))}$$
$$\leq C a(\boldsymbol{\theta}) \frac{\left(\|\boldsymbol{\theta}\|^{M-1} + 1\right)^2}{\phi(\mathcal{G}(\boldsymbol{\theta}))} \leq C a(\boldsymbol{\theta}) \frac{\|\boldsymbol{\theta}\|^{2M-2}}{\phi(\mathcal{G}(\boldsymbol{\theta}))}.$$

Here, the upper bounds on $\|\nabla \mathcal{G}(\boldsymbol{\theta})\|$ and $\|\nabla^2 \mathcal{G}(\boldsymbol{\theta})\|$ are due to Lemma F.7. We thus obtain that

$$\|\nabla A(\boldsymbol{\theta})\| \leq C \left( a(\boldsymbol{\theta}) \frac{\|\boldsymbol{\theta}\|^{2M-2}}{\phi(\mathcal{G}(\boldsymbol{\theta}))} + M a(\boldsymbol{\theta}) \frac{\|\boldsymbol{\theta}\|^{M-2}}{\phi(\mathcal{G}(\boldsymbol{\theta}))} + M \frac{a(\boldsymbol{\theta})}{\|\boldsymbol{\theta}\|^2} \right) \leq C a(\boldsymbol{\theta}) \|\boldsymbol{\theta}\|^{2M-2}$$

for sufficiently large $\|\boldsymbol{\theta}\|$. Further since $|\mathcal{G}(\boldsymbol{\theta}_{t+\epsilon}) - \mathcal{G}(\boldsymbol{\theta}_t)| = o(|\mathcal{G}(\boldsymbol{\theta}_t)|)$ as $t \to \infty$, we know that for sufficiently large $t$, $a(\boldsymbol{\theta}_{t+\epsilon}) \leq 2 a(\boldsymbol{\theta}_t)$, thus leading to

$$\|\nabla A(\boldsymbol{\theta}_{t+\epsilon})\| \leq C a(\boldsymbol{\theta}_{t+\epsilon}) \rho_{t+\epsilon}^{2M-2} \leq C a(\boldsymbol{\theta}_t) \rho_t^{2M-2}.$$

It finally follows that

$$\|A(\boldsymbol{\theta}_{t+\alpha}) - A(\boldsymbol{\theta}_t)\| \leq \eta \|\nabla \mathcal{L}(\boldsymbol{\theta}_t)\| \|\nabla A(\boldsymbol{\theta}_{t+\epsilon})\| \leq C \eta \|\nabla \mathcal{L}_t\| a_t \rho_t^{2M-2}$$
$$\leq C \eta \mathcal{L}_t a_t \rho_t^{3M-3} \leq C \eta \mathcal{G}_t a_t \exp(-\mathsf{p}_a(\rho_t)) \rho_t^{3M-3}$$
$$\leq \frac{M a_t}{\rho_t} \cdot \frac{2}{\phi(\mathcal{G}_t)} \leq \|A(\boldsymbol{\theta}_t)\|$$

for sufficiently large $t$, where the last inequality is just (82), and the second-to-last one is because of $\mathcal{G}_t \phi(\mathcal{G}_t) \leq C$ for any $\mathcal{G}_t < 1/ne^2$. This finally leads to

$$\left\| \nabla a(\boldsymbol{\theta}_{t+\alpha}) - M a(\boldsymbol{\theta}_{t+\alpha}) \frac{\boldsymbol{\theta}_{t+\alpha}}{\|\boldsymbol{\theta}_{t+\alpha}\|^2} \right\| \leq 2 \left\| \nabla a(\boldsymbol{\theta}_t) - M a(\boldsymbol{\theta}_t) \frac{\boldsymbol{\theta}_t}{\|\boldsymbol{\theta}_t\|^2} \right\|,$$

completing the proof of Lemma F.17. $\qquad\square$

**Lemma F.18** (Inequalities between $a_t$ and $\gamma^{\mathsf{GD}}(\boldsymbol{\theta}_t)$)**.** *Under Assumption 5, for all sufficiently large $t \geq s$,*

$$\|\partial_r a_t\| \geq M\gamma^{\mathsf{GD}}(\boldsymbol{\theta}_s)\rho_t^{M-1}, \tag{83}$$

*and*

$$\|\partial_r \gamma^{\mathsf{GD}}(\boldsymbol{\theta}_t)\| \leq \frac{C_1 M \log n + C_2 \mathsf{p}'(\rho_t)}{\rho_t^{M+1}}. \tag{84}$$

*Combining these two inequalities, there exists a threshold $s_1 > s > 0$, for all sufficiently large $t \geq s_1$, we have*

$$\|\partial_r a_t\| \geq \gamma^{\mathsf{GD}}(\boldsymbol{\theta}_s)\rho_t^{M+1/2}\|\partial_r \gamma^{\mathsf{GD}}(\boldsymbol{\theta}_t)\|. \tag{85}$$

*Proof.* First, note that from the proof of Lemma F.16, we have

$$\|\partial_r a_t\| = \frac{1}{\rho_t}\langle \nabla a_t, \boldsymbol{\theta}_t \rangle \geq \frac{Ma_t}{\rho_t} = M\gamma^{\mathsf{GD}}(\boldsymbol{\theta}_t)\rho_t^{M-1} \geq M\gamma^{\mathsf{GD}}(\boldsymbol{\theta}_s)\rho_t^{M-1}, \tag{86}$$

which proves (83).

Second, again from the proof of Lemma F.16, we get

$$
\begin{aligned}
\|\partial_r \gamma^{\mathsf{GD}}(\boldsymbol{\theta}_t)\| &= \frac{1}{\rho_t^{M+1}}\left( \langle \nabla a_t, \boldsymbol{\theta}_t \rangle - Ma_t \right) \\
&= \frac{1}{\rho_t^{M+1}}\left( a_t |\Phi'(\mathcal{G}_t)| \left( e^{\mathsf{p}_a(\rho_t)}v_t - \mathcal{G}_t \mathsf{p}'_a(\rho_t)\rho_t \right) - Ma_t \right) \\
&\overset{(i)}{\leq} \frac{1}{\rho_t^{M+1}}\left( a_t |\Phi'(\mathcal{G}_t)|\mathcal{G}_t \left( M\log\frac{1}{\mathcal{L}_t} + \mathsf{p}'(\rho_t) - \mathsf{p}'_a(\rho_t)\rho_t \right) - Ma_t \right) \\
&= \frac{1}{\rho_t^{M+1}}\left( a_t |\Phi'(\mathcal{G}_t)|\mathcal{G}_t \left( M\log n + M\phi(\mathcal{G}_t) + 2\mathsf{p}'(\rho_t) \right) - Ma_t \right) \\
&= \frac{1}{\rho_t^{M+1}}\left( a_t |\Phi'(\mathcal{G}_t)|\mathcal{G}_t \left( M\log n + 2\mathsf{p}'(\rho_t) \right) + Ma_t \left( |\Phi'(\mathcal{G}_t)|\mathcal{G}_t\phi(\mathcal{G}_t) - 1 \right) \right) \\
&\leq \frac{1}{\rho_t^{M+1}}\left( C_1 \left( M\log n + 2\mathsf{p}'(\rho_t) \right) + C_2 M \right) \\
&\leq \frac{1}{\rho_t^{M+1}}\left( C'_1 M\log n + C'_2 \mathsf{p}'(\rho_t) \right),
\end{aligned}
$$

where

$$C_1 = \sup_{0<x<1/ne^3} e^{\Phi(x)}|\Phi'(x)|x, \quad C_2 = \sup_{0<x<1/ne^3} e^{\Phi(x)}\left( |\Phi'(x)|x\phi(x) - 1 \right)$$

are positive constants (easily seen from the definition of $\Phi$), and $(i)$ follows from lemma F.8.

Finally, the proof of (85) follows the same way as (32). $\qquad\square$

Now we will use Lemmas C.19 and C.20 to prove Lemma F.15.

*Proof of Lemma F.15.* Recall that we have the following decompositon in (78):

$$\gamma^{\mathsf{GD}}(\boldsymbol{\theta}_{t+1}) - \gamma^{\mathsf{GD}}(\boldsymbol{\theta}_t) \geq c\eta\Big( \|\partial_r \gamma^{\mathsf{GD}}(\boldsymbol{\theta}_t)\|\|\partial_r \mathcal{L}_t\| + \|\partial_\perp \gamma^{\mathsf{GD}}(\boldsymbol{\theta}_t)\|\|\partial_\perp \mathcal{L}_t\|\Big).$$

Two cases will be considered in this proof:

- Case 1: $\|\partial_r \gamma^{\mathsf{GD}}(\boldsymbol{\theta}_t)\|\|\partial_r \mathcal{L}_t\|$ is larger, and we will apply Lemma C.20 for construction.

- Case 2: $\|\partial_\perp \gamma^{\mathsf{GD}}(\boldsymbol{\theta}_t)\|\|\partial_\perp \mathcal{L}_t\|$ is larger, and we will apply Lemma C.19 for construction.

The two cases will be determined by the ratio of $\|\partial_\perp \gamma^{\mathrm{GD}}(\boldsymbol{\theta}_t)\|$ and $\|\partial_r \gamma^{\mathrm{GD}}(\boldsymbol{\theta}_t)\|$. In case 1, we assume that:

$$\|\partial_\perp \gamma^{\mathrm{GD}}(\boldsymbol{\theta}_t)\| \le \rho_t^{\frac{1}{8}} \|\partial_r \gamma^{\mathrm{GD}}(\boldsymbol{\theta}_t)\|. \tag{87}$$

For case 2, the condition is:

$$\|\partial_\perp \gamma^{\mathrm{GD}}(\boldsymbol{\theta}_t)\| \ge \rho_t^{\frac{1}{8}} \|\partial_r \gamma^{\mathrm{GD}}(\boldsymbol{\theta}_t)\|. \tag{88}$$

**Case 1.** By (85), we have

$$
\begin{aligned}
\|\partial_r a_t\| &\ge \gamma^{\mathrm{GD}}(\boldsymbol{\theta}_s)\rho_t^{M+\frac{1}{2}} \|\partial_r \gamma^{\mathrm{GD}}(\boldsymbol{\theta}_t)\| \\
&\ge \gamma^{\mathrm{GD}}(\boldsymbol{\theta}_s)\rho_t^{M+\frac{3}{8}} \|\partial_\perp \gamma^{\mathrm{GD}}(\boldsymbol{\theta}_t)\| && \text{By (34)} \\
&\ge \gamma^{\mathrm{GD}}(\boldsymbol{\theta}_s)\rho_t^{\frac{3}{8}} \|\partial_\perp a_t\|. && \text{By Lemma F.16.}
\end{aligned} \tag{89}
$$

Now we need to transfer this inequality to the ratio of $\|\partial_\perp \mathcal{L}_t\|$ and $\|\partial_r \mathcal{L}_t\|$. Note that

$$
\begin{aligned}
\|\partial_r a_t\| = \frac{1}{\rho_t} \langle \nabla a_t, \boldsymbol{\theta}_t \rangle &= -\frac{\langle \nabla \mathcal{L}_t, \boldsymbol{\theta}_t \rangle}{\mathcal{L}_t \rho_t} - \mathsf{p}'_a(\rho_t) \\
&\le -\frac{\langle \nabla \mathcal{L}_t, \boldsymbol{\theta}_t \rangle}{\mathcal{L}_t \rho_t} = \frac{1}{\mathcal{L}_t} \|\partial_r \mathcal{L}_t\|.
\end{aligned}
$$

On the other hand, we have

$$\|\partial_\perp a_t\| = \frac{1}{\mathcal{L}_t} \|\partial_\perp \mathcal{L}_t\|.$$

Combining these two inequalities and pluging them into (89), we have

$$\|\partial_r \mathcal{L}_t\| \ge \gamma^{\mathrm{GD}}(\boldsymbol{\theta}_s)\rho_t^{\frac{3}{8}} \|\partial_\perp \mathcal{L}_t\|. \tag{90}$$

On the other hand, we know that there exists $s_2 > s > 0$ such that for a.e. $t \ge s_2$, we have $\rho_t > 1$. Hence, we have

$$\|\partial \gamma^{\mathrm{GD}}(\boldsymbol{\theta}_t)\| \le \|\partial_r \gamma^{\mathrm{GD}}(\boldsymbol{\theta}_t)\| + \|\partial_\perp \gamma^{\mathrm{GD}}(\boldsymbol{\theta}_t)\| \le 2\rho_t^{\frac{1}{8}} \|\partial_r \gamma^{\mathrm{GD}}(\boldsymbol{\theta}_t)\|. \tag{91}$$

Therefore, in terms of (78), we have

$$
\begin{aligned}
\gamma^{\mathrm{GD}}(\boldsymbol{\theta}_{t+1}) - \gamma^{\mathrm{GD}}(\boldsymbol{\theta}_t) &\ge c\eta \|\partial_r \gamma^{\mathrm{GD}}(\boldsymbol{\theta}_t)\| \|\partial_r \mathcal{L}_t\| \\
&\ge c\eta \frac{\gamma^{\mathrm{GD}}(\boldsymbol{\theta}_s)}{2} \rho_t^{\frac{1}{4}} \|\partial \gamma^{\mathrm{GD}}(\boldsymbol{\theta}_t)\| \|\partial_\perp \mathcal{L}_t\| && \text{By (90) and (91)} \\
&\ge c\rho_t^{\frac{5}{4}} \|\partial \gamma^{\mathrm{GD}}(\boldsymbol{\theta}_t)\| \zeta_t. && \text{By Lemma F.16}
\end{aligned} \tag{92}
$$

Now we invoke Lemma C.20 to construct the desingularizing function. We apply it to the definable function $\gamma_* - \gamma^{\mathrm{GD}}(\boldsymbol{\theta})$ with $\lambda = \frac{1}{4}$. Then there exists $\nu_1 > 0$ and a definable desingularizing function $\Psi_1$ on $[0, \nu_1)$ such that

$$8\Psi'_1(\gamma_* - \gamma^{\mathrm{GD}}(\boldsymbol{\theta}))\|\boldsymbol{\theta}\|^{5/4} \|\partial \gamma^{\mathrm{GD}}(\boldsymbol{\theta})\| \ge 1, \quad \text{if } \gamma^{\mathrm{GD}}(\boldsymbol{\theta}) \ge \gamma_* - \nu_1.$$

Further, Lemma F.17 implies that there exists a constant $c > 0$, such that for all $\alpha \in [0, 1]$,

$$8\Psi'_1(\gamma_* - \gamma^{\mathrm{GD}}(t + \alpha))\rho_t^{5/4} \|\partial \gamma^{\mathrm{GD}}(\boldsymbol{\theta}_t)\| \ge c, \quad \text{if } \gamma^{\mathrm{GD}}(\boldsymbol{\theta}_t) \ge \gamma_* - \nu_1. \tag{93}$$

Plugging (92) into the above inequality, we have

$$8\Psi'_1(\gamma_* - \gamma^{\mathrm{GD}}(\boldsymbol{\theta}_{t+\alpha}))\left(\gamma^{\mathrm{GD}}(\boldsymbol{\theta}_{t+1}) - \gamma^{\mathrm{GD}}(\boldsymbol{\theta}_t)\right) \ge c\zeta_t, \quad \text{if } \gamma^{\mathrm{GD}}(\boldsymbol{\theta}_t) \ge \gamma_* - \nu_1.$$

This completes the proof for case 1.

**Case 2.** By lemma F.16, we have

$$\gamma^{\mathsf{GD}}(\boldsymbol{\theta}_{t+1}) - \gamma^{\mathsf{GD}}(\boldsymbol{\theta}_t) \geq c\eta \|\partial_\perp \gamma^{\mathsf{GD}}(\boldsymbol{\theta}_t)\| \|\partial_\perp \mathcal{L}_t\| \geq c\rho_t \|\partial_\perp \gamma^{\mathsf{GD}}(\boldsymbol{\theta}_t)\| \zeta_t. \tag{94}$$

For a.e. $t \geq s_1 > s > 0$, we have

$$\|\partial_\perp \gamma^{\mathsf{GD}}(\boldsymbol{\theta}_t)\| \geq \rho_t^{\frac{1}{8}} \|\partial_r \gamma^{\mathsf{GD}}(\boldsymbol{\theta}_t)\| \geq \|\partial_r \gamma^{\mathsf{GD}}(\boldsymbol{\theta}_t)\|.$$

This leads to

$$\|\partial_\perp \gamma^{\mathsf{GD}}(\boldsymbol{\theta}_t)\| \geq \frac{1}{2}\|\partial \gamma^{\mathsf{GD}}(\boldsymbol{\theta}_t)\|.$$

Plugging this into (94), we have

$$\gamma^{\mathsf{GD}}(\boldsymbol{\theta}_{t+1}) - \gamma^{\mathsf{GD}}(\boldsymbol{\theta}_t) \geq \frac{c\rho_t}{2}\|\partial \gamma^{\mathsf{GD}}(\boldsymbol{\theta}_t)\| \zeta_t. \tag{95}$$

We invoke Lemma C.19 to construct the desingularizing function. We apply it to the definable function $\gamma_* - \gamma^{\mathsf{GD}}(\boldsymbol{\theta})$ with $c = 1$ and $\eta = \frac{1}{8}$. Then there exists $\nu_2 > 0$ and a definable desingularizing function $\Psi_2$ on $[0, \nu_2)$ such that

$$\Psi_2'(\gamma_* - \gamma^{\mathsf{GD}}(\boldsymbol{\theta}))\|\boldsymbol{\theta}\| \|\partial \gamma^{\mathsf{GD}}(\boldsymbol{\theta})\| \geq 1, \quad \text{if } \gamma^{\mathsf{GD}}(\boldsymbol{\theta}) \geq \gamma_* - \nu_2.$$

Further, Lemma F.17 implies that there exists a constant $c > 0$, such that for all $\alpha \in [0, 1]$,

$$\Psi_2'(\gamma_* - \gamma^{\mathsf{GD}}(t + \alpha))\rho_t \|\partial \gamma^{\mathsf{GD}}(\boldsymbol{\theta}_t)\| \geq c, \quad \text{if } \gamma^{\mathsf{GD}}(\boldsymbol{\theta}_t) \geq \gamma_* - \nu_2. \tag{96}$$

Plugging (95) into the above inequality, we have

$$2\Psi_2'(\gamma_* - \gamma^{\mathsf{GD}}(\boldsymbol{\theta}_{t+\alpha}))\left(\gamma^{\mathsf{GD}}(\boldsymbol{\theta}_{t+1}) - \gamma^{\mathsf{GD}}(\boldsymbol{\theta}_t)\right) \geq c\zeta_t, \quad \text{if } \gamma^{\mathsf{GD}}(\boldsymbol{\theta}_t) \geq \gamma_* - \nu_2.$$

This completes the proof for case 2.

**Combining the two cases.** Since $\Psi_1' - \Psi_2'$ is a definable function, it's nonnegative or nonpositive on some interval $(0, \nu)$. Let $\Psi = \max\{\Psi_1, \Psi_2\}$. Then, we have for a.e. large enought $t$ such that $\rho_t \geq 1$ and $\gamma^{\mathsf{GD}}(\boldsymbol{\theta}_t) \geq \gamma_* - \nu$, and $\log n + 2p_a(\rho_t) \leq \rho_t^{M-\frac{1}{2}}$, it holds that

$$c\Psi'(\gamma_* - \gamma^{\mathsf{GD}}(\boldsymbol{\theta}_{t+\alpha}))\left(\gamma^{\mathsf{GD}}(\boldsymbol{\theta}_{t+1}) - \gamma^{\mathsf{GD}}(\boldsymbol{\theta}_t)\right) \geq \zeta_t,$$

for some constant $c > 0$. The final conclusion follows directly from the Lagrange mean value theorem. This completes the proof of Lemma F.15. $\qquad \square$

### F.5. KKT Convergence: Proof of Theorem 6.2 (Part 2)

The main idea is to verify the KKT conditions. Recall that the optimization problem (P) is defined as follows:

$$\min \|\boldsymbol{\theta}\|_2^2, \quad \text{s.t. } y_i f_{\mathsf{H}}(\boldsymbol{\theta}; \mathbf{x}_i) \geq 1 \text{ for all } i \in [n].$$

Following the notations in Appendix C.8, we have

$$\bar{f}_i(\boldsymbol{\theta}) = y_i f(\boldsymbol{\theta}; \mathbf{x}_i), \quad \bar{f}_{\min}(\boldsymbol{\theta}) = \min_{i \in [n]} \bar{f}_i(\boldsymbol{\theta}),$$

$$\bar{f}_{\mathsf{H},i}(\boldsymbol{\theta}) = y_i f_{\mathsf{H}}(\boldsymbol{\theta}; \mathbf{x}_i), \quad \bar{f}_{\mathsf{H},\min}(\boldsymbol{\theta}) = \min_{i \in [n]} \bar{f}_{\mathsf{H},i}(\boldsymbol{\theta}),$$

$$\mathbf{h}_i(\boldsymbol{\theta}) = \partial \bar{f}_i(\boldsymbol{\theta}), \quad \mathbf{h}(\boldsymbol{\theta}) = \frac{1}{n}\sum_{i=1}^{n} e^{-\bar{f}_i(\boldsymbol{\theta})}\mathbf{h}_i(\boldsymbol{\theta}),$$

$$\mathbf{h}_{\mathsf{H},i}(\boldsymbol{\theta}) = \partial \bar{f}_{\mathsf{H},i}(\boldsymbol{\theta}), \quad \mathbf{h}_{\mathsf{H}}(\boldsymbol{\theta}) = \frac{1}{n}\sum_{i=1}^{n} e^{-\bar{f}_i(\boldsymbol{\theta})}\mathbf{h}_{\mathsf{H},i}(\boldsymbol{\theta}).$$

We are going to verify that $\hat{\boldsymbol{\theta}}_t := \boldsymbol{\theta}_t / (\bar{f}_{\mathsf{H},\min}(\boldsymbol{\theta}_t))^{1/M}$ satisfies the two conditions in Definition 8, i.e.,

1. $\|\boldsymbol{\theta} - \sum_{i=1}^n \lambda_i \mathbf{h}_{\mathsf{H},i}(\boldsymbol{\theta})\| \le \epsilon$, where $\mathbf{h}_{\mathsf{H},i}(\boldsymbol{\theta}) = \partial \bar{f}_{\mathsf{H},i}(\boldsymbol{\theta})$ for all $i \in [n]$;

2. For any $i \in [n]$, $\lambda_i\left(\bar{f}_{\mathsf{H},i}(\boldsymbol{\theta}) - 1\right) \le \delta$.

Recall that $\lambda_i$ and $\beta$ are constructed as follows:

$$\lambda_i(\boldsymbol{\theta}) := \frac{\bar{f}_{\mathsf{H},\min}^{1-2/M} \rho_t e^{-\bar{f}_i(\boldsymbol{\theta})}}{n\|\mathbf{h}_{\mathsf{H}}(\boldsymbol{\theta})\|}, \quad \beta_t := \frac{\langle \boldsymbol{\theta}_t, \mathbf{h}_{\mathsf{H}}(\boldsymbol{\theta}_t)\rangle}{\|\boldsymbol{\theta}_t\|\|\mathbf{h}_{\mathsf{H}}(\boldsymbol{\theta}_t)\|}.$$

Since our model assumptions are the same as those of the GF case, Lemma C.21, Lemma C.23 and Lemma C.24 hold here. Hence, the second condition is satisfied. All we need to show is that

$$\beta_t = \frac{\langle \boldsymbol{\theta}_t, \mathbf{h}_{\mathsf{H}}(\boldsymbol{\theta}_t)\rangle}{\|\boldsymbol{\theta}_t\|\|\mathbf{h}_{\mathsf{H}}(\boldsymbol{\theta}_t)\|} \to 1.$$

Similarly, we will only show a subsequence of $\beta$ goes to 0. Note that by Assumption 3 and Proposition 5.1, there is a function $\mathsf{r}(x) = o(x^{M-1})$ as $x \to \infty$, such that for almost every $\boldsymbol{\theta}_t$ and any $i \in [n]$, we have

$$\|\nabla \bar{f}_i(\boldsymbol{\theta}_t) - \nabla \bar{f}_{\mathsf{H},i}(\boldsymbol{\theta}_t)\| \le \mathsf{r}(\|\boldsymbol{\theta}_t\|) = \mathsf{r}(\rho_t).$$

**Lemma F.19** (Bound of $\beta$ in GD). *Under Assumptions 3 and 5, we have for any $t_2 > t_1 > s$,*

$$\sum_{t=t_1}^{t_2} \left[\frac{1-p_1(t)}{(p_2(t)+\beta_t)^2} - 1\right] \cdot \log \frac{\rho_{t+1}}{\rho_t} \le \frac{1}{M} \log \frac{\gamma^{\mathsf{GD}}(\boldsymbol{\theta}_{t_2})}{\gamma^{\mathsf{GD}}(\boldsymbol{\theta}_{t_1})}, \tag{97}$$

*where*

$$p_1(t) = \frac{2\mathsf{r}(\rho_t)}{M\gamma^{\mathsf{GD}}(\boldsymbol{\theta}_s)\rho_t^{M-1}}, \quad p_2(t) = \frac{\mathsf{p}_a(\rho_t)}{M\gamma^{\mathsf{GD}}(\boldsymbol{\theta}_s)\rho_t^M}.$$

*Proof.* Note that in Theorem F.9, we have shown:

$$\log \gamma^{\mathsf{GD}}(\boldsymbol{\theta}_{t+1}) - \log \gamma^{\mathsf{GD}}(\boldsymbol{\theta}_t) \ge \log \frac{\rho_{t+1}}{\rho_t} \frac{M\rho_t^2\|\partial_\perp \mathcal{L}_t\|_2^2}{v_t^2}.$$

We will give a lower bound of $\frac{M\rho_t^2\|\partial_\perp \mathcal{L}_t\|_2^2}{v_t^2}$. For the denominator, we have

$$v_t = \langle -\nabla \mathcal{L}_t, \boldsymbol{\theta}_t\rangle = \frac{1}{n}\sum_{i=1}^n e^{-\bar{f}_i(\boldsymbol{\theta}_t)}\langle \nabla \bar{f}_i(\boldsymbol{\theta}_t), \boldsymbol{\theta}_t\rangle \le \langle \mathbf{h}_{\mathsf{H}}(\boldsymbol{\theta}_t), \boldsymbol{\theta}_t\rangle + 2M\mathcal{L}_t\mathsf{p}_a(\rho_t). \tag{98}$$

For the numerator, we have

$$\|\partial_\perp \mathcal{L}_t\|_2^2 = \|\nabla \mathcal{L}_t\|_2^2 - \langle \nabla \mathcal{L}_t, \tilde{\boldsymbol{\theta}}_t\rangle^2 = \|\mathbf{h}(\boldsymbol{\theta}_t)\|^2 - \langle \mathbf{h}(\boldsymbol{\theta}_t), \tilde{\boldsymbol{\theta}}_t\rangle^2. \tag{99}$$

Similarly, we have

$$\|\mathbf{h}(\boldsymbol{\theta}_t) - \mathbf{h}_{\mathsf{H}}(\boldsymbol{\theta}_t)\| \le \frac{1}{n}\sum_{i=1}^n e^{-\bar{f}_i(\boldsymbol{\theta}_t)}\|\mathbf{h}_i(\boldsymbol{\theta}_t) - \mathbf{h}_{\mathsf{H},i}(\boldsymbol{\theta}_t)\| \le \mathcal{L}_t\mathsf{r}(\rho_t).$$

And we can get

$$\|\mathbf{h}(\boldsymbol{\theta}_t)\| \ge \|\mathbf{h}_{\mathsf{H}}(\boldsymbol{\theta}_t)\| - \|\mathbf{h}(\boldsymbol{\theta}_t) - \mathbf{h}_{\mathsf{H}}(\boldsymbol{\theta}_t)\| \ge \|\mathbf{h}_{\mathsf{H}}(\boldsymbol{\theta}_t)\| - \mathcal{L}_t\mathsf{r}(\rho_t).$$

In what follows, we will use $\mathbf{h}$ and $\mathbf{h}_{\mathsf{H}}$ as shorthands for $\mathbf{h}(\boldsymbol{\theta}_t)$ and $\mathbf{h}_{\mathsf{H}}(\boldsymbol{\theta}_t)$, repsectively. As $\rho_t \to \infty$,

$$\|\mathbf{h}_{\mathsf{H}}\| \ge \frac{\langle \mathbf{h}_{\mathsf{H}}, \boldsymbol{\theta}_t\rangle}{\rho_t} = \frac{M\sum_{i=1}^n e^{-\bar{f}_i(\boldsymbol{\theta}_t)} \bar{f}_{\mathsf{H},i}(\boldsymbol{\theta}_t)}{n\rho_t}$$

$$\ge \frac{M\sum_{i=1}^n e^{-\bar{f}_i(\boldsymbol{\theta}_t)}\left(\bar{f}_i(\boldsymbol{\theta}_t) - \mathsf{p}_a(\rho_t)\right)}{n\rho_t}$$

$$\geq \frac{M\mathcal{L}_t\phi(\mathcal{G}_t)}{\rho_t} \geq M\mathcal{L}_t\gamma^{\mathsf{GD}}(\boldsymbol{\theta}_t)\rho_t^{M-1}$$

$$\geq M\mathcal{L}_t\gamma^{\mathsf{GD}}(\boldsymbol{\theta}_s)\rho_t^{M-1} \geq 2\mathcal{L}_t\mathsf{r}(\rho_t) \tag{100}$$

for sufficiently large $\rho_t$, since $\mathsf{r}(\rho_t) = o(\rho_t^{M-1})$ as $t \to \infty$. Therefore, there exists $s_5 > s > 0$ such that for a.e. $t \geq s_5$, we have

$$\begin{aligned} \|\mathbf{h}\|^2 &= \|\mathbf{h}_{\mathsf{H}}\|^2 + 2\langle\mathbf{h}_{\mathsf{H}}, \mathbf{h} - \mathbf{h}_{\mathsf{H}}\rangle + \|\mathbf{h} - \mathbf{h}_{\mathsf{H}}\|^2 \\ &\geq \|\mathbf{h}_{\mathsf{H}}\|^2 + 2\langle\mathbf{h}_{\mathsf{H}}, \mathbf{h} - \mathbf{h}_{\mathsf{H}}\rangle \\ &\geq \|\mathbf{h}_{\mathsf{H}}\|^2(1 - 2\|\mathbf{h}_{\mathsf{H}}\|^{-1}\mathcal{L}_t\mathsf{r}(\rho_t)) \\ &\geq \|\mathbf{h}_{\mathsf{H}}\|^2\left(1 - \frac{2\mathsf{r}(\rho_t)}{M\gamma^{\mathsf{GD}}(\boldsymbol{\theta}_s)\rho_t^{M-1}}\right). \end{aligned} \tag{101}$$

At last, we have

$$\langle\mathbf{h}, \tilde{\boldsymbol{\theta}}_t\rangle = v_t/\rho_t \leq \langle\mathbf{h}_{\mathsf{H}}, \tilde{\boldsymbol{\theta}}_t\rangle + \mathcal{L}_t\mathsf{p}_a(\rho_t)/\rho_t. \tag{102}$$

Plugging (101) and (102) into (99), we can give a lower bound for $\|\partial_\perp\mathcal{L}_t\|_2^2$,

$$\|\partial_\perp\mathcal{L}_t\|_2^2 \geq \|\mathbf{h}_{\mathsf{H}}\|_2^2\left(1 - \frac{2\mathsf{r}(\rho_t)}{M\gamma^{\mathsf{GD}}(\boldsymbol{\theta}_s)\rho_t^{M-1}}\right) - \left(\langle\mathbf{h}_{\mathsf{H}}, \tilde{\boldsymbol{\theta}}_t\rangle + \mathcal{L}_t\mathsf{p}_a(\rho_t)/\rho_t\right)^2. \tag{103}$$

Combining (98) and (103), we have

$$\begin{aligned} \frac{\rho_t^2\|\partial_\perp\mathcal{L}_t\|_2^2}{v_t^2} &\geq \frac{\|\mathbf{h}_{\mathsf{H}}\|_2^2\left\{1 - 2\mathsf{r}(\rho_t)/\left[M\gamma^{\mathsf{GD}}(\boldsymbol{\theta}_s)\rho_t^{M-1}\right]\right\} - \left(\langle\mathbf{h}_{\mathsf{H}}, \tilde{\boldsymbol{\theta}}_t\rangle + \mathcal{L}_t\mathsf{p}_a(\rho_t)/\rho_t\right)^2}{\left(\langle\mathbf{h}_{\mathsf{H}}, \tilde{\boldsymbol{\theta}}_t\rangle + \mathcal{L}_t\mathsf{p}_a(\rho_t)/\rho_t\right)^2} \\ &= \frac{\|\mathbf{h}_{\mathsf{H}}\|_2^2\left\{1 - 2\mathsf{r}(\rho_t)/\left[M\gamma^{\mathsf{GD}}(\boldsymbol{\theta}_s)\rho_t^{M-1}\right]\right\}}{\left(\langle\mathbf{h}_{\mathsf{H}}, \tilde{\boldsymbol{\theta}}_t\rangle + \mathcal{L}_t\mathsf{p}_a(\rho_t)/\rho_t\right)^2} - 1 \\ &= \frac{1 - 2\mathsf{r}(\rho_t)/\left[M\gamma^{\mathsf{GD}}(\boldsymbol{\theta}_s)\rho_t^{M-1}\right]}{\left(\beta_t + \mathcal{L}_t\mathsf{p}_a(\rho_t)/\rho_t\|\mathbf{h}_{\mathsf{H}}\|^{-1}\right)^2} - 1 \\ &\geq \frac{1 - 2\mathsf{r}(\rho_t)/(M\gamma^{\mathsf{GD}}(\boldsymbol{\theta}_s)\rho_t^{M-1})}{\left(\beta_t + \mathsf{p}_a(\rho_t)/(M\gamma^{\mathsf{GD}}(\boldsymbol{\theta}_s)\rho_t^M)\right)^2} - 1 \\ &= \frac{1 - p_1(t)}{\left(\beta_t + p_2(t)\right)^2} - 1. \end{aligned}$$

This completes the proof of Lemma F.19. $\qquad\qquad\square$

Once we have this, we can show the following analogous results. We omit their proofs since they are completely similar to those of Lemma C.27 and Lemma C.28.

**Lemma F.20** ($\beta$ converges to 1)**.** *Under Assumptions 3 and 5, there exists a sequence $t_k$ such that $\lim_{k\to\infty}\beta_{t_k} \to 1$.*

**Lemma F.21** (Approximate KKT point)**.** *Under Assumptions 3 and 5, there exists a sequence $t_k$ such that $\tilde{\boldsymbol{\theta}}_{t_k}$ is an $(\epsilon_k, \delta_k)$-KKT point of (P) for all $k \in \mathbb{N}$, where $\epsilon_k \to 0$ and $\delta_k \to 0$ as $k \to \infty$.*

We are now in position to prove the KKT convergence. Appplying Lemma F.21, we have a sequence $\{t_k\}$ such that $\tilde{\boldsymbol{\theta}}_{t_k}$ is an $(\epsilon_k, \delta_k)$-KKT point of (P) for all $k \in \mathbb{N}$, where $\epsilon_k \to 0$ and $\delta_k \to 0$ as $k \to \infty$. Since $\tilde{\boldsymbol{\theta}}_{t_k}$ converges in direction, $\tilde{\boldsymbol{\theta}}_{t_k}$ will converge to the same direction as one of the KKT points of (P). By Theorem 6.2, we know $\tilde{\boldsymbol{\theta}}_{t_k}$ also converges to the limit $\boldsymbol{\theta}_*$. Hence, $\boldsymbol{\theta}_*/\left(\bar{f}_{\mathsf{H,min}}(\boldsymbol{\theta}_*)\right)^{1/M}$ is a KKT point of (P). This completes the proof of Theorem 6.2.

# G. Additional Lemmas

**Lemma G.1.** *The function $\pi : \mathbb{R}^n \to \mathbb{R}$:*

$$\pi(v) = \log\left(\frac{1}{n}\sum_{i=1}^{n} \exp(-v_i)\right)$$

*is convex.*

*Proof.* Let $u_i = e^{-x_i}, v_i = e^{-y_i}$. So

$$\pi\big(\theta x_i + (1-\theta)y_i\big) = \log\left(\frac{1}{n}\sum_{i=1}^{n}\exp(-\theta x_i - (1-\theta)y_i)\right) = \log\left(\frac{1}{n}\sum_{i=1}^{n} u_i^\theta v_i^{1-\theta}\right).$$

From Hölder's inequality, we have

$$\sum_{i=1}^{n} x_i y_i \leq \left(\sum_{i=1}^{n} x_i^{\frac{1}{\theta}}\right)^{\theta}\left(\sum_{i=1}^{n} y_i^{\frac{1}{1-\theta}}\right)^{(1-\theta)}.$$

Therefore,

$$\log\left(\frac{1}{n}\sum_{i=1}^{n} u_i^\theta v_i^{1-\theta}\right) \leq \log\left[\left(\frac{1}{n}\sum_{i=1}^{n} u_i^{\theta \cdot \frac{1}{\theta}}\right)^{\theta} \cdot \left(\frac{1}{n}\sum_{i=1}^{n} v_i^{(1-\theta)\cdot\frac{1}{1-\theta}}\right)^{1-\theta}\right].$$

The right formula can be reduced to:

$$\theta\log\left(\frac{1}{n}\sum_{i=1}^{n} u_i\right) + (1-\theta)\log\left(\frac{1}{n}\sum_{i=1}^{n} v_i\right).$$

Therefore, we get:

$$\pi(\theta x + (1-\theta)y) \leq \theta\pi(x) + (1-\theta)\pi(y).$$

This completes the proof of Lemma G.1. $\qquad\square$

**Lemma G.2.** *Let $s > 1$ and $S(x) = \int_s^x \frac{1}{(\log t)^{2-2/M}} dt$. Then,*

$$S(x) = \Theta\left(\frac{x}{(\log x)^{2-2/M}}\right), \quad S^{-1}(y) = \Theta\left(y(\log y)^{2-2/M}\right).$$

*Proof.* First, we bound the rate of $S(x)$. Note that

$$S(x) = \int_s^x \frac{1}{(\log t)^{2-2/M}} dt = \int_s^x \frac{(\log t)^{2/M}}{(\log t)^2} dt \leq (\log x)^{2/M}\int_s^x \frac{1}{(\log t)^2} dt = \mathcal{O}\left(\frac{x}{(\log x)^{2-2/M}}\right).$$

On the other hand, we have

$$S(x) \geq \int_{\sqrt{x}}^x \frac{1}{(\log t)^{2-2/M}} dt \geq ((\log x)/2)^{2/M}\int_{\sqrt{x}}^x \frac{1}{(\log t)^2} dt = \Omega\left(\frac{x}{(\log x)^{2-2/M}}\right).$$

Now we bound the rate of $S^{-1}(y)$. Let $x = S^{-1}(y)$ for $y \geq 0$. By the previous results, we have $x \to \infty$ as $y \to \infty$. Besides, we know that $y = S(x) = \Theta\left(\frac{x}{(\log x)^{2-2/M}}\right)$. Taking logarithm on both sides, we have $\log y = \Theta(\log x)$. Therefore,

$$y(\log y)^{2-2/M} = \Theta(y(\log x)^{2-2/M}) = \Theta(x).$$

This implies that $x = \Theta(y(\log y)^{2-2/M})$. This completes the proof of Lemma G.2. $\qquad\square$

**Lemma G.3.** *Given two integers $m, n \in \mathbb{Z}_+$ and a constant $s < \frac{1}{ne^2}$, let $S(x) := \int_x^s \frac{1}{t^2 (\log(nt))^{2-2/M}} \mathrm{d}t$ for $x < \frac{s}{2}$. Then,*

$$S(x) = \Theta\left(\frac{1}{x(\log(nx))^{2-2/M}}\right), \quad S^{-1}(y) = \Theta\left(\frac{1}{y(\log(ny))^{2-2/M}}\right).$$

*Proof.* We prove the LHS rate first. Now we can bound the rate of $S(x)$. Note that

$$S(x) = \int_x^s \frac{1}{t^2 (\log(nt))^{2-2/M}} \mathrm{d}t \geq \frac{1}{(\log(nx))^{2-2/M}} \int_x^s \frac{1}{t^2} \mathrm{d}t$$

$$= \frac{1}{(\log(nx))^{2-2/M}} \left(\frac{1}{x} - \frac{1}{s}\right) = \Omega\left(\frac{1}{x(\log(nx))^{2-2/M}}\right).$$

Besides, we know that

$$S(x) = \int_x^s \frac{1}{t^2 (\log(nt))^{2-2/M}} \mathrm{d}t$$

$$\leq (\log(nx))^{2/M} \int_x^s \frac{1}{t^2 (\log(nt))^2} \mathrm{d}t$$

$$\leq (\log(nx))^{2/M} \left(-\frac{3}{t(\log(nt))^2}\Big|_x^s\right)$$

$$\leq \frac{3(\log(nx))^{2/M}}{x(\log(nx))^2} = \mathcal{O}\left(\frac{1}{x(\log(nx))^{2-2/M}}\right).$$

Combine them, we get that

$$S(x) = \Theta\left(\frac{1}{x(\log(nx))^{2-2/M}}\right).$$

Let $y = S(x)$. Then, we have $\log(x) = \Theta(-\log(y))$. Hence, we have

$$x = \frac{1}{y(\log(nx))^{2-2/M}} = \Theta\left(\frac{1}{y(\log(ny))^{2-2/M}}\right).$$

This completes the proof of Lemma G.3.

$\square$

**Lemma G.4.** *Given $0 < \gamma < 1$, when $t \geq \left(4/\gamma(\log(4/\gamma))\right)^4$, we have*

$$\frac{\log t}{2\gamma} \leq t^{\frac{1}{4}}.$$

*Proof.* Let $z = t^{\frac{1}{4}}$. Then, we want to show:

$$g(z) := \gamma z - 2\log z \geq 0.$$

Note that $g'(z) = \gamma - \frac{2}{z}$. When $z \geq 2/\gamma$, $g(z)$ is increasing. Furthermore, we have

$$g(4/\gamma \log(4/\gamma)) = 4\log(4/\gamma) - 2\log(4/\gamma) - 2\log\log(4/\gamma) = 2\log\frac{4/\gamma}{\log(4/\gamma)} > 0.$$

Since $4/\gamma \log(4/\gamma)) > 2/\gamma$, we have finished the proof of Lemma G.4.

$\square$

**Lemma G.5** (Moore-Osgood Theorem). *Assume that a series of functions $(f_n(x))_{n=1}^\infty$ converge uniformly to $f(x)$ in $(-a, a)$ for some $a > 0$, and for any $n$, $\lim_{x \to 0} f_n(x) = L_n$ exists. Then, both $\lim_{x \to 0} f(x)$ and $\lim_{n \to \infty} L_n$ exist and are equal, namely*

$$\lim_{n \to \infty} L_n = \lim_{n \to \infty} \lim_{x \to 0} f_n(x) = \lim_{x \to 0} \lim_{n \to \infty} f_n(x) = \lim_{x \to 0} f(x).$$

*Proof.* Due to uniform convergence, for any $\epsilon > 0$ there exist $N(\epsilon) \in \mathbb{N}$, such that: for all $x \in (-a, a)\backslash\{0\}$, $n, m > N$ implies $|f_n(x) - f_m(x)| < \frac{\epsilon}{3}$. As $x \to 0$, we have $|L_n - L_m| < \frac{\epsilon}{3}$, which means that $L_n$ is a Cauchy sequence which converges to a limit $L$. In addition, as $m \to \infty$, we have $|L_n - L| < \frac{\epsilon}{3}$. On the other hand, if we take $m \to \infty$ first, we have $|f_n(x) - f(x)| < \frac{\epsilon}{3}$. By the existence of pointwise limit, for any $\epsilon > 0$ and $n > N$, there exist $\delta(\epsilon, n) > 0$, such that $0 < |x| < \delta$ implies $|f_n(x) - L_n| < \frac{\epsilon}{3}$. Then for that fixed $n, 0 < |x| < \delta$ implies $|f(x) - L| \leq |f(x) - f_n(x)| + |f_n(x) - L_n| + |L_n - L| \leq \epsilon$. This proves that $\lim_{x \to 0} f(x) = L = \lim_{n \to \infty} L_n$. $\qquad\square$

