# OpenReview forum: "Implicit Bias of Gradient Descent for Non-Homogeneous Deep Networks"
_ICML.cc/2025/Conference — ICML 2025 poster_

### Official Review · Reviewer_ueC9 · 2025-03-09

**Overall Recommendation:** 3

**Summary:**

This paper studies the implicit bias of gradient descent for separable classification task and non-homogeneous model. The class of non-homogeneous models studied in this paper is quite general and covers many of the common deep learning models. The results of this paper shows that, if the model's difference to a homogeneous function is bounded, then the iterates of gradient flow/descent converges in direction to a maximum-margin solution. Most importantly, this paper extends the analysis of implicit bias from homogeneous models to a much more general class of non-homogeneous models that are relevant to deep learning.

**Claims And Evidence:**

The theorem statements are clear and easy to understand. And the paper devotes significant length into illustrating the relevance and generality of its results. I do not have any major issue regarding the claims of the paper.

The main results are not too surprising because they are intuitively the direct extension of the results on homogeneous models. And the high-level proof strategy seems to follow the existing literature. But of course the generalization to these "nearly-homogenous" function class would be very technically involved. In light of this, the theorems all make sense and are solid contribution to the literature.

One big issue, but I think can be fixed easily:
In the definition of $(M, N)$-nearly-homogeneous functions, $M, N$ are assumed to be positive, but Example 4.1.C has $M = 0$?

A few small issues:
1. some of the "theorems," such 5.1, 5.2 and 5.3, should be downgraded to a lemma or proposition. Otherwise the results look too crowded. 2. the definitions and assumptions are bit scattered and so an index of notations/definitions would be a good addition to the appendix.
3. I would like to see $M$-nearly-homogeneous and $(M, N)$-nearly-homogeneous be unified into one definition.
4. I do not quite get how the discussion on o-minimal structure connects with rest of the paper, the author should elaborate on this.

**Essential References Not Discussed:**

I believe that author should also mention this work:

Du, Simon, Jason Lee, Haochuan Li, Liwei Wang, and Xiyu Zhai. "Gradient descent finds global minima of deep neural networks." In International conference on machine learning, pp. 1675-1685. PMLR, 2019.

**Experimental Designs Or Analyses:**

n/a, this is a theory paper

**Methods And Evaluation Criteria:**

n/a, this is a theory paper

**Other Comments Or Suggestions:**

see above

**Other Strengths And Weaknesses:**

see ab

**Questions For Authors:**

see above

**Relation To Broader Scientific Literature:**

This work shows the implicit bias of gradient descent for a large class deep learning models. Many of the existing work in this topic only apply to limited settings such as linear models, deep linear networks, or 2-layer relu networks, etc, so this paper is definitely a very valuable addition.

As I previously discussed, the technical work of this paper is very profound but the high-level strategy is already present in the literature. So, I don' think this work offers new insights on the understanding of implicit bias. But given that the technically work is clearly a step above the current literature, I think a weak accept would be most appropriate for this paper.

**Theoretical Claims:**

see above

---

> ### Author Rebuttal · Authors · 2025-03-31
>
> Thank you for your feedback. For the technical novelty, please refer to the section **Technical novelty** in our response to the reviewer R5kD. For insights of our results, please refer to the section **Insights of our results** in our response to the reviewer NQfK. Below, we address your other questions.
>
> ---
>
> **Q1.** “One big issue, but I think it can be fixed easily: In the definition of (M,N)-nearly-homogeneous functions, M,N are assumed to be positive, but Example 4.1.C has M=0?”
>
> **A1.** Thank you for pointing this out. This issue can be solved by relaxing our near-(M,N)-homogeneity definition to allow $M=0$. Specifically, we call a block s(θ; x) near-(0, N)-homogeneous, if it is independent of θ and near-N-homogeneous in $x$. We will clarify this in the revision.
>
> ---
>
> **Q2.** “Some of the ‘theorems,’ such as 5.1, 5.2, and 5.3, should be downgraded to a lemma or proposition. … The definitions and assumptions are a bit scattered, so an index of notations/definitions would be a good addition to the appendix.”
>
> **A2.**  We will revise the paper according to your suggestions. Thank you.
>
> ---
>
> **Q3.** “I would like to see M-nearly-homogeneous and (M, N)-nearly-homogeneous be unified into one definition.”
>
> **A3.** While these two definitions can be unified, we prefer to keep them separate for the sake of clarity. Since our main results in Sections 3, 5, and 6 only make use of near-M-homogeneity, unifying these two definitions would make these results harder to unpack for the readers.
>
> ---
>
> **Q4.** “I do not quite get how the discussion on o-minimal structure connects with the rest of the paper. The author should elaborate on this.”
>
> **A4.** We will add a paragraph to clarify the role of o-minimal structures. Specifically, the o-minimal structure enables the chain rule of Clarke’s subdifferentials (see Lemma A.6), the existence of a desingularizing function (see Lemma C.14), and Kurdyka–Łojasiewicz inequalities (see Lemmas C.19 and C.20). These are crucial tools in our analysis.
>
> ---
>
> **Q5.** “As I previously discussed, the technical work of this paper is very profound, but the high-level strategy is already present in the literature. So, I don't think this work offers new insights on the understanding of implicit bias.”
>
> **A5.** We respectfully disagree with these comments. For the technical novelty, please refer to the section **Technical novelty** in our response to the reviewer R5kD. For insights of our results, please refer to the section **Insights of our results** in our response to the reviewer NQfK.
>
> ---
>
> **Q6.** “I believe that author should also mention this work … ‘Gradient descent finds global minima of deep neural networks.’....”
>
> **A6.** We will consider discussing this work in the revision. Based on our understanding, the paper you mentioned mainly uses neural tangent kernel (NTK) style techniques to study global convergence. This seems to be quite different from our focus: characterizing the asymptotic implicit bias of GD/GF in near-homogeneous models assuming a strong separability condition. We appreciate it if the reviewer could further elaborate on the connections between this work and ours.
>
> ---

---

> > ### Comment · Reviewer_ueC9 · 2025-04-02
> >
> > I thank the authors for their detailed response and I appreciate their efforts to address the concerns I have raised.
> >
> > Regarding A6, I would like see this discussion be included in the paper because the differences may not be immediately obvious the readers.
> >
> > Overall, I think the contribution of this paper is solid even though it can be hard to digest, and I will keep my score.

---

> > > ### Author Response · Authors · 2025-04-03
> > >
> > > Thank you for recognizing our contribution!  We will add a discussion on the differences between our work and the NTK papers in the revision. With the discussions on the "Insights of our results" and "Technical novelty" from our responses to Reviewers NQfK and R5kD, we believe our contribution has been made very clear. If you find any specific place hard to digest, please let us know, and we will be happy to provide further clarification!

---

### Official Review · Reviewer_R5kD · 2025-03-09

**Overall Recommendation:** 4

**Summary:**

The main contribution of this work is a generalization of previous theoretical results on the implicit bias of gradient descent for homogeneous networks to the case including non-homogeneous networks that satisfy a mild near-homogeneous condition, such as linear layers with an additional bias term (i.e., $Ax + b$) which are not homogeneous but empirically shown to converge to max-margin solution in previous works.

In general, I think the theoretical results in this paper are significant, in the sense that they make the study of implicit bias of gradient descent for classification problem (this topic has been studied for a long time) more comprehensive.

**Claims And Evidence:**

The claims are supported by the mathematical proof.

**Essential References Not Discussed:**

The discussion of essential references is sufficient.

**Experimental Designs Or Analyses:**

This paper does not contain experimental results.

**Methods And Evaluation Criteria:**

Not applicable, as this paper does not have experimental results.

**Other Comments Or Suggestions:**

1. The main results are using exponential loss only, while through the whole paper it is replaced implicitly by the notation $\ell$. Then my question is whether the formulation of exponential loss is necessary.  Can the results still be valid for multi-classification with cross-entropy loss? This is the case for homogeneous networks. Will the non-homogeneity bring any additional difficulties?

2. I think it would be best to have some experimental results to make this paper more complete, but I understand that the current version is also fine.

**Other Strengths And Weaknesses:**

A minor weakness is that there lacks experimental evidence in this paper.

In addition, as many techniques have already been discussed in previous works (since the implicit bias of gradient descent is an important topic), I think the core contribution of this work is how to handle the difficulties brought by the non-homogeneity, hence it would be better for the authors to have a section to discuss the technical novelty.

**Questions For Authors:**

What are the main difficulties when solving the non-homogeneity compared to previous works and how do the authors introduce new techniques to solve them?

**Relation To Broader Scientific Literature:**

The contributions are mainly related to the implicit bias of gradient descent, where the most prominent results in my view are the convergence to a KKT of margin-maximization problem (Lyu & Li, 2019) and the alignment and directional convergence (Ji & Telgarsky, 2020) for homogeneous deep neural networks.

**Theoretical Claims:**

I checked the outline of the proofs, which appears to be correct to me.

---

> ### Author Rebuttal · Authors · 2025-03-31
>
> Thank you for supporting our paper! We address your questions first and then discuss our technical novelty at the end of this response.
>
> ---
>
> **Q1.** “...my question is whether the formulation of exponential loss is necessary. Can the results still be valid for multi-classification with cross-entropy loss?...”
>
> **A1.** Our results can be extended to other loss functions with an exponential tail such as logistic loss. This is because our analysis focuses on the late training regime assuming that the predictor already classifies all data very well. In this regime, only the tail property of the loss function matters. Additionally, using arguments in [Lyu & Li 2020, Appendix G], there is no mathematical difficulty extending our results to multi-class settings using cross entropy loss. We will explain these in detail in the revision.
>
> ---
> ---
>
> ## **Technical novelty**
> We make significant technical innovations in our analysis. Below, we discuss them in our GF and GD analysis, respectively.
>
>
> ### A. Technical novelty in GF analysis
>
> In our gradient flow analysis, we make the following innovations rather than simply combining our near-homogeneity conditions and techniques from [Lyu and Li 2020, Ji and Telgarsky 2020]
>
> **1. Margin improvement.** [Lyu and Li 2020] directly analyzed the smoothed margin (see their equation (3)). This does not work for non-homogenous predictors. Instead, we have to analyze a modified margin (see (6) in lines 213-215). Identifying the correct modification of the margin involves a sharp balance of many error terms. This is quite technical as we aim to cover as many non-homogeneous functions as possible; behind the scenes, we solve multiple ODEs to identify the right formula. Coming up with this modified margin is a significant technical contribution.
>
> **2.Directional convergence.** The analysis in [Ji and Telgarsky 2020] heavily relies on a specific form of GF defined using the minimal norm subgradient; they then analyzed the corresponding spherical and radial components (see their Lemms C.2 and C.3)). This again fails to work for non-homogenous predictors. To address this issue, we have to consider a different form of GF defined with a special subgradient, which enables a special spherical and radial decomposition as shown in equations (21) and (22) and Lemma C.17. Moreover, we have to use an advanced property of the o-minimal structure to show that the choice of sub-gradient does not affect the global path property of GF (see Lemma A.6).
>
> **3.KKT convergence.** Our KKT convergence proof for non-homogenous predictors is new. Although KKT conditions are well established for a homogeneous predictor that maximizes the margin, it is a priori unclear why an optimal non-homogenous predictor admits KKT conditions. To address this issue, we first need to prove the existence of a sufficiently good homogenization of a near-homogeneous predictor (see Theorem 5.1). Then we come up with a set of KKT conditions of a margin maximization program — however, this is not defined using the original predictor — this is defined using its homogenization. Because of this discrepancy, we have to design and analyze new dual variables that involve both the non-homogenous predictor and its homogenization (see equation (44)).
>
>
> ### B. Technical novelty in GD analysis
>
> We solve many additional technical challenges when extending our GF analysis to GD. Due to the space limit, we highlight two of them:
>
> **4.Stepsize condition.** Although [Lyu and Li 2020] handled GD in the homogenous case, their stepsize assumption depends on a “constant” $C_{\eta}$ (see their Assumption (S5) in Appendix E.1), which turns out to be function of the initial margin (see their Page 32), preventing the stepsize to be large. In comparison, our assumptions on the stepsize are weak, determined by an explicit and interpretable separability condition (see Assumption 5). To achieve this, we have to conduct a technical, yet much tighter, analysis of the GD path (see our Theorem F.9 and Lemma F.10). This is beyond the techniques from [Lyu and Li 2020].
>
> **5.Directional convergence.** Our directional convergence analysis for GD is new. Note that [Ji and Telgarsky 2020] only analyzed GF but not GD, where tools from differential equations are unavailable. To address this issue, we construct an arc by connecting all GD iterates using line segments. Even so, analyzing the directional limit of this arc is extremely technical, involving careful estimations of the spherical and the radial parts of this arc (see Lemmas F.16 and F.17, which are not needed for GF). These additional technical challenges do not exist in the GF analysis but must be addressed in the GD analysis.
> We will highlight these in the revision.

---

> > ### Comment · Reviewer_R5kD · 2025-04-04
> >
> > I thank the authors for the detailed response. I do not have further questions, and I support "Accept" as before.

---

### Official Review · Reviewer_NQfK · 2025-03-14

**Overall Recommendation:** 4

**Summary:**

The paper characterizes the implicit bias of non-homogeneous deep models trained with gradient-flow or gradient-descent to minimize an exponential loss, under some seperability and near-homogeneity conditions. The results are extensions of previous works that found similar implicit bias in strictly homogeneous models.


## Update After Rebuttal
I have increased my score, assuming that the authors will incorporate the changes discussed during the rebuttal into the final version of the paper.

**Claims And Evidence:**

The paper is theoretical in nature, and supports its claims with clear and convincing proofs.

**Essential References Not Discussed:**

I do not think that there are any essential references not discussed.

**Experimental Designs Or Analyses:**

There are no experiments in the paper.

**Methods And Evaluation Criteria:**

Yes.

**Other Comments Or Suggestions:**

1. Possible typo in line 136-l.

2. Typo in line 557.

3. Typo in line 598.

4. Typo in line 344-r.

5. Possible typo in line 885.

6. Is there an intuitive meaning to $p_a$? If so, it should be discussed briefly in the main paper.

7. Possible typo in line 1043.

8. Typo in line 4008.

9. Line 1187 — $\zeta_t$ is not the curve swept by the normalized parameters, but its length.

10. Line 1212 — which proof does “We skip the proof here” refer to? Seems to be regarding Lemma C.14
which is proved later on, in page 27.

11. Line 1624 — what does “for all $i \in [n]$” refer to? Is the supremum also being taken over $i$?

12. Proof of Lemma C.22 — there is a mismatch between the value of $\delta$ in line 1630, and the RHS in line 1681.

13. Starting from page 35, there seems to be a change in the naming of the assumptions, e.g. (A1)-(A3), (B1)-(B3), which does not match the main text and is not clearly addressed.

14. Line 1925 — “Proposition” should be “Lemma”.

**Other Strengths And Weaknesses:**

The paper is well organized, and clearly written.

**Questions For Authors:**

As shown in Theorem 5.1, Assumption 1 implies that the model is asymptotically $M$-homogeneous, and gives conditions for the increase of the norm of the parameters in Lemma C.4. In light of this, Assumption 2 seems to be “assuming what needs to be proven”. Specifically, it assumes that the parameters are already in a regime where (1) the dominant term is of order $>M-1$ (since $deg (p_a) = M-1$, and this exponent bounds the loss), and (2) due to Lemma C.4, that the norm of the parameters shall increase such that this behavior persists. Together, this implies that the main result applies only in the regime where the model is already “practically” homogeneous.

This is explicitly said in line 198-r.

In light of this, the contributions of the paper given previous results seem marginal, as, in addition, and as far as I noticed, there is no significant technical novelty beyond that of [1, 2].

Can you please clarify whether this observation is correct? Is there some delicate technical issue that I am missing?

[1] Lyu, Kaifeng, and Jian Li. "Gradient Descent Maximizes the Margin of Homogeneous Neural Networks." International Conference on Learning Representations.

[2] Ji, Ziwei, and Matus Telgarsky. "Directional convergence and alignment in deep learning." Advances in Neural Information Processing Systems 33 (2020): 17176-17186.

**Relation To Broader Scientific Literature:**

The paper contributes to the study of the implicit bias of gradient-based training of deep neural networks. Specifically, the paper extends previous results from the strictly-homogeneous setting to the nearly-homogeneous setting, thus extending the types of models to which the results apply.

**Theoretical Claims:**

I checked the proofs in Sections C.1-C.4, only briefly read the proofs in the other appendices, and did not find any fundamental issues.

---

> ### Author Rebuttal · Authors · 2025-03-31
>
> We thank the reviewer for their detailed comments and will fix all grammar issues and typos in the revision. For the technical novelty, please refer to the section **Technical novelty** in our response to the reviewer R5kD. For insights of our results, please refer to the end of this response. Below, we address your other questions.
>
> ---
>
> **Q1.** "Is there an intuitive meaning to $p_a$? If so, it should be discussed briefly in the main paper."
>
> **A1.** $p_a$ is used in our Assumption 2. This choice of $p_a$ guarantees that the class of near-homogeneous functions under Assumptions 1 and 2 all behave close enough to a homogeneous function along the GF path for $t\ge s$. Concretely, as shown in Lines 945-950, $p_a$ is chosen such that $g:= f - p_a$ satisfies a one-sided inequality of the homogeneous condition (see equation (3) in Line 152). This is a crucial design choice that enables our proof.
>
> ---
>
> **Q2.** "Possible typo in line 136-l, line 885 and line 1043."
>
> **A2.** Thanks for pointing these out. These are all typos. (i) The period in line 136-l should be a comma; (ii) $\log \phi$ in line 885 should be replaced by $\phi$; (iii) In line 1043, $G$ in the denominator should be $G_t$. We will make sure to correct them and other typos in the revision.
>
> ---
>
> **Q3.** Starting from page 35, there seems to be a change in the naming of the assumptions, e.g. (A1)-(A3), (B1)-(B3), which does not match the main text and is not clearly addressed.
>
> **A3.** We will fix this in the revision.
>
> ---
>
> **Q4.** “Line 1212 — which proof does “We skip the proof here” refer to? Seems to be regarding Lemma C.14 which is proved later on, in page 27.”
>
> **A4.** You are correct. The proof of Lemma C.14 is on page 27. We will polish this part (and all the appendix) carefully in the revision.
>
> ---
>
> **Q5.** “Line 1624 — what does “for all $i \in [n]$” refer to? Is the supremum also being taken over $i$?”
>
> **A5.** This is a typo. In this place, we define $B\coloneqq [ (\gamma^{GF}(\theta_s)]^{-1/M}$. Then we can verify that
> $$
>  \sup_{t \ge s, i \in [n]} \bar{f}_{M,i}^{-1/M} (\theta_t) \cdot \\|\theta_t\\|_2 \le B.
> $$
> This suffices for the remaining proof. We will fix this in the revision.
>
> ---
>
> **Q6.** "Proof of Lemma C.22 — there is a mismatch between the value of  in line 1630, and the RHS in line 1681."
>
> **A6.** This is a typo. The correct formula for $\delta$ is
> $$
> \delta :=  n B^2 \frac{1+2 p_a(\rho_t)}{M \bar f_{M,\min }(\theta_t)}.
> $$
>
> ---
>
> **Q7.** "In light of this, the contributions of the paper given previous results seem marginal, as, in addition, and as far as I noticed, there is no significant technical novelty beyond that of [1, 2]. "
>
> **A7.** We respectfully disagree with your assertions that our contributions are marginal and our technical novelty is not significant. For the technical novelty, please refer to the section **Technical novelty** in our response to the reviewer R5kD. For insights of our results, please refer to the end of this response.
>
> ---
> ---
>
> ## **Insights of our results**
>
> We highlight insights from our results in the following three aspects.
>
> **1. A good definition = good insights.** As a first step towards understanding implicit bias for non-homogeneous models, identifying the proper function classes for which meaningful theoretical insights can be extracted is already a challenging task. There are a few attempts in prior literature, but they seem to be far less fundamental than ours (see paragraph “Non-homogeneous Predictors” in Section 1.1). Our Definition 1 provides a natural quantification of the homogeneity error, covers a large class of networks, and has rich implications on the implicit bias. Such a powerful definition is rare, which already carries significant insights in our opinion.
>
> **2. Insights for understanding implicit bias.** Our results suggest that a near-homogeneous predictor has the same implicit bias as its companion homogenization (see Theorem 3.4 and 5.1). This provides an important insight: understanding the implicit bias of a non-homogenous predictor can be reduced to understanding that of its homogenization. Compared to a generic non-homogenous model, its homogenization is much simpler to study (despite still being challenging).
>
> **3. Broader implications beyond DL theory.** To the best of our knowledge, our near-homogeneity definition is novel and cannot be found even in pure math literature. As homogeneity is a fundamental math concept, widely used in many areas (such as PDE, harmonic analysis, and semi-algebraic geometry), our notion of near-homogeneity could motivate broader mathematical research beyond deep learning theory. In this regard, we believe technical tools established in our paper, integrating the near-homogeneity with o-minimal structures and non-smooth analysis (see also section **Technical novelty** in our response to reviewer R5kD), might be of broader interest.
> We will include these discussions in the revision.

---

> > ### Comment · Reviewer_NQfK · 2025-04-05
> >
> > I thank the authors for the detailed response.
> >
> > I agree with the authors' claims regarding the insights and technical novelty, in particular about the novelty of the definitions, and will keep my score.
> >
> > In light of this discussion, I think that future revisions of the paper can benefit from a more thorough discussion of the necessity of Assumption 2. In particular, the assumption can be separated into two components — 1) a separability condition that is also used for homogeneous predictors, which has the intuitive interpretation of perfectly classifying the training set, and 2) an assumption on the margin/magnitude of the predictor. Is the additional assumption only an artifact of the analysis, or is it a fundamental requirement for near-homogeneous models?

---

> > > ### Author Response · Authors · 2025-04-06
> > >
> > > We are glad that you agree with the insights of our results and our technical novelties. If you have any further concerns, please let us know, and we will be happy to discuss them more.
> > >
> > > Your interpretation of our Assumption 2 is correct. Regarding your follow-up question, we argue that our Assumption 2 is necessary (in the worst-case sense) for genetic near-homogeneous models to exhibit implicit bias.
> > >
> > > To see this, consider the following simple example: $\theta\in \mathbb{R}$, $(x,y)=(1,1)$, $p_a(|\theta|) = M |\theta|^{M-1}$ for an odd integer $M\ge 3$, and $f(\theta) = \theta^M+p_a(|\theta|)$.
> > > It is clear that such a predictor satisfies Assumption 1. Moreover, our Assumption 2 is equivalent to
> > > $$L(\theta_s) = \exp( - f(\theta_s) ) < \exp( -  p_a(|\theta_s|) )
> > > \quad \Longleftrightarrow\quad  \theta_s^M > 0
> > > \quad \Longleftrightarrow \quad \theta_s > 0.
> > > $$
> > > If the above condition does not hold, then $\theta_s \le 0$. Notice that $0$ is a stationary point for $L(\theta)$. So GF initialized from $\theta_s$ cannot produce positive parameters in the future, that is, $\theta_t \le 0$ for every $t\ge s$. Hence, GF cannot minimize the loss or exhibit any implicit bias suggested by our theorems.
> > >
> > > This explanation should clarify your concerns. We will add this discussion in the revision.

---

### Official Review · Reviewer_Uxz2 · 2025-03-14

**Overall Recommendation:** 3

**Summary:**

This paper establishes the asymptotic implicit bias of gradient descent for generic non-homogeneous deep networks under exponential loss. Specifically, the authors show that (1) the normalized margin increases nearly monotonically, (2) the direction of the parameters converges, and (3) the directional limit satisfies the KKT conditions of the margin maximization problem.

**Claims And Evidence:**

Yes.

**Essential References Not Discussed:**

In the subarea of implicit bias, the key contribution of this paper is its consideration of a more general non-homogeneous model instead of a homogeneous one. Regarding the related work on homogeneous predictors, there are also special cases, such as shallow neural networks, with similar results. I think it would be helpful if the authors also referenced those papers.

**Experimental Designs Or Analyses:**

No experiments.

**Methods And Evaluation Criteria:**

Yes.

**Other Comments Or Suggestions:**

A small suggestion about the writing style: Since most people in this subarea are already familiar with the key results and proof techniques from previous works, I find the explanation of the implications of the theorems in this paper somewhat plain and less informative. In many places, the paper simply states that the results generalize previous work, but that is something the audience likely already knows. Personally, I would appreciate more discussion on the key insights and motivation behind introducing the near-homogeneity measure. For example, how does defining a polynomial upper bound make the proof more tractable compared to previous works? This kind of explanation would help readers better assess the contribution of this paper—specifically, whether the introduction of this new concept brings any novel analytical tools to the field.

**Other Strengths And Weaknesses:**

Strengths:
1. Generalization to Non-Homogeneous Models: The paper extends prior results from homogeneous models to a more general non-homogeneous setting, contributing to a broader understanding of implicit bias.
2. Interesting Definition of Near-Homogeneity: The introduction of the near-homogeneity measure (Definition 1) is novel and could provide insights into implicit bias, depending on its broader applicability.

Weaknesses:
1. Limited Insight into Key Design Choices: While the paper presents generalizations of existing results, the motivation behind certain technical definitions—such as the choice of a polynomial upper bound—could be better justified. Explaining how this choice impacts proof tractability or brings new analytical tools would strengthen the contribution.

**Questions For Authors:**

1. I am wondering whether the upper bound on the degree of the polynomial in Definition 1 can be relaxed—either to some function of M,  or even made independent of M. If not, why does the upper bound have to be M, and is there any intuition behind this choice?
2. Why is it helpful to define the upper bound as a polynomial of the weight norm? Did you use any specific tools for handling polynomials? If not, why not simply set the upper bound as $O(||\theta||_2^M)$ instead of introducing the polynomial concept $p'(||\theta||_2)$?

**Relation To Broader Scientific Literature:**

The major contribution of this paper is to generalize previous results on simpler models to similar conclusions for a more general model. Additionally, I find the near-homogeneity definition (Definition 1) somewhat interesting, though I am unsure whether it was designed to make the proof more tractable or if it provides deeper insight.

**Theoretical Claims:**

No, but the results look reasonable to me.

---

> ### Author Rebuttal · Authors · 2025-03-31
>
> Thank you for your thoughtful comments and suggestions on the writing style. We will include more discussions on intuitions and technical innovations for our theorems in the revision. For the technical novelty, please refer to the section **Technical novelty** in our response to the reviewer R5kD. For insights of our results, please refer to the section **Insights of our results** in our response to the reviewer NQfK. Below, we address your other questions.
>
> ---
>
> **Q1.** “I am wondering whether the upper bound on the degree of the polynomial in Definition 1 can be relaxed…why does the upper bound have to be $M$, and is there any intuition behind this choice?”
>
> **A1.** In Definition 1, the upper bound of the degrees of the polynomials have to be $M$. We justify this from the following two aspects.
>
> First, we discuss its importance. If we relax the degree upper bound from $M$ to $M+1$, then every sufficiently smooth function $f(\theta; x)$ that is uniformly bounded by $O(\\\| \\theta \\\|\_2\^M)$ for large $\\\| \\theta \\\|\_2$ satisfies Definition 1. This includes predictors that do not admit a homogenization (see Theorem 5.1). Note that homogenization plays a crucial role in our analysis. For instance, without homogenization, it seems impossible to even define a KKT problem, let alone prove the implicit bias of KKT directional convergence of GF/GD.
>
> Second, we explain the intuition for choosing $M$ as the degree upper bound. In Definition 1, the polynomial $p’$ quantifies the deviation of $f$ from an $M$-homogeneous function (see $f_M$ in Theorem 5.1). Given that we require $f$ to be “near-homogeneous”, it is natural to assume that the discrepancy between $f$ and $f_M$ (quantified by $p’$) is of an order lower than $f_M$. The natural assumption, $\deg p \leq M$, suffices for this purpose.
>
> We will clarify these in the revision.
>
> ---
>
> **Q2.** “Why is it helpful to define the upper bound as a polynomial of the weight norm? Did you use any specific tools for handling polynomials? If not, why not simply set the upper bound as $O( \\| \\theta \\|_2^M)$ instead of introducing the polynomial concept $p' ( \\| \\theta \\|_2) $?”
>
> **A2.** Good question. But before we address it, we would like to point out that the error upper bound is $p' ( \\| \\theta \\|\_2) = O(\\| \\theta \\|\_2^{M-1})$ in our Definition 1. So in your question, the error upper bound should be replaced by $o( \\| \\theta \\|_2^{M})$ instead of $O( \\| \\theta \\|_2^{M})$. We suspect this is a typo. But please let us know if we misunderstood your question.
>
> We choose to define the upper bound as $p’$ in Definition 1, primarily for the simplicity of the exposition. This choice allows us to explicitly define the function $p_a$ (see Eq. (4)), which streamlines the theorem statements and their proof by making the inequalities more explicit and tractable.
>
> Note that our results are not limited to this specific polynomial form. If the upper bound in Definition 1 is replaced by a function uniformly bounded by $O(\\\| \\theta \\\|\_2^{M-1})$ for large $\\\| \\theta \\\|\_2$, our analysis still goes through when Assumption 2 is adjusted accordingly. However, we do not feel this provides new information compared to our current version.
>
> In general, if the error upper bound is only controlled by $o( \\\| θ \\\|\_2\^{M})$, we expect additional regularity conditions are needed to carry out the analysis. As our main focus is neural networks, we feel our current definition is both clean and sufficiently broad. We leave it as future work to further extend our results.
> We will include these discussions in the revision.
>
> ---

---

### Decision · Program_Chairs · 2025-05-01

**Decision:**

Accept (poster)

**Comment:**

This paper studies the implicit bias of non-homogeneous deep networks under exponential loss, and it provides a generalization of earlier results for homogeneous networks leveraging a near-homogeneity condition. In particular, the main results are that the normalized margin increases almost monotonically, the parameter direction converges and KKT conditions are satisfied by the directional limit.

All the reviewers agree that the results are interesting, novel and rather strong. The generalization to non-homogeneous networks is relevant, even with the caveat of a near-homogeneity requirement that may be a bit artificial. The authors' feedback has resolved most issues and all reviewers are positive about accepting the paper (with varying levels of enthusiasm). I concur: this is clearly a paper that should be accepted to ICML and will be a nice contribution to the conference. I would like to encourage the authors to incorporate the feedback received throughout the review process in the camera ready.